# Self-Normalizing Neural Networks

**Günter Klambauer**  **Thomas Unterthiner**  **Andreas Mayr**

**Sepp Hochreiter**
LIT AI Lab & Institute of Bioinformatics,
Johannes Kepler University Linz
A-4040 Linz, Austria
{klambauer,unterthiner,mayr,hochreit}@bioinf.jku.at

## Abstract

Deep Learning has revolutionized vision via convolutional neural networks (CNNs) and natural language processing via recurrent neural networks (RNNs). However, success stories of Deep Learning with standard feed-forward neural networks (FNNs) are rare. FNNs that perform well are typically shallow and, therefore cannot exploit many levels of abstract representations. We introduce self-normalizing neural networks (SNNs) to enable high-level abstract representations. While batch normalization requires explicit normalization, neuron activations of SNNs automatically converge towards zero mean and unit variance. The activation function of SNNs are "scaled exponential linear units" (SELUs), which induce self-normalizing properties. Using the Banach fixed-point theorem, we prove that activations close to zero mean and unit variance that are propagated through many network layers will converge towards zero mean and unit variance — even under the presence of noise and perturbations. This convergence property of SNNs allows to (1) train deep networks with many layers, (2) employ strong regularization schemes, and (3) to make learning highly robust. Furthermore, for activations not close to unit variance, we prove an upper and lower bound on the variance, thus, vanishing and exploding gradients are impossible. We compared SNNs on (a) 121 tasks from the UCI machine learning repository, on (b) drug discovery benchmarks, and on (c) astronomy tasks with standard FNNs, and other machine learning methods such as random forests and support vector machines. For FNNs we considered (i) ReLU networks without normalization, (ii) batch normalization, (iii) layer normalization, (iv) weight normalization, (v) highway networks, and (vi) residual networks. SNNs significantly outperformed all competing FNN methods at 121 UCI tasks, outperformed all competing methods at the Tox21 dataset, and set a new record at an astronomy data set. The winning SNN architectures are often very deep.

## 1 Introduction

Deep Learning has set new records at different benchmarks and led to various commercial applications [21, 26]. Recurrent neural networks (RNNs) [15] achieved new levels at speech and natural language processing, for example at the TIMIT benchmark [10] or at language translation [29], and are already employed in mobile devices [24]. RNNs have won handwriting recognition challenges (Chinese and Arabic handwriting) [26, 11, 4] and Kaggle challenges, such as the "Grasp-and Lift EEG" competition. Their counterparts, convolutional neural networks (CNNs) [20] excel at vision and video tasks. CNNs are on par with human dermatologists at the visual detection of skin cancer [8]. The visual processing for self-driving cars is based on CNNs [16], as is the visual input to AlphaGo which has beaten one

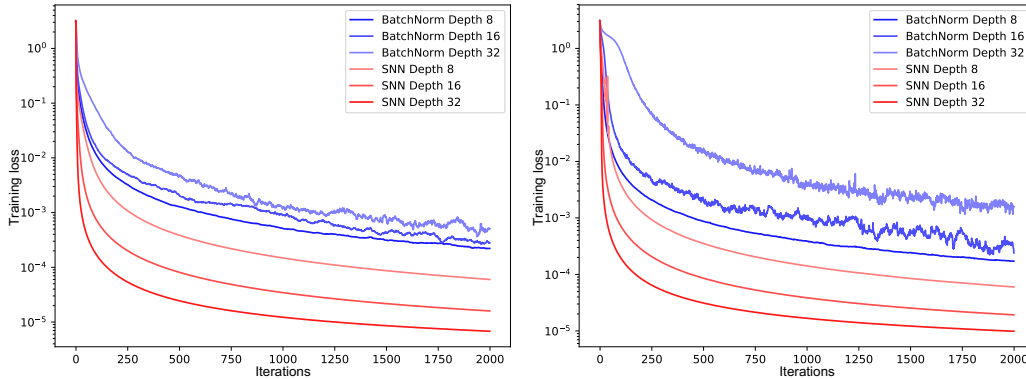

Figure 1: The left panel and the right panel show the training error (y-axis) for feed-forward neural networks (FNNs) with batch normalization (BatchNorm) and self-normalizing networks (SNN) across update steps (x-axis) on the MNIST dataset the CIFAR10 dataset, respectively. We tested networks with 8, 16, and 32 layers and learning rate 1e-5. FNNs with batch normalization exhibit high variance due to perturbations. In contrast, SNNs do not suffer from high variance as they are more robust to perturbations and learn faster.

of the best human GO players [27]. At vision challenges, CNNs are constantly winning, for example at the large ImageNet competition [19, 13], but also almost all Kaggle vision challenges, such as the "Diabetic Retinopathy" and the "Right Whale" challenges [7, 12].

However, looking at Kaggle challenges that are not related to vision or sequential tasks, gradient boosting, random forests, or support vector machines (SVMs) are winning most of the competitions. Deep Learning is notably absent, and for the few cases where FNNs won, they are shallow. For example, the HIGGS challenge, the Merck Molecular Activity challenge, and the Tox21 Data challenge were all won by FNNs with at most four hidden layers. Surprisingly, it is hard to find success stories with FNNs that have many hidden layers, though they would allow for different levels of abstract representations of the input [2].

To robustly train very deep CNNs, batch normalization evolved into a standard to normalize neuron activations to zero mean and unit variance [17]. Layer normalization [1] also ensures zero mean and unit variance, while weight normalization [25] ensures zero mean and unit variance if in the previous layer the activations have zero mean and unit variance. Natural neural networks [6] also aim at normalizing the variance of activations by reparametrization of the weights. However, training with normalization techniques is perturbed by stochastic gradient descent (SGD), stochastic regularization (like dropout), and the estimation of the normalization parameters. Both RNNs and CNNs can stabilize learning via weight sharing, therefore they are less prone to these perturbations. In contrast, FNNs trained with normalization techniques suffer from these perturbations and have high variance in the training error (see Figure 1). This high variance hinders learning and slows it down. Furthermore, strong regularization, such as dropout, is not possible as it would further increase the variance which in turn would lead to divergence of the learning process. We believe that this sensitivity to perturbations is the reason that FNNs are less successful than RNNs and CNNs.

Self-normalizing neural networks (SNNs) are robust to perturbations and do not have high variance in their training errors (see Figure 1). SNNs push neuron activations to zero mean and unit variance thereby leading to the same effect as batch normalization, which enables to robustly learn many layers. SNNs are based on scaled exponential linear units "SELUs" which induce self-normalizing properties like variance stabilization which in turn avoids exploding and vanishing gradients.

## 2 Self-normalizing Neural Networks (SNNs)

**Normalization and SNNs.** For a neural network with activation function $f$, we consider two consecutive layers that are connected by a weight matrix $\boldsymbol{W}$. Since the input to a neural network is a random variable, the activations $\boldsymbol{x}$ in the lower layer, the network inputs $\boldsymbol{z} = \boldsymbol{W}\boldsymbol{x}$, and the activations $\boldsymbol{y} = f(\boldsymbol{z})$ in the higher layer are random variables as well. We assume that all activations

$x_i$ of the lower layer have mean $\mu := \mathrm{E}(x_i)$ and variance $\nu := \mathrm{Var}(x_i)$. An activation $y$ in the higher layer has mean $\tilde{\mu} := \mathrm{E}(y)$ and variance $\tilde{\nu} := \mathrm{Var}(y)$. Here $\mathrm{E}(.)$ denotes the expectation and $\mathrm{Var}(.)$ the variance of a random variable. A single activation $y = f(z)$ has net input $z = \boldsymbol{w}^T\boldsymbol{x}$. For $n$ units with activation $x_i, 1 \leqslant i \leqslant n$ in the lower layer, we define $n$ times the mean of the weight vector $\boldsymbol{w} \in \mathbb{R}^n$ as $\omega := \sum_{i=1}^{n} w_i$ and $n$ times the second moment as $\tau := \sum_{i=1}^{n} w_i^2$.

We consider the mapping $g$ that maps mean and variance of the activations from one layer to mean and variance of the activations in the next layer $g : (\mu, \nu) \mapsto (\tilde{\mu}, \tilde{\nu})$. Normalization techniques like batch, layer, or weight normalization ensure a mapping $g$ that keeps $(\mu, \nu)$ and $(\tilde{\mu}, \tilde{\nu})$ close to predefined values, typically $(0, 1)$.

**Definition 1** (Self-normalizing neural net). *A neural network is self-normalizing if it possesses a mapping $g : \Omega \mapsto \Omega$ for each activation $y$ that maps mean and variance from one layer to the next and has a stable and attracting fixed point depending on $(\omega, \tau)$ in $\Omega$. Furthermore, the mean and the variance remain in the domain $\Omega$, that is $g(\Omega) \subseteq \Omega$, where $\Omega = \{(\mu, \nu) \mid \mu \in [\mu_{\min}, \mu_{\max}], \nu \in [\nu_{\min}, \nu_{\max}]\}$. When iteratively applying the mapping $g$, each point within $\Omega$ converges to this fixed point.*

Therefore, we consider activations of a neural network to be normalized, if both their mean and their variance across samples are within predefined intervals. If mean and variance of $\boldsymbol{x}$ are already within these intervals, then also mean and variance of $\boldsymbol{y}$ remain in these intervals, i.e., the normalization is transitive across layers. Within these intervals, the mean and variance both converge to a fixed point if the mapping $g$ is applied iteratively.

Therefore, SNNs keep normalization of activations when propagating them through layers of the network. The normalization effect is observed across layers of a network: in each layer the activations are getting closer to the fixed point. The normalization effect can also observed be for two fixed layers across learning steps: perturbations of lower layer activations or weights are damped in the higher layer by drawing the activations towards the fixed point. If for all $y$ in the higher layer, $\omega$ and $\tau$ of the corresponding weight vector are the same, then the fixed points are also the same. In this case we have a unique fixed point for all activations $y$. Otherwise, in the more general case, $\omega$ and $\tau$ differ for different $y$ but the mean activations are drawn into $[\mu_{\min}, \mu_{\max}]$ and the variances are drawn into $[\nu_{\min}, \nu_{\max}]$.

**Constructing Self-Normalizing Neural Networks.** We aim at constructing self-normalizing neural networks by adjusting the properties of the function $g$. Only two design choices are available for the function $g$: (1) the activation function and (2) the initialization of the weights.

For the activation function, we propose "scaled exponential linear units" (SELUs) to render a FNN as self-normalizing. The SELU activation function is given by

$$\mathrm{selu}(x) \;=\; \lambda \begin{cases} x & \text{if } x > 0 \\ \alpha e^x - \alpha & \text{if } x \leqslant 0 \end{cases} . \tag{1}$$

SELUs allow to construct a mapping $g$ with properties that lead to SNNs. SNNs cannot be derived with (scaled) rectified linear units (ReLUs), sigmoid units, $\tanh$ units, and leaky ReLUs. The activation function is required to have (1) negative and positive values for controlling the mean, (2) saturation regions (derivatives approaching zero) to dampen the variance if it is too large in the lower layer, (3) a slope larger than one to increase the variance if it is too small in the lower layer, (4) a continuous curve. The latter ensures a fixed point, where variance damping is equalized by variance increasing. We met these properties of the activation function by multiplying the exponential linear unit (ELU) [5] with $\lambda > 1$ to ensure a slope larger than one for positive net inputs.

For the weight initialization, we propose $\omega = 0$ and $\tau = 1$ for all units in the higher layer. The next paragraphs will show the advantages of this initialization. Of course, during learning these assumptions on the weight vector will be violated. However, we can prove the self-normalizing property even for weight vectors that are not normalized, therefore, the self-normalizing property can be kept during learning and weight changes.

**Deriving the Mean and Variance Mapping Function $g$.** We assume that the $x_i$ are independent from each other but share the same mean $\mu$ and variance $\nu$. Of course, the independence assumptions is not fulfilled in general. We will elaborate on the independence assumption below. The network

input $z$ in the higher layer is $z = \boldsymbol{w}^T \boldsymbol{x}$ for which we can infer the following moments $\mathrm{E}(z) = \sum_{i=1}^n w_i \, \mathrm{E}(x_i) = \mu \, \omega$ and $\mathrm{Var}(z) = \mathrm{Var}(\sum_{i=1}^n w_i \, x_i) = \nu \, \tau$, where we used the independence of the $x_i$. The net input $z$ is a weighted sum of independent, but not necessarily identically distributed variables $x_i$, for which the central limit theorem (CLT) states that $z$ approaches a normal distribution: $z \sim \mathcal{N}(\mu\omega, \sqrt{\nu\tau})$ with density $p_\mathrm{N}(z; \mu\omega, \sqrt{\nu\tau})$. According to the CLT, the larger $n$, the closer is $z$ to a normal distribution. For Deep Learning, broad layers with hundreds of neurons $x_i$ are common. Therefore the assumption that $z$ is normally distributed is met well for most currently used neural networks (see Supplementary Figure S7). The function $g$ maps the mean and variance of activations in the lower layer to the mean $\tilde{\mu} = \mathrm{E}(y)$ and variance $\tilde{\nu} = \mathrm{Var}(y)$ of the activations $y$ in the next layer:

$$g : \begin{pmatrix} \mu \\ \nu \end{pmatrix} \mapsto \begin{pmatrix} \tilde{\mu} \\ \tilde{\nu} \end{pmatrix} : \quad \tilde{\mu}(\mu, \omega, \nu, \tau) = \int_{-\infty}^{\infty} \mathrm{selu}(z) \, p_\mathrm{N}(z; \mu\omega, \sqrt{\nu\tau}) \, \mathrm{d}z \tag{2}$$

$$\tilde{\nu}(\mu, \omega, \nu, \tau) = \int_{-\infty}^{\infty} \mathrm{selu}(z)^2 \, p_\mathrm{N}(z; \mu\omega, \sqrt{\nu\tau}) \, \mathrm{d}z - \tilde{\mu}^2 \,.$$

These integrals can be analytically computed and lead to following mappings of the moments:

$$\tilde{\mu} = \frac{1}{2}\lambda \left( (\mu\omega) \operatorname{erf}\left( \frac{\mu\omega}{\sqrt{2}\sqrt{\nu\tau}} \right) + \right. \tag{3}$$

$$\left. \alpha \, e^{\mu\omega + \frac{\nu\tau}{2}} \operatorname{erfc}\left( \frac{\mu\omega + \nu\tau}{\sqrt{2}\sqrt{\nu\tau}} \right) - \alpha \operatorname{erfc}\left( \frac{\mu\omega}{\sqrt{2}\sqrt{\nu\tau}} \right) + \sqrt{\frac{2}{\pi}}\sqrt{\nu\tau}e^{-\frac{(\mu\omega)^2}{2(\nu\tau)}} + \mu\omega \right)$$

$$\tilde{\nu} = \frac{1}{2}\lambda^2 \left( ((\mu\omega)^2 + \nu\tau)\left( 2 - \operatorname{erfc}\left( \frac{\mu\omega}{\sqrt{2}\sqrt{\nu\tau}} \right) \right) + \alpha^2\left( -2e^{\mu\omega + \frac{\nu\tau}{2}} \operatorname{erfc}\left( \frac{\mu\omega + \nu\tau}{\sqrt{2}\sqrt{\nu\tau}} \right) \right. \right. \tag{4}$$

$$\left. \left. + e^{2(\mu\omega + \nu\tau)} \operatorname{erfc}\left( \frac{\mu\omega + 2\nu\tau}{\sqrt{2}\sqrt{\nu\tau}} \right) + \operatorname{erfc}\left( \frac{\mu\omega}{\sqrt{2}\sqrt{\nu\tau}} \right) \right) + \sqrt{\frac{2}{\pi}}(\mu\omega)\sqrt{\nu\tau}e^{-\frac{(\mu\omega)^2}{2(\nu\tau)}} \right) - (\tilde{\mu})^2$$

**Stable and Attracting Fixed Point $(0, 1)$ for Normalized Weights.** We assume a normalized weight vector $\boldsymbol{w}$ with $\omega = 0$ and $\tau = 1$. Given a fixed point $(\mu, \nu)$, we can solve equations Eq. (3) and Eq. (4) for $\alpha$ and $\lambda$. We chose the fixed point $(\mu, \nu) = (0, 1)$, which is typical for activation normalization. We obtain the fixed point equations $\tilde{\mu} = \mu = 0$ and $\tilde{\nu} = \nu = 1$ that we solve for $\alpha$ and $\lambda$ and obtain the solutions $\alpha_{01} \approx 1.6733$ and $\lambda_{01} \approx 1.0507$, where the subscript 01 indicates that these are the parameters for fixed point $(0, 1)$. The analytical expressions for $\alpha_{01}$ and $\lambda_{01}$ are given in Supplementary Eq. (8). We are interested whether the fixed point $(\mu, \nu) = (0, 1)$ is stable and attracting. If the Jacobian of $g$ has a norm smaller than 1 at the fixed point, then $g$ is a contraction mapping and the fixed point is stable. The (2x2)-Jacobian $\mathcal{J}(\mu, \nu)$ of $g : (\mu, \nu) \mapsto (\tilde{\mu}, \tilde{\nu})$ evaluated at the fixed point $(0, 1)$ with $\alpha_{01}$ and $\lambda_{01}$ is $\mathcal{J}(0, 1) = ((0.0, 0.088834), (0.0, 0.782648))$. The spectral norm of $\mathcal{J}(0, 1)$ (its largest singular value) is $0.7877 < 1$. That means $g$ is a contraction mapping around the fixed point $(0, 1)$ (the mapping is depicted in Figure 2). Therefore, $(0, 1)$ is a stable fixed point of the mapping $g$. The norm of the Jacobian also determines the convergence rate as a consequence of the Banach fixed point theorem. The convergence rate around the fixed point $(0,1)$ is about $0.78$. In general, the convergence rate depends on $\omega, \mu, \nu, \tau$ and is between $0.78$ and $1$.

**Stable and Attracting Fixed Points for Unnormalized Weights.** A normalized weight vector $\boldsymbol{w}$ cannot be ensured during learning. For SELU parameters $\alpha = \alpha_{01}$ and $\lambda = \lambda_{01}$, we show in the next theorem that if $(\omega, \tau)$ is close to $(0, 1)$, then $g$ still has an attracting and stable fixed point that is close to $(0, 1)$. Thus, in the general case there still exists a stable fixed point which, however, depends on $(\omega, \tau)$. If we restrict $(\mu, \nu, \omega, \tau)$ to certain intervals, then we can show that $(\mu, \nu)$ is mapped to the respective intervals. Next we present the central theorem of this paper, from which follows that SELU networks are self-normalizing under mild conditions on the weights.

**Theorem 1** (Stable and Attracting Fixed Points). *We assume $\alpha = \alpha_{01}$ and $\lambda = \lambda_{01}$. We restrict the range of the variables to the following intervals $\mu \in [-0.1, 0.1]$, $\omega \in [-0.1, 0.1]$, $\nu \in [0.8, 1.5]$, and $\tau \in [0.95, 1.1]$, that define the functions' domain $\Omega$. For $\omega = 0$ and $\tau = 1$, the mapping Eq. (2) has the stable fixed point $(\mu, \nu) = (0, 1)$, whereas for other $\omega$ and $\tau$ the mapping Eq. (2) has a stable and attracting fixed point depending on $(\omega, \tau)$ in the $(\mu, \nu)$-domain: $\mu \in [-0.03106, 0.06773]$ and*

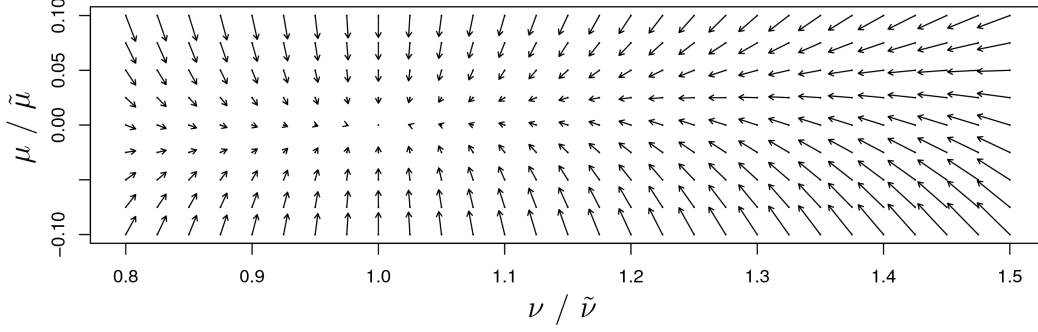

Figure 2: For $\omega = 0$ and $\tau = 1$, the mapping $g$ of mean $\mu$ ($x$-axis) and variance $\nu$ ($y$-axis) to the next layer's mean $\tilde{\mu}$ and variance $\tilde{\nu}$ is depicted. Arrows show in which direction $(\mu, \nu)$ is mapped by $g : (\mu, \nu) \mapsto (\tilde{\mu}, \tilde{\nu})$. The fixed point of the mapping $g$ is $(0, 1)$.

$\nu \in [0.80009, 1.48617]$. *All points within the $(\mu, \nu)$-domain converge when iteratively applying the mapping Eq.* (2) *to this fixed point.*

*Proof.* We provide a proof sketch (see detailed proof in Supplementary Material). With the Banach fixed point theorem we show that there exists a unique attracting and stable fixed point. To this end, we have to prove that a) $g$ is a contraction mapping and b) that the mapping stays in the domain, that is, $g(\Omega) \subseteq \Omega$. The spectral norm of the Jacobian of $g$ can be obtained via an explicit formula for the largest singular value for a $2 \times 2$ matrix. $g$ is a contraction mapping if its spectral norm is smaller than 1. We perform a computer-assisted proof to evaluate the largest singular value on a fine grid and ensure the precision of the computer evaluation by an error propagation analysis of the implemented algorithms on the according hardware. Singular values between grid points are upper bounded by the mean value theorem. To this end, we bound the derivatives of the formula for the largest singular value with respect to $\omega, \tau, \mu, \nu$. Then we apply the mean value theorem to pairs of points, where one is on the grid and the other is off the grid. This shows that for all values of $\omega, \tau, \mu, \nu$ in the domain $\Omega$, the spectral norm of $g$ is smaller than one. Therefore, $g$ is a contraction mapping on the domain $\Omega$. Finally, we show that the mapping $g$ stays in the domain $\Omega$ by deriving bounds on $\tilde{\mu}$ and $\tilde{\nu}$. Hence, the Banach fixed-point theorem holds and there exists a unique fixed point in $\Omega$ that is attained. $\square$

Consequently, feed-forward neural networks with many units in each layer and with the SELU activation function are self-normalizing (see definition 1), which readily follows from Theorem 1. To give an intuition, the main property of SELUs is that they damp the variance for negative net inputs and increase the variance for positive net inputs. The variance damping is stronger if net inputs are further away from zero while the variance increase is stronger if net inputs are close to zero. Thus, for large variance of the activations in the lower layer the damping effect is dominant and the variance decreases in the higher layer. Vice versa, for small variance the variance increase is dominant and the variance increases in the higher layer.

However, we cannot guarantee that mean and variance remain in the domain $\Omega$. Therefore, we next treat the case where $(\mu, \nu)$ are outside $\Omega$. It is especially crucial to consider $\nu$ because this variable has much stronger influence than $\mu$. Mapping $\nu$ across layers to a high value corresponds to an exploding gradient, since the Jacobian of the activation of high layers with respect to activations in lower layers has large singular values. Analogously, mapping $\nu$ across layers to a low value corresponds to an vanishing gradient. Bounding the mapping of $\nu$ from above and below would avoid both exploding and vanishing gradients. Theorem 2 states that the variance of neuron activations of SNNs is bounded from above, and therefore ensures that SNNs learn robustly and do not suffer from exploding gradients.

**Theorem 2** (Decreasing $\nu$). *For $\lambda = \lambda_{01}$, $\alpha = \alpha_{01}$ and the domain $\Omega^+$: $-1 \leqslant \mu \leqslant 1$, $-0.1 \leqslant \omega \leqslant 0.1$, $3 \leqslant \nu \leqslant 16$, and $0.8 \leqslant \tau \leqslant 1.25$, we have for the mapping of the variance $\tilde{\nu}(\mu, \omega, \nu, \tau, \lambda, \alpha)$ given in Eq.* (4)*: $\tilde{\nu}(\mu, \omega, \nu, \tau, \lambda_{01}, \alpha_{01}) < \nu$.*

The proof can be found in Supplementary Material. Thus, when mapped across many layers, the variance in the interval $[3, 16]$ is mapped to a value below 3. Consequently, all fixed points $(\mu, \nu)$

of the mapping $g$ (Eq. (2)) have $\nu < 3$. Analogously, Theorem 3 states that the variance of neuron activations of SNNs is bounded from below, and therefore ensures that SNNs do not suffer from vanishing gradients.

**Theorem 3** (Increasing $\nu$). *We consider $\lambda = \lambda_{01}$, $\alpha = \alpha_{01}$ and the domain $\Omega^-$: $-0.1 \leqslant \mu \leqslant 0.1$, and $-0.1 \leqslant \omega \leqslant 0.1$. For the domain $0.02 \leqslant \nu \leqslant 0.16$ and $0.8 \leqslant \tau \leqslant 1.25$ as well as for the domain $0.02 \leqslant \nu \leqslant 0.24$ and $0.9 \leqslant \tau \leqslant 1.25$, the mapping of the variance $\tilde{\nu}(\mu, \omega, \nu, \tau, \lambda, \alpha)$ given in Eq. (4) increases: $\tilde{\nu}(\mu, \omega, \nu, \tau, \lambda_{01}, \alpha_{01}) > \nu$.*

The proof can be found in the Supplementary Material. All fixed points $(\mu, \nu)$ of the mapping $g$ (Eq. (2)) ensure for $0.8 \leqslant \tau$ that $\tilde{\nu} > 0.16$ and for $0.9 \leqslant \tau$ that $\tilde{\nu} > 0.24$. Consequently, the variance mapping Eq. (4) ensures a lower bound on the variance $\nu$. Therefore SELU networks control the variance of the activations and push it into an interval, whereafter the mean and variance move toward the fixed point. Thus, SELU networks are steadily normalizing the variance and subsequently normalizing the mean, too. In all experiments, we observed that self-normalizing neural networks push the mean and variance of activations into the domain $\Omega$.

**Initialization.** Since SNNs have a fixed point at zero mean and unit variance for normalized weights $\omega = \sum_{i=1}^{n} w_i = 0$ and $\tau = \sum_{i=1}^{n} w_i^2 = 1$ (see above), we initialize SNNs such that these constraints are fulfilled in expectation. We draw the weights from a Gaussian distribution with $\mathrm{E}(w_i) = 0$ and variance $\mathrm{Var}(w_i) = 1/n$. Uniform and truncated Gaussian distributions with these moments led to networks with similar behavior. The "MSRA initialization" is similar since it uses zero mean and variance $2/n$ to initialize the weights [14]. The additional factor 2 counters the effect of rectified linear units.

**New Dropout Technique.** Standard dropout randomly sets an activation $x$ to zero with probability $1 - q$ for $0 < q \leqslant 1$. In order to preserve the mean, the activations are scaled by $1/q$ during training. If $x$ has mean $\mathrm{E}(x) = \mu$ and variance $\mathrm{Var}(x) = \nu$, and the dropout variable $d$ follows a binomial distribution $B(1, q)$, then the mean $\mathrm{E}(1/qdx) = \mu$ is kept. Dropout fits well to rectified linear units, since zero is in the low variance region and corresponds to the default value. For scaled exponential linear units, the default and low variance value is $\lim_{x \to -\infty} \mathrm{selu}(x) = -\lambda\alpha = \alpha'$. Therefore, we propose "alpha dropout", that randomly sets inputs to $\alpha'$. The new mean and new variance is $\mathrm{E}(xd + \alpha'(1 - d)) = q\mu + (1 - q)\alpha'$, and $\mathrm{Var}(xd + \alpha'(1 - d)) = q((1 - q)(\alpha' - \mu)^2 + \nu)$. We aim at keeping mean and variance to their original values after "alpha dropout", in order to ensure the self-normalizing property even for "alpha dropout". The affine transformation $a(xd + \alpha'(1 - d)) + b$ allows to determine parameters $a$ and $b$ such that mean and variance are kept to their values: $\mathrm{E}(a(x \cdot d + \alpha'(1 - d)) + b) = \mu$ and $\mathrm{Var}(a(x \cdot d + \alpha'(1 - d)) + b) = \nu$. In contrast to dropout, $a$ and $b$ will depend on $\mu$ and $\nu$, however our SNNs converge to activations with zero mean and unit variance. With $\mu = 0$ and $\nu = 1$, we obtain $a = \left(q + \alpha'^2 q(1 - q)\right)^{-1/2}$ and $b = -\left(q + \alpha'^2 q(1 - q)\right)^{-1/2} \left((1 - q)\alpha'\right)$. The parameters $a$ and $b$ only depend on the dropout rate $1 - q$ and the most negative activation $\alpha'$. Empirically, we found that dropout rates $1 - q = 0.05$ or $0.10$ lead to models with good performance. "Alpha-dropout" fits well to scaled exponential linear units by randomly setting activations to the negative saturation value.

**Applicability of the central limit theorem and independence assumption.** In the derivative of the mapping (Eq. (2)), we used the central limit theorem (CLT) to approximate the network inputs $z = \sum_{i=1}^{n} w_i x_i$ with a normal distribution. We justified normality because network inputs represent a weighted sum of the inputs $x_i$, where for Deep Learning $n$ is typically large. The Berry-Esseen theorem states that the convergence rate to normality is $n^{-1/2}$ [18]. In the classical version of the CLT, the random variables have to be independent and identically distributed, which typically does not hold for neural networks. However, the Lyapunov CLT does not require the variable to be identically distributed anymore. Furthermore, even under weak dependence, sums of random variables converge in distribution to a Gaussian distribution [3].

**Optimizers.** Empirically, we found that SGD, momentum, Adadelta and Adamax worked well for training SNNs, whereas for Adam we had to adjust the parameters ($\beta_2 = 0.99$, $\epsilon = 0.01$) to obtain proficient networks.

# 3 Experiments

We compare SNNs to other deep networks at different benchmarks. Hyperparameters such as number of layers (blocks), neurons per layer, learning rate, and dropout rate, are adjusted by grid-search for each dataset on a separate validation set (see Supplementary Section S4). We compare the following FNN methods: **(1) "MSRAinit":** FNNs without normalization and with ReLU activations and "Microsoft weight initialization" [14]. **(2) "BatchNorm":** FNNs with batch normalization [17]. **(3) "LayerNorm":** FNNs with layer normalization [1]. **(4) "WeightNorm":** FNNs with weight normalization [25]. **(5) "Highway":** Highway networks [28]. **(6) "ResNet":** Residual networks [13] adapted to FNNs using residual blocks with 2 or 3 layers with rectangular or diavolo shape. **(7) "SNNs":** Self normalizing networks with SELUs with $\alpha = \alpha_{01}$ and $\lambda = \lambda_{01}$ and the proposed dropout technique and initialization strategy.

**121 UCI Machine Learning Repository datasets.**  The benchmark comprises 121 classification datasets from the UCI Machine Learning repository [9] from diverse application areas, such as physics, geology, or biology. The size of the datasets ranges between 10 and 130,000 data points and the number of features from 4 to 250. In abovementioned work [9], there were methodological mistakes [30] which we avoided here. Each compared FNN method was optimized with respect to its architecture and hyperparameters on a validation set that was then removed from the subsequent analysis. The selected hyperparameters served to evaluate the methods in terms of accuracy on the pre-defined test sets. The accuracies are reported in the Supplementary Table S8. We ranked the methods by their accuracy for each prediction task and compared their average ranks. SNNs significantly outperform all competing networks in pairwise comparisons (paired Wilcoxon test across datasets) as reported in Table 1 (left panel).

Table 1: **Left:** Comparison of seven FNNs on 121 UCI tasks. We consider the average rank difference to rank 4, which is the average rank of seven methods with random predictions. The first column gives the method, the second the average rank difference, and the last the $p$-value of a paired Wilcoxon test whether the difference to the best performing method is significant. SNNs significantly outperform all other methods. **Right:** Comparison of 24 machine learning methods (ML) on the UCI datasets with more than 1000 data points. The first column gives the method, the second the average rank difference to rank 12.5, and the last the $p$-value of a paired Wilcoxon test whether the difference to the best performing method is significant. Methods that were significantly worse than the best method are marked with "*". SNNs outperform all competing methods.

| FNN method comparison | | | ML method comparison | | |
|---|---|---|---|---|---|
| Method | avg. rank diff. | $p$-value | Method | avg. rank diff. | $p$-value |
| SNN | -0.756 | | SNN | -6.7 | |
| MSRAinit | -0.240* | 2.7e-02 | SVM | -6.4 | 5.8e-01 |
| LayerNorm | -0.198* | 1.5e-02 | RandomForest | -5.9 | 2.1e-01 |
| Highway | 0.021* | 1.9e-03 | MSRAinit | -5.4* | 4.5e-03 |
| ResNet | 0.273* | 5.4e-04 | LayerNorm | -5.3 | 7.1e-02 |
| WeightNorm | 0.397* | 7.8e-07 | Highway | -4.6* | 1.7e-03 |
| BatchNorm | 0.504* | 3.5e-06 | . . . | . . . | . . . |

We further included 17 machine learning methods representing diverse method groups [9] in the comparison and the grouped the data sets into "small" and "large" data sets (for details see Supplementary Section S4.2). On 75 small datasets with less than 1000 data points, random forests and SVMs outperform SNNs and other FNNs. On 46 larger datasets with at least 1000 data points, SNNs show the highest performance followed by SVMs and random forests (see right panel of Table 1, for complete results see Supplementary Tables S9 and S10). Overall, SNNs have outperformed state of the art machine learning methods on UCI datasets with more than 1,000 data points.

Typically, hyperparameter selection chose SNN architectures that were much deeper than the selected architectures of other FNNs, with an average depth of 10.8 layers, compared to average depths of 6.0 for BatchNorm, 3.8 WeightNorm, 7.0 LayerNorm, 5.9 Highway, and 7.1 for MSRAinit networks. For ResNet, the average number of blocks was 6.35. SNNs with many more than 4 layers often provide the best predictive accuracies across all neural networks.

**Drug discovery: The Tox21 challenge dataset.** The Tox21 challenge dataset comprises about 12,000 chemical compounds whose twelve toxic effects have to be predicted based on their chemical structure. We used the validation sets of the challenge winners for hyperparameter selection (see Supplementary Section S4) and the challenge test set for performance comparison. We repeated the whole evaluation procedure 5 times to obtain error bars. The results in terms of average AUC are given in Table 2. In 2015, the challenge organized by the US NIH was won by an ensemble of shallow ReLU FNNs which achieved an AUC of 0.846 [23]. Besides FNNs, this ensemble also contained random forests and SVMs. Single SNNs came close with an AUC of 0.845±0.003. The best performing SNNs have 8 layers, compared to the runner-ups ReLU networks with layer normalization with 2 and 3 layers. Also batchnorm and weightnorm networks, typically perform best with shallow networks of 2 to 4 layers (Table 2). The deeper the networks, the larger the difference in performance between SNNs and other methods (see columns 5–8 of Table 2). The best performing method is an SNN with 8 layers.

Table 2: Comparison of FNNs at the Tox21 challenge dataset in terms of AUC. The rows represent different methods and the columns different network depth and for ResNets the number of residual blocks (6 and 32 blocks were omitted due to computational constraints). The deeper the networks, the more prominent is the advantage of SNNs. The best networks are SNNs with 8 layers.

| method | #layers / #blocks | | | | | | |
|---|---|---|---|---|---|---|---|
| | 2 | 3 | 4 | 6 | 8 | 16 | 32 |
| SNN | $83.7_{\pm 0.3}$ | $\mathbf{84.4}_{\pm 0.5}$ | $\mathbf{84.2}_{\pm 0.4}$ | $\mathbf{83.9}_{\pm 0.5}$ | $\mathbf{84.5}_{\pm 0.2}$ | $\mathbf{83.5}_{\pm 0.5}$ | $\mathbf{82.5}_{\pm 0.7}$ |
| Batchnorm | $80.0_{\pm 0.5}$ | $79.8_{\pm 1.6}$ | $77.2_{\pm 1.1}$ | $77.0_{\pm 1.7}$ | $75.0_{\pm 0.9}$ | $73.7_{\pm 2.0}$ | $76.0_{\pm 1.1}$ |
| WeightNorm | $83.7_{\pm 0.8}$ | $82.9_{\pm 0.8}$ | $82.2_{\pm 0.9}$ | $82.5_{\pm 0.6}$ | $81.9_{\pm 1.2}$ | $78.1_{\pm 1.3}$ | $56.6_{\pm 2.6}$ |
| LayerNorm | $\mathbf{84.3}_{\pm 0.3}$ | $84.3_{\pm 0.5}$ | $84.0_{\pm 0.2}$ | $82.5_{\pm 0.8}$ | $80.9_{\pm 1.8}$ | $78.7_{\pm 2.3}$ | $78.8_{\pm 0.8}$ |
| Highway | $83.3_{\pm 0.9}$ | $83.0_{\pm 0.5}$ | $82.6_{\pm 0.9}$ | $82.4_{\pm 0.8}$ | $80.3_{\pm 1.4}$ | $80.3_{\pm 2.4}$ | $79.6_{\pm 0.8}$ |
| MSRAinit | $82.7_{\pm 0.4}$ | $81.6_{\pm 0.9}$ | $81.1_{\pm 1.7}$ | $80.6_{\pm 0.6}$ | $80.9_{\pm 1.1}$ | $80.2_{\pm 1.1}$ | $80.4_{\pm 1.9}$ |
| ResNet | $82.2_{\pm 1.1}$ | $80.0_{\pm 2.0}$ | $80.5_{\pm 1.2}$ | $81.2_{\pm 0.7}$ | $81.8_{\pm 0.6}$ | $81.2_{\pm 0.6}$ | na |

**Astronomy: Prediction of pulsars in the HTRU2 dataset.** Since a decade, machine learning methods have been used to identify pulsars in radio wave signals [22]. Recently, the High Time Resolution Universe Survey (HTRU2) dataset has been released with 1,639 real pulsars and 16,259 spurious signals. Currently, the highest AUC value of a 10-fold cross-validation is 0.976 which has been achieved by Naive Bayes classifiers followed by decision tree C4.5 with 0.949 and SVMs with 0.929. We used eight features constructed by the PulsarFeatureLab as used previously [22]. We assessed the performance of FNNs using 10-fold nested cross-validation, where the hyperparameters were selected in the inner loop on a validation set (for details see Supplementary Section S4). Table 3 reports the results in terms of AUC. SNNs outperform all other methods and have pushed the state-of-the-art to an AUC of 0.98.

Table 3: Comparison of FNNs and reference methods at HTRU2 in terms of AUC. The first, fourth and seventh column give the method, the second, fifth and eight column the AUC averaged over 10 cross-validation folds, and the third and sixth column the $p$-value of a paired Wilcoxon test of the AUCs against the best performing method across the 10 folds. FNNs achieve better results than Naive Bayes (NB), C4.5, and SVM. SNNs exhibit the best performance and set a new record.

| | FNN methods | | | FNN methods | | | ref. methods | |
|---|---|---|---|---|---|---|---|---|
| method | AUC | $p$-value | method | AUC | $p$-value | method | AUC |
| SNN | $0.9803_{\pm 0.010}$ | | | | | | |
| MSRAinit | $0.9791_{\pm 0.010}$ | 3.5e-01 | LayerNorm | $0.9762*_{\pm 0.011}$ | 1.4e-02 | NB | 0.976 |
| WeightNorm | $0.9786*_{\pm 0.010}$ | 2.4e-02 | BatchNorm | $0.9760_{\pm 0.013}$ | 6.5e-02 | C4.5 | 0.946 |
| Highway | $0.9766*_{\pm 0.009}$ | 9.8e-03 | ResNet | $0.9753*_{\pm 0.010}$ | 6.8e-03 | SVM | 0.929 |

*SNNs and convolutional neural networks.* In initial experiments with CNNs, we found that SELU activations work well at image classification tasks: On MNIST, SNN-CNNs (2x Conv, MaxPool, 2x fully-connected, 30 Epochs) reach 99.2%±0.1 accuracy (ReLU: 99.2%±0.1) and on CIFAR10 (2x Conv, MaxPool, 2x Conv, MaxPool, 2x fully-connected, 200 Epochs) SNN-CNNs reach 82.5±0.8%

accuracy (ReLU: 76.1±1.0%). This finding unsurprising since even standard ELUs without the self-normalizing property have been shown to improve CNN training and accuracy[5].

## 4    Conclusion

To summarize, self-normalizing networks work well with the following configuration:

- SELU activation with parameters $\lambda \approx 1.0507$ and $\alpha \approx 1.6733$,
- inputs normalized to zero mean and unit variance,
- network weights initialized with variance $1/n$, and
- regularization with "alpha-dropout".

We have introduced self-normalizing neural networks for which we have proved that neuron activations are pushed towards zero mean and unit variance when propagated through the network. Additionally, for activations not close to unit variance, we have proved an upper and lower bound on the variance mapping. Consequently, SNNs do not face vanishing and exploding gradient problems. Therefore, SNNs work well for architectures with many layers, allowed us to introduce a novel regularization scheme and learn very robustly. On 121 UCI benchmark datasets, SNNs have outperformed other FNNs with and without normalization techniques, such as batch, layer, and weight normalization, or specialized architectures, such as Highway or Residual networks. SNNs also yielded the best results on drug discovery and astronomy tasks. The best performing SNN architectures are typically very deep in contrast to other FNNs.

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
