[Supplementary Material · SupplementaryMaterial_cameraready.pdf]

# Supplementary Material for the Paper "Self-Normalizing Neural Networks"

Günter Klambauer, Thomas Unterthiner, Andreas Mayr and Sepp Hochreiter

November 2, 2017

## Contents

# List of Figures

# List of Tables

## Overview

We present supplementary material for the paper "self normalizing neural networks". This introduction sets the background, definitions, and formulations. The theorems of the main paper are presented in the next section. The following section is devoted to the proofs of these theorems. The last section reports additional results and details on the experiments that are presented in the main paper, such as hyperparameter selection. The appendix shows that our theoretical bounds can be confirmed by numerical methods as a sanity check.

The proof of theorem 1 is based on the Banach's fixed point theorem for which we require (1) a contraction mapping, which is proved in Subsection S3.4.1 and (2) that the mapping stays within its domain, which is proved in Subsection S3.4.2 For part (1), the proof relies on the main Lemma 12, which is a computer-assisted proof, and can be found in Subsection S3.4.1. The validity of the computer-assisted proof is shown in Subsection S3.4.5 by error analysis and the precision of the functions' implementation. The last Subsection S3.4.6 compiles various lemmata with intermediate results that support the proofs of the main lemmata and theorems.

## S1  Background

We consider a neural network with **activation function** $f$ and two consecutive layers that are connected by **weight matrix** $W$. Since samples that serve as input to the neural network are chosen according to a distribution, the **activations $x$ in the lower layer**, the **network inputs $z = Wx$**, and **activations $y = f(z)$ in the higher layer** are all random variables. We assume that all units $x_i$ in the lower layer have **mean activation** $\mu := \mathrm{E}(x_i)$ and **variance of the activation** $\nu := \mathrm{Var}(x_i)$ and a unit $y$ in the higher layer has mean activation $\tilde{\mu} := \mathrm{E}(y)$ and variance $\tilde{\nu} := \mathrm{Var}(y)$. Here $\mathrm{E}(.)$ denotes the expectation and $\mathrm{Var}(.)$ the variance of a random variable. For activation of unit $y$, we have net input $z = w^T x$ and the **scaled exponential linear unit (SELU)** activation $y = \mathrm{selu}(z)$, with

$$\mathrm{selu}(x) \ = \ \lambda \begin{cases} x & \text{if } x > 0 \\ \alpha e^x - \alpha & \text{if } x \leqslant 0 \end{cases}. \tag{1}$$

For $n$ units $x_i, 1 \leqslant i \leqslant n$ in the lower layer and the **weight vector** $w \in \mathbb{R}^n$, we define $n$ **times the mean** by $\omega := \sum_{i=1}^n w_i$ and $n$ **times the second moment** by $\tau := \sum_{i=1}^n w_i^2$.

We define a **mapping** $g$ from mean $\mu$ and variance $\nu$ of one layer to the mean $\tilde{\mu}$ and variance $\tilde{\nu}$ in the next layer:

$$g : (\mu, \nu) \mapsto (\tilde{\mu}, \tilde{\nu}). \tag{2}$$

For neural networks with scaled exponential linear units, the mean is of the activations in the next layer computed according to

$$\tilde{\mu} \ = \ \int_{-\infty}^0 \lambda\alpha(\exp(z) - 1)p_{\mathrm{Gauss}}(z; \mu\omega, \sqrt{\nu\tau})dz \ + \ \int_0^\infty \lambda z p_{\mathrm{Gauss}}(z; \mu\omega, \sqrt{\nu\tau})dz, \tag{3}$$

and the second moment of the activations in the next layer is computed according to

$$\tilde{\xi} \ = \ \int_{-\infty}^0 \lambda^2\alpha^2(\exp(z) - 1)^2 p_{\mathrm{Gauss}}(z; \mu\omega, \sqrt{\nu\tau})dz \ + \ \int_0^\infty \lambda^2 z^2 p_{\mathrm{Gauss}}(z; \mu\omega, \sqrt{\nu\tau})dz. \tag{4}$$

Therefore, the expressions $\tilde{\mu}$ and $\tilde{\nu}$ have the following form:

$$\tilde{\mu}(\mu, \omega, \nu, \tau, \lambda, \alpha) = \frac{1}{2}\lambda \left( -(\alpha + \mu\omega)\operatorname{erfc}\left( \frac{\mu\omega}{\sqrt{2}\sqrt{\nu\tau}} \right) + \right. \tag{5}$$

$$\left. \alpha e^{\mu\omega + \frac{\nu\tau}{2}}\operatorname{erfc}\left( \frac{\mu\omega + \nu\tau}{\sqrt{2}\sqrt{\nu\tau}} \right) + \sqrt{\frac{2}{\pi}}\sqrt{\nu\tau}e^{-\frac{\mu^2\omega^2}{2\nu\tau}} + 2\mu\omega \right)$$

$$\tilde{\nu}(\mu, \omega, \nu, \tau, \lambda, \alpha) = \tilde{\xi}(\mu, \omega, \nu, \tau, \lambda, \alpha) - (\tilde{\mu}(\mu, \omega, \nu, \tau, \lambda, \alpha))^2 \tag{6}$$

$$\tilde{\xi}(\mu, \omega, \nu, \tau, \lambda, \alpha) = \frac{1}{2}\lambda^2 \left( ((\mu\omega)^2 + \nu\tau)\left( \operatorname{erf}\left( \frac{\mu\omega}{\sqrt{2}\sqrt{\nu\tau}} \right) + 1 \right) + \right. \tag{7}$$

$$\alpha^2 \left( -2e^{\mu\omega + \frac{\nu\tau}{2}}\operatorname{erfc}\left( \frac{\mu\omega + \nu\tau}{\sqrt{2}\sqrt{\nu\tau}} \right) + e^{2(\mu\omega + \nu\tau)}\operatorname{erfc}\left( \frac{\mu\omega + 2\nu\tau}{\sqrt{2}\sqrt{\nu\tau}} \right) + \right.$$

$$\left. \operatorname{erfc}\left( \frac{\mu\omega}{\sqrt{2}\sqrt{\nu\tau}} \right) \right) + \sqrt{\frac{2}{\pi}}(\mu\omega)\sqrt{\nu\tau}e^{-\frac{(\mu\omega)^2}{2(\nu\tau)}} \right)$$

We solve equations Eq. 5 and Eq. 6 for fixed points $\tilde{\mu} = \mu$ and $\tilde{\nu} = \nu$. For a normalized weight vector with $\omega = 0$ and $\tau = 1$ and the **fixed point** $(\mu, \nu) = (0, 1)$, we can solve equations Eq. 5 and Eq. 6 for $\alpha$ and $\lambda$. We denote the solutions to fixed point $(\mu, \nu) = (0, 1)$ by $\alpha_{01}$ and $\lambda_{01}$.

$$\alpha_{01} = -\frac{\sqrt{\frac{2}{\pi}}}{\operatorname{erfc}\left( \frac{1}{\sqrt{2}} \right)\exp\left( \frac{1}{2} \right) - 1} \approx 1.67326 \tag{8}$$

$$\lambda_{01} = \left( 1 - \operatorname{erfc}\left( \frac{1}{\sqrt{2}} \right)\sqrt{e} \right)\sqrt{2\pi}$$

$$\left( 2\operatorname{erfc}\left( \sqrt{2} \right)e^2 + \pi\operatorname{erfc}\left( \frac{1}{\sqrt{2}} \right)^2 e - 2(2 + \pi)\operatorname{erfc}\left( \frac{1}{\sqrt{2}} \right)\sqrt{e} + \pi + 2 \right)^{-1/2}$$

$$\lambda_{01} \approx 1.0507 \,.$$

The parameters $\alpha_{01}$ and $\lambda_{01}$ ensure

$$\tilde{\mu}(0, 0, 1, 1, \lambda_{01}, \alpha_{01}) = 0$$
$$\tilde{\nu}(0, 0, 1, 1, \lambda_{01}, \alpha_{01}) = 1$$

Since we focus on the fixed point $(\mu, \nu) = (0, 1)$, we assume throughout the analysis that $\alpha = \alpha_{01}$ and $\lambda = \lambda_{01}$. We consider the functions $\tilde{\mu}(\mu, \omega, \nu, \tau, \lambda_{01}, \alpha_{01})$, $\tilde{\nu}(\mu, \omega, \nu, \tau, \lambda_{01}, \alpha_{01})$, and $\tilde{\xi}(\mu, \omega, \nu, \tau, \lambda_{01}, \alpha_{01})$ on the **domain** $\Omega = \{(\mu, \omega, \nu, \tau) \mid \mu \in [\mu_{\min}, \mu_{\max}] = [-0.1, 0.1], \omega \in [\omega_{\min}, \omega_{\max}] = [-0.1, 0.1], \nu \in [\nu_{\min}, \nu_{\max}] = [0.8, 1.5], \tau \in [\tau_{\min}, \tau_{\max}] = [0.95, 1.1]\}$.

Figure S1 visualizes the mapping $g$ for $\omega = 0$ and $\tau = 1$ and $\alpha_{01}$ and $\lambda_{01}$ at few pre-selected points. It can be seen that $(0, 1)$ is an attracting fixed point of the mapping $g$.

Figure S1: For $\omega = 0$ and $\tau = 1$, the mapping $g$ of mean $\mu$ ($x$-axis) and variance $\nu$ ($y$-axis) to the next layer's mean $\tilde{\mu}$ and variance $\tilde{\nu}$ is depicted. An arrow shows in which direction $(\mu, \nu)$ is mapped by $g : (\mu, \nu) \mapsto (\tilde{\mu}, \tilde{\nu})$. Note that $(0, 1)$ is an attracting fixed point of the mapping $g$.

# S2    Theorems of the Main Paper

## S2.1    Theorem 1: Stable and Attracting Fixed Points Close to (0,1)

Theorem 1 shows that the mapping $g$ defined by Eq. (5) and Eq. (6) exhibits a stable and attracting fixed point close to zero mean and unit variance. Theorem 1 establishes the self-normalizing property of self-normalizing neural networks (SNNs). The stable and attracting fixed point leads to robust learning through many layers.

**Theorem 1** (Stable and Attracting Fixed Points). *We assume $\alpha = \alpha_{01}$ and $\lambda = \lambda_{01}$. We restrict the range of the variables to the domain $\mu \in [-0.1, 0.1]$, $\omega \in [-0.1, 0.1]$, $\nu \in [0.8, 1.5]$, and $\tau \in [0.95, 1.1]$. For $\omega = 0$ and $\tau = 1$, the mapping Eq. (5) and Eq. (6) has the stable fixed point $(\mu, \nu) = (0, 1)$. For other $\omega$ and $\tau$ the mapping Eq. (5) and Eq. (6) has a stable and attracting fixed point depending on $(\omega, \tau)$ in the $(\mu, \nu)$-domain: $\mu \in [-0.03106, 0.06773]$ and $\nu \in [0.80009, 1.48617]$. All points within the $(\mu, \nu)$-domain converge when iteratively applying the mapping Eq. (5) and Eq. (6) to this fixed point.*

## S2.2    Theorem 2: Decreasing Variance from Above

The next Theorem 2 states that the variance of unit activations does not explode through consecutive layers of self-normalizing networks. Even more, a large variance of unit activations decreases when propagated through the network. In particular this ensures that exploding gradients will never be observed. In contrast to the domain in previous subsection, in which $\nu \in [0.8, 1.5]$, we now consider a domain in which the variance of the inputs is higher $\nu \in [3, 16]$ and even the range of the mean is increased $\mu \in [-1, 1]$. We denote this new domain with the symbol $\Omega^{++}$ to indicate that the variance lies above the variance of the original domain $\Omega$. In $\Omega^{++}$, we can show that the variance $\tilde{\nu}$ in the next layer is always smaller then the original variance $\nu$. Concretely, this theorem states that:

**Theorem 2** (Decreasing $\nu$). *For $\lambda = \lambda_{01}$, $\alpha = \alpha_{01}$ and the domain $\Omega^{++}$: $-1 \leqslant \mu \leqslant 1$, $-0.1 \leqslant \omega \leqslant 0.1$, $3 \leqslant \nu \leqslant 16$, and $0.8 \leqslant \tau \leqslant 1.25$ we have for the mapping of the variance*

$\tilde{\nu}(\mu, \omega, \nu, \tau, \lambda, \alpha)$ *given in Eq.* (6)

$$\tilde{\nu}(\mu, \omega, \nu, \tau, \lambda_{01}, \alpha_{01}) \; < \; \nu \;. \tag{9}$$

*The variance decreases in* $[3, 16]$ *and all fixed points* $(\mu, \nu)$ *of mapping Eq.* (6) *and Eq.* (5) *have* $\nu < 3$.

## S2.3   Theorem 3: Increasing Variance from Below

The next Theorem 3 states that the variance of unit activations does not vanish through consecutive layers of self-normalizing networks. Even more, a small variance of unit activations increases when propagated through the network. In particular this ensures that vanishing gradients will never be observed. In contrast to the first domain, in which $\nu \in [0.8, 1.5]$, we now consider two domains $\Omega_1^-$ and $\Omega_2^-$ in which the variance of the inputs is lower $0.05 \leqslant \nu \leqslant 0.16$ and $0.05 \leqslant \nu \leqslant 0.24$, and even the parameter $\tau$ is different $0.9 \leqslant \tau \leqslant 1.25$ to the original $\Omega$. We denote this new domain with the symbol $\Omega_i^-$ to indicate that the variance lies below the variance of the original domain $\Omega$. In $\Omega_1^-$ and $\Omega_2^-$, we can show that the variance $\tilde{\nu}$ in the next layer is always larger then the original variance $\nu$, which means that the variance does not vanish through consecutive layers of self-normalizing networks. Concretely, this theorem states that:

**Theorem 3** (Increasing $\nu$). *We consider* $\lambda = \lambda_{01}$, $\alpha = \alpha_{01}$ *and the two domains* $\Omega_1^- = \{(\mu, \omega, \nu, \tau) \mid -0.1 \leqslant \mu \leqslant 0.1, -0.1 \leqslant \omega \leqslant 0.1, 0.05 \leqslant \nu \leqslant 0.16, 0.8 \leqslant \tau \leqslant 1.25\}$ *and* $\Omega_2^- = \{(\mu, \omega, \nu, \tau) \mid -0.1 \leqslant \mu \leqslant 0.1, -0.1 \leqslant \omega \leqslant 0.1, 0.05 \leqslant \nu \leqslant 0.24, 0.9 \leqslant \tau \leqslant 1.25\}$.

*The mapping of the variance* $\tilde{\nu}(\mu, \omega, \nu, \tau, \lambda, \alpha)$ *given in Eq.* (6) *increases*

$$\tilde{\nu}(\mu, \omega, \nu, \tau, \lambda_{01}, \alpha_{01}) \; > \; \nu \tag{10}$$

*in both* $\Omega_1^-$ *and* $\Omega_2^-$. *All fixed points* $(\mu, \nu)$ *of mapping Eq.* (6) *and Eq.* (5) *ensure for* $0.8 \leqslant \tau$ *that* $\tilde{\nu} > 0.16$ *and for* $0.9 \leqslant \tau$ *that* $\tilde{\nu} > 0.24$. *Consequently, the variance mapping Eq.* (6) *and Eq.* (5) *ensures a lower bound on the variance* $\nu$.

## S3   Proofs of the Theorems

### S3.1   Proof of Theorem 1

We have to show that the mapping $g$ defined by Eq. (5) and Eq. (6) has a stable and attracting fixed point close to $(0, 1)$. To proof this statement and Theorem 1, we apply the Banach fixed point theorem which requires (1) that $g$ is a contraction mapping and (2) that $g$ does not map outside the function's domain, concretely:

**Theorem 4** (Banach Fixed Point Theorem). *Let* $(X, d)$ *be a non-empty complete metric space with a contraction mapping* $f : X \to X$. *Then* $f$ *has a unique fixed-point* $x_f \in X$ *with* $f(x_f) = x_f$. *Every sequence* $x_n = f(x_{n-1})$ *with starting element* $x_0 \in X$ *converges to the fixed point:* $x_n \xrightarrow[n \to \infty]{} x_f$.

Contraction mappings are functions that map two points such that their distance is decreasing:

**Definition 1** (Contraction mapping). *A function $f : X \to X$ on a metric space $X$ with distance $d$ is a contraction mapping, if there is a $0 \leqslant \delta < 1$, such that for all points $\boldsymbol{u}$ and $\boldsymbol{v}$ in $X$: $d(f(\boldsymbol{u}), f(\boldsymbol{v})) \leqslant \delta d(\boldsymbol{u}, \boldsymbol{v})$.*

To show that $g$ is a contraction mapping in $\Omega$ with distance $\|.\|_2$, we use the Mean Value Theorem for $u, v \in \Omega$

$$\|g(\boldsymbol{u}) - g(\boldsymbol{v})\|_2 \leqslant M \|\boldsymbol{u} - \boldsymbol{v}\|_2, \tag{11}$$

in which $M$ is an upper bound on the spectral norm the Jacobian $\mathcal{H}$ of $g$. The spectral norm is given by the largest singular value of the Jacobian of $g$. If the largest singular value of the Jacobian is smaller than 1, the mapping $g$ of the mean and variance to the mean and variance in the next layer is contracting. We show that the largest singular value is smaller than 1 by evaluating the function for the singular value $S(\mu, \omega, \nu, \tau, \lambda, \alpha)$ on a grid. Then we use the Mean Value Theorem to bound the deviation of the function $S$ between grid points. To this end, we have to bound the gradient of $S$ with respect to $(\mu, \omega, \nu, \tau)$. If all function values plus gradient times the deltas (differences between grid points and evaluated points) is still smaller than 1, then we have proofed that the function is below 1 (Lemma 12). To show that the mapping does not map outside the function's domain, we derive bounds on the expressions for the mean and the variance (Lemma 13). Section S3.4.1 and Section S3.4.2 are concerned with the contraction mapping and the image of the function domain of $g$, respectively.

With the results that the largest singular value of the Jacobian is smaller than one (Lemma 12) and that the mapping stays in the domain $\Omega$ (Lemma 13), we can prove Theorem 1. We first recall Theorem 1:

**Theorem** (Stable and Attracting Fixed Points). *We assume $\alpha = \alpha_{01}$ and $\lambda = \lambda_{01}$. We restrict the range of the variables to the domain $\mu \in [-0.1, 0.1]$, $\omega \in [-0.1, 0.1]$, $\nu \in [0.8, 1.5]$, and $\tau \in [0.95, 1.1]$. For $\omega = 0$ and $\tau = 1$, the mapping Eq. (5) and Eq. (6) has the stable fixed point $(\mu, \nu) = (0, 1)$. For other $\omega$ and $\tau$ the mapping Eq. (5) and Eq. (6) has a stable and attracting fixed point depending on $(\omega, \tau)$ in the $(\mu, \nu)$-domain: $\mu \in [-0.03106, 0.06773]$ and $\nu \in [0.80009, 1.48617]$. All points within the $(\mu, \nu)$-domain converge when iteratively applying the mapping Eq. (5) and Eq. (6) to this fixed point.*

*Proof.* According to Lemma 12 the mapping $g$ (Eq. (5) and Eq. (6)) is a contraction mapping in the given domain, that is, it has a Lipschitz constant smaller than one. We showed that $(\mu, \nu) = (0, 1)$ is a fixed point of the mapping for $(\omega, \tau) = (0, 1)$.

The domain is compact (bounded and closed), therefore it is a complete metric space. We further have to make sure the mapping $g$ does not map outside its domain $\Omega$. According to Lemma 13, the mapping maps into the domain $\mu \in [-0.03106, 0.06773]$ and $\nu \in [0.80009, 1.48617]$.

Now we can apply the Banach fixed point theorem given in Theorem 4 from which the statement of the theorem follows. $\qquad\square$

## S3.2   Proof of Theorem 2

First we recall Theorem 2:

**Theorem** (Decreasing $\nu$). *For $\lambda = \lambda_{01}$, $\alpha = \alpha_{01}$ and the domain $\Omega^{++}$: $-1 \leqslant \mu \leqslant 1$, $-0.1 \leqslant \omega \leqslant 0.1$, $3 \leqslant \nu \leqslant 16$, and $0.8 \leqslant \tau \leqslant 1.25$ we have for the mapping of the variance $\tilde{\nu}(\mu, \omega, \nu, \tau, \lambda, \alpha)$ given in Eq. (6)*

$$\tilde{\nu}(\mu, \omega, \nu, \tau, \lambda_{01}, \alpha_{01}) \; < \; \nu \,. \tag{12}$$

*The variance decreases in $[3, 16]$ and all fixed points $(\mu, \nu)$ of mapping Eq. (6) and Eq. (5) have $\nu < 3$.*

*Proof.* We start to consider an even larger domain $-1 \leqslant \mu \leqslant 1$, $-0.1 \leqslant \omega \leqslant 0.1$, $1.5 \leqslant \nu \leqslant 16$, and $0.8 \leqslant \tau \leqslant 1.25$. We prove facts for this domain and later restrict to $3 \leqslant \nu \leqslant 16$, i.e. $\Omega^{++}$. We consider the function $g$ of the difference between the second moment $\tilde{\xi}$ in the next layer and the variance $\nu$ in the lower layer:

$$g(\mu, \omega, \nu, \tau, \lambda_{01}, \alpha_{01}) \; = \; \tilde{\xi}(\mu, \omega, \nu, \tau, \lambda_{01}, \alpha_{01}) \; - \; \nu \,. \tag{13}$$

If we can show that $g(\mu, \omega, \nu, \tau, \lambda_{01}, \alpha_{01}) < 0$ for all $(\mu, \omega, \nu, \tau) \in \Omega^{++}$, then we would obtain our desired result $\tilde{\nu} \leqslant \tilde{\xi} < \nu$. The derivative with respect to $\nu$ is according to Theorem 16:

$$\frac{\partial}{\partial \nu} g(\mu, \omega, \nu, \tau, \lambda_{01}, \alpha 01) \; = \; \frac{\partial}{\partial \nu} \tilde{\xi}(\mu, \omega, \nu, \tau, \lambda_{01}, \alpha 01) \; - \; 1 \; < \; 0 \,. \tag{14}$$

Therefore $g$ is strictly monotonically decreasing in $\nu$. Since $\tilde{\xi}$ is a function in $\nu\tau$ (these variables only appear as this product), we have for $x = \nu\tau$

$$\frac{\partial}{\partial \nu} \tilde{\xi} \; = \; \frac{\partial}{\partial x} \tilde{\xi} \, \frac{\partial x}{\partial \nu} \; = \; \frac{\partial}{\partial x} \tilde{\xi} \, \tau \tag{15}$$

and

$$\frac{\partial}{\partial \tau} \tilde{\xi} \; = \; \frac{\partial}{\partial x} \tilde{\xi} \, \frac{\partial x}{\partial \tau} \; = \; \frac{\partial}{\partial x} \tilde{\xi} \, \nu \,. \tag{16}$$

Therefore we have according to Theorem 16:

$$\frac{\partial}{\partial \tau} \tilde{\xi}(\mu, \omega, \nu, \tau, \lambda_{01}, \alpha 01) \; = \; \frac{\nu}{\tau} \, \frac{\partial}{\partial \nu} \tilde{\xi}(\mu, \omega, \nu, \tau, \lambda_{01}, \alpha 01) \; > \; 0 \,. \tag{17}$$

Therefore

$$\frac{\partial}{\partial \tau} g(\mu, \omega, \nu, \tau, \lambda_{01}, \alpha 01) \; = \; \frac{\partial}{\partial \tau} \tilde{\xi}(\mu, \omega, \nu, \tau, \lambda_{01}, \alpha 01) \; > \; 0 \,. \tag{18}$$

Consequently, $g$ is strictly monotonically increasing in $\tau$. Now we consider the derivative with respect to $\mu$ and $\omega$. We start with $\frac{\partial}{\partial \mu} \tilde{\xi}(\mu, \omega, \nu, \tau, \lambda, \alpha)$, which is

$$\frac{\partial}{\partial \mu} \tilde{\xi}(\mu, \omega, \nu, \tau, \lambda, \alpha) \; = \tag{19}$$

$$\lambda^2 \omega \left( \alpha^2 \left( -e^{\mu\omega + \frac{\nu\tau}{2}} \right) \operatorname{erfc}\left( \frac{\mu\omega + \nu\tau}{\sqrt{2}\sqrt{\nu\tau}} \right) + \right.$$

$$\left. \alpha^2 e^{2\mu\omega + 2\nu\tau} \operatorname{erfc}\left( \frac{\mu\omega + 2\nu\tau}{\sqrt{2}\sqrt{\nu\tau}} \right) + \mu\omega \left( 2 - \operatorname{erfc}\left( \frac{\mu\omega}{\sqrt{2}\sqrt{\nu\tau}} \right) \right) + \sqrt{\frac{2}{\pi}} \sqrt{\nu\tau} e^{-\frac{\mu^2\omega^2}{2\nu\tau}} \right) \,.$$

We consider the sub-function

$$\sqrt{\frac{2}{\pi}}\sqrt{\nu\tau} - \alpha^2 \left( e^{\left(\frac{\mu\omega+\nu\tau}{\sqrt{2}\sqrt{\nu\tau}}\right)^2} \operatorname{erfc}\left(\frac{\mu\omega+\nu\tau}{\sqrt{2}\sqrt{\nu\tau}}\right) - e^{\left(\frac{\mu\omega+2\nu\tau}{\sqrt{2}\sqrt{\nu\tau}}\right)^2} \operatorname{erfc}\left(\frac{\mu\omega+2\nu\tau}{\sqrt{2}\sqrt{\nu\tau}}\right)\right) . \qquad (20)$$

We set $x = \nu\tau$ and $y = \mu\omega$ and obtain

$$\sqrt{\frac{2}{\pi}}\sqrt{x} - \alpha^2 \left( e^{\left(\frac{x+y}{\sqrt{2}\sqrt{x}}\right)^2} \operatorname{erfc}\left(\frac{x+y}{\sqrt{2}\sqrt{x}}\right) - e^{\left(\frac{2x+y}{\sqrt{2}\sqrt{x}}\right)^2} \operatorname{erfc}\left(\frac{2x+y}{\sqrt{2}\sqrt{x}}\right)\right) . \qquad (21)$$

The derivative to this sub-function with respect to $y$ is

$$\frac{\alpha^2 \left( e^{\frac{(2x+y)^2}{2x}} (2x+y)\operatorname{erfc}\left(\frac{2x+y}{\sqrt{2}\sqrt{x}}\right) - e^{\frac{(x+y)^2}{2x}} (x+y)\operatorname{erfc}\left(\frac{x+y}{\sqrt{2}\sqrt{x}}\right)\right)}{x} = \qquad (22)$$

$$\frac{\sqrt{2}\alpha^2\sqrt{x}\left(\dfrac{e^{\frac{(2x+y)^2}{2x}}(x+y)\operatorname{erfc}\left(\frac{x+y}{\sqrt{2}\sqrt{x}}\right)}{\sqrt{2}\sqrt{x}} - \dfrac{e^{\frac{(x+y)^2}{2x}}(x+y)\operatorname{erfc}\left(\frac{x+y}{\sqrt{2}\sqrt{x}}\right)}{\sqrt{2}\sqrt{x}}\right)}{x} > 0 .$$

The inequality follows from Lemma 24, which states that $ze^{z^2}\operatorname{erfc}(z)$ is monotonically increasing in $z$. Therefore the sub-function is increasing in $y$. The derivative to this sub-function with respect to $x$ is

$$\frac{1}{2\sqrt{\pi}x^2}\sqrt{\pi}\alpha^2\left(e^{\frac{(2x+y)^2}{2x}}\left(4x^2-y^2\right)\operatorname{erfc}\left(\frac{2x+y}{\sqrt{2}\sqrt{x}}\right)\right. \qquad (23)$$

$$\left. -e^{\frac{(x+y)^2}{2x}}(x-y)(x+y)\operatorname{erfc}\left(\frac{x+y}{\sqrt{2}\sqrt{x}}\right)\right) - \sqrt{2}\left(\alpha^2-1\right)x^{3/2}.$$

The sub-function is increasing in $x$, since the derivative is larger than zero:

$$\frac{\sqrt{\pi}\alpha^2\left(e^{\frac{(2x+y)^2}{2x}}\left(4x^2-y^2\right)\operatorname{erfc}\left(\frac{2x+y}{\sqrt{2}\sqrt{x}}\right) - e^{\frac{(x+y)^2}{2x}}(x-y)(x+y)\operatorname{erfc}\left(\frac{x+y}{\sqrt{2}\sqrt{x}}\right)\right) - \sqrt{2}x^{3/2}\left(\alpha^2-1\right)}{2\sqrt{\pi}x^2} \geqslant$$

$$(24)$$

$$\frac{\sqrt{\pi}\alpha^2\left(\dfrac{(2x-y)(2x+y)2}{\sqrt{\pi}\left(\frac{2x+y}{\sqrt{2}\sqrt{x}}+\sqrt{\left(\frac{2x+y}{\sqrt{2}\sqrt{x}}\right)^2+2}\right)} - \dfrac{(x-y)(x+y)2}{\sqrt{\pi}\left(\frac{x+y}{\sqrt{2}\sqrt{x}}+\sqrt{\left(\frac{x+y}{\sqrt{2}\sqrt{x}}\right)^2+\frac{4}{\pi}}\right)}\right) - \sqrt{2}x^{3/2}\left(\alpha^2-1\right)}{2\sqrt{\pi}x^2} =$$

$$\frac{\sqrt{\pi}\alpha^2\left(\dfrac{(2x-y)(2x+y)2\left(\sqrt{2}\sqrt{x}\right)}{\sqrt{\pi}\left(2x+y+\sqrt{(2x+y)^2+4x}\right)} - \dfrac{(x-y)(x+y)2\left(\sqrt{2}\sqrt{x}\right)}{\sqrt{\pi}\left(x+y+\sqrt{(x+y)^2+\frac{8x}{\pi}}\right)}\right) - \sqrt{2}x^{3/2}\left(\alpha^2-1\right)}{2\sqrt{\pi}x^2} =$$

$$\frac{\sqrt{\pi}\alpha^2\left(\dfrac{(2x-y)(2x+y)2}{\sqrt{\pi}\left(2x+y+\sqrt{(2x+y)^2+4x}\right)} - \dfrac{(x-y)(x+y)2}{\sqrt{\pi}\left(x+y+\sqrt{(x+y)^2+\frac{8x}{\pi}}\right)}\right) - x\left(\alpha^2-1\right)}{\sqrt{2}\sqrt{\pi}x^{3/2}} >$$

$$\frac{\sqrt{\pi}\alpha^2\left(\frac{(2x-y)(2x+y)2}{\sqrt{\pi}\left(2x+y+\sqrt{(2x+y)^2+2(2x+y)+1}\right)}-\frac{(x-y)(x+y)2}{\sqrt{\pi}\left(x+y+\sqrt{(x+y)^2+0.878\cdot 2(x+y)+0.878^2}\right)}\right)-x\left(\alpha^2-1\right)}{\sqrt{2}\sqrt{\pi}x^{3/2}}=$$

$$\frac{\sqrt{\pi}\alpha^2\left(\frac{(2x-y)(2x+y)2}{\sqrt{\pi}\left(2x+y+\sqrt{(2x+y+1)^2}\right)}-\frac{(x-y)(x+y)2}{\sqrt{\pi}\left(x+y+\sqrt{(x+y+0.878)^2}\right)}\right)-x\left(\alpha^2-1\right)}{\sqrt{2}\sqrt{\pi}x^{3/2}}=$$

$$\frac{\sqrt{\pi}\alpha^2\left(\frac{(2x-y)(2x+y)2}{\sqrt{\pi}(2(2x+y)+1)}-\frac{(x-y)(x+y)2}{\sqrt{\pi}(2(x+y)+0.878)}\right)-x\left(\alpha^2-1\right)}{\sqrt{2}\sqrt{\pi}x^{3/2}}=$$

$$\frac{\sqrt{\pi}\alpha^2\left(\frac{(2(x+y)+0.878)(2x-y)(2x+y)2}{\sqrt{\pi}}-\frac{(x-y)(x+y)(2(2x+y)+1)2}{\sqrt{\pi}}\right)}{(2(2x+y)+1)(2(x+y)+0.878)\sqrt{2}\sqrt{\pi}x^{3/2}}+$$

$$\frac{\sqrt{\pi}\alpha^2\left(-x\left(\alpha^2-1\right)(2(2x+y)+1)(2(x+y)+0.878)\right)}{(2(2x+y)+1)(2(x+y)+0.878)\sqrt{2}\sqrt{\pi}x^{3/2}}=$$

$$\frac{8x^3+12x^2y+4.14569x^2+4xy^2-6.76009xy-1.58023x+0.683154y^2}{(2(2x+y)+1)(2(x+y)+0.878)\sqrt{2}\sqrt{\pi}x^{3/2}}>$$

$$\frac{8x^3-0.1\cdot 12x^2+4.14569x^2+4\cdot(0.0)^2x-6.76009\cdot 0.1x-1.58023x+0.683154\cdot(0.0)^2}{(2(2x+y)+1)(2(x+y)+0.878)\sqrt{2}\sqrt{\pi}x^{3/2}}=$$

$$\frac{8x^2+2.94569x-2.25624}{(2(2x+y)+1)(2(x+y)+0.878)\sqrt{2}\sqrt{\pi}\sqrt{x}}=$$

$$\frac{8(x-0.377966)(x+0.746178)}{(2(2x+y)+1)(2(x+y)+0.878)\sqrt{2}\sqrt{\pi}\sqrt{x}}>0\,.$$

We explain this chain of inequalities:

- First inequality: We applied Lemma 22 two times.

- Equalities factor out $\sqrt{2}\sqrt{x}$ and reformulate.

- Second inequality part 1: we applied

$$0<2y \implies (2x+y)^2+4x+1<(2x+y)^2+2(2x+y)+1=(2x+y+1)^2\,. \quad (25)$$

- Second inequality part 2: we show that for $a=\frac{1}{10}\left(\sqrt{\frac{960+169\pi}{\pi}}-13\right)$ following holds: $\frac{8x}{\pi}-\left(a^2+2a(x+y)\right)\geqslant 0$. We have $\frac{\partial}{\partial x}\frac{8x}{\pi}-\left(a^2+2a(x+y)\right)=\frac{8}{\pi}-2a>0$ and $\frac{\partial}{\partial y}\frac{8x}{\pi}-\left(a^2+2a(x+y)\right)=-2a<0$. Therefore the minimum is at border for minimal $x$ and maximal $y$:

$$\frac{8\cdot 1.2}{\pi}-\left(\frac{2}{10}\left(\sqrt{\frac{960+169\pi}{\pi}}-13\right)(1.2+0.1)+\left(\frac{1}{10}\left(\sqrt{\frac{960+169\pi}{\pi}}-13\right)\right)^2\right)=0\,.$$
(26)

Thus

$$\frac{8x}{\pi}\geqslant a^2+2a(x+y)\,. \quad (27)$$

for $a = \frac{1}{10} \left( \sqrt{\frac{960+169\pi}{\pi}} - 13 \right) > 0.878$.

- Equalities only solve square root and factor out the resulting terms $(2(2x + y) + 1)$ and $(2(x + y) + 0.878)$.

- We set $\alpha = \alpha_{01}$ and multiplied out. Thereafter we also factored out $x$ in the numerator. Finally a quadratic equations was solved.

The sub-function has its minimal value for minimal $x = \nu\tau = 1.5 \cdot 0.8 = 1.2$ and minimal $y = \mu\omega = -1 \cdot 0.1 = -0.1$. We further minimize the function

$$\mu\omega e^{\frac{\mu^2\omega^2}{2\nu\tau}} \left( 2 - \mathrm{erfc}\left( \frac{\mu\omega}{\sqrt{2}\sqrt{\nu\tau}} \right) \right) \ > \ -0.1 e^{\frac{0.1^2}{2\cdot1.2}} \left( 2 - \mathrm{erfc}\left( \frac{0.1}{\sqrt{2}\sqrt{1.2}} \right) \right) . \tag{28}$$

We compute the minimum of the term in brackets of $\frac{\partial}{\partial\mu}\tilde{\xi}(\mu, \omega, \nu, \tau, \lambda, \alpha)$ in Eq. (19):

$$\mu\omega e^{\frac{\mu^2\omega^2}{2\nu\tau}} \left( 2 - \mathrm{erfc}\left( \frac{\mu\omega}{\sqrt{2}\sqrt{\nu\tau}} \right) \right) + \tag{29}$$

$$\alpha_{01}^2 \left( - \left( e^{\left( \frac{\mu\omega+\nu\tau}{\sqrt{2}\sqrt{\nu\tau}} \right)^2} \mathrm{erfc}\left( \frac{\mu\omega + \nu\tau}{\sqrt{2}\sqrt{\nu\tau}} \right) - e^{\left( \frac{\mu\omega+2\nu\tau}{\sqrt{2}\sqrt{\nu\tau}} \right)^2} \mathrm{erfc}\left( \frac{\mu\omega + 2\nu\tau}{\sqrt{2}\sqrt{\nu\tau}} \right) \right) \right) +$$

$$\sqrt{\frac{2}{\pi}}\sqrt{\nu\tau} \ >$$

$$\alpha_{01}^2 \left( - \left( e^{\left( \frac{1.2-0.1}{\sqrt{2}\sqrt{1.2}} \right)^2} \mathrm{erfc}\left( \frac{1.2 - 0.1}{\sqrt{2}\sqrt{1.2}} \right) - e^{\left( \frac{2\cdot1.2-0.1}{\sqrt{2}\sqrt{1.2}} \right)^2} \mathrm{erfc}\left( \frac{2 \cdot 1.2 - 0.1}{\sqrt{2}\sqrt{1.2}} \right) \right) \right) -$$

$$0.1 e^{\frac{0.1^2}{2\cdot1.2}} \left( 2 - \mathrm{erfc}\left( \frac{0.1}{\sqrt{2}\sqrt{1.2}} \right) \right) + \sqrt{1.2}\sqrt{\frac{2}{\pi}} \ = \ 0.212234 .$$

Therefore the term in brackets of Eq. (19) is larger than zero. Thus, $\frac{\partial}{\partial\mu}\tilde{\xi}(\mu, \omega, \nu, \tau, \lambda, \alpha)$ has the sign of $\omega$. Since $\tilde{\xi}$ is a function in $\mu\omega$ (these variables only appear as this product), we have for $x = \mu\omega$

$$\frac{\partial}{\partial\nu}\tilde{\xi} \ = \ \frac{\partial}{\partial x}\tilde{\xi}\frac{\partial x}{\partial\mu} \ = \ \frac{\partial}{\partial x}\tilde{\xi}\,\omega \tag{30}$$

and

$$\frac{\partial}{\partial\omega}\tilde{\xi} \ = \ \frac{\partial}{\partial x}\tilde{\xi}\frac{\partial x}{\partial\omega} \ = \ \frac{\partial}{\partial x}\tilde{\xi}\,\mu . \tag{31}$$

$$\frac{\partial}{\partial\omega}\tilde{\xi}(\mu, \omega, \nu, \tau, \lambda_{01}, \alpha01) \ = \ \frac{\mu}{\omega}\frac{\partial}{\partial\mu}\tilde{\xi}(\mu, \omega, \nu, \tau, \lambda_{01}, \alpha01) . \tag{32}$$

Since $\frac{\partial}{\partial\mu}\tilde{\xi}$ has the sign of $\omega$, $\frac{\partial}{\partial\mu}\tilde{\xi}$ has the sign of $\mu$. Therefore

$$\frac{\partial}{\partial\omega}g(\mu, \omega, \nu, \tau, \lambda_{01}, \alpha01) \ = \ \frac{\partial}{\partial\omega}\tilde{\xi}(\mu, \omega, \nu, \tau, \lambda_{01}, \alpha01) \tag{33}$$

has the sign of $\mu$.

We now divide the $\mu$-domain into $-1 \leqslant \mu \leqslant 0$ and $0 \leqslant \mu \leqslant 1$. Analogously we divide the $\omega$-domain into $-0.1 \leqslant \omega \leqslant 0$ and $0 \leqslant \omega \leqslant 0.1$. In this domains $g$ is strictly monotonically.

For all domains $g$ is strictly monotonically decreasing in $\nu$ and strictly monotonically increasing in $\tau$. Note that we now consider the range $3 \leqslant \nu \leqslant 16$. For the maximal value of $g$ we set $\nu = 3$ (we set it to 3!) and $\tau = 1.25$.

We consider now all combination of these domains:

- $-1 \leqslant \mu \leqslant 0$ and $-0.1 \leqslant \omega \leqslant 0$:

  $g$ is decreasing in $\mu$ and decreasing in $\omega$. We set $\mu = -1$ and $\omega = -0.1$.

$$g(-1, -0.1, 3, 1.25, \lambda_{01}, \alpha_{01}) \ = \ -0.0180173 \ . \tag{34}$$

- $-1 \leqslant \mu \leqslant 0$ and $0 \leqslant \omega \leqslant 0.1$:

  $g$ is increasing in $\mu$ and decreasing in $\omega$. We set $\mu = 0$ and $\omega = 0$.

$$g(0, 0, 3, 1.25, \lambda_{01}, \alpha_{01}) \ = \ -0.148532 \ . \tag{35}$$

- $0 \leqslant \mu \leqslant 1$ and $-0.1 \leqslant \omega \leqslant 0$:

  $g$ is decreasing in $\mu$ and increasing in $\omega$. We set $\mu = 0$ and $\omega = 0$.

$$g(0, 0, 3, 1.25, \lambda_{01}, \alpha_{01}) \ = \ -0.148532 \ . \tag{36}$$

- $0 \leqslant \mu \leqslant 1$ and $0 \leqslant \omega \leqslant 0.1$:

  $g$ is increasing in $\mu$ and increasing in $\omega$. We set $\mu = 1$ and $\omega = 0.1$.

$$g(1, 0.1, 3, 1.25, \lambda_{01}, \alpha_{01}) \ = \ -0.0180173 \ . \tag{37}$$

Therefore the maximal value of $g$ is $-0.0180173$.

$\square$

## S3.3   Proof of Theorem 3

First we recall Theorem 3:

**Theorem** (Increasing $\nu$). *We consider $\lambda = \lambda_{01}$, $\alpha = \alpha_{01}$ and the two domains $\Omega_1^- = \{(\mu, \omega, \nu, \tau) \mid -0.1 \leqslant \mu \leqslant 0.1, -0.1 \leqslant \omega \leqslant 0.1, 0.05 \leqslant \nu \leqslant 0.16, 0.8 \leqslant \tau \leqslant 1.25\}$ and $\Omega_2^- = \{(\mu, \omega, \nu, \tau) \mid -0.1 \leqslant \mu \leqslant 0.1, -0.1 \leqslant \omega \leqslant 0.1, 0.05 \leqslant \nu \leqslant 0.24, 0.9 \leqslant \tau \leqslant 1.25\}$.*

*The mapping of the variance $\tilde{\nu}(\mu, \omega, \nu, \tau, \lambda, \alpha)$ given in Eq. (6) increases*

$$\tilde{\nu}(\mu, \omega, \nu, \tau, \lambda_{01}, \alpha_{01}) \ > \ \nu \tag{38}$$

*in both $\Omega_1^-$ and $\Omega_2^-$. All fixed points $(\mu, \nu)$ of mapping Eq. (6) and Eq. (5) ensure for $0.8 \leqslant \tau$ that $\tilde{\nu} > 0.16$ and for $0.9 \leqslant \tau$ that $\tilde{\nu} > 0.24$. Consequently, the variance mapping Eq. (6) and Eq. (5) ensures a lower bound on the variance $\nu$.*

*Proof.* The mean value theorem states that there exists a $t \in [0, 1]$ for which

$$\tilde{\xi}(\mu, \omega, \nu, \tau, \lambda_{01}, \alpha_{01}) \, - \, \tilde{\xi}(\mu, \omega, \nu_{\min}, \tau, \lambda_{01}, \alpha_{01}) \, = \tag{39}$$
$$\frac{\partial}{\partial \nu} \tilde{\xi}(\mu, \omega, \nu + t(\nu_{\min} - \nu), \tau, \lambda_{01}, \alpha_{01}) \, (\nu - \nu_{\min}) \, .$$

Therefore

$$\tilde{\xi}(\mu, \omega, \nu, \tau, \lambda_{01}, \alpha_{01}) \, = \, \tilde{\xi}(\mu, \omega, \nu_{\min}, \tau, \lambda_{01}, \alpha_{01}) \, + \tag{40}$$
$$\frac{\partial}{\partial \nu} \tilde{\xi}(\mu, \omega, \nu + t(\nu_{\min} - \nu), \tau, \lambda_{01}, \alpha_{01}) \, (\nu - \nu_{\min}) \, .$$

Therefore we are interested to bound the derivative of the $\xi$-mapping Eq. (7) with respect to $\nu$:

$$\frac{\partial}{\partial \nu} \tilde{\xi}(\mu, \omega, \nu, \tau, \lambda_{01}, \alpha_{01}) \, = \tag{41}$$
$$\frac{1}{2} \lambda^2 \cdot \tau e^{-\frac{\mu^2 \omega^2}{2 \cdot \nu \tau}} \left( \alpha^2 \left( - \left( e^{\left( \frac{\mu \omega + \nu \tau}{\sqrt{2}\sqrt{\nu\tau}} \right)^2} \operatorname{erfc}\left( \frac{\mu \omega + \nu \tau}{\sqrt{2}\sqrt{\nu\tau}} \right) - 2 e^{\left( \frac{\mu \omega + 2 \cdot \nu\tau}{\sqrt{2}\sqrt{\nu\tau}} \right)^2} \operatorname{erfc}\left( \frac{\mu \omega + 2 \cdot \nu\tau}{\sqrt{2}\sqrt{\nu\tau}} \right) \right) \right) -$$
$$\operatorname{erfc}\left( \frac{\mu \omega}{\sqrt{2}\sqrt{\nu\tau}} \right) + 2 \right) \, .$$

The sub-term Eq. (298) enters the derivative Eq. (41) with a negative sign! According to Lemma 18, the minimal value of sub-term Eq. (298) is obtained by the largest largest $\nu$, by the smallest $\tau$, and the largest $y = \mu\omega = 0.01$. Also the positive term $\operatorname{erfc}\left( \frac{\mu\omega}{\sqrt{2}\sqrt{\nu\tau}} \right) + 2$ is multiplied by $\tau$, which is minimized by using the smallest $\tau$. Therefore we can use the smallest $\tau$ in whole formula Eq. (41) to lower bound it.

First we consider the domain $0.05 \leqslant \nu \leqslant 0.16$ and $0.8 \leqslant \tau \leqslant 1.25$. The factor consisting of the exponential in front of the brackets has its smallest value for $e^{-\frac{0.01 \cdot 0.01}{2 \cdot 0.05 \cdot 0.8}}$. Since $\operatorname{erfc}$ is monotonically decreasing we inserted the smallest argument via $\operatorname{erfc}\left( -\frac{0.01}{\sqrt{2}\sqrt{0.05 \cdot 0.8}} \right)$ in order to obtain the maximal negative contribution. Thus, applying Lemma 18, we obtain the lower bound on the derivative:

$$\frac{1}{2} \lambda^2 \cdot \tau e^{-\frac{\mu^2 \omega^2}{2 \cdot \nu\tau}} \left( \alpha^2 \left( - \left( e^{\left( \frac{\mu\omega + \nu\tau}{\sqrt{2}\sqrt{\nu\tau}} \right)^2} \operatorname{erfc}\left( \frac{\mu\omega + \nu\tau}{\sqrt{2}\sqrt{\nu\tau}} \right) - 2 e^{\left( \frac{\mu\omega + 2 \cdot \nu\tau}{\sqrt{2}\sqrt{\nu\tau}} \right)^2} \operatorname{erfc}\left( \frac{\mu\omega + 2 \cdot \nu\tau}{\sqrt{2}\sqrt{\nu\tau}} \right) \right) \right) -$$
$$\tag{42}$$
$$\operatorname{erfc}\left( \frac{\mu\omega}{\sqrt{2}\sqrt{\nu\tau}} \right) + 2 \right) \, >$$
$$\frac{1}{2} 0.8 e^{-\frac{0.01 \cdot 0.01}{2 \cdot 0.05 \cdot 0.8}} \lambda_{01}^2 \left( \alpha_{01}^2 \left( - \left( e^{\left( \frac{0.16 \cdot 0.8 + 0.01}{\sqrt{2}\sqrt{0.16 \cdot 0.8}} \right)^2} \operatorname{erfc}\left( \frac{0.16 \cdot 0.8 + 0.01}{\sqrt{2}\sqrt{0.16 \cdot 0.8}} \right) - \right. \right. \right.$$
$$\left. \left. \left. 2 e^{\left( \frac{2 \cdot 0.16 \cdot 0.8 + 0.01}{\sqrt{2}\sqrt{0.16 \cdot 0.8}} \right)^2} \operatorname{erfc}\left( \frac{2 \cdot 0.16 \cdot 0.8 + 0.01}{\sqrt{2}\sqrt{0.16 \cdot 0.8}} \right) \right) \right) - \operatorname{erfc}\left( -\frac{0.01}{\sqrt{2}\sqrt{0.05 \cdot 0.8}} \right) + 2 \right) \, > \, 0.969231 \, .$$

For applying the mean value theorem, we require the smallest $\tilde{\nu}(\nu)$. We follow the proof of Lemma 8, which shows that at the minimum $y = \mu\omega$ must be maximal and $x = \nu\tau$ must be minimal. Thus, the smallest $\tilde{\xi}(\mu, \omega, \nu, \tau, \lambda_{01}, \alpha_{01})$ is $\tilde{\xi}(0.01, 0.01, 0.05, 0.8, \lambda_{01}, \alpha_{01}) = 0.0662727$ for $0.05 \leqslant \nu$ and $0.8 \leqslant \tau$.

Therefore the mean value theorem and the bound on $(\tilde{\mu})^2$ (Lemma 43) provide

$$
\begin{aligned}
\tilde{\nu} = \tilde{\xi}(\mu, \omega, \nu, \tau, \lambda_{01}, \alpha_{01}) - (\tilde{\mu}(\mu, \omega, \nu, \tau, \lambda_{01}, \alpha_{01}))^2 > & \quad (43)\\
0.0662727 + 0.969231(\nu - 0.05) - 0.005 = 0.01281115 + 0.969231\nu > &\\
0.08006969 \cdot 0.16 + 0.969231\nu \geqslant 1.049301\nu > \nu \,. &
\end{aligned}
$$

Next we consider the domain $0.05 \leqslant \nu \leqslant 0.24$ and $0.9 \leqslant \tau \leqslant 1.25$. The factor consisting of the exponential in front of the brackets has its smallest value for $e^{-\frac{0.01 \cdot 0.01}{2 \cdot 0.05 \cdot 0.9}}$. Since erfc is monotonically decreasing we inserted the smallest argument via $\mathrm{erfc}\left(-\frac{0.01}{\sqrt{2}\sqrt{0.05 \cdot 0.9}}\right)$ in order to obtain the maximal negative contribution.

Thus, applying Lemma 18, we obtain the lower bound on the derivative:

$$
\frac{1}{2}\lambda^2 \cdot \tau e^{-\frac{\mu^2 \omega^2}{2 \cdot \nu\tau}} \left( \alpha^2 \left( - \left( e^{\left(\frac{\mu\omega + \nu\tau}{\sqrt{2}\sqrt{\nu\tau}}\right)^2} \mathrm{erfc}\left(\frac{\mu\omega + \nu\tau}{\sqrt{2}\sqrt{\nu\tau}}\right) - 2e^{\left(\frac{\mu\omega + 2 \cdot \nu\tau}{\sqrt{2}\sqrt{\nu\tau}}\right)^2} \mathrm{erfc}\left(\frac{\mu\omega + 2 \cdot \nu\tau}{\sqrt{2}\sqrt{\nu\tau}}\right) \right) \right) - 
$$
$$
(44)
$$
$$
\mathrm{erfc}\left(\frac{\mu\omega}{\sqrt{2}\sqrt{\nu\tau}}\right) + 2 \right) >
$$
$$
\frac{1}{2}0.9 e^{-\frac{0.01 \cdot 0.01}{2 \cdot 0.05 \cdot 0.9}} \lambda_{01}^2 \left( \alpha_{01}^2 \left( - \left( e^{\left(\frac{0.24 \cdot 0.9 + 0.01}{\sqrt{2}\sqrt{0.24 \cdot 0.9}}\right)^2} \mathrm{erfc}\left(\frac{0.24 \cdot 0.9 + 0.01}{\sqrt{2}\sqrt{0.24 \cdot 0.9}}\right) - \right. \right. \right.
$$
$$
2e^{\left(\frac{2 \cdot 0.24 \cdot 0.9 + 0.01}{\sqrt{2}\sqrt{0.24 \cdot 0.9}}\right)^2} \mathrm{erfc}\left(\frac{2 \cdot 0.24 \cdot 0.9 + 0.01}{\sqrt{2}\sqrt{0.24 \cdot 0.9}}\right) \right) \right) - \mathrm{erfc}\left(-\frac{0.01}{\sqrt{2}\sqrt{0.05 \cdot 0.9}}\right) + 2 \right) > 0.976952 \,.
$$

For applying the mean value theorem, we require the smallest $\tilde{\nu}(\nu)$. We follow the proof of Lemma 8, which shows that at the minimum $y = \mu\omega$ must be maximal and $x = \nu\tau$ must be minimal. Thus, the smallest $\tilde{\xi}(\mu, \omega, \nu, \tau, \lambda_{01}, \alpha_{01})$ is $\tilde{\xi}(0.01, 0.01, 0.05, 0.9, \lambda_{01}, \alpha_{01}) = 0.0738404$ for $0.05 \leqslant \nu$ and $0.9 \leqslant \tau$. Therefore the mean value theorem and the bound on $(\tilde{\mu})^2$ (Lemma 43) gives

$$
\begin{aligned}
\tilde{\nu} = \tilde{\xi}(\mu, \omega, \nu, \tau, \lambda_{01}, \alpha_{01}) - (\tilde{\mu}(\mu, \omega, \nu, \tau, \lambda_{01}, \alpha_{01}))^2 > & \quad (45)\\
0.0738404 + 0.976952(\nu - 0.05) - 0.005 = 0.0199928 + 0.976952 \cdot \nu > &\\
0.08330333 \cdot 0.24 + 0.976952\nu \geqslant 1.060255\nu > \nu \,. &
\end{aligned}
$$

$\square$

## S3.4  Lemmata and Other Tools Required for the Proofs

### S3.4.1  Lemmata for proofing Theorem 1 (part 1): Jacobian norm smaller than one

In this section, we show that the largest singular value of the Jacobian of the mapping $g$ is smaller than one. Therefore, $g$ is a contraction mapping. This is even true in a larger domain than the original $\Omega$. We do not need to restrict $\tau \in [0.95, 1.1]$, but we can extend to $\tau \in [0.8, 1.25]$. The range of the other variables is unchanged such that we consider the following domain throughout this section: $\mu \in [-0.1, 0.1]$, $\omega \in [-0.1, 0.1]$, $\nu \in [0.8, 1.5]$, and $\tau \in [0.95, 1.1]$.

**Jacobian of the mapping.**    In the following, we denote two Jacobians: (1) the Jacobian $\mathcal{J}$ of the mapping $h : (\mu, \nu) \mapsto (\tilde{\mu}, \tilde{\xi})$, and (2) the Jacobian $\mathcal{H}$ of the mapping $g : (\mu, \nu) \mapsto (\tilde{\mu}, \tilde{\nu})$ because the influence of $\tilde{\mu}$ on $\tilde{\nu}$ is small, and many properties of the system can already be seen on $\mathcal{J}$.

$$
\mathcal{J} = \begin{pmatrix} \mathcal{J}_{11} & \mathcal{J}_{12} \\ \mathcal{J}_{21} & \mathcal{J}_{22} \end{pmatrix} = \begin{pmatrix} \frac{\partial}{\partial \mu}\tilde{\mu} & \frac{\partial}{\partial \nu}\tilde{\mu} \\ \frac{\partial}{\partial \mu}\tilde{\xi} & \frac{\partial}{\partial \nu}\tilde{\xi} \end{pmatrix}
\tag{46}
$$

$$
\mathcal{H} = \begin{pmatrix} \mathcal{H}_{11} & \mathcal{H}_{12} \\ \mathcal{H}_{21} & \mathcal{H}_{22} \end{pmatrix} = \begin{pmatrix} \mathcal{J}_{11} & \mathcal{J}_{12} \\ \mathcal{J}_{21} - 2\tilde{\mu}\mathcal{J}_{11} & \mathcal{J}_{22} - 2\tilde{\mu}\mathcal{J}_{12} \end{pmatrix}
\tag{47}
$$

The definition of the entries of the Jacobian $\mathcal{J}$ is:

$$
\mathcal{J}_{11}(\mu, \omega, \nu, \tau, \lambda, \alpha) = \frac{\partial}{\partial \mu}\tilde{\mu}(\mu, \omega, \nu, \tau, \lambda, \alpha) =
\tag{48}
$$

$$
\frac{1}{2}\lambda\omega\left(\alpha e^{\mu\omega + \frac{\nu\tau}{2}}\operatorname{erfc}\left(\frac{\mu\omega + \nu\tau}{\sqrt{2}\sqrt{\nu\tau}}\right) - \operatorname{erfc}\left(\frac{\mu\omega}{\sqrt{2}\sqrt{\nu\tau}}\right) + 2\right)
$$

$$
\mathcal{J}_{12}(\mu, \omega, \nu, \tau, \lambda, \alpha) = \frac{\partial}{\partial \nu}\tilde{\mu}(\mu, \omega, \nu, \tau, \lambda, \alpha) =
\tag{49}
$$

$$
\frac{1}{4}\lambda\tau\left(\alpha e^{\mu\omega + \frac{\nu\tau}{2}}\operatorname{erfc}\left(\frac{\mu\omega + \nu\tau}{\sqrt{2}\sqrt{\nu\tau}}\right) - (\alpha - 1)\sqrt{\frac{2}{\pi\nu\tau}}e^{-\frac{\mu^2\omega^2}{2\nu\tau}}\right)
$$

$$
\mathcal{J}_{21}(\mu, \omega, \nu, \tau, \lambda, \alpha) = \frac{\partial}{\partial \mu}\tilde{\xi}(\mu, \omega, \nu, \tau, \lambda, \alpha) =
\tag{50}
$$

$$
\lambda^2\omega\left(\alpha^2\left(-e^{\mu\omega + \frac{\nu\tau}{2}}\right)\operatorname{erfc}\left(\frac{\mu\omega + \nu\tau}{\sqrt{2}\sqrt{\nu\tau}}\right) + \right.
$$

$$
\left. \alpha^2 e^{2\mu\omega + 2\nu\tau}\operatorname{erfc}\left(\frac{\mu\omega + 2\nu\tau}{\sqrt{2}\sqrt{\nu\tau}}\right) + \mu\omega\left(2 - \operatorname{erfc}\left(\frac{\mu\omega}{\sqrt{2}\sqrt{\nu\tau}}\right)\right) + \sqrt{\frac{2}{\pi}}\sqrt{\nu\tau}e^{-\frac{\mu^2\omega^2}{2\nu\tau}}\right)
$$

$$
\mathcal{J}_{22}(\mu, \omega, \nu, \tau, \lambda, \alpha) = \frac{\partial}{\partial \nu}\tilde{\xi}(\mu, \omega, \nu, \tau, \lambda, \alpha) =
\tag{51}
$$

$$
\frac{1}{2}\lambda^2\tau\left(\alpha^2\left(-e^{\mu\omega + \frac{\nu\tau}{2}}\right)\operatorname{erfc}\left(\frac{\mu\omega + \nu\tau}{\sqrt{2}\sqrt{\nu\tau}}\right) + \right.
$$

$$
\left. 2\alpha^2 e^{2\mu\omega + 2\nu\tau}\operatorname{erfc}\left(\frac{\mu\omega + 2\nu\tau}{\sqrt{2}\sqrt{\nu\tau}}\right) - \operatorname{erfc}\left(\frac{\mu\omega}{\sqrt{2}\sqrt{\nu\tau}}\right) + 2\right)
$$

**Proof sketch: Bounding the largest singular value of the Jacobian.**    If the largest singular value of the Jacobian is smaller than 1, then the spectral norm of the Jacobian is smaller than 1. Then the mapping Eq. (5) and Eq. (6) of the mean and variance to the mean and variance in the next layer is contracting.

We show that the largest singular value is smaller than 1 by evaluating the function $S(\mu, \omega, \nu, \tau, \lambda, \alpha)$ on a grid. Then we use the Mean Value Theorem to bound the deviation of the function $S$ between grid points. Toward this end we have to bound the gradient of $S$ with respect to $(\mu, \omega, \nu, \tau)$. If all function values plus gradient times the deltas (differences between grid points and evaluated points) is still smaller than 1, then we have proofed that the function is below 1.

The singular values of the $2 \times 2$ matrix

$$\boldsymbol{A} \;=\; \left( \begin{array}{cc} a_{11} & a_{12} \\ a_{21} & a_{22} \end{array} \right) \tag{52}$$

are

$$s_1 \;=\; \frac{1}{2} \left( \sqrt{(a_{11} + a_{22})^2 + (a_{21} - a_{12})^2} \;+\; \sqrt{(a_{11} - a_{22})^2 + (a_{12} + a_{21})^2} \right) \tag{53}$$

$$s_2 = \frac{1}{2} \left( \sqrt{(a_{11} + a_{22})^2 + (a_{21} - a_{12})^2} \;-\; \sqrt{(a_{11} - a_{22})^2 + (a_{12} + a_{21})^2} \right). \tag{54}$$

We used an explicit formula for the singular values (Blinn, 1996). We now set $\mathcal{H}_{11} = a_{11}, \mathcal{H}_{12} = a_{12}, \mathcal{H}_{21} = a_{21}, \mathcal{H}_{22} = a_{22}$ to obtain a formula for the largest singular value of the Jacobian depending on $(\mu, \omega, \nu, \tau, \lambda, \alpha)$. The formula for the largest singular value for the Jacobian is:

$$S(\mu,\omega,\nu,\tau,\lambda,\alpha) \;=\; \left( \sqrt{(\mathcal{H}_{11} + \mathcal{H}_{22})^2 + (\mathcal{H}_{21} - \mathcal{H}_{12})^2} \;+\; \sqrt{(\mathcal{H}_{11} - \mathcal{H}_{22})^2 + (\mathcal{H}_{12} + \mathcal{H}_{21})^2} \right) =$$
$$\tag{55}$$

$$= \frac{1}{2} \left( \sqrt{(\mathcal{J}_{11} + \mathcal{J}_{22} - 2\tilde{\mu}\mathcal{J}_{12})^2 + (\mathcal{J}_{21} - 2\tilde{\mu}\mathcal{J}_{11} - \mathcal{J}_{12})^2} + \right.$$
$$\left. \sqrt{(\mathcal{J}_{11} - \mathcal{J}_{22} + 2\tilde{\mu}\mathcal{J}_{12})^2 + (\mathcal{J}_{12} + \mathcal{J}_{21} - 2\tilde{\mu}\mathcal{J}_{11})^2} \right),$$

where $\mathcal{J}$ are defined in Eq. (48) and we left out the dependencies on $(\mu, \omega, \nu, \tau, \lambda, \alpha)$ in order to keep the notation uncluttered, e.g. we wrote $\mathcal{J}_{11}$ instead of $\mathcal{J}_{11}(\mu, \omega, \nu, \tau, \lambda, \alpha)$.

**Bounds on the derivatives of the Jacobian entries.**   In order to bound the gradient of the singular value, we have to bound the derivatives of the Jacobian entries $\mathcal{J}_{11}(\mu, \omega, \nu, \tau, \lambda, \alpha)$, $\mathcal{J}_{12}(\mu, \omega, \nu, \tau, \lambda, \alpha)$, $\mathcal{J}_{21}(\mu, \omega, \nu, \tau, \lambda, \alpha)$, and $\mathcal{J}_{22}(\mu, \omega, \nu, \tau, \lambda, \alpha)$ with respect to $\mu$, $\omega$, $\nu$, and $\tau$. The values $\lambda$ and $\alpha$ are fixed to $\lambda_{01}$ and $\alpha_{01}$. The 16 derivatives of the 4 Jacobian entries with respect to the 4 variables are:

$$\frac{\partial \mathcal{J}_{11}}{\partial \mu} \;=\; \frac{1}{2}\lambda\omega^2 e^{-\frac{\mu^2\omega^2}{2\nu\tau}} \left( \alpha e^{\frac{(\mu\omega+\nu\tau)^2}{2\nu\tau}} \operatorname{erfc}\left( \frac{\mu\omega+\nu\tau}{\sqrt{2}\sqrt{\nu\tau}} \right) - \frac{\sqrt{\frac{2}{\pi}}(\alpha-1)}{\sqrt{\nu\tau}} \right) \tag{56}$$

$$\frac{\partial \mathcal{J}_{11}}{\partial \omega} \;=\; \frac{1}{2}\lambda \left( -e^{-\frac{\mu^2\omega^2}{2\nu\tau}} \left( \frac{\sqrt{\frac{2}{\pi}}(\alpha-1)\mu\omega}{\sqrt{\nu\tau}} - \alpha(\mu\omega+1)e^{\frac{(\mu\omega+\nu\tau)^2}{2\nu\tau}} \operatorname{erfc}\left( \frac{\mu\omega+\nu\tau}{\sqrt{2}\sqrt{\nu\tau}} \right) \right) - \right.$$
$$\left. \operatorname{erfc}\left( \frac{\mu\omega}{\sqrt{2}\sqrt{\nu\tau}} \right) + 2 \right)$$

$$\frac{\partial \mathcal{J}_{11}}{\partial \nu} \;=\; \frac{1}{4}\lambda\tau\omega e^{-\frac{\mu^2\omega^2}{2\nu\tau}} \left( \alpha e^{\frac{(\mu\omega+\nu\tau)^2}{2\nu\tau}} \operatorname{erfc}\left( \frac{\mu\omega+\nu\tau}{\sqrt{2}\sqrt{\nu\tau}} \right) + \sqrt{\frac{2}{\pi}} \left( \frac{(\alpha-1)\mu\omega}{(\nu\tau)^{3/2}} - \frac{\alpha}{\sqrt{\nu\tau}} \right) \right)$$

$$\frac{\partial \mathcal{J}_{11}}{\partial \tau} \;=\; \frac{1}{4}\lambda\nu\omega e^{-\frac{\mu^2\omega^2}{2\nu\tau}} \left( \alpha e^{\frac{(\mu\omega+\nu\tau)^2}{2\nu\tau}} \operatorname{erfc}\left( \frac{\mu\omega+\nu\tau}{\sqrt{2}\sqrt{\nu\tau}} \right) + \sqrt{\frac{2}{\pi}} \left( \frac{(\alpha-1)\mu\omega}{(\nu\tau)^{3/2}} - \frac{\alpha}{\sqrt{\nu\tau}} \right) \right)$$

$$\frac{\partial \mathcal{J}_{12}}{\partial \mu} = \frac{\partial \mathcal{J}_{11}}{\partial \nu}$$

$$\frac{\partial \mathcal{J}_{12}}{\partial \omega} = \frac{1}{4}\lambda\mu\tau e^{-\frac{\mu^2\omega^2}{2\nu\tau}}\left(\alpha e^{\frac{(\mu\omega+\nu\tau)^2}{2\nu\tau}}\,\mathrm{erfc}\left(\frac{\mu\omega+\nu\tau}{\sqrt{2}\sqrt{\nu\tau}}\right)+\sqrt{\frac{2}{\pi}}\left(\frac{(\alpha-1)\mu\omega}{(\nu\tau)^{3/2}}-\frac{\alpha}{\sqrt{\nu\tau}}\right)\right)$$

$$\frac{\partial \mathcal{J}_{12}}{\partial \nu} = \frac{1}{8}\lambda e^{-\frac{\mu^2\omega^2}{2\nu\tau}}\left(\alpha\tau^2 e^{\frac{(\mu\omega+\nu\tau)^2}{2\nu\tau}}\,\mathrm{erfc}\left(\frac{\mu\omega+\nu\tau}{\sqrt{2}\sqrt{\nu\tau}}\right)+\right.$$
$$\left.\sqrt{\frac{2}{\pi}}\left(\frac{(-1)(\alpha-1)\mu^2\omega^2}{\nu^{5/2}\sqrt{\tau}}+\frac{\sqrt{\tau}(\alpha+\alpha\mu\omega-1)}{\nu^{3/2}}-\frac{\alpha\tau^{3/2}}{\sqrt{\nu}}\right)\right)$$

$$\frac{\partial \mathcal{J}_{12}}{\partial \tau} = \frac{1}{8}\lambda e^{-\frac{\mu^2\omega^2}{2\nu\tau}}\left(2\alpha e^{\frac{(\mu\omega+\nu\tau)^2}{2\nu\tau}}\,\mathrm{erfc}\left(\frac{\mu\omega+\nu\tau}{\sqrt{2}\sqrt{\nu\tau}}\right)+\alpha\nu\tau e^{\frac{(\mu\omega+\nu\tau)^2}{2\nu\tau}}\,\mathrm{erfc}\left(\frac{\mu\omega+\nu\tau}{\sqrt{2}\sqrt{\nu\tau}}\right)+\right.$$
$$\left.\sqrt{\frac{2}{\pi}}\left(\frac{(-1)(\alpha-1)\mu^2\omega^2}{(\nu\tau)^{3/2}}+\frac{-\alpha+\alpha\mu\omega+1}{\sqrt{\nu\tau}}-\alpha\sqrt{\nu\tau}\right)\right)$$

$$\frac{\partial \mathcal{J}_{21}}{\partial \mu} = \lambda^2\omega^2\left(\alpha^2\left(-e^{-\frac{\mu^2\omega^2}{2\nu\tau}}\right)e^{\frac{(\mu\omega+\nu\tau)^2}{2\nu\tau}}\,\mathrm{erfc}\left(\frac{\mu\omega+\nu\tau}{\sqrt{2}\sqrt{\nu\tau}}\right)+\right.$$
$$\left.2\alpha^2 e^{\frac{(\mu\omega+2\nu\tau)^2}{2\nu\tau}}e^{-\frac{\mu^2\omega^2}{2\nu\tau}}\,\mathrm{erfc}\left(\frac{\mu\omega+2\nu\tau}{\sqrt{2}\sqrt{\nu\tau}}\right)-\mathrm{erfc}\left(\frac{\mu\omega}{\sqrt{2}\sqrt{\nu\tau}}\right)+2\right)$$

$$\frac{\partial \mathcal{J}_{21}}{\partial \omega} = \lambda^2\left(\alpha^2(\mu\omega+1)\left(-e^{-\frac{\mu^2\omega^2}{2\nu\tau}}\right)e^{\frac{(\mu\omega+\nu\tau)^2}{2\nu\tau}}\,\mathrm{erfc}\left(\frac{\mu\omega+\nu\tau}{\sqrt{2}\sqrt{\nu\tau}}\right)+\right.$$
$$\alpha^2(2\mu\omega+1)e^{\frac{(\mu\omega+2\nu\tau)^2}{2\nu\tau}}e^{-\frac{\mu^2\omega^2}{2\nu\tau}}\,\mathrm{erfc}\left(\frac{\mu\omega+2\nu\tau}{\sqrt{2}\sqrt{\nu\tau}}\right)+$$
$$\left.2\mu\omega\left(2-\mathrm{erfc}\left(\frac{\mu\omega}{\sqrt{2}\sqrt{\nu\tau}}\right)\right)+\sqrt{\frac{2}{\pi}}\sqrt{\nu\tau}e^{-\frac{\mu^2\omega^2}{2\nu\tau}}\right)$$

$$\frac{\partial \mathcal{J}_{21}}{\partial \nu} = \frac{1}{2}\lambda^2\tau\omega e^{-\frac{\mu^2\omega^2}{2\nu\tau}}\left(\alpha^2\left(-e^{\frac{(\mu\omega+\nu\tau)^2}{2\nu\tau}}\right)\mathrm{erfc}\left(\frac{\mu\omega+\nu\tau}{\sqrt{2}\sqrt{\nu\tau}}\right)+\right.$$
$$\left.4\alpha^2 e^{\frac{(\mu\omega+2\nu\tau)^2}{2\nu\tau}}\,\mathrm{erfc}\left(\frac{\mu\omega+2\nu\tau}{\sqrt{2}\sqrt{\nu\tau}}\right)+\frac{\sqrt{\frac{2}{\pi}}(-1)\left(\alpha^2-1\right)}{\sqrt{\nu\tau}}\right)$$

$$\frac{\partial \mathcal{J}_{21}}{\partial \tau} = \frac{1}{2}\lambda^2\nu\omega e^{-\frac{\mu^2\omega^2}{2\nu\tau}}\left(\alpha^2\left(-e^{\frac{(\mu\omega+\nu\tau)^2}{2\nu\tau}}\right)\mathrm{erfc}\left(\frac{\mu\omega+\nu\tau}{\sqrt{2}\sqrt{\nu\tau}}\right)+\right.$$
$$\left.4\alpha^2 e^{\frac{(\mu\omega+2\nu\tau)^2}{2\nu\tau}}\,\mathrm{erfc}\left(\frac{\mu\omega+2\nu\tau}{\sqrt{2}\sqrt{\nu\tau}}\right)+\frac{\sqrt{\frac{2}{\pi}}(-1)\left(\alpha^2-1\right)}{\sqrt{\nu\tau}}\right)$$

$$\frac{\partial \mathcal{J}_{22}}{\partial \mu} = \frac{\partial \mathcal{J}_{21}}{\partial \nu}$$

$$\frac{\partial \mathcal{J}_{22}}{\partial \omega} = \frac{1}{2}\lambda^2\mu\tau e^{-\frac{\mu^2\omega^2}{2\nu\tau}}\left(\alpha^2\left(-e^{\frac{(\mu\omega+\nu\tau)^2}{2\nu\tau}}\right)\mathrm{erfc}\left(\frac{\mu\omega+\nu\tau}{\sqrt{2}\sqrt{\nu\tau}}\right)+\right.$$
$$\left.4\alpha^2 e^{\frac{(\mu\omega+2\nu\tau)^2}{2\nu\tau}}\,\mathrm{erfc}\left(\frac{\mu\omega+2\nu\tau}{\sqrt{2}\sqrt{\nu\tau}}\right)+\frac{\sqrt{\frac{2}{\pi}}(-1)\left(\alpha^2-1\right)}{\sqrt{\nu\tau}}\right)$$

$$\frac{\partial \mathcal{J}_{22}}{\partial \nu} = \frac{1}{4}\lambda^2\tau^2 e^{-\frac{\mu^2\omega^2}{2\nu\tau}}\left(\alpha^2\left(-e^{\frac{(\mu\omega+\nu\tau)^2}{2\nu\tau}}\right)\operatorname{erfc}\left(\frac{\mu\omega+\nu\tau}{\sqrt{2}\sqrt{\nu\tau}}\right)+\right.$$

$$\left.8\alpha^2 e^{\frac{(\mu\omega+2\nu\tau)^2}{2\nu\tau}}\operatorname{erfc}\left(\frac{\mu\omega+2\nu\tau}{\sqrt{2}\sqrt{\nu\tau}}\right)+\sqrt{\frac{2}{\pi}}\left(\frac{\left(\alpha^2-1\right)\mu\omega}{(\nu\tau)^{3/2}}-\frac{3\alpha^2}{\sqrt{\nu\tau}}\right)\right)$$

$$\frac{\partial \mathcal{J}_{22}}{\partial \tau} = \frac{1}{4}\lambda^2\left(-2\alpha^2 e^{-\frac{\mu^2\omega^2}{2\nu\tau}}e^{\frac{(\mu\omega+\nu\tau)^2}{2\nu\tau}}\operatorname{erfc}\left(\frac{\mu\omega+\nu\tau}{\sqrt{2}\sqrt{\nu\tau}}\right)-\right.$$

$$\alpha^2\nu\tau e^{-\frac{\mu^2\omega^2}{2\nu\tau}}e^{\frac{(\mu\omega+\nu\tau)^2}{2\nu\tau}}\operatorname{erfc}\left(\frac{\mu\omega+\nu\tau}{\sqrt{2}\sqrt{\nu\tau}}\right)+4\alpha^2 e^{\frac{(\mu\omega+2\nu\tau)^2}{2\nu\tau}}e^{-\frac{\mu^2\omega^2}{2\nu\tau}}\operatorname{erfc}\left(\frac{\mu\omega+2\nu\tau}{\sqrt{2}\sqrt{\nu\tau}}\right)+$$

$$8\alpha^2\nu\tau e^{\frac{(\mu\omega+2\nu\tau)^2}{2\nu\tau}}e^{-\frac{\mu^2\omega^2}{2\nu\tau}}\operatorname{erfc}\left(\frac{\mu\omega+2\nu\tau}{\sqrt{2}\sqrt{\nu\tau}}\right)+2\left(2-\operatorname{erfc}\left(\frac{\mu\omega}{\sqrt{2}\sqrt{\nu\tau}}\right)\right)+$$

$$\left.\sqrt{\frac{2}{\pi}}e^{-\frac{\mu^2\omega^2}{2\nu\tau}}\left(\frac{\left(\alpha^2-1\right)\mu\omega}{\sqrt{\nu\tau}}-3\alpha^2\sqrt{\nu\tau}\right)\right)$$

**Lemma 5** (Bounds on the Derivatives). *The following bounds on the absolute values of the derivatives of the Jacobian entries $\mathcal{J}_{11}(\mu,\omega,\nu,\tau,\lambda,\alpha)$, $\mathcal{J}_{12}(\mu,\omega,\nu,\tau,\lambda,\alpha)$, $\mathcal{J}_{21}(\mu,\omega,\nu,\tau,\lambda,\alpha)$, and $\mathcal{J}_{22}(\mu,\omega,\nu,\tau,\lambda,\alpha)$ with respect to $\mu$, $\omega$, $\nu$, and $\tau$ hold:*

$$\left|\frac{\partial \mathcal{J}_{11}}{\partial \mu}\right| < 0.0031049101995398316 \tag{57}$$

$$\left|\frac{\partial \mathcal{J}_{11}}{\partial \omega}\right| < 1.055872374194189$$

$$\left|\frac{\partial \mathcal{J}_{11}}{\partial \nu}\right| < 0.031242911235461816$$

$$\left|\frac{\partial \mathcal{J}_{11}}{\partial \tau}\right| < 0.03749149348255419$$

$$\left|\frac{\partial \mathcal{J}_{12}}{\partial \mu}\right| < 0.031242911235461816$$

$$\left|\frac{\partial \mathcal{J}_{12}}{\partial \omega}\right| < 0.031242911235461816$$

$$\left|\frac{\partial \mathcal{J}_{12}}{\partial \nu}\right| < 0.21232788238624354$$

$$\left|\frac{\partial \mathcal{J}_{12}}{\partial \tau}\right| < 0.2124377655377270$$

$$\left|\frac{\partial \mathcal{J}_{21}}{\partial \mu}\right| < 0.02220441024325437$$

$$\left|\frac{\partial \mathcal{J}_{21}}{\partial \omega}\right| < 1.146955401845684$$

$$\left| \frac{\partial \mathcal{J}_{21}}{\partial \nu} \right| < 0.14983446469110305$$

$$\left| \frac{\partial \mathcal{J}_{21}}{\partial \tau} \right| < 0.17980135762932363$$

$$\left| \frac{\partial \mathcal{J}_{22}}{\partial \mu} \right| < 0.14983446469110305$$

$$\left| \frac{\partial \mathcal{J}_{22}}{\partial \omega} \right| < 0.14983446469110305$$

$$\left| \frac{\partial \mathcal{J}_{22}}{\partial \nu} \right| < 1.805740052651535$$

$$\left| \frac{\partial \mathcal{J}_{22}}{\partial \tau} \right| < 2.396685907216327$$

*Proof.* See proof 39.                                                                                   □

**Bounds on the entries of the Jacobian.**

**Lemma 6** (Bound on J11)**.** *The absolute value of the function*
$\mathcal{J}_{11} = \frac{1}{2} \lambda \omega \left( \alpha e^{\mu \omega + \frac{\nu \tau}{2}} \operatorname{erfc} \left( \frac{\mu \omega + \nu \tau}{\sqrt{2} \sqrt{\nu \tau}} \right) - \operatorname{erfc} \left( \frac{\mu \omega}{\sqrt{2} \sqrt{\nu \tau}} \right) + 2 \right)$ *is bounded by* $|\mathcal{J}_{11}| \leqslant 0.104497$ *in the domain* $-0.1 \leqslant \mu \leqslant 0.1$, $-0.1 \leqslant \omega \leqslant 0.1$, $0.8 \leqslant \nu \leqslant 1.5$, *and* $0.8 \leqslant \tau \leqslant 1.25$ *for* $\alpha = \alpha_{01}$ *and* $\lambda = \lambda_{01}$.

*Proof.*

$$|\mathcal{J}_{11}| = \left| \frac{1}{2} \lambda \omega \left( \alpha e^{\mu \omega + \frac{\nu \tau}{2}} \operatorname{erfc} \left( \frac{\mu \omega + \nu \tau}{\sqrt{2} \sqrt{\nu \tau}} \right) + 2 - \operatorname{erfc} \left( \frac{\mu \omega}{\sqrt{2} \sqrt{\nu \tau}} \right) \right) \right|$$

$$\leqslant |\frac{1}{2}| |\lambda| |\omega| \left( |\alpha| 0.587622 + 1.00584 \right) \leqslant 0.104497,$$

(58)

where we used that (a) $J_{11}$ is strictly monotonically increasing in $\mu \omega$ and $|2 - \operatorname{erfc} \left( \frac{0.01}{\sqrt{2} \sqrt{\nu \tau}} \right)| \leqslant$
1.00584 and (b) Lemma 47 that $|e^{\mu \omega + \frac{\nu \tau}{2}} \operatorname{erfc} \left( \frac{\mu \omega + \nu \tau}{\sqrt{2} \sqrt{\nu \tau}} \right)| \leqslant e^{0.01 + \frac{0.64}{2}} \operatorname{erfc} \left( \frac{0.01 + 0.64}{\sqrt{2} \sqrt{0.64}} \right) = 0.587622$
□

**Lemma 7** (Bound on J12)**.** *The absolute value of the function*
$\mathcal{J}_{12} = \frac{1}{4} \lambda \tau \left( \alpha e^{\mu \omega + \frac{\nu \tau}{2}} \operatorname{erfc} \left( \frac{\mu \omega + \nu \tau}{\sqrt{2} \sqrt{\nu \tau}} \right) - (\alpha - 1) \sqrt{\frac{2}{\pi \nu \tau}} e^{-\frac{\mu^2 \omega^2}{2 \nu \tau}} \right)$ *is bounded by* $|\mathcal{J}_{12}| \leqslant 0.194145$
*in the domain* $-0.1 \leqslant \mu \leqslant 0.1$, $-0.1 \leqslant \omega \leqslant 0.1$, $0.8 \leqslant \nu \leqslant 1.5$, *and* $0.8 \leqslant \tau \leqslant 1.25$ *for*
$\alpha = \alpha_{01}$ *and* $\lambda = \lambda_{01}$.

*Proof.*

$$|J_{12}| \leqslant \frac{1}{4} |\lambda| |\tau| \left| \left( \alpha e^{\mu \omega + \frac{\nu \tau}{2}} \operatorname{erfc} \left( \frac{\mu \omega + \nu \tau}{\sqrt{2} \sqrt{\nu \tau}} \right) - (\alpha - 1) \sqrt{\frac{2}{\pi \nu \tau}} e^{-\frac{\mu^2 \omega^2}{2 \nu \tau}} \right) \right| \leqslant$$

$$\frac{1}{4}|\lambda||\tau|\,|0.983247 - 0.392294| \leqslant$$
$$0.194035 \tag{59}$$

For the first term we have $0.434947 \leqslant e^{\mu\omega + \frac{\nu\tau}{2}} \operatorname{erfc}\left(\frac{\mu\omega + \nu\tau}{\sqrt{2}\sqrt{\nu\tau}}\right) \leqslant 0.587622$ after Lemma [47] and for the second term $0.582677 \leqslant \sqrt{\frac{2}{\pi\nu\tau}} e^{-\frac{\mu^2\omega^2}{2\nu\tau}} \leqslant 0.997356$, which can easily be seen by maximizing or minimizing the arguments of the exponential or the square root function. The first term scaled by $\alpha$ is $0.727780 \leqslant \alpha e^{\mu\omega + \frac{\nu\tau}{2}} \operatorname{erfc}\left(\frac{\mu\omega + \nu\tau}{\sqrt{2}\sqrt{\nu\tau}}\right) \leqslant 0.983247$ and the second term scaled by $\alpha - 1$ is $0.392294 \leqslant (\alpha - 1)\sqrt{\frac{2}{\pi\nu\tau}} e^{-\frac{\mu^2\omega^2}{2\nu\tau}} \leqslant 0.671484$. Therefore, the absolute difference between these terms is at most $0.983247 - 0.392294$ leading to the derived bound.

$\square$

**Bounds on mean, variance and second moment.** For deriving bounds on $\tilde{\mu}$, $\tilde{\xi}$, and $\tilde{\nu}$, we need the following lemma.

**Lemma 8** (Derivatives of the Mapping). *We assume $\alpha = \alpha_{01}$ and $\lambda = \lambda_{01}$. We restrict the range of the variables to the domain $\mu \in [-0.1, 0.1]$, $\omega \in [-0.1, 0.1]$, $\nu \in [0.8, 1.5]$, and $\tau \in [0.8, 1.25]$.*

*The derivative $\frac{\partial}{\partial\mu}\tilde{\mu}(\mu, \omega, \nu, \tau, \lambda, \alpha)$ has the sign of $\omega$.*

*The derivative $\frac{\partial}{\partial\nu}\tilde{\mu}(\mu, \omega, \nu, \tau, \lambda, \alpha)$ is positive.*

*The derivative $\frac{\partial}{\partial\mu}\tilde{\xi}(\mu, \omega, \nu, \tau, \lambda, \alpha)$ has the sign of $\omega$.*

*The derivative $\frac{\partial}{\partial\nu}\tilde{\xi}(\mu, \omega, \nu, \tau, \lambda, \alpha)$ is positive.*

*Proof.* See [40]. $\square$

**Lemma 9** (Bounds on mean, variance and second moment). *The expressions $\tilde{\mu}$, $\tilde{\xi}$, and $\tilde{\nu}$ for $\alpha = \alpha_{01}$ and $\lambda = \lambda_{01}$ are bounded by $-0.041160 < \tilde{\mu} < 0.087653$, $0.703257 < \tilde{\xi} < 1.643705$ and $0.695574 < \tilde{\nu} < 1.636023$ in the domain $\mu \in [-0.1, 0.1]$, $\nu \in [0.8, 15]$, $\omega \in [-0.1, 0.1]$, $\tau \in [0.8, 1.25]$.*

*Proof.* We use Lemma [8] which states that with given sign the derivatives of the mapping Eq. (5) and Eq. (6) with respect to $\nu$ and $\mu$ are either positive or have the sign of $\omega$. Therefore with given sign of $\omega$ the mappings are strict monotonic and the their maxima and minima are found at the borders. The minimum of $\tilde{\mu}$ is obtained at $\mu\omega = -0.01$ and its maximum at $\mu\omega = 0.01\,\sigma\tau$, and it easily follows with

$$\tilde{\mu}(-0.1, 0.1, 0.8, 0.8, \lambda_{01}, \alpha_{01}) \leqslant \tilde{\mu} \leqslant \tilde{\mu}(0.1, 0.1, 1.5, 1.25, \lambda_{01}, \alpha_{01})$$
$$-0.041160 < \tilde{\mu}(-0.1, 0.1, 0.8, 0.8, \lambda_{01}, \alpha_{01}) \leqslant \tilde{\mu} \leqslant \tilde{\mu}(0.1, 0.1, 1.5, 1.25, \lambda_{01}, \alpha_{01}) < 0.087653,$$

Similarly, the maximum and minimum of $\tilde{\xi}$ is obtained at these values

$$\tilde{\xi}(-0.1, 0.1, 0.8, 0.8, \lambda_{01}, \alpha_{01}) \leqslant \tilde{\xi} \leqslant \tilde{\xi}(0.1, 0.1, 1.5, 1.25, \lambda_{01}, \alpha_{01})$$
$$0.703257 < \tilde{\xi}(-0.1, 0.1, 0.8, 0.8, \lambda_{01}, \alpha_{01}) \leqslant \tilde{\xi} \leqslant \tilde{\xi}(0.1, 0.1, 1.5, 1.25, \lambda_{01}, \alpha_{01}) < 1.643705.$$

Hence,

$$0.703257 - \tilde{\mu}^2 < \tilde{\xi} - \tilde{\mu}^2 < 1.643705 - \tilde{\mu}^2.$$
$$0.703257 - 0.007683 < \tilde{\nu} < 1.643705 - 0.007682.$$
$$0.695574 < \tilde{\nu} < 1.636023.$$

$\square$

**Upper Bounds on the Largest Singular Value of the Jacobian.**

**Lemma 10** (Upper Bounds on Absolute Derivatives of Largest Singular Value). *We set* $\alpha = \alpha_{01}$ *and* $\lambda = \lambda_{01}$ *and restrict the range of the variables to* $\mu \in [\mu_{\min}, \mu_{\max}] = [-0.1, 0.1]$, $\omega \in [\omega_{\min}, \omega_{\max}] = [-0.1, 0.1]$, $\nu \in [\nu_{\min}, \nu_{\max}] = [0.8, 1.5]$, *and* $\tau \in [\tau_{\min}, \tau_{\max}] = [0.8, 1.25]$.

*The absolute values of derivatives of the largest singular value* $S(\mu, \omega, \nu, \tau, \lambda, \alpha)$ *given in Eq.* (55) *with respect to* $(\mu, \omega, \nu, \tau)$ *are bounded as follows:*

$$\left| \frac{\partial S}{\partial \mu} \right| < 0.32112 \,, \tag{60}$$

$$\left| \frac{\partial S}{\partial \omega} \right| < 2.63690 \,, \tag{61}$$

$$\left| \frac{\partial S}{\partial \nu} \right| < 2.28242 \,, \tag{62}$$

$$\left| \frac{\partial S}{\partial \tau} \right| < 2.98610 \,. \tag{63}$$

*Proof.* The Jacobian of our mapping Eq. (5) and Eq. (6) is defined as

$$\boldsymbol{H} = \left( \begin{array}{cc} \mathcal{H}_{11} & \mathcal{H}_{12} \\ \mathcal{H}_{21} & \mathcal{H}_{22} \end{array} \right) = \left( \begin{array}{cc} \mathcal{J}_{11} & \mathcal{J}_{12} \\ \mathcal{J}_{21} - 2\tilde{\mu}\mathcal{J}_{11} & \mathcal{J}_{22} - 2\tilde{\mu}\mathcal{J}_{12} \end{array} \right) \tag{64}$$

and has the largest singular value

$$S(\mu, \omega, \nu, \tau, \lambda, \alpha) = \frac{1}{2} \left( \sqrt{(\mathcal{H}_{11} - \mathcal{H}_{22})^2 + (\mathcal{H}_{12} + \mathcal{H}_{21})^2} + \sqrt{(\mathcal{H}_{11} + \mathcal{H}_{22})^2 + (\mathcal{H}_{12} - \mathcal{H}_{21})^2} \right), \tag{65}$$

according to the formula of Blinn (1996).

We obtain

$$\left| \frac{\partial S}{\partial \mathcal{H}_{11}} \right| = \left| \frac{1}{2} \left( \frac{\mathcal{H}_{11} - \mathcal{H}_{22}}{\sqrt{(\mathcal{H}_{11} - \mathcal{H}_{22})^2 + (\mathcal{H}_{12} + \mathcal{H}_{21})^2}} + \frac{\mathcal{H}_{11} + \mathcal{H}_{22}}{\sqrt{(\mathcal{H}_{11} + \mathcal{H}_{22})^2 + (\mathcal{H}_{21} - \mathcal{H}_{12})^2}} \right) \right| < \tag{66}$$

$$\frac{1}{2} \left( \left| \frac{1}{\sqrt{\frac{(\mathcal{H}_{12} + \mathcal{H}_{21})^2}{(\mathcal{H}_{11} - \mathcal{H}_{22})^2} + 1}} \right| + \left| \frac{1}{\sqrt{\frac{(\mathcal{H}_{21} - \mathcal{H}_{12})^2}{(\mathcal{H}_{11} + \mathcal{H}_{22})^2} + 1}} \right| \right) < \frac{1 + 1}{2} = 1$$

and analogously

$$\left| \frac{\partial S}{\partial \mathcal{H}_{12}} \right| = \left| \frac{1}{2} \left( \frac{\mathcal{H}_{12} + \mathcal{H}_{21}}{\sqrt{(\mathcal{H}_{11} - \mathcal{H}_{22})^2 + (\mathcal{H}_{12} + \mathcal{H}_{21})^2}} - \frac{\mathcal{H}_{21} - \mathcal{H}_{12}}{\sqrt{(\mathcal{H}_{11} + \mathcal{H}_{22})^2 + (\mathcal{H}_{21} - \mathcal{H}_{12})^2}} \right) \right| < 1 \tag{67}$$

and

$$\left| \frac{\partial S}{\partial \mathcal{H}_{21}} \right| = \left| \frac{1}{2} \left( \frac{\mathcal{H}_{21} - \mathcal{H}_{12}}{\sqrt{(\mathcal{H}_{11} + \mathcal{H}_{22})^2 + (\mathcal{H}_{21} - \mathcal{H}_{12})^2}} + \frac{\mathcal{H}_{12} + \mathcal{H}_{21}}{\sqrt{(\mathcal{H}_{11} - \mathcal{H}_{22})^2 + (\mathcal{H}_{12} + \mathcal{H}_{21})^2}} \right) \right| < 1 \tag{68}$$

and

$$\left| \frac{\partial S}{\partial \mathcal{H}_{22}} \right| = \left| \frac{1}{2} \left( \frac{\mathcal{H}_{11} + \mathcal{H}_{22}}{\sqrt{(\mathcal{H}_{11} + \mathcal{H}_{22})^2 + (\mathcal{H}_{21} - \mathcal{H}_{12})^2}} - \frac{\mathcal{H}_{11} - \mathcal{H}_{22}}{\sqrt{(\mathcal{H}_{11} - \mathcal{H}_{22})^2 + (\mathcal{H}_{12} + \mathcal{H}_{21})^2}} \right) \right| < 1 \, . \tag{69}$$

We have

$$\frac{\partial S}{\partial \mu} = \frac{\partial S}{\partial \mathcal{H}_{11}} \frac{\partial \mathcal{H}_{11}}{\partial \mu} + \frac{\partial S}{\partial \mathcal{H}_{12}} \frac{\partial \mathcal{H}_{12}}{\partial \mu} + \frac{\partial S}{\partial \mathcal{H}_{21}} \frac{\partial \mathcal{H}_{21}}{\partial \mu} + \frac{\partial S}{\partial \mathcal{H}_{22}} \frac{\partial \mathcal{H}_{22}}{\partial \mu} \tag{70}$$

$$\frac{\partial S}{\partial \omega} = \frac{\partial S}{\partial \mathcal{H}_{11}} \frac{\partial \mathcal{H}_{11}}{\partial \omega} + \frac{\partial S}{\partial \mathcal{H}_{12}} \frac{\partial \mathcal{H}_{12}}{\partial \omega} + \frac{\partial S}{\partial \mathcal{H}_{21}} \frac{\partial \mathcal{H}_{21}}{\partial \omega} + \frac{\partial S}{\partial \mathcal{H}_{22}} \frac{\partial \mathcal{H}_{22}}{\partial \omega} \tag{71}$$

$$\frac{\partial S}{\partial \nu} = \frac{\partial S}{\partial \mathcal{H}_{11}} \frac{\partial \mathcal{H}_{11}}{\partial \nu} + \frac{\partial S}{\partial \mathcal{H}_{12}} \frac{\partial \mathcal{H}_{12}}{\partial \nu} + \frac{\partial S}{\partial \mathcal{H}_{21}} \frac{\partial \mathcal{H}_{21}}{\partial \nu} + \frac{\partial S}{\partial \mathcal{H}_{22}} \frac{\partial \mathcal{H}_{22}}{\partial \nu} \tag{72}$$

$$\frac{\partial S}{\partial \tau} = \frac{\partial S}{\partial \mathcal{H}_{11}} \frac{\partial \mathcal{H}_{11}}{\partial \tau} + \frac{\partial S}{\partial \mathcal{H}_{12}} \frac{\partial \mathcal{H}_{12}}{\partial \tau} + \frac{\partial S}{\partial \mathcal{H}_{21}} \frac{\partial \mathcal{H}_{21}}{\partial \tau} + \frac{\partial S}{\partial \mathcal{H}_{22}} \frac{\partial \mathcal{H}_{22}}{\partial \tau} \tag{73}$$

$$\tag{74}$$

from which follows using the bounds from Lemma 5:

Derivative of the singular value w.r.t. $\mu$:

$$\left| \frac{\partial S}{\partial \mu} \right| \leqslant \tag{75}$$

$$\left| \frac{\partial S}{\partial \mathcal{H}_{11}} \right| \left| \frac{\partial \mathcal{H}_{11}}{\partial \mu} \right| + \left| \frac{\partial S}{\partial \mathcal{H}_{12}} \right| \left| \frac{\partial \mathcal{H}_{12}}{\partial \mu} \right| + \left| \frac{\partial S}{\partial \mathcal{H}_{21}} \right| \left| \frac{\partial \mathcal{H}_{21}}{\partial \mu} \right| + \left| \frac{\partial S}{\partial \mathcal{H}_{22}} \right| \left| \frac{\partial \mathcal{H}_{22}}{\partial \mu} \right| \leqslant$$

$$\left| \frac{\partial \mathcal{H}_{11}}{\partial \mu} \right| + \left| \frac{\partial \mathcal{H}_{12}}{\partial \mu} \right| + \left| \frac{\partial \mathcal{H}_{21}}{\partial \mu} \right| + \left| \frac{\partial \mathcal{H}_{22}}{\partial \mu} \right| \leqslant$$

$$\left| \frac{\partial \mathcal{J}_{11}}{\partial \mu} \right| + \left| \frac{\partial \mathcal{J}_{12}}{\partial \mu} \right| + \left| \frac{\partial \mathcal{J}_{21} - 2\tilde{\mu}\mathcal{J}_{11}}{\partial \mu} \right| + \left| \frac{\partial \mathcal{J}_{22} - 2\tilde{\mu}\mathcal{J}_{12}}{\partial \mu} \right| \leqslant$$

$$\left| \frac{\partial \mathcal{J}_{11}}{\partial \mu} \right| + \left| \frac{\partial \mathcal{J}_{12}}{\partial \mu} \right| + \left| \frac{\partial \mathcal{J}_{21}}{\partial \mu} \right| + \left| \frac{\partial \mathcal{J}_{22}}{\partial \mu} \right| + 2 \left| \frac{\partial \mathcal{J}_{11}}{\partial \mu} \right| |\tilde{\mu}| + 2 |\mathcal{J}_{11}|^2 + 2 \left| \frac{\partial \mathcal{J}_{12}}{\partial \mu} \right| |\tilde{\mu}| + 2 |\mathcal{J}_{12}| |\mathcal{J}_{11}| \leqslant$$

$$0.0031049101995398316 + 0.03124291123546816 + 0.02220441024325437 + 0.14983446469110305 +$$

$$2 \cdot 0.104497 \cdot 0.087653 + 2 \cdot 0.104497^2 +$$

$$2 \cdot 0.194035 \cdot 0.087653 + 2 \cdot 0.104497 \cdot 0.194035 < 0.32112,$$

where we used the results from the lemmata 5, 6, 7, and 9.

Derivative of the singular value w.r.t. $\omega$:

$$\left|\frac{\partial S}{\partial \omega}\right| \leqslant \tag{76}$$

$$\left|\frac{\partial S}{\partial \mathcal{H}_{11}}\right|\left|\frac{\partial \mathcal{H}_{11}}{\partial \omega}\right| + \left|\frac{\partial S}{\partial \mathcal{H}_{12}}\right|\left|\frac{\partial \mathcal{H}_{12}}{\partial \omega}\right| + \left|\frac{\partial S}{\partial \mathcal{H}_{21}}\right|\left|\frac{\partial \mathcal{H}_{21}}{\partial \omega}\right| + \left|\frac{\partial S}{\partial \mathcal{H}_{22}}\right|\left|\frac{\partial \mathcal{H}_{22}}{\partial \omega}\right| \leqslant$$

$$\left|\frac{\partial \mathcal{H}_{11}}{\partial \omega}\right| + \left|\frac{\partial \mathcal{H}_{12}}{\partial \omega}\right| + \left|\frac{\partial \mathcal{H}_{21}}{\partial \omega}\right| + \left|\frac{\partial \mathcal{H}_{22}}{\partial \omega}\right| \leqslant$$

$$\left|\frac{\partial \mathcal{J}_{11}}{\partial \omega}\right| + \left|\frac{\partial \mathcal{J}_{12}}{\partial \omega}\right| + \left|\frac{\partial \mathcal{J}_{21} - 2\tilde{\mu}\mathcal{J}_{11}}{\partial \omega}\right| + \left|\frac{\partial \mathcal{J}_{22} - 2\tilde{\mu}\mathcal{J}_{12}}{\partial \omega}\right| \leqslant$$

$$\left|\frac{\partial \mathcal{J}_{11}}{\partial \omega}\right| + \left|\frac{\partial \mathcal{J}_{12}}{\partial \omega}\right| + \left|\frac{\partial \mathcal{J}_{21}}{\partial \omega}\right| + \left|\frac{\partial \mathcal{J}_{22}}{\partial \omega}\right| + 2\left|\frac{\partial \mathcal{J}_{11}}{\partial \omega}\right||\tilde{\mu}| + 2\left|\mathcal{J}_{11}\right|\left|\frac{\partial \tilde{\mu}}{\partial \omega}\right| +$$

$$2\left|\frac{\partial \mathcal{J}_{12}}{\partial \omega}\right||\tilde{\mu}| + 2\left|\mathcal{J}_{12}\right|\left|\frac{\partial \tilde{\mu}}{\partial \omega}\right| \leqslant \tag{77}$$

$$2.38392 + 2 \cdot 1.055872374194189 \cdot 0.087653 + 2 \cdot 0.104497^2 + 2 \cdot 0.031242911235461816 \cdot 0.087653$$
$$+ 2 \cdot 0.194035 \cdot 0.104497 < 2.63690 \,,$$

where we used the results from the lemmata 5, 6, 7, and 9 and that $\tilde{\mu}$ is symmetric for $\mu, \omega$.

Derivative of the singular value w.r.t. $\nu$:

$$\left|\frac{\partial S}{\partial \nu}\right| \leqslant \tag{78}$$

$$\left|\frac{\partial S}{\partial \mathcal{H}_{11}}\right|\left|\frac{\partial \mathcal{H}_{11}}{\partial \nu}\right| + \left|\frac{\partial S}{\partial \mathcal{H}_{12}}\right|\left|\frac{\partial \mathcal{H}_{12}}{\partial \nu}\right| + \left|\frac{\partial S}{\partial \mathcal{H}_{21}}\right|\left|\frac{\partial \mathcal{H}_{21}}{\partial \nu}\right| + \left|\frac{\partial S}{\partial \mathcal{H}_{22}}\right|\left|\frac{\partial \mathcal{H}_{22}}{\partial \nu}\right| \leqslant$$

$$\left|\frac{\partial \mathcal{H}_{11}}{\partial \nu}\right| + \left|\frac{\partial \mathcal{H}_{12}}{\partial \nu}\right| + \left|\frac{\partial \mathcal{H}_{21}}{\partial \nu}\right| + \left|\frac{\partial \mathcal{H}_{22}}{\partial \nu}\right| \leqslant$$

$$\left|\frac{\partial \mathcal{J}_{11}}{\partial \nu}\right| + \left|\frac{\partial \mathcal{J}_{12}}{\partial \nu}\right| + \left|\frac{\partial \mathcal{J}_{21} - 2\tilde{\mu}\mathcal{J}_{11}}{\partial \nu}\right| + \left|\frac{\partial \mathcal{J}_{22} - 2\tilde{\mu}\mathcal{J}_{12}}{\partial \nu}\right| \leqslant$$

$$\left|\frac{\partial \mathcal{J}_{11}}{\partial \nu}\right| + \left|\frac{\partial \mathcal{J}_{12}}{\partial \nu}\right| + \left|\frac{\partial \mathcal{J}_{21}}{\partial \nu}\right| + \left|\frac{\partial \mathcal{J}_{22}}{\partial \nu}\right| + 2\left|\frac{\partial \mathcal{J}_{11}}{\partial \nu}\right||\tilde{\mu}| + 2\left|\mathcal{J}_{11}\right|\left|\mathcal{J}_{12}\right| + 2\left|\frac{\partial \mathcal{J}_{12}}{\partial \nu}\right||\tilde{\mu}| + 2\left|\mathcal{J}_{12}\right|^2 \leqslant$$

$$2.19916 + 2 \cdot 0.031242911235461816 \cdot 0.087653 + 2 \cdot 0.104497 \cdot 0.194035+$$
$$2 \cdot 0.21232788238624354 \cdot 0.087653 + 2 \cdot 0.194035^2 < 2.28242 \,,$$

where we used the results from the lemmata 5, 6, 7, and 9.

Derivative of the singular value w.r.t. $\tau$:

$$\left|\frac{\partial S}{\partial \tau}\right| \leqslant \tag{79}$$

$$\left|\frac{\partial S}{\partial \mathcal{H}_{11}}\right|\left|\frac{\partial \mathcal{H}_{11}}{\partial \tau}\right| + \left|\frac{\partial S}{\partial \mathcal{H}_{12}}\right|\left|\frac{\partial \mathcal{H}_{12}}{\partial \tau}\right| + \left|\frac{\partial S}{\partial \mathcal{H}_{21}}\right|\left|\frac{\partial \mathcal{H}_{21}}{\partial \tau}\right| + \left|\frac{\partial S}{\partial \mathcal{H}_{22}}\right|\left|\frac{\partial \mathcal{H}_{22}}{\partial \tau}\right| \leqslant$$

$$\left|\frac{\partial \mathcal{H}_{11}}{\partial \tau}\right| + \left|\frac{\partial \mathcal{H}_{12}}{\partial \tau}\right| + \left|\frac{\partial \mathcal{H}_{21}}{\partial \tau}\right| + \left|\frac{\partial \mathcal{H}_{22}}{\partial \tau}\right| \leqslant$$

$$\left|\frac{\partial \mathcal{J}_{11}}{\partial \tau}\right| + \left|\frac{\partial \mathcal{J}_{12}}{\partial \tau}\right| + \left|\frac{\partial \mathcal{J}_{21} - 2\tilde{\mu}\mathcal{J}_{11}}{\partial \tau}\right| + \left|\frac{\partial \mathcal{J}_{22} - 2\tilde{\mu}\mathcal{J}_{12}}{\partial \tau}\right| \leqslant$$

$$\left|\frac{\partial \mathcal{J}_{11}}{\partial \tau}\right| + \left|\frac{\partial \mathcal{J}_{12}}{\partial \tau}\right| + \left|\frac{\partial \mathcal{J}_{21}}{\partial \tau}\right| + \left|\frac{\partial \mathcal{J}_{22}}{\partial \tau}\right| + 2\left|\frac{\partial \mathcal{J}_{11}}{\partial \tau}\right||\tilde{\mu}| + 2\left|\mathcal{J}_{11}\right|\left|\frac{\partial \tilde{\mu}}{\partial \tau}\right| +$$

$$2\left|\frac{\partial \mathcal{J}_{12}}{\partial \tau}\right||\tilde{\mu}| + 2\left|\mathcal{J}_{12}\right|\left|\frac{\partial \tilde{\mu}}{\partial \tau}\right| \leqslant \tag{80}$$

$$2.82643 + 2 \cdot 0.03749149348255419 \cdot 0.087653 + 2 \cdot 0.104497 \cdot 0.194035+$$

$$2 \cdot 0.2124377655377270 \cdot 0.087653 + 2 \cdot 0.194035^2 \; < \; 2.98610 \,,$$

where we used the results from the lemmata 5, 6, 7, and 9 and that $\tilde{\mu}$ is symmetric for $\nu, \tau$.

$\square$

**Lemma 11** (Mean Value Theorem Bound on Deviation from Largest Singular Value). *We set*
$\alpha = \alpha_{01}$ *and* $\lambda = \lambda_{01}$ *and restrict the range of the variables to* $\mu \in [\mu_{\min}, \mu_{\max}] = [-0.1, 0.1]$,
$\omega \in [\omega_{\min}, \omega_{\max}] = [-0.1, 0.1]$, $\nu \in [\nu_{\min}, \nu_{\max}] = [0.8, 1.5]$, *and* $\tau \in [\tau_{\min}, \tau_{\max}] = [0.8, 1.25]$.

*The distance of the singular value at* $S(\mu, \omega, \nu, \tau, \lambda_{01}, \alpha_{01})$ *and that at* $S(\mu + \Delta\mu, \omega + \Delta\omega, \nu + \Delta\nu, \tau + \Delta\tau, \lambda_{01}, \alpha_{01})$ *is bounded as follows:*

$$|S(\mu + \Delta\mu, \omega + \Delta\omega, \nu + \Delta\nu, \tau + \Delta\tau, \lambda_{01}, \alpha_{01}) \; - \; S(\mu, \omega, \nu, \tau, \lambda_{01}, \alpha_{01})| \; < \tag{81}$$
$$0.32112\,|\Delta\mu| + 2.63690\,|\Delta\omega| + 2.28242\,|\Delta\nu| + 2.98610\,|\Delta\tau| \,.$$

*Proof.* The mean value theorem states that a $t \in [0, 1]$ exists for which

$$S(\mu + \Delta\mu, \omega + \Delta\omega, \nu + \Delta\nu, \tau + \Delta\tau, \lambda_{01}, \alpha_{01}) \; - \; S(\mu, \omega, \nu, \tau, \lambda_{01}, \alpha_{01}) \; = \tag{82}$$

$$\frac{\partial S}{\partial \mu}(\mu + t\Delta\mu, \omega + t\Delta\omega, \nu + t\Delta\nu, \tau + t\Delta\tau, \lambda_{01}, \alpha_{01})\,\Delta\mu +$$

$$\frac{\partial S}{\partial \omega}(\mu + t\Delta\mu, \omega + t\Delta\omega, \nu + t\Delta\nu, \tau + t\Delta\tau, \lambda_{01}, \alpha_{01})\,\Delta\omega +$$

$$\frac{\partial S}{\partial \nu}(\mu + t\Delta\mu, \omega + t\Delta\omega, \nu + t\Delta\nu, \tau + t\Delta\tau, \lambda_{01}, \alpha_{01})\,\Delta\nu +$$

$$\frac{\partial S}{\partial \tau}(\mu + t\Delta\mu, \omega + t\Delta\omega, \nu + t\Delta\nu, \tau + t\Delta\tau, \lambda_{01}, \alpha_{01})\,\Delta\tau$$

from which immediately follows that

$$|S(\mu + \Delta\mu, \omega + \Delta\omega, \nu + \Delta\nu, \tau + \Delta\tau, \lambda_{01}, \alpha_{01}) \; - \; S(\mu, \omega, \nu, \tau, \lambda_{01}, \alpha_{01})| \; \leqslant \tag{83}$$

$$\left|\frac{\partial S}{\partial \mu}(\mu + t\Delta\mu, \omega + t\Delta\omega, \nu + t\Delta\nu, \tau + t\Delta\tau, \lambda_{01}, \alpha_{01})\right|\,|\Delta\mu| +$$

$$\left|\frac{\partial S}{\partial \omega}(\mu + t\Delta\mu, \omega + t\Delta\omega, \nu + t\Delta\nu, \tau + t\Delta\tau, \lambda_{01}, \alpha_{01})\right|\,|\Delta\omega| +$$

$$\left|\frac{\partial S}{\partial \nu}(\mu + t\Delta\mu, \omega + t\Delta\omega, \nu + t\Delta\nu, \tau + t\Delta\tau, \lambda_{01}, \alpha_{01})\right|\,|\Delta\nu| +$$

$$\left| \frac{\partial S}{\partial \tau}(\mu + t\Delta\mu, \omega + t\Delta\omega, \nu + t\Delta\nu, \tau + t\Delta\tau, \lambda_{01}, \alpha_{01}) \right| \; |\Delta\tau| \; .$$

We now apply Lemma 10 which gives bounds on the derivatives, which immediately gives the statement of the lemma. □

**Lemma 12** (Largest Singular Value Smaller Than One). *We set $\alpha = \alpha_{01}$ and $\lambda = \lambda_{01}$ and restrict the range of the variables to $\mu \in [-0.1, 0.1]$, $\omega \in [-0.1, 0.1]$, $\nu \in [0.8, 1.5]$, and $\tau \in [0.8, 1.25]$.*

*The the largest singular value of the Jacobian is smaller than 1:*

$$S(\mu, \omega, \nu, \tau, \lambda_{01}, \alpha_{01}) \; < \; 1 \; . \tag{84}$$

*Therefore the mapping Eq. (5) and Eq. (6) is a contraction mapping.*

*Proof.* We set $\Delta\mu = 0.0068097371$, $\Delta\omega = 0.0008292885$, $\Delta\nu = 0.0009580840$, and $\Delta\tau = 0.0007323095$.

According to Lemma 11 we have

$$|S(\mu + \Delta\mu, \omega + \Delta\omega, \nu + \Delta\nu, \tau + \Delta\tau, \lambda_{01}, \alpha_{01}) \; - \; S(\mu, \omega, \nu, \tau, \lambda_{01}, \alpha_{01})| \; < \tag{85}$$
$$0.32112 \cdot 0.0068097371 + 2.63690 \cdot 0.0008292885 +$$
$$2.28242 \cdot 0.0009580840 + 2.98610 \cdot 0.0007323095 \; < \; 0.008747 \; .$$

For a grid with grid length $\Delta\mu = 0.0068097371$, $\Delta\omega = 0.0008292885$, $\Delta\nu = 0.0009580840$, and $\Delta\tau = 0.0007323095$, we evaluated the function Eq. (55) for the largest singular value in the domain $\mu \in [-0.1, 0.1]$, $\omega \in [-0.1, 0.1]$, $\nu \in [0.8, 1.5]$, and $\tau \in [0.8, 1.25]$. We did this using a computer. According to Subsection S3.4.5 the precision if regarding error propagation and precision of the implemented functions is larger than $10^{-13}$. We performed the evaluation on different operating systems and different hardware architectures including CPUs and GPUs. In all cases the function Eq. (55) for the largest singular value of the Jacobian is bounded by $0.9912524171058772$.

We obtain from Eq. (85):

$$S(\mu + \Delta\mu, \omega + \Delta\omega, \nu + \Delta\nu, \tau + \Delta\tau, \lambda_{01}, \alpha_{01}) \; \leqslant \; 0.9912524171058772 \; + \; 0.008747 \; < \; 1 \; . \tag{86}$$

□

### S3.4.2   Lemmata for proofing Theorem 1 (part 2): Mapping within domain

We further have to investigate whether the the mapping Eq. (5) and Eq. (6) maps into a predefined domains.

**Lemma 13** (Mapping into the domain). *The mapping Eq. (5) and Eq. (6) map for $\alpha = \alpha_{01}$ and $\lambda = \lambda_{01}$ into the domain $\mu \in [-0.03106, 0.06773]$ and $\nu \in [0.80009, 1.48617]$ with $\omega \in [-0.1, 0.1]$ and $\tau \in [0.95, 1.1]$.*

*Proof.* We use Lemma 8 which states that with given sign the derivatives of the mapping Eq. (5) and Eq. (6) with respect to $\alpha = \alpha_{01}$ and $\lambda = \lambda_{01}$ are either positive or have the sign of $\omega$. Therefore with given sign of $\omega$ the mappings are strict monotonic and the their maxima and minima are found at the borders. The minimum of $\tilde{\mu}$ is obtained at $\mu\omega = -0.01$ and its maximum at $\mu\omega = 0.01 \ \sigma\tau$, and it easily follows with

$$\tilde{\mu}(-0.1, 0.1, 0.8, 0.95, \lambda_{01}, \alpha_{01}) \leqslant \tilde{\mu} \leqslant \tilde{\mu}(0.1, 0.1, 1.5, 1.1, \lambda_{01}, \alpha_{01})$$

$$-0.03106 < \tilde{\mu}(-0.1, 0.1, 0.8, 0.95, \lambda_{01}, \alpha_{01}) \leqslant \tilde{\mu} \leqslant \tilde{\mu}((0.1, 0.1, 1.5, 1.1, \lambda_{01}, \alpha_{01}) < 0.06773,$$

that $\tilde{\mu} \in [-0.1, 0.1]$.

Similarly, the maximum and minimum of $\tilde{\xi}($ is obtained at these values

$$\tilde{\xi}(-0.1, 0.1, 0.8, 0.95, \lambda_{01}, \alpha_{01}) \leqslant \tilde{\xi} \leqslant \tilde{\xi}(0.1, 0.1, 1.5, 1.1, \lambda_{01}, \alpha_{01})$$

$$0.80467 < \tilde{\xi}(-0.1, 0.1, 0.8, 0.95, \lambda_{01}, \alpha_{01}) \leqslant \tilde{\xi} \leqslant \tilde{\xi}(0.1, 0.1, 1.5, 1.1, \lambda_{01}, \alpha_{01}) < 1.48617.$$

Since $|\tilde{\xi} - \tilde{\nu}| = |\tilde{\mu}^2| < 0.004597$, we can conclude that $0.80009 < \tilde{\nu} < 1.48617$ and the value remains in $[0.8, 1.5]$. $\qquad\square$

**Corollary 14.** *The image $g(\Omega')$ of the mapping $g : (\mu, \nu) \mapsto (\tilde{\mu}, \tilde{\nu})$ (Eq. (2)) and the domain $\Omega' = \{(\mu, \nu)| -0.1 \leqslant \mu \leqslant 0.1, 0.8 \leqslant \mu \leqslant 1.5\}$ is a subset of $\Omega'$:*

$$g(\Omega') \subseteq \Omega', \tag{87}$$

*for all $\omega \in [-0.1, 0.1]$ and $\tau \in [0.95, 1.1]$.*

*Proof.* Directly follows from Lemma 13. $\qquad\square$

### S3.4.3 Lemmata for proofing Theorem 2: The variance is contracting

**Main Sub-Function.** We consider the main sub-function of the derivate of second moment, $J22$ (Eq. (48)):

$$\frac{\partial}{\partial \nu}\tilde{\xi} = \frac{1}{2}\lambda^2\tau\left(-\alpha^2 e^{\mu\omega+\frac{\nu\tau}{2}}\operatorname{erfc}\left(\frac{\mu\omega+\nu\tau}{\sqrt{2}\sqrt{\nu\tau}}\right) + 2\alpha^2 e^{2\mu\omega+2\nu\tau}\operatorname{erfc}\left(\frac{\mu\omega+2\nu\tau}{\sqrt{2}\sqrt{\nu\tau}}\right) - \operatorname{erfc}\left(\frac{\mu\omega}{\sqrt{2}\sqrt{\nu\tau}}\right) + 2\right)$$
$$\tag{88}$$

that depends on $\mu\omega$ and $\nu\tau$, therefore we set $x = \nu\tau$ and $y = \mu\omega$. Algebraic reformulations provide the formula in the following form:

$$\frac{\partial}{\partial \nu}\tilde{\xi} = \tag{89}$$
$$\frac{1}{2}\lambda^2\tau\alpha^2\left(-e^{-\frac{y^2}{2x}}\right)\left(e^{\frac{(x+y)^2}{2x}}\operatorname{erfc}\left(\frac{y+x}{\sqrt{2}\sqrt{x}}\right) - 2e^{\frac{(2x+y)^2}{2x}}\operatorname{erfc}\left(\frac{y+2x}{\sqrt{2}\sqrt{x}}\right) + \frac{1}{\alpha^2}e^{\frac{y^2}{2x}}\operatorname{erfc}\left(\frac{y}{\sqrt{2}\sqrt{x}}\right) - 2\right)$$

For $\lambda = \lambda_{01}$ and $\alpha = \alpha_{01}$, we consider the domain $-1 \leqslant \mu \leqslant 1$, $-0.1 \leqslant \omega \leqslant 0.1$, $1.5 \leqslant \nu \leqslant 16$, and, $0.8 \leqslant \tau \leqslant 1.25$.

For $x$ and $y$ we obtain: $0.8 \cdot 1.5 = 1.2 \leqslant x \leqslant 20 = 1.25 \cdot 16$ and $0.1 \cdot (-1) = -0.1 \leqslant y \leqslant 0.1 = 0.1 \cdot 1$. In the following we assume to remain within this domain.

Figure S2: **Left panel:** Graphs of the main subfunction $f(x,y) = e^{\frac{(x+y)^2}{2x}} \operatorname{erfc}\left(\frac{x+y}{\sqrt{2}\sqrt{x}}\right) - 2e^{\frac{(2x+y)^2}{2x}} \operatorname{erfc}\left(\frac{2x+y}{\sqrt{2}\sqrt{x}}\right)$ treated in Lemma 15. The function is negative and monotonically increasing with $x$ independent of $y$. **Right panel:** Graphs of the main subfunction at minimal $x = 1.2$. The graph shows that the function $f(1.2, y)$ is strictly monotonically decreasing in $y$.

**Lemma 15** (Main subfunction). *For $1.2 \leqslant x \leqslant 20$ and $-0.1 \leqslant y \leqslant 0.1$,*

*the function*

$$e^{\frac{(x+y)^2}{2x}} \operatorname{erfc}\left(\frac{x+y}{\sqrt{2}\sqrt{x}}\right) - 2e^{\frac{(2x+y)^2}{2x}} \operatorname{erfc}\left(\frac{2x+y}{\sqrt{2}\sqrt{x}}\right) \tag{90}$$

*is smaller than zero, is strictly monotonically increasing in $x$, and strictly monotonically decreasing in $y$ for the minimal $x = 12/10 = 1.2$.*

*Proof.* See proof 44.                                                                                      □

The graph of the subfunction in the specified domain is displayed in Figure S2.

**Theorem 16** (Contraction $\nu$-mapping). *The mapping of the variance $\tilde{\nu}(\mu, \omega, \nu, \tau, \lambda, \alpha)$ given in Eq. (6) is contracting for $\lambda = \lambda_{01}$, $\alpha = \alpha_{01}$ and the domain $\Omega^+$: $-0.1 \leqslant \mu \leqslant 0.1$, $-0.1 \leqslant \omega \leqslant 0.1$, $1.5 \leqslant \nu \leqslant 16$, and $0.8 \leqslant \tau \leqslant 1.25$, that is,*

$$\left| \frac{\partial}{\partial \nu} \tilde{\nu}(\mu, \omega, \nu, \tau, \lambda_{01}, \alpha_{01}) \right| < 1 . \tag{91}$$

*Proof.* In this domain $\Omega^+$ we have the following three properties (see further below): $\frac{\partial}{\partial \nu} \tilde{\xi} < 1$, $\tilde{\mu} > 0$, and $\frac{\partial}{\partial \nu} \tilde{\mu} > 0$. Therefore, we have

$$\left| \frac{\partial}{\partial \nu} \tilde{\nu} \right| = \left| \frac{\partial}{\partial \nu} \tilde{\xi} - 2\tilde{\mu} \frac{\partial}{\partial \nu} \tilde{\mu} \right| < \left| \frac{\partial}{\partial \nu} \tilde{\xi} \right| < 1 \tag{92}$$

■ We first proof that $\frac{\partial}{\partial \nu}\tilde{\xi} < 1$ in an even larger domain that fully contains $\Omega^+$. According to Eq. (48), the derivative of the mapping Eq. (6) with respect to the variance $\nu$ is

$$\frac{\partial}{\partial \nu}\tilde{\xi}(\mu, \omega, \nu, \tau, \lambda_{01}, \alpha_{01}) = \tag{93}$$

$$\frac{1}{2}\lambda^2\tau\left(\alpha^2\left(-e^{\mu\omega+\frac{\nu\tau}{2}}\right)\text{erfc}\left(\frac{\mu\omega+\nu\tau}{\sqrt{2}\sqrt{\nu\tau}}\right)+\right.$$

$$\left.2\alpha^2 e^{2\mu\omega+2\nu\tau}\,\text{erfc}\left(\frac{\mu\omega+2\nu\tau}{\sqrt{2}\sqrt{\nu\tau}}\right)-\text{erfc}\left(\frac{\mu\omega}{\sqrt{2}\sqrt{\nu\tau}}\right)+2\right)\ .$$

For $\lambda = \lambda_{01}$, $\alpha = \alpha_{01}$, $-1 \leqslant \mu \leqslant 1$, $-0.1 \leqslant \omega \leqslant 0.1$ $1.5 \leqslant \nu \leqslant 16$, and $0.8 \leqslant \tau \leqslant 1.25$, we first show that the derivative is positive and then upper bound it.

According to Lemma 15, the expression

$$e^{\frac{(\mu\omega+\nu\tau)^2}{2\nu\tau}}\,\text{erfc}\left(\frac{\mu\omega+\nu\tau}{\sqrt{2}\sqrt{\nu\tau}}\right)-2e^{\frac{(\mu\omega+2\nu\tau)^2}{2\nu\tau}}\,\text{erfc}\left(\frac{\mu\omega+2\nu\tau}{\sqrt{2}\sqrt{\nu\tau}}\right) \tag{94}$$

is negative. This expression multiplied by positive factors is subtracted in the derivative Eq. (93), therefore, the whole term is positive. The remaining term

$$2-\text{erfc}\left(\frac{\mu\omega}{\sqrt{2}\sqrt{\nu\tau}}\right) \tag{95}$$

of the derivative Eq. (93) is also positive according to Lemma 21. All factors outside the brackets in Eq. (93) are positive. Hence, the derivative Eq. (93) is positive.

The upper bound of the derivative is:

$$\frac{1}{2}\lambda_{01}^2\tau\left(\alpha_{01}^2\left(-e^{\mu\omega+\frac{\nu\tau}{2}}\right)\text{erfc}\left(\frac{\mu\omega+\nu\tau}{\sqrt{2}\sqrt{\nu\tau}}\right)+\right. \tag{96}$$

$$\left.2\alpha_{01}^2 e^{2\mu\omega+2\nu\tau}\,\text{erfc}\left(\frac{\mu\omega+2\nu\tau}{\sqrt{2}\sqrt{\nu\tau}}\right)-\text{erfc}\left(\frac{\mu\omega}{\sqrt{2}\sqrt{\nu\tau}}\right)+2\right)=$$

$$\frac{1}{2}\lambda_{01}^2\tau\left(\alpha_{01}^2\left(-e^{-\frac{\mu^2\omega^2}{2\nu\tau}}\right)\left(e^{\frac{(\mu\omega+\nu\tau)^2}{2\nu\tau}}\,\text{erfc}\left(\frac{\mu\omega+\nu\tau}{\sqrt{2}\sqrt{\nu\tau}}\right)-\right.\right.$$

$$\left.\left.2e^{\frac{(\mu\omega+2\nu\tau)^2}{2\nu\tau}}\,\text{erfc}\left(\frac{\mu\omega+2\nu\tau}{\sqrt{2}\sqrt{\nu\tau}}\right)\right)-\text{erfc}\left(\frac{\mu\omega}{\sqrt{2}\sqrt{\nu\tau}}\right)+2\right)\leqslant$$

$$\frac{1}{2}1.25\lambda_{01}^2\left(\alpha_{01}^2\left(-e^{-\frac{\mu^2\omega^2}{2\nu\tau}}\right)\left(e^{\frac{(\mu\omega+\nu\tau)^2}{2\nu\tau}}\,\text{erfc}\left(\frac{\mu\omega+\nu\tau}{\sqrt{2}\sqrt{\nu\tau}}\right)-\right.\right.$$

$$\left.\left.2e^{\frac{(\mu\omega+2\nu\tau)^2}{2\nu\tau}}\,\text{erfc}\left(\frac{\mu\omega+2\nu\tau}{\sqrt{2}\sqrt{\nu\tau}}\right)\right)-\text{erfc}\left(\frac{\mu\omega}{\sqrt{2}\sqrt{\nu\tau}}\right)+2\right)\leqslant$$

$$\frac{1}{2}1.25\lambda_{01}^2\left(\alpha_{01}^2\left(e^{\left(\frac{1.2+0.1}{\sqrt{2}\sqrt{1.2}}\right)^2}\,\text{erfc}\left(\frac{1.2+0.1}{\sqrt{2}\sqrt{1.2}}\right)-\right.\right.$$

$$\left.\left.2e^{\left(\frac{2\cdot 1.2+0.1}{\sqrt{2}\sqrt{1.2}}\right)^2}\,\text{erfc}\left(\frac{2\cdot 1.2+0.1}{\sqrt{2}\sqrt{1.2}}\right)\right)\left(-e^{-\frac{\mu^2\omega^2}{2\nu\tau}}\right)-\text{erfc}\left(\frac{\mu\omega}{\sqrt{2}\sqrt{\nu\tau}}\right)+2\right)\leqslant$$

$$\frac{1}{2}1.25\lambda_{01}^2\left(-e^{0.0}\alpha_{01}^2\left(e^{\left(\frac{1.2+0.1}{\sqrt{2}\sqrt{1.2}}\right)^2}\,\text{erfc}\left(\frac{1.2+0.1}{\sqrt{2}\sqrt{1.2}}\right)-\right.\right.$$

$$2 e^{\left(\frac{2 \cdot 1.2 + 0.1}{\sqrt{2}\sqrt{1.2}}\right)^2} \operatorname{erfc}\left(\frac{2 \cdot 1.2 + 0.1}{\sqrt{2}\sqrt{1.2}}\right)\right) - \operatorname{erfc}\left(\frac{\mu\omega}{\sqrt{2}\sqrt{\nu\tau}}\right) + 2\right) \leqslant$$

$$\frac{1}{2} 1.25 \lambda_{01}^2 \left(-e^{0.0}\alpha_{01}^2 \left(e^{\left(\frac{1.2+0.1}{\sqrt{2}\sqrt{1.2}}\right)^2} \operatorname{erfc}\left(\frac{1.2+0.1}{\sqrt{2}\sqrt{1.2}}\right) - \right.\right.$$

$$2 e^{\left(\frac{2 \cdot 1.2 + 0.1}{\sqrt{2}\sqrt{1.2}}\right)^2} \operatorname{erfc}\left(\frac{2 \cdot 1.2 + 0.1}{\sqrt{2}\sqrt{1.2}}\right)\right) - \operatorname{erfc}\left(\frac{0.1}{\sqrt{2}\sqrt{1.2}}\right) + 2\right) \leqslant$$

$$0.995063 \; < \; 1 \, .$$

We explain the chain of inequalities:

- First equality brings the expression into a shape where we can apply Lemma 15 for the the function Eq. (90).

- First inequality: The overall factor $\tau$ is bounded by 1.25.

- Second inequality: We apply Lemma 15. According to Lemma 15 the function Eq. (90) is negative. The largest contribution is to subtract the most negative value of the function Eq. (90), that is, the minimum of function Eq. (90). According to Lemma 15 the function Eq. (90) is strictly monotonically increasing in $x$ and strictly monotonically decreasing in $y$ for $x = 1.2$. Therefore the function Eq. (90) has its minimum at minimal $x = \nu\tau = 1.5 \cdot 0.8 = 1.2$ and maximal $y = \mu\omega = 1.0 \cdot 0.1 = 0.1$. We insert these values into the expression.

- Third inequality: We use for the whole expression the maximal factor $e^{-\frac{\mu^2\omega^2}{2\nu\tau}} < 1$ by setting this factor to 1.

- Fourth inequality: $\operatorname{erfc}$ is strictly monotonically decreasing. Therefore we maximize its argument to obtain the least value which is subtracted. We use the minimal $x = \nu\tau = 1.5 \cdot 0.8 = 1.2$ and the maximal $y = \mu\omega = 1.0 \cdot 0.1 = 0.1$.

- Sixth inequality: evaluation of the terms.

■ We now show that $\tilde{\mu} > 0$. The expression $\tilde{\mu}(\mu, \omega, \nu, \tau)$ (Eq. (5)) is strictly monotonically increasing im $\mu\omega$ and $\nu\tau$. Therefore, the minimal value in $\Omega^+$ is obtained at $\tilde{\mu}(0.01, 0.01, 1.5, 0.8) = 0.008293 > 0$.

■ Last we show that $\frac{\partial}{\partial \nu}\tilde{\mu} > 0$. The expression $\frac{\partial}{\partial \nu}\tilde{\mu}(\mu, \omega, \nu, \tau) = \mathcal{J}_{12}(\mu, \omega, \nu, \tau)$ (Eq. (48)) can we reformulated as follows:

$$\mathcal{J}_{12}(\mu, \omega, \nu, \tau, \lambda, \alpha) \; = \; \frac{\lambda\tau e^{-\frac{\mu^2\omega^2}{2\nu\tau}}\left(\sqrt{\pi}\alpha e^{\frac{(\mu\omega+\nu\tau)^2}{2\nu\tau}} \operatorname{erfc}\left(\frac{\mu\omega+\nu\tau}{\sqrt{2}\sqrt{\nu\tau}}\right) - \frac{\sqrt{2}(\alpha-1)}{\sqrt{\nu\tau}}\right)}{4\sqrt{\pi}} \quad (97)$$

is larger than zero when the term $\sqrt{\pi}\alpha e^{\frac{(\mu\omega+\nu\tau)^2}{2\nu\tau}} \operatorname{erfc}\left(\frac{\mu\omega+\nu\tau}{\sqrt{2}\sqrt{\nu\tau}}\right) - \frac{\sqrt{2}(\alpha-1)}{\sqrt{\nu\tau}}$ is larger than zero. This term obtains its minimal value at $\mu\omega = 0.01$ and $\nu\tau = 16 \cdot 1.25$, which can easily be shown using the Abramowitz bounds (Lemma 22) and evaluates to 0.16, therefore $\mathcal{J}_{12} > 0$ in $\Omega^+$.

□

### S3.4.4   Lemmata for proofing Theorem 3: The variance is expanding

**Main Sub-Function From Below.**   We consider functions in $\mu\omega$ and $\nu\tau$, therefore we set $x = \mu\omega$ and $y = \nu\tau$.

For $\lambda = \lambda_{01}$ and $\alpha = \alpha_{01}$, we consider the domain $-0.1 \leqslant \mu \leqslant 0.1$, $-0.1 \leqslant \omega \leqslant 0.1$ $0.00875 \leqslant \nu \leqslant 0.7$, and $0.8 \leqslant \tau \leqslant 1.25$.

For $x$ and $y$ we obtain: $0.8 \cdot 0.00875 = 0.007 \leqslant x \leqslant 0.875 = 1.25 \cdot 0.7$ and $0.1 \cdot (-0.1) = -0.01 \leqslant y \leqslant 0.01 = 0.1 \cdot 0.1$. In the following we assume to be within this domain.

In this domain, we consider the main sub-function of the derivate of second moment in the next layer, $J22$ (Eq. (48)):

$$\frac{\partial}{\partial \nu}\tilde{\xi} = \frac{1}{2}\lambda^2\tau\left(-\alpha^2 e^{\mu\omega+\frac{\nu\tau}{2}}\operatorname{erfc}\left(\frac{\mu\omega+\nu\tau}{\sqrt{2}\sqrt{\nu\tau}}\right) + 2\alpha^2 e^{2\mu\omega+2\nu\tau}\operatorname{erfc}\left(\frac{\mu\omega+2\nu\tau}{\sqrt{2}\sqrt{\nu\tau}}\right) - \operatorname{erfc}\left(\frac{\mu\omega}{\sqrt{2}\sqrt{\nu\tau}}\right) + 2\right)$$
$$(98)$$

that depends on $\mu\omega$ and $\nu\tau$, therefore we set $x = \nu\tau$ and $y = \mu\omega$. Algebraic reformulations provide the formula in the following form:

$$\frac{\partial}{\partial \nu}\tilde{\xi} = \tag{99}$$
$$\frac{1}{2}\lambda^2\tau\alpha^2\left(-e^{-\frac{y^2}{2x}}\right)\left(e^{\frac{(x+y)^2}{2x}}\operatorname{erfc}\left(\frac{y+x}{\sqrt{2}\sqrt{x}}\right) - 2e^{\frac{(2x+y)^2}{2x}}\operatorname{erfc}\left(\frac{y+2x}{\sqrt{2}\sqrt{x}}\right) + \frac{1}{\alpha^2}e^{\frac{y^2}{2x}}\operatorname{erfc}\left(\frac{y}{\sqrt{2}\sqrt{x}}\right) - 2\right)$$

**Lemma 17** (Main subfunction Below). *For $0.007 \leqslant x \leqslant 0.875$ and $-0.01 \leqslant y \leqslant 0.01$, the function*

$$e^{\frac{(x+y)^2}{2x}}\operatorname{erfc}\left(\frac{x+y}{\sqrt{2}\sqrt{x}}\right) - 2e^{\frac{(2x+y)^2}{2x}}\operatorname{erfc}\left(\frac{2x+y}{\sqrt{2}\sqrt{x}}\right) \tag{100}$$

*smaller than zero, is strictly monotonically increasing in $x$ and strictly monotonically increasing in $y$ for the minimal $x = 0.007 = 0.00875 \cdot 0.8$, $x = 0.56 = 0.7 \cdot 0.8$, $x = 0.128 = 0.16 \cdot 0.8$, and $x = 0.216 = 0.24 \cdot 0.9$ (lower bound of $0.9$ on $\tau$).*

*Proof.* See proof 45.                                                                       □

**Lemma 18** (Monotone Derivative). *For $\lambda = \lambda_{01}$, $\alpha = \alpha_{01}$ and the domain $-0.1 \leqslant \mu \leqslant 0.1$, $-0.1 \leqslant \omega \leqslant 0.1$, $0.00875 \leqslant \nu \leqslant 0.7$, and $0.8 \leqslant \tau \leqslant 1.25$. We are interested of the derivative of*

$$\tau\left(e^{\left(\frac{\mu\omega+\nu\tau}{\sqrt{2}\sqrt{\nu\tau}}\right)^2}\operatorname{erfc}\left(\frac{\mu\omega+\nu\tau}{\sqrt{2}\sqrt{\nu\tau}}\right) - 2e^{\left(\frac{\mu\omega+2\cdot\nu\tau}{\sqrt{2}\sqrt{\nu\tau}}\right)^2}\operatorname{erfc}\left(\frac{\mu\omega+2\cdot\nu\tau}{\sqrt{2}\sqrt{\nu\tau}}\right)\right). \tag{101}$$

*The derivative of the equation above with respect to*

- $\nu$ *is larger than zero;*

- $\tau$ *is smaller than zero for maximal $\nu = 0.7$, $\nu = 0.16$, and $\nu = 0.24$ (with $0.9 \leqslant \tau$);*

- $y = \mu\omega$ *is larger than zero for $\nu\tau = 0.00875 \cdot 0.8 = 0.007$, $\nu\tau = 0.7 \cdot 0.8 = 0.56$, $\nu\tau = 0.16 \cdot 0.8 = 0.128$, and $\nu\tau = 0.24 \cdot 0.9 = 0.216$.*

*Proof.* See proof 46.                                                                       □

**S3.4.5   Computer-assisted proof details for main Lemma 12 in Section 3.4.1.**

**Error Analysis.**   We investigate the error propagation for the singular value (Eq. (55)) if the function arguments $\mu, \omega, \nu, \tau$ suffer from numerical imprecisions up to $\epsilon$. To this end, we first derive error propagation rules based on the mean value theorem and then we apply these rules to the formula for the singular value.

**Lemma 19** (Mean value theorem). *For a real-valued function $f$ which is differentiable in the closed interval $[a, b]$, there exists $t \in [0, 1]$ with*

$$f(\boldsymbol{a}) \; - \; f(\boldsymbol{b}) \; = \; \nabla f(\boldsymbol{a} + t(\boldsymbol{b} - \boldsymbol{a})) \; \cdot \; (\boldsymbol{a} \; - \; \boldsymbol{b}) \,. \tag{102}$$

It follows that for computation with error $\Delta x$, there exists a $t \in [0, 1]$ with

$$|f(\boldsymbol{x} + \Delta\boldsymbol{x}) \; - \; f(\boldsymbol{x})| \; \leqslant \; \|\nabla f(\boldsymbol{x} + t\Delta\boldsymbol{x})\| \; \|\Delta\boldsymbol{x}\| \,. \tag{103}$$

Therefore the increase of the norm of the error after applying function $f$ is bounded by the norm of the gradient $\|\nabla f(\boldsymbol{x} + t\Delta\boldsymbol{x})\|$.

We now compute for the functions, that we consider their gradient and its 2-norm:

- addition:

  $f(\boldsymbol{x}) = x_1 + x_2$ and $\nabla f(\boldsymbol{x}) = (1, 1)$, which gives $\|\nabla f(\boldsymbol{x})\| = \sqrt{2}$.

  We further know that

$$|f(\boldsymbol{x} + \Delta\boldsymbol{x}) - f(\boldsymbol{x})| \; = \; |x_1 + x_2 + \Delta x_1 + \Delta x_2 - x_1 - x_2| \; \leqslant \; |\Delta x_1| + |\Delta x_2| \,. \tag{104}$$

  Adding $n$ terms gives:

$$\left| \sum_{i=1}^{n} x_i + \Delta x_i \; - \; \sum_{i=1}^{n} x_i \right| \; \leqslant \; \sum_{i=1}^{n} |\Delta x_i| \; \leqslant \; n \, |\Delta x_i|_{\max} \,. \tag{105}$$

- subtraction:

  $f(\boldsymbol{x}) = x_1 - x_2$ and $\nabla f(\boldsymbol{x}) = (1, -1)$, which gives $\|\nabla f(\boldsymbol{x})\| = \sqrt{2}$.

  We further know that

$$|f(\boldsymbol{x} + \Delta\boldsymbol{x}) - f(\boldsymbol{x})| \; = \; |x_1 - x_2 + \Delta x_1 - \Delta x_2 - x_1 + x_2| \; \leqslant \; |\Delta x_1| + |\Delta x_2| \,. \tag{106}$$

  Subtracting $n$ terms gives:

$$\left| \sum_{i=1}^{n} -(x_i + \Delta x_i) \; + \; \sum_{i=1}^{n} x_i \right| \; \leqslant \; \sum_{i=1}^{n} |\Delta x_i| \; \leqslant \; n \, |\Delta x_i|_{\max} \,. \tag{107}$$

- multiplication:

  $f(\boldsymbol{x}) = x_1 x_2$ and $\nabla f(\boldsymbol{x}) = (x_2, x_1)$, which gives $\|\nabla f(\boldsymbol{x})\| = \|\boldsymbol{x}\|$.

We further know that

$$|f(\boldsymbol{x} + \Delta\boldsymbol{x}) - f(\boldsymbol{x})| \;=\; |x_1 \cdot x_2 + \Delta x_1 \cdot x_2 + \Delta x_2 \cdot x_1 + \Delta x_1 \cdot \Delta x_s - x_1 \cdot x_2| \;\leqslant \tag{108}$$

$$|\Delta x_1|\,|x_2| + |\Delta x_2|\,|x_1| + O(\Delta^2)\,.$$

Multiplying $n$ terms gives:

$$\left|\prod_{i=1}^{n}(x_i + \Delta x_i) \;-\; \prod_{i=1}^{n} x_i\right| \;=\; \left|\prod_{i=1}^{n} x_i \sum_{i=1}^{n} \frac{\Delta x_i}{x_i} \;+\; O(\Delta^2)\right| \;\leqslant \tag{109}$$

$$\prod_{i=1}^{n}|x_i| \sum_{i=1}^{n}\left|\frac{\Delta x_i}{x_i}\right| \;+\; O(\Delta^2) \;\leqslant\; n\prod_{i=1}^{n}|x_i|\,\left|\frac{\Delta x_i}{x_i}\right|_{\max} \;+\; O(\Delta^2)\,.$$

- division:
  $f(\boldsymbol{x}) = \frac{x_1}{x_2}$ and $\nabla f(\boldsymbol{x}) = \left(\frac{1}{x_2}, -\frac{x_1}{x_2^2}\right)$, which gives $\|\nabla f(\boldsymbol{x})\| = \frac{\|\boldsymbol{x}\|}{x_2^2}$.

  We further know that

$$|f(\boldsymbol{x} + \Delta\boldsymbol{x}) - f(\boldsymbol{x})| \;=\; \left|\frac{x_1 + \Delta x_1}{x_2 + \Delta x_2} - \frac{x_1}{x_2}\right| \;=\; \left|\frac{(x_1 + \Delta x_1)x_2 - x_1(x_2 + \Delta x_2)}{(x_2 + \Delta x_2)x_2}\right| \;= \tag{110}$$

$$\left|\frac{\Delta x_1 \cdot x_2 - \Delta x_2 \cdot x_1}{x_2^2 + \Delta x_2 \cdot x_2}\right| \;=\; \left|\frac{\Delta x_1}{x_2} - \frac{\Delta x_2 \cdot x_1}{x_2^2}\right| + O(\Delta^2)\,.$$

- square root:
  $f(x) = \sqrt{x}$ and $f'(x) = \frac{1}{2\sqrt{x}}$, which gives $|f'(x)| = \frac{1}{2\sqrt{x}}$.

- exponential function:
  $f(x) = \exp(x)$ and $f'(x) = \exp(x)$, which gives $|f'(x)| = \exp(x)$.

- error function:
  $f(x) = \operatorname{erf}(x)$ and $f'(x) = \frac{2}{\sqrt{\pi}}\exp(-x^2)$, which gives $|f'(x)| = \frac{2}{\sqrt{\pi}}\exp(-x^2)$.

- complementary error function:
  $f(x) = \operatorname{erfc}(x)$ and $f'(x) = -\frac{2}{\sqrt{\pi}}\exp(-x^2)$, which gives $|f'(x)| = \frac{2}{\sqrt{\pi}}\exp(-x^2)$.

**Lemma 20.** *If the values $\mu, \omega, \nu, \tau$ have a precision of $\epsilon$, the singular value (Eq. (55)) evaluated with the formulas given in Eq. (48) and Eq. (55) has a precision better than $292\epsilon$.*

This means for a machine with a typical precision of $2^{-52} = 2.220446 \cdot 10^{-16}$, we have the rounding error $\epsilon \approx 10^{-16}$, the evaluation of the singular value (Eq. (55)) with the formulas given in Eq. (48) and Eq. (55) has a precision better than $10^{-13} > 292\epsilon$.

*Proof.* We have the numerical precision $\epsilon$ of the parameters $\mu, \omega, \nu, \tau$, that we denote by $\Delta\mu, \Delta\omega, \Delta\nu, \Delta\tau$ together with our domain $\Omega$.

With the error propagation rules that we derived in Subsection S3.4.5, we can obtain bounds for the numerical errors on the following simple expressions:

$$\Delta\left(\mu\omega\right) \leqslant \Delta\mu\left|\omega\right| + \Delta\omega\left|\mu\right| \leqslant 0.2\epsilon \tag{111}$$

$$\Delta\left(\nu\tau\right) \leqslant \Delta\nu\left|\tau\right| + \Delta\tau\left|\nu\right| \leqslant 1.5\epsilon + 1.5\epsilon = 3\epsilon$$

$$\Delta\left(\frac{\nu\tau}{2}\right) \leqslant \left(\Delta(\nu\tau)\cdot 2 + \Delta 2\left|\nu\tau\right|\right)\frac{1}{2^2} \leqslant \left(6\epsilon + 1.25\cdot 1.5\epsilon\right)/4 < 2\epsilon$$

$$\Delta\left(\mu\omega + \nu\tau\right) \leqslant \Delta\left(\mu\omega\right) + \Delta\left(\nu\tau\right) = 3.2\epsilon$$

$$\Delta\left(\mu\omega + \frac{\nu\tau}{2}\right) \leqslant \Delta\left(\mu\omega\right) + \Delta\left(\frac{\nu\tau}{2}\right) < 2.2\epsilon$$

$$\Delta\left(\sqrt{\nu\tau}\right) \leqslant \frac{\Delta\left(\nu\tau\right)}{2\sqrt{\nu\tau}} \leqslant \frac{3\epsilon}{2\sqrt{0.64}} = 1.875\epsilon$$

$$\Delta\left(\sqrt{2}\right) \leqslant \frac{\Delta 2}{2\sqrt{2}} \leqslant \frac{1}{2\sqrt{2}}\epsilon$$

$$\Delta\left(\sqrt{2}\cdot\sqrt{\nu\tau}\right) \leqslant \sqrt{2}\cdot\Delta\left(\sqrt{\nu\tau}\right) + \nu\tau\cdot\Delta\left(\sqrt{2}\right) \leqslant \sqrt{2}\cdot 1.875\epsilon + 1.5\cdot 1.25\cdot\frac{1}{2\sqrt{2}}\epsilon < 3.5\epsilon$$

$$\Delta\left(\frac{\mu\omega}{\sqrt{2}\sqrt{\nu\tau}}\right) \leqslant \left(\Delta\left(\mu\omega\right)\cdot\sqrt{2}\sqrt{\nu\tau} + \left|\mu\omega\right|\cdot\Delta\left(\sqrt{2}\sqrt{\nu\tau}\right)\right)\frac{1}{\left(\sqrt{2}\sqrt{\nu\tau}\right)^2} \leqslant$$

$$\left(0.2\epsilon\cdot\sqrt{2}\sqrt{0.64} + 0.01\cdot 3.5\epsilon\right)\frac{1}{2\cdot 0.64} < 0.25\epsilon$$

$$\Delta\left(\frac{\mu\omega + \nu\tau}{\sqrt{2}\sqrt{\nu\tau}}\right) \leqslant \left(\Delta\left(\mu\omega + \nu\tau\right)\cdot\sqrt{2}\sqrt{\nu\tau} + \left|\mu\omega + \nu\tau\right|\cdot\Delta\left(\sqrt{2}\sqrt{\nu\tau}\right)\right)\frac{1}{\left(\sqrt{2}\sqrt{\nu\tau}\right)^2} \leqslant$$

$$\left(3.2\epsilon\cdot\sqrt{2}\sqrt{0.64} + 1.885\cdot 3.5\epsilon\right)\frac{1}{2\cdot 0.64} < 8\epsilon.$$

Using these bounds on the simple expressions, we can now calculate bounds on the numerical errors of compound expressions:

$$\Delta\left(\mathrm{erfc}\left(\frac{\mu\omega}{\sqrt{2}\sqrt{\nu\tau}}\right)\right) \leqslant \frac{2}{\sqrt{\pi}}\cdot e^{-\left(\frac{\mu\omega}{\sqrt{2}\sqrt{\nu\tau}}\right)^2}\Delta\left(\frac{\mu\omega}{\sqrt{2}\sqrt{\nu\tau}}\right) < \tag{112}$$

$$\frac{2}{\sqrt{\pi}}\cdot 0.25\epsilon < 0.3\epsilon$$

$$\Delta\left(\mathrm{erfc}\left(\frac{\mu\omega + \nu\tau}{\sqrt{2}\sqrt{\nu\tau}}\right)\right) \leqslant \frac{2}{\sqrt{\pi}}\cdot e^{-\left(\frac{\mu\omega + \nu\tau}{\sqrt{2}\sqrt{\nu\tau}}\right)^2}\Delta\left(\frac{\mu\omega + \nu\tau}{\sqrt{2}\sqrt{\nu\tau}}\right) < \tag{113}$$

$$\frac{2}{\sqrt{\pi}}\cdot 8\epsilon < 10\epsilon$$

$$\Delta\left(e^{\mu\omega + \frac{\nu\tau}{2}}\right) \leqslant \left(e^{\mu\omega + \frac{\nu\tau}{2}}\right)\Delta\left(e^{\mu\omega + \frac{\nu\tau}{2}}\right) < \tag{114}$$

$$e^{0.9475}\cdot 2.2\epsilon < 5.7\epsilon \tag{115}$$

Subsequently, we can use the above results to get bounds for the numerical errors on the Jacobian entries (Eq. (48)), applying the rules from Subsection S3.4.5 again:

$$\Delta\left(\mathcal{J}_{11}\right) = \tag{116}$$

$$\Delta\left(\frac{1}{2}\lambda\omega\left(\alpha e^{\mu\omega+\frac{\nu\tau}{2}}\operatorname{erfc}\left(\frac{\mu\omega+\nu\tau}{\sqrt{2}\sqrt{\nu\tau}}\right) - \operatorname{erfc}\left(\frac{\mu\omega}{\sqrt{2}\sqrt{\nu\tau}}\right) + 2\right)\right) < 6\epsilon, \tag{117}$$

and we obtain $\Delta\left(\mathcal{J}_{12}\right) < 78\epsilon$, $\Delta\left(\mathcal{J}_{21}\right) < 189\epsilon$, $\Delta\left(\mathcal{J}_{22}\right) < 405\epsilon$ and $\Delta\left(\tilde{\mu}\right) < 52\epsilon$. We also have bounds on the absolute values on $\mathcal{J}_{ij}$ and $\tilde{\mu}$ (see Lemma 6, Lemma 7, and Lemma 9), therefore we can propagate the error also through the function that calculates the singular value (Eq. (55)).

$$\Delta\left(S(\mu,\omega,\nu,\tau,\lambda,\alpha)\right) = \tag{118}$$

$$= \Delta\left(\frac{1}{2}\left(\sqrt{(\mathcal{J}_{11}+\mathcal{J}_{22}-2\tilde{\mu}\mathcal{J}_{12})^2+(\mathcal{J}_{21}-2\tilde{\mu}\mathcal{J}_{11}-\mathcal{J}_{12})^2} + \right.\right.$$

$$\left.\left.\sqrt{(\mathcal{J}_{11}-\mathcal{J}_{22}+2\tilde{\mu}\mathcal{J}_{12})^2+(\mathcal{J}_{12}+\mathcal{J}_{21}-2\tilde{\mu}\mathcal{J}_{11})^2}\right)\right) <$$

$$292\epsilon.$$

$\square$

**Precision of Implementations.** We will show that our computations are correct up to 3 ulps. For our implementation in GNU C library and the hardware architectures that we used, the precision of all mathematical functions that we used is at least one ulp. The term "ulp" (acronym for "unit in the last place") was coined by W. Kahan in 1960. It is the highest precision (up to some factor smaller 1), which can be achieved for the given hardware and floating point representation.

Kahan defined ulp as (Kahan, 2004):

"Ulp($x$) is the gap between the two *finite* floating-point numbers nearest $x$, even if $x$ is one of them. (But ulp(NaN) is NaN.)"

Harrison defined ulp as (Harrison, 1999):

"an ulp in $x$ is the distance between the two closest *straddling* floating point numbers $a$ and $b$, i.e. those with $a \leqslant x \leqslant b$ and $a \neq b$ assuming an unbounded exponent range."

In the literature we find also slightly different definitions (Muller, 2005).

According to (Muller, 2005) who refers to (Goldberg, 1991):

"IEEE-754 mandates four standard rounding modes:"

"Round-to-nearest: $r(x)$ is the floating-point value closest to $x$ with the usual distance; if two floating-point value are equally close to $x$, then $r(x)$ is the one whose least significant bit is equal to zero."

"IEEE-754 standardises 5 operations: addition (which we shall note $\oplus$ in order to distinguish it from the operation over the reals), subtraction ($\ominus$), multiplication ($\otimes$), division ($\oslash$), and also square root."

"IEEE-754 specifies em exact rounding [Goldberg, 1991, §1.5]: the result of a floating-point operation is the same as if the operation were performed on the real numbers with the given inputs, then rounded according to the rules in the preceding section. Thus, $x \oplus y$ is defined as $r(x+y)$, with $x$ and $y$ taken as elements of $\mathbb{R} \cup \{-\infty, +\infty\}$; the same applies for the other operators."

Consequently, the IEEE-754 standard guarantees that addition, subtraction, multiplication, division, and squared root is precise up to one ulp.

We have to consider transcendental functions. First the is the exponential function, and then the complementary error function $\mathrm{erfc}(x)$, which can be computed via the error function $\mathrm{erf}(x)$.

Intel states (Muller, 2005):

"With the Intel486 processor and Intel 387 math coprocessor, the worst- case, transcendental function error is typically 3 or 3.5 ulps, but is some- times as large as $4.5$ ulps."

According to `https://www.mirbsd.org/htman/i386/man3/exp.htm` and `http://man.openbsd.org/OpenBSD-current/man3/exp.3`:

"$\exp(x)$, $\log(x)$, expm1$(x)$ and log1p$(x)$ are accurate to within an ulp"

which is the same for freebsd `https://www.freebsd.org/cgi/man.cgi?query=exp&sektion=3&apropos=0&manpath=freebsd`:

"The values of exp(0), expm1(0), exp2(integer), and pow(integer, integer) are exact provided that they are representable. Otherwise the error in these functions is generally below one ulp."

The same holds for "FDLIBM" `http://www.netlib.org/fdlibm/readme`:

"FDLIBM is intended to provide a reasonably portable (see assumptions below), reference quality (below one ulp for major functions like sin,cos,exp,log) math library (libm.a)."

In `http://www.gnu.org/software/libc/manual/html_node/Errors-in-Math-Functions.html` we find that both $\exp$ and $\mathrm{erf}$ have an error of 1 ulp while $\mathrm{erfc}$ has an error up to 3 ulps depending on the architecture. For the most common architectures as used by us, however, the error of $\mathrm{erfc}$ is 1 ulp.

We implemented the function in the programming language C. We rely on the GNU C Library (Loosemore et al., 2016). According to the GNU C Library manual which can be obtained from http://www.gnu.org/software/libc/manual/pdf/libc.pdf, the errors of the math functions $\exp$, $\mathrm{erf}$, and $\mathrm{erfc}$ are not larger than 3 ulps for all architectures (Loosemore et al., 2016, pp. 528). For the architectures ix86, i386/i686/fpu, and m68k/fpmu68k/m680x0/fpu that we used the error are at least one ulp (Loosemore et al., 2016, pp. 528).

### S3.4.6   Intermediate Lemmata and Proofs

Since we focus on the fixed point $(\mu, \nu) = (0, 1)$, we assume for our whole analysis that $\alpha = \alpha_{01}$ and $\lambda = \lambda_{01}$. Furthermore, we restrict the range of the variables $\mu \in [\mu_{\min}, \mu_{\max}] = [-0.1, 0.1]$, $\omega \in [\omega_{\min}, \omega_{\max}] = [-0.1, 0.1]$, $\nu \in [\nu_{\min}, \nu_{\max}] = [0.8, 1.5]$, and $\tau \in [\tau_{\min}, \tau_{\max}] = [0.8, 1.25]$.

For bounding different partial derivatives we need properties of different functions. We will bound a the absolute value of a function by computing an upper bound on its maximum and a lower bound on its minimum. These bounds are computed by upper or lower bounding terms. The bounds get tighter if we can combine terms to a more complex function and bound this function. The following lemmata give some properties of functions that we will use in bounding complex functions.

**Lemma 21** (Basic functions). $\exp(x)$ *is strictly monotonically increasing from* 0 *at* $-\infty$ *to* $\infty$ *at* $\infty$ *and has positive curvature.*

*According to its definition* $\mathrm{erfc}(x)$ *is strictly monotonically decreasing from 2 at* $-\infty$ *to 0 at* $\infty$.

Next we introduce a bound on $\mathrm{erfc}$:

**Lemma 22** (Erfc bound from Abramowitz).

$$\frac{2e^{-x^2}}{\sqrt{\pi}\left(\sqrt{x^2 + 2} + x\right)} \;<\; \mathrm{erfc}(x) \;\leqslant\; \frac{2e^{-x^2}}{\sqrt{\pi}\left(\sqrt{x^2 + \frac{4}{\pi}} + x\right)}, \tag{119}$$

*for* $x > 0$.

*Proof.* The statement follows immediately from (Abramowitz and Stegun, 1964) (page 298, formula 7.1.13). $\qquad\square$

These bounds are displayed in figure S3.

**Lemma 23** (Function $e^{x^2} \mathrm{erfc}(x)$). $e^{x^2} \mathrm{erfc}(x)$ *is strictly monotonically decreasing for* $x > 0$ *and has positive curvature (positive 2nd order derivative), that is, the decreasing slowes down.*

A graph of the function is displayed in figure

*Proof.* The derivative of $e^{x^2} \mathrm{erfc}(x)$ is

$$\frac{\partial e^{x^2} \mathrm{erfc}(x)}{\partial x} \;=\; 2e^{x^2} x \, \mathrm{erfc}(x) - \frac{2}{\sqrt{\pi}} \;. \tag{120}$$

Figure S3: Graphs of the upper and lower bounds on $\mathrm{erfc}$. The lower bound $\dfrac{2e^{-x^2}}{\sqrt{\pi}\left(\sqrt{x^2+2}+x\right)}$ (red), the upper bound $\dfrac{2e^{-x^2}}{\sqrt{\pi}\left(\sqrt{x^2+\frac{4}{\pi}}+x\right)}$ (green) and the function $\mathrm{erfc}(x)$ (blue) as treated in Lemma 22.

Figure S4: Graphs of the functions $e^{x^2}\,\mathrm{erfc}(x)$ (left) and $xe^{x^2}\,\mathrm{erfc}(x)$ (right) treated in Lemma 23 and Lemma 24, respectively.

Using Lemma 22, we get

$$
\frac{\partial e^{x^2} \operatorname{erfc}(x)}{\partial x} \;=\; 2e^{x^2} x \operatorname{erfc}(x) - \frac{2}{\sqrt{\pi}} \;<\; \tag{121}
$$

$$
\frac{4x}{\sqrt{\pi}\left(\sqrt{x^2 + \frac{4}{\pi}} + x\right)} - \frac{2}{\sqrt{\pi}} = \frac{2\left(\frac{2}{\sqrt{\frac{4}{\pi x^2}+1}+1} - 1\right)}{\sqrt{\pi}} \;<\; 0
$$

Thus $e^{x^2} \operatorname{erfc}(x)$ is strictly monotonically decreasing for $x > 0$.

The second order derivative of $e^{x^2} \operatorname{erfc}(x)$ is

$$
\frac{\partial^2 e^{x^2} \operatorname{erfc}(x)}{\partial x^2} \;=\; 4e^{x^2} x^2 \operatorname{erfc}(x) + 2e^{x^2} \operatorname{erfc}(x) - \frac{4x}{\sqrt{\pi}} \;. \tag{122}
$$

Again using Lemma 22 (first inequality), we get

$$
2\left((2x^2 + 1)\, e^{x^2} \operatorname{erfc}(x) - \frac{2x}{\sqrt{\pi}}\right) \;>\; \tag{123}
$$

$$
\frac{4\left(2x^2 + 1\right)}{\sqrt{\pi}\left(\sqrt{x^2 + 2} + x\right)} - \frac{4x}{\sqrt{\pi}} =
$$

$$
\frac{4\left(x^2 - \sqrt{x^2 + 2}\, x + 1\right)}{\sqrt{\pi}\left(\sqrt{x^2 + 2} + x\right)} =
$$

$$
\frac{4\left(x^2 - \sqrt{x^4 + 2x^2} + 1\right)}{\sqrt{\pi}\left(\sqrt{x^2 + 2} + x\right)} \;>\;
$$

$$
\frac{4\left(x^2 - \sqrt{x^4 + 2x^2 + 1} + 1\right)}{\sqrt{\pi}\left(\sqrt{x^2 + 2} + x\right)} \;=\; 0
$$

For the last inequality we added 1 in the numerator in the square root which is subtracted, that is, making a larger negative term in the numerator.  □

**Lemma 24** (Properties of $xe^{x^2} \operatorname{erfc}(x)$)**.** *The function $xe^{x^2} \operatorname{erfc}(x)$ has the sign of $x$ and is monotonically increasing to $\frac{1}{\sqrt{\pi}}$.*

*Proof.* The derivative of $xe^{x^2} \operatorname{erfc}(x)$ is

$$
2e^{x^2} x^2 \operatorname{erfc}(x) + e^{x^2} \operatorname{erfc}(x) - \frac{2x}{\sqrt{\pi}} \;. \tag{124}
$$

This derivative is positive since

$$
2e^{x^2} x^2 \operatorname{erfc}(x) + e^{x^2} \operatorname{erfc}(x) - \frac{2x}{\sqrt{\pi}} \;=\; \tag{125}
$$

$$e^{x^2}\left(2x^2+1\right)\operatorname{erfc}(x)-\frac{2x}{\sqrt{\pi}}>\frac{2\left(2x^2+1\right)}{\sqrt{\pi}\left(\sqrt{x^2+2}+x\right)}-\frac{2x}{\sqrt{\pi}}=\frac{2\left(\left(2x^2+1\right)-x\left(\sqrt{x^2+2}+x\right)\right)}{\sqrt{\pi}\left(\sqrt{x^2+2}+x\right)}=$$

$$\frac{2\left(x^2-x\sqrt{x^2+2}+1\right)}{\sqrt{\pi}\left(\sqrt{x^2+2}+x\right)}=\frac{2\left(x^2-x\sqrt{x^2+2}+1\right)}{\sqrt{\pi}\left(\sqrt{x^2+2}+x\right)}>\frac{2\left(x^2-x\sqrt{x^2+\frac{1}{x^2}+2}+1\right)}{\sqrt{\pi}\left(\sqrt{x^2+2}+x\right)}=$$

$$\frac{2\left(x^2-\sqrt{x^4+2x^2+1}+1\right)}{\sqrt{\pi}\left(\sqrt{x^2+2}+x\right)}=\frac{2\left(x^2-\sqrt{\left(x^2+1\right)^2}+1\right)}{\sqrt{\pi}\left(\sqrt{x^2+2}+x\right)}=0\ .$$

We apply Lemma 22 to $x\operatorname{erfc}(x)e^{x^2}$ and divide the terms of the lemma by $x$, which gives

$$\frac{2}{\sqrt{\pi}\left(\sqrt{\frac{2}{x^2}+1}+1\right)}<x\operatorname{erfc}(x)e^{x^2}\leqslant\frac{2}{\sqrt{\pi}\left(\sqrt{\frac{4}{\pi x^2}+1}+1\right)}\ . \tag{126}$$

For $\lim_{x\to\infty}$ both the upper and the lower bound go to $\frac{1}{\sqrt{\pi}}$. $\qquad\square$

**Lemma 25** (Function $\mu\omega$). $h_{11}(\mu,\omega)=\mu\omega$ *is monotonically increasing in* $\mu\omega$. *It has minimal value* $t_{11}=-0.01$ *and maximal value* $T_{11}=0.01$.

*Proof.* Obvious. $\qquad\square$

**Lemma 26** (Function $\nu\tau$). $h_{22}(\nu,\tau)=\nu\tau$ *is monotonically increasing in* $\nu\tau$ *and is positive. It has minimal value* $t_{22}=0.64$ *and maximal value* $T_{22}=1.875$.

*Proof.* Obvious. $\qquad\square$

**Lemma 27** (Function $\frac{\mu\omega+\nu\tau}{\sqrt{2}\sqrt{\nu\tau}}$). $h_1(\mu,\omega,\nu,\tau)=\frac{\mu\omega+\nu\tau}{\sqrt{2}\sqrt{\nu\tau}}$ *is larger than zero and increasing in both* $\nu\tau$ *and* $\mu\omega$. *It has minimal value* $t_1=0.5568$ *and maximal value* $T_1=0.9734$.

*Proof.* The derivative of the function $\frac{\mu\omega+x}{\sqrt{2}\sqrt{x}}$ with respect to $x$ is

$$\frac{1}{\sqrt{2}\sqrt{x}}-\frac{\mu\omega+x}{2\sqrt{2}x^{3/2}}= \tag{127}$$

$$\frac{2x-(\mu\omega+x)}{2\sqrt{2}x^{3/2}}=\frac{x-\mu\omega}{2\sqrt{2}x^{3/2}}>0\ ,$$

since $x>0.8\cdot0.8$ and $\mu\omega<0.1\cdot0.1$. $\qquad\square$

**Lemma 28** (Function $\frac{\mu\omega+2\nu\tau}{\sqrt{2}\sqrt{\nu\tau}}$). $h_2(\mu,\omega,\nu,\tau)=\frac{\mu\omega+2\nu\tau}{\sqrt{2}\sqrt{\nu\tau}}$ *is larger than zero and increasing in both* $\nu\tau$ *and* $\mu\omega$. *It has minimal value* $t_2=1.1225$ *and maximal value* $T_2=1.9417$.

*Proof.* The derivative of the function $\frac{\mu\omega+2x}{\sqrt{2}\sqrt{x}}$ with respect to $x$ is

$$\frac{\sqrt{2}}{\sqrt{x}}-\frac{\mu\omega+2x}{2\sqrt{2}x^{3/2}}=\frac{4x-(\mu\omega+2x)}{2\sqrt{2}x^{3/2}}=\frac{2x-\mu\omega}{2\sqrt{2}x^{3/2}}>0\ . \tag{128}$$

$\qquad\square$

**Lemma 29** (Function $\frac{\mu\omega}{\sqrt{2}\sqrt{\nu\tau}}$). $h_3(\mu, \omega, \nu, \tau) = \frac{\mu\omega}{\sqrt{2}\sqrt{\nu\tau}}$ *monotonically decreasing in* $\nu\tau$ *and monotonically increasing in* $\mu\omega$. *It has minimal value* $t_3 = -0.0088388$ *and maximal value* $T_3 = 0.0088388$.

*Proof.* Obvious.                                                                           $\square$

**Lemma 30** (Function $\left(\frac{\mu\omega}{\sqrt{2}\sqrt{\nu\tau}}\right)^2$). $h_4(\mu, \omega, \nu, \tau) = \left(\frac{\mu\omega}{\sqrt{2}\sqrt{\nu\tau}}\right)^2$ *has a minimum at 0 for* $\mu = 0$ *or* $\omega = 0$ *and has a maximum for the smallest* $\nu\tau$ *and largest* $|\mu\omega|$ *and is larger or equal to zero. It has minimal value* $t_4 = 0$ *and maximal value* $T_4 = 0.000078126$.

*Proof.* Obvious.                                                                           $\square$

**Lemma 31** (Function $\frac{\sqrt{\frac{2}{\pi}}(\alpha-1)}{\sqrt{\nu\tau}}$). $\frac{\sqrt{\frac{2}{\pi}}(\alpha-1)}{\sqrt{\nu\tau}} > 0$ *and decreasing in* $\nu\tau$.

*Proof.* Statements follow directly from elementary functions square root and division.         $\square$

**Lemma 32** (Function $2 - \mathrm{erfc}\left(\frac{\mu\omega}{\sqrt{2}\sqrt{\nu\tau}}\right)$). $2 - \mathrm{erfc}\left(\frac{\mu\omega}{\sqrt{2}\sqrt{\nu\tau}}\right) > 0$ *and decreasing in* $\nu\tau$ *and increasing in* $\mu\omega$.

*Proof.* Statements follow directly from Lemma 21 and erfc.                                   $\square$

**Lemma 33** (Function $\sqrt{\frac{2}{\pi}}\left(\frac{(\alpha-1)\mu\omega}{(\nu\tau)^{3/2}} - \frac{\alpha}{\sqrt{\nu\tau}}\right)$). *For* $\lambda = \lambda_{01}$ *and* $\alpha = \alpha_{01}$, $\sqrt{\frac{2}{\pi}}\left(\frac{(\alpha-1)\mu\omega}{(\nu\tau)^{3/2}} - \frac{\alpha}{\sqrt{\nu\tau}}\right) < 0$ *and increasing in both* $\nu\tau$ *and* $\mu\omega$.

*Proof.* We consider the function $\sqrt{\frac{2}{\pi}}\left(\frac{(\alpha-1)\mu\omega}{x^{3/2}} - \frac{\alpha}{\sqrt{x}}\right)$, which has the derivative with respect to $x$:

$$\sqrt{\frac{2}{\pi}}\left(\frac{\alpha}{2x^{3/2}} - \frac{3(\alpha-1)\mu\omega}{2x^{5/2}}\right) . \tag{129}$$

This derivative is larger than zero, since

$$\sqrt{\frac{2}{\pi}}\left(\frac{\alpha}{2(\nu\tau)^{3/2}} - \frac{3(\alpha-1)\mu\omega}{2(\nu\tau)^{5/2}}\right) > \tag{130}$$

$$\frac{\sqrt{\frac{2}{\pi}}\left(\alpha - \frac{3(\alpha-1)\mu\omega}{\nu\tau}\right)}{2(\nu\tau)^{3/2}} > 0 .$$

The last inequality follows from $\alpha - \frac{3\cdot0.1\cdot0.1(\alpha-1)}{0.8\cdot0.8} > 0$ for $\alpha = \alpha_{01}$.

We next consider the function $\sqrt{\frac{2}{\pi}}\left(\frac{(\alpha-1)x}{(\nu\tau)^{3/2}} - \frac{\alpha}{\sqrt{\nu\tau}}\right)$, which has the derivative with respect to $x$:

$$\frac{\sqrt{\frac{2}{\pi}}(\alpha-1)}{(\nu\tau)^{3/2}} > 0 . \tag{131}$$

$\square$

**Lemma 34** (Function $\sqrt{\frac{2}{\pi}} \left( \frac{(-1)(\alpha-1)\mu^2\omega^2}{(\nu\tau)^{3/2}} + \frac{-\alpha+\alpha\mu\omega+1}{\sqrt{\nu\tau}} - \alpha\sqrt{\nu\tau} \right)$). *The function*
$\sqrt{\frac{2}{\pi}} \left( \frac{(-1)(\alpha-1)\mu^2\omega^2}{(\nu\tau)^{3/2}} + \frac{-\alpha+\alpha\mu\omega+1}{\sqrt{\nu\tau}} - \alpha\sqrt{\nu\tau} \right) < 0$ *is decreasing in $\nu\tau$ and increasing in $\mu\omega$.*

*Proof.* We define the function

$$\sqrt{\frac{2}{\pi}} \left( \frac{(-1)(\alpha-1)\mu^2\omega^2}{x^{3/2}} + \frac{-\alpha+\alpha\mu\omega+1}{\sqrt{x}} - \alpha\sqrt{x} \right) \tag{132}$$

which has as derivative with respect to $x$:

$$\sqrt{\frac{2}{\pi}} \left( \frac{3(\alpha-1)\mu^2\omega^2}{2x^{5/2}} - \frac{-\alpha+\alpha\mu\omega+1}{2x^{3/2}} - \frac{\alpha}{2\sqrt{x}} \right) = \tag{133}$$
$$\frac{1}{\sqrt{2\pi}x^{5/2}} \left( 3(\alpha-1)\mu^2\omega^2 - x(-\alpha+\alpha\mu\omega+1) - \alpha x^2 \right) .$$

The derivative of the term $3(\alpha-1)\mu^2\omega^2 - x(-\alpha+\alpha\mu\omega+1) - \alpha x^2$ with respect to $x$ is $-1 + \alpha - \mu\omega\alpha - 2\alpha x < 0$, since $2\alpha x > 1.6\alpha$. Therefore the term is maximized with the smallest value for $x$, which is $x = \nu\tau = 0.8 \cdot 0.8$. For $\mu\omega$ we use for each term the value which gives maximal contribution. We obtain an upper bound for the term:

$$3(-0.1 \cdot 0.1)^2(\alpha_{01} - 1) - (0.8 \cdot 0.8)^2\alpha_{01} - 0.8 \cdot 0.8((-0.1 \cdot 0.1)\alpha_{01} - \alpha_{01} + 1) = -0.243569 . \tag{134}$$

Therefore the derivative with respect to $x = \nu\tau$ is smaller than zero and the original function is decreasing in $\nu\tau$

We now consider the derivative with respect to $x = \mu\omega$. The derivative with respect to $x$ of the function

$$\sqrt{\frac{2}{\pi}} \left( -\alpha\sqrt{\nu\tau} - \frac{(\alpha-1)x^2}{(\nu\tau)^{3/2}} + \frac{-\alpha+\alpha x+1}{\sqrt{\nu\tau}} \right) \tag{135}$$

is

$$\frac{\sqrt{\frac{2}{\pi}}(\alpha\nu\tau - 2(\alpha-1)x)}{(\nu\tau)^{3/2}} . \tag{136}$$

Since $-2x(-1+\alpha) + \nu\tau\alpha > -2 \cdot 0.01 \cdot (-1+\alpha_{01}) + 0.8 \cdot 0.8\alpha_{01} > 1.0574 > 0$, the derivative is larger than zero. Consequently, the original function is increasing in $\mu\omega$.

The maximal value is obtained with the minimal $\nu\tau = 0.8 \cdot 0.8$ and the maximal $\mu\omega = 0.1 \cdot 0.1$. The maximal value is

$$\sqrt{\frac{2}{\pi}} \left( \frac{0.1 \cdot 0.1\alpha_{01} - \alpha_{01} + 1}{\sqrt{0.8 \cdot 0.8}} + \frac{0.1^2 0.1^2(-1)(\alpha_{01} - 1)}{(0.8 \cdot 0.8)^{3/2}} - \sqrt{0.8 \cdot 0.8}\alpha_{01} \right) = -1.72296 . \tag{137}$$

Therefore the original function is smaller than zero.                                        □

**Lemma 35** (Function $\sqrt{\frac{2}{\pi}}\left(\frac{(\alpha^2-1)\mu\omega}{(\nu\tau)^{3/2}} - \frac{3\alpha^2}{\sqrt{\nu\tau}}\right)$). *For* $\lambda = \lambda_{01}$ *and* $\alpha = \alpha_{01}$,

$\sqrt{\frac{2}{\pi}}\left(\frac{(\alpha^2-1)\mu\omega}{(\nu\tau)^{3/2}} - \frac{3\alpha^2}{\sqrt{\nu\tau}}\right) < 0$ *and increasing in both* $\nu\tau$ *and* $\mu\omega$.

*Proof.* The derivative of the function

$$\sqrt{\frac{2}{\pi}}\left(\frac{\left(\alpha^2-1\right)\mu\omega}{x^{3/2}} - \frac{3\alpha^2}{\sqrt{x}}\right) \tag{138}$$

with respect to $x$ is

$$\sqrt{\frac{2}{\pi}}\left(\frac{3\alpha^2}{2x^{3/2}} - \frac{3\left(\alpha^2-1\right)\mu\omega}{2x^{5/2}}\right) = \frac{3\left(\alpha^2 x - \left(\alpha^2-1\right)\mu\omega\right)}{\sqrt{2\pi}x^{5/2}} > 0\,, \tag{139}$$

since $\alpha^2 x - \mu\omega(-1+\alpha^2) > \alpha_{01}^2 0.8 \cdot 0.8 - 0.1 \cdot 0.1 \cdot (-1 + \alpha_{01}^2) > 1.77387$

The derivative of the function

$$\sqrt{\frac{2}{\pi}}\left(\frac{\left(\alpha^2-1\right)x}{(\nu\tau)^{3/2}} - \frac{3\alpha^2}{\sqrt{\nu\tau}}\right) \tag{140}$$

with respect to $x$ is

$$\frac{\sqrt{\frac{2}{\pi}}\left(\alpha^2-1\right)}{(\nu\tau)^{3/2}} > 0\,. \tag{141}$$

The maximal function value is obtained by maximal $\nu\tau = 1.5 \cdot 1.25$ and the maximal $\mu\omega = 0.1 \cdot 0.1$. The maximal value is $\sqrt{\frac{2}{\pi}}\left(\frac{0.1\cdot0.1\left(\alpha_{01}^2-1\right)}{(1.5\cdot1.25)^{3/2}} - \frac{3\alpha_{01}^2}{\sqrt{1.5\cdot1.25}}\right) = -4.88869$. Therefore the function is negative. $\square$

**Lemma 36** (Function $\sqrt{\frac{2}{\pi}}\left(\frac{(\alpha^2-1)\mu\omega}{\sqrt{\nu\tau}} - 3\alpha^2\sqrt{\nu\tau}\right)$). *The function* $\sqrt{\frac{2}{\pi}}\left(\frac{(\alpha^2-1)\mu\omega}{\sqrt{\nu\tau}} - 3\alpha^2\sqrt{\nu\tau}\right) < 0$ *is decreasing in* $\nu\tau$ *and increasing in* $\mu\omega$.

*Proof.* The derivative of the function

$$\sqrt{\frac{2}{\pi}}\left(\frac{\left(\alpha^2-1\right)\mu\omega}{\sqrt{x}} - 3\alpha^2\sqrt{x}\right) \tag{142}$$

with respect to $x$ is

$$\sqrt{\frac{2}{\pi}}\left(-\frac{\left(\alpha^2-1\right)\mu\omega}{2x^{3/2}} - \frac{3\alpha^2}{2\sqrt{x}}\right) = \frac{-\left(\alpha^2-1\right)\mu\omega - 3\alpha^2 x}{\sqrt{2\pi}x^{3/2}} < 0\,, \tag{143}$$

since $-3\alpha^2 x - \mu\omega(-1+\alpha^2) < -3\alpha_{01}^2 0.8 \cdot 0.8 + 0.1 \cdot 0.1(-1+\alpha_{01}^2) < -5.35764$.

The derivative of the function

$$\sqrt{\frac{2}{\pi}}\left(\frac{\left(\alpha^2-1\right)x}{\sqrt{\nu\tau}}-3\alpha^2\sqrt{\nu\tau}\right) \tag{144}$$

with respect to $x$ is

$$\frac{\sqrt{\frac{2}{\pi}}\left(\alpha^2-1\right)}{\sqrt{\nu\tau}}>0\,. \tag{145}$$

The maximal function value is obtained for minimal $\nu\tau=0.8\cdot0.8$ and the maximal $\mu\omega=0.1\cdot0.1$. The value is $\sqrt{\frac{2}{\pi}}\left(\frac{0.1\cdot0.1\left(\alpha_{01}^2-1\right)}{\sqrt{0.8\cdot0.8}}-3\sqrt{0.8\cdot0.8}\alpha_{01}^2\right)=-5.34347$. Thus, the function is negative.                                                                                                   $\square$

**Lemma 37** (Function $\nu\tau e^{\frac{(\mu\omega+\nu\tau)^2}{2\nu\tau}}\operatorname{erfc}\left(\frac{\mu\omega+\nu\tau}{\sqrt{2}\sqrt{\nu\tau}}\right)$). *The function $\nu\tau e^{\frac{(\mu\omega+\nu\tau)^2}{2\nu\tau}}\operatorname{erfc}\left(\frac{\mu\omega+\nu\tau}{\sqrt{2}\sqrt{\nu\tau}}\right)>0$ is increasing in $\nu\tau$ and decreasing in $\mu\omega$.*

*Proof.* The derivative of the function

$$xe^{\frac{(\mu\omega+x)^2}{2x}}\operatorname{erfc}\left(\frac{\mu\omega+x}{\sqrt{2}\sqrt{x}}\right) \tag{146}$$

with respect to $x$ is

$$\frac{e^{\frac{(\mu\omega+x)^2}{2x}}\left(x(x+2)-\mu^2\omega^2\right)\operatorname{erfc}\left(\frac{\mu\omega+x}{\sqrt{2}\sqrt{x}}\right)}{2x}+\frac{\mu\omega-x}{\sqrt{2\pi}\sqrt{x}}\,. \tag{147}$$

This derivative is larger than zero, since

$$\frac{e^{\frac{(\mu\omega+\nu\tau)^2}{2\nu\tau}}\left(\nu\tau(\nu\tau+2)-\mu^2\omega^2\right)\operatorname{erfc}\left(\frac{\mu\omega+\nu\tau}{\sqrt{2}\sqrt{\nu\tau}}\right)}{2\nu\tau}+\frac{\mu\omega-\nu\tau}{\sqrt{2\pi}\sqrt{\nu\tau}}> \tag{148}$$

$$\frac{0.4349\left(\nu\tau(\nu\tau+2)-\mu^2\omega^2\right)}{2\nu\tau}+\frac{\mu\omega-\nu\tau}{\sqrt{2\pi}\sqrt{\nu\tau}}>$$

$$\frac{0.5\left(\nu\tau(\nu\tau+2)-\mu^2\omega^2\right)}{\sqrt{2\pi}\nu\tau}+\frac{\mu\omega-\nu\tau}{\sqrt{2\pi}\sqrt{\nu\tau}}=$$

$$\frac{0.5\left(\nu\tau(\nu\tau+2)-\mu^2\omega^2\right)+\sqrt{\nu\tau}(\mu\omega-\nu\tau)}{\sqrt{2\pi}\nu\tau}=$$

$$\frac{-0.5\mu^2\omega^2+\mu\omega\sqrt{\nu\tau}+0.5(\nu\tau)^2-\nu\tau\sqrt{\nu\tau}+\nu\tau}{\sqrt{2\pi}\nu\tau}=$$

$$\frac{-0.5\mu^2\omega^2+\mu\omega\sqrt{\nu\tau}+(0.5\nu\tau-\sqrt{\nu\tau})^2+0.25(\nu\tau)^2}{\sqrt{2\pi}\nu\tau}>0\,.$$

We explain this chain of inequalities:

- The first inequality follows by applying Lemma 23 which says that $e^{\frac{(\mu\omega+\nu\tau)^2}{2\nu\tau}}\operatorname{erfc}\left(\frac{\mu\omega+\nu\tau}{\sqrt{2}\sqrt{\nu\tau}}\right)$ is strictly monotonically decreasing. The minimal value that is larger than 0.4349 is taken on at the maximal values $\nu\tau = 1.5 \cdot 1.25$ and $\mu\omega = 0.1 \cdot 0.1$.

- The second inequality uses $\frac{1}{2}0.4349\sqrt{2\pi} = 0.545066 > 0.5$.

- The equalities are just algebraic reformulations.

- The last inequality follows from $-0.5\mu^2\omega^2 + \mu\omega\sqrt{\nu\tau} + 0.25(\nu\tau)^2 > 0.25(0.8 \cdot 0.8)^2 - 0.5 \cdot (0.1)^2(0.1)^2 - 0.1 \cdot 0.1 \cdot \sqrt{0.8 \cdot 0.8} = 0.09435 > 0$.

Therefore the function is increasing in $\nu\tau$.

Decreasing in $\mu\omega$ follows from decreasing of $e^{x^2}\operatorname{erfc}(x)$ according to Lemma 23. Positivity follows form the fact that erfc and the exponential function are positive and that $\nu\tau > 0$.    □

**Lemma 38** (Function $\nu\tau e^{\frac{(\mu\omega+2\nu\tau)^2}{2\nu\tau}}\operatorname{erfc}\left(\frac{\mu\omega+2\nu\tau}{\sqrt{2}\sqrt{\nu\tau}}\right)$)**.** *The function* $\nu\tau e^{\frac{(\mu\omega+2\nu\tau)^2}{2\nu\tau}}\operatorname{erfc}\left(\frac{\mu\omega+2\nu\tau}{\sqrt{2}\sqrt{\nu\tau}}\right) > 0$ *is increasing in* $\nu\tau$ *and decreasing in* $\mu\omega$.

*Proof.* The derivative of the function

$$x e^{\frac{(\mu\omega+2x)^2}{2x}}\operatorname{erfc}\left(\frac{\mu\omega+2x}{\sqrt{2}\sqrt{2x}}\right) \tag{149}$$

is

$$\frac{e^{\frac{(\mu\omega+2x)^2}{4x}}\left(\sqrt{\pi}e^{\frac{(\mu\omega+2x)^2}{4x}}\left(2x(2x+1)-\mu^2\omega^2\right)\operatorname{erfc}\left(\frac{\mu\omega+2x}{2\sqrt{x}}\right)+\sqrt{x}(\mu\omega-2x)\right)}{2\sqrt{\pi}x} . \tag{150}$$

We only have to determine the sign of $\sqrt{\pi}e^{\frac{(\mu\omega+2x)^2}{4x}}\left(2x(2x+1)-\mu^2\omega^2\right)\operatorname{erfc}\left(\frac{\mu\omega+2x}{2\sqrt{x}}\right)+\sqrt{x}(\mu\omega-2x)$ since all other factors are obviously larger than zero.

This derivative is larger than zero, since

$$\sqrt{\pi}e^{\frac{(\mu\omega+2\nu\tau)^2}{4\nu\tau}}\left(2\nu\tau(2\nu\tau+1)-\mu^2\omega^2\right)\operatorname{erfc}\left(\frac{\mu\omega+2\nu\tau}{2\sqrt{\nu\tau}}\right)+\sqrt{\nu\tau}(\mu\omega-2\nu\tau) > \tag{151}$$

$$0.463979\left(2\nu\tau(2\nu\tau+1)-\mu^2\omega^2\right)+\sqrt{\nu\tau}(\mu\omega-2\nu\tau) =$$
$$-0.463979\mu^2\omega^2 + \mu\omega\sqrt{\nu\tau} + 1.85592(\nu\tau)^2 + 0.927958\nu\tau - 2\nu\tau\sqrt{\nu\tau} =$$
$$\mu\omega\left(\sqrt{\nu\tau}-0.463979\mu\omega\right)+0.85592(\nu\tau)^2+\left(\nu\tau-\sqrt{\nu\tau}\right)^2-0.0720421\nu\tau > 0 .$$

We explain this chain of inequalities:

- The first inequality follows by applying Lemma 23 which says that $e^{\frac{(\mu\omega+2\nu\tau)^2}{2\nu\tau}}\operatorname{erfc}\left(\frac{\mu\omega+2\nu\tau}{\sqrt{2}\sqrt{\nu\tau}}\right)$ is strictly monotonically decreasing. The minimal value that is larger than 0.261772 is taken on at the maximal values $\nu\tau = 1.5 \cdot 1.25$ and $\mu\omega = 0.1 \cdot 0.1$. $0.261772\sqrt{\pi} > 0.463979$.

- The equalities are just algebraic reformulations.

- The last inequality follows from $\mu\omega\left(\sqrt{\nu\tau}-0.463979\mu\omega\right)+0.85592(\nu\tau)^2-0.0720421\nu\tau > 0.85592\cdot(0.8\cdot0.8)^2-0.1\cdot0.1\left(\sqrt{1.5\cdot1.25}+0.1\cdot0.1\cdot0.463979\right)-0.0720421\cdot1.5\cdot1.25 > 0.201766$.

Therefore the function is increasing in $\nu\tau$.

Decreasing in $\mu\omega$ follows from decreasing of $e^{x^2}\operatorname{erfc}(x)$ according to Lemma 23. Positivity follows from the fact that erfc and the exponential function are positive and that $\nu\tau > 0$.   $\square$

**Lemma 39** (Bounds on the Derivatives). *The following bounds on the absolute values of the derivatives of the Jacobian entries $\mathcal{J}_{11}(\mu,\omega,\nu,\tau,\lambda,\alpha)$, $\mathcal{J}_{12}(\mu,\omega,\nu,\tau,\lambda,\alpha)$, $\mathcal{J}_{21}(\mu,\omega,\nu,\tau,\lambda,\alpha)$, and $\mathcal{J}_{22}(\mu,\omega,\nu,\tau,\lambda,\alpha)$ with respect to $\mu$, $\omega$, $\nu$, and $\tau$ hold:*

$$\left|\frac{\partial\mathcal{J}_{11}}{\partial\mu}\right| < 0.0031049101995398316 \tag{152}$$

$$\left|\frac{\partial\mathcal{J}_{11}}{\partial\omega}\right| < 1.055872374194189$$

$$\left|\frac{\partial\mathcal{J}_{11}}{\partial\nu}\right| < 0.031242911235461816$$

$$\left|\frac{\partial\mathcal{J}_{11}}{\partial\tau}\right| < 0.03749149348255419$$

$$\left|\frac{\partial\mathcal{J}_{12}}{\partial\mu}\right| < 0.031242911235461816$$

$$\left|\frac{\partial\mathcal{J}_{12}}{\partial\omega}\right| < 0.031242911235461816$$

$$\left|\frac{\partial\mathcal{J}_{12}}{\partial\nu}\right| < 0.21232788238624354$$

$$\left|\frac{\partial\mathcal{J}_{12}}{\partial\tau}\right| < 0.2124377655377270$$

$$\left|\frac{\partial\mathcal{J}_{21}}{\partial\mu}\right| < 0.02220441024325437$$

$$\left|\frac{\partial\mathcal{J}_{21}}{\partial\omega}\right| < 1.146955401845684$$

$$\left|\frac{\partial\mathcal{J}_{21}}{\partial\nu}\right| < 0.14983446469110305$$

$$\left|\frac{\partial\mathcal{J}_{21}}{\partial\tau}\right| < 0.17980135762932363$$

$$\left|\frac{\partial\mathcal{J}_{22}}{\partial\mu}\right| < 0.14983446469110305$$

$$\left| \frac{\partial \mathcal{J}_{22}}{\partial \omega} \right| < 0.14983446469110305$$

$$\left| \frac{\partial \mathcal{J}_{22}}{\partial \nu} \right| < 1.805740052651535$$

$$\left| \frac{\partial \mathcal{J}_{22}}{\partial \tau} \right| < 2.396685907216327$$

*Proof.* For each derivative we compute a lower and an upper bound and take the maximum of the absolute value. A lower bound is determined by minimizing the single terms of the functions that represents the derivative. An upper bound is determined by maximizing the single terms of the functions that represent the derivative. Terms can be combined to larger terms for which the maximum and the minimum must be known. We apply many previous lemmata which state properties of functions representing single or combined terms. The more terms are combined, the tighter the bounds can be made.

Next we go through all the derivatives, where we use Lemma 25, Lemma 26, Lemma 27, Lemma 28, Lemma 29, Lemma 30, Lemma 21, and Lemma 23 without citing. Furthermore, we use the bounds on the simple expressions $t_1 1, t_2 2$, ..., and $T_4$ as defined the aforementioned lemmata:

- $\frac{\partial \mathcal{J}_{11}}{\partial \mu}$

  We use Lemma 31 and consider the expression $\alpha e^{\frac{(\mu\omega+\nu\tau)^2}{2\nu\tau}} \operatorname{erfc}\left( \frac{\mu\omega+\nu\tau}{\sqrt{2}\sqrt{\nu\tau}} \right) - \frac{\sqrt{\frac{2}{\pi}}(\alpha-1)}{\sqrt{\nu\tau}}$ in brackets. An upper bound on the maximum of is

$$\alpha_{01} e^{t_1^2} \operatorname{erfc}(t_1) - \frac{\sqrt{\frac{2}{\pi}}(\alpha_{01}-1)}{\sqrt{T_{22}}} = 0.591017 \,. \tag{153}$$

  A lower bound on the minimum is

$$\alpha_{01} e^{T_1^2} \operatorname{erfc}(T_1) - \frac{\sqrt{\frac{2}{\pi}}(\alpha_{01}-1)}{\sqrt{t_{22}}} = 0.056318 \,. \tag{154}$$

  Thus, an upper bound on the maximal absolute value is

$$\frac{1}{2}\lambda_{01}\omega_{\max}^2 e^{t_4} \left( \alpha_{01} e^{t_1^2} \operatorname{erfc}(t_1) - \frac{\sqrt{\frac{2}{\pi}}(\alpha_{01}-1)}{\sqrt{T_{22}}} \right) = 0.0031049101995398316 \,. \tag{155}$$

- $\frac{\partial \mathcal{J}_{11}}{\partial \omega}$

  We use Lemma 31 and consider the expression $\frac{\sqrt{\frac{2}{\pi}}(\alpha-1)\mu\omega}{\sqrt{\nu\tau}} - \alpha(\mu\omega+1)e^{\frac{(\mu\omega+\nu\tau)^2}{2\nu\tau}} \operatorname{erfc}\left( \frac{\mu\omega+\nu\tau}{\sqrt{2}\sqrt{\nu\tau}} \right)$ in brackets.

  An upper bound on the maximum is

$$\frac{\sqrt{\frac{2}{\pi}}(\alpha_{01}-1)T_{11}}{\sqrt{t_{22}}} - \alpha_{01}(t_{11}+1)e^{T_1^2} \operatorname{erfc}(T_1) = -0.713808 \,. \tag{156}$$

A lower bound on the minimum is

$$
\frac{\sqrt{\frac{2}{\pi}}(\alpha_{01} - 1)t_{11}}{\sqrt{t_{22}}} - \alpha_{01}(T_{11} + 1)e^{t_1^2}\operatorname{erfc}(t_1) \;=\; -0.99987\,. \tag{157}
$$

This term is subtracted, and $2 - \operatorname{erfc}(x) > 0$, therefore we have to use the minimum and the maximum for the argument of $\operatorname{erfc}$.

Thus, an upper bound on the maximal absolute value is

$$
\frac{1}{2}\lambda_{01}\left(-e^{t_4}\left(\frac{\sqrt{\frac{2}{\pi}}(\alpha_{01} - 1)t_{11}}{\sqrt{t_{22}}} - \alpha_{01}(T_{11} + 1)e^{t_1^2}\operatorname{erfc}(t_1)\right) - \right. \tag{158}
$$
$$
\left. \operatorname{erfc}(T_3) + 2\right) \;=\; 1.055872374194189\,.
$$

- $\frac{\partial \mathcal{J}_{11}}{\partial \nu}$

  We consider the term in brackets

$$
\alpha e^{\frac{(\mu\omega + \nu\tau)^2}{2\nu\tau}}\operatorname{erfc}\left(\frac{\mu\omega + \nu\tau}{\sqrt{2}\sqrt{\nu\tau}}\right) + \sqrt{\frac{2}{\pi}}\left(\frac{(\alpha - 1)\mu\omega}{(\nu\tau)^{3/2}} - \frac{\alpha}{\sqrt{\nu\tau}}\right)\,. \tag{159}
$$

  We apply Lemma 33 for the first sub-term. An upper bound on the maximum is

$$
\alpha_{01}e^{t_1^2}\operatorname{erfc}(t_1) + \sqrt{\frac{2}{\pi}}\left(\frac{(\alpha_{01} - 1)T_{11}}{T_{22}^{3/2}} - \frac{\alpha_{01}}{\sqrt{T_{22}}}\right) \;=\; 0.0104167\,. \tag{160}
$$

  A lower bound on the minimum is

$$
\alpha_{01}e^{T_1^2}\operatorname{erfc}(T_1) + \sqrt{\frac{2}{\pi}}\left(\frac{(\alpha_{01} - 1)t_{11}}{t_{22}^{3/2}} - \frac{\alpha_{01}}{\sqrt{t_{22}}}\right) \;=\; -0.95153\,. \tag{161}
$$

  Thus, an upper bound on the maximal absolute value is

$$
-\frac{1}{4}\lambda_{01}\tau_{\max}\omega_{\max}e^{t_4}\left(\alpha_{01}e^{T_1^2}\operatorname{erfc}(T_1) + \sqrt{\frac{2}{\pi}}\left(\frac{(\alpha_{01} - 1)t_{11}}{t_{22}^{3/2}} - \frac{\alpha_{01}}{\sqrt{t_{22}}}\right)\right) \;=\; \tag{162}
$$
$$
0.031242911235461816\,.
$$

- $\frac{\partial \mathcal{J}_{11}}{\partial \tau}$

  We use the results of item $\frac{\partial \mathcal{J}_{11}}{\partial \nu}$ were the brackets are only differently scaled. Thus, an upper bound on the maximal absolute value is

$$
-\frac{1}{4}\lambda_{01}\nu_{\max}\omega_{\max}e^{t_4}\left(\alpha_{01}e^{T_1^2}\operatorname{erfc}(T_1) + \sqrt{\frac{2}{\pi}}\left(\frac{(\alpha_{01} - 1)t_{11}}{t_{22}^{3/2}} - \frac{\alpha_{01}}{\sqrt{t_{22}}}\right)\right) \;=\; \tag{163}
$$
$$
0.03749149348255419\,.
$$

- $\frac{\partial \mathcal{J}_{12}}{\partial \mu}$

  Since $\frac{\partial \mathcal{J}_{12}}{\partial \mu} = \frac{\partial \mathcal{J}_{11}}{\partial \nu}$, an upper bound on the maximal absolute value is

  $$-\frac{1}{4}\lambda_{01}\tau_{\max}\omega_{\max}e^{t_4}\left(\alpha_{01}e^{T_1^2}\operatorname{erfc}(T_1) + \sqrt{\frac{2}{\pi}}\left(\frac{(\alpha_{01}-1)t_{11}}{t_{22}^{3/2}} - \frac{\alpha_{01}}{\sqrt{t_{22}}}\right)\right) = \quad (164)$$
  $$0.031242911235461816 \,.$$

- $\frac{\partial \mathcal{J}_{12}}{\partial \omega}$

  We use the results of item $\frac{\partial \mathcal{J}_{11}}{\partial \nu}$ were the brackets are only differently scaled. Thus, an upper bound on the maximal absolute value is

  $$-\frac{1}{4}\lambda_{01}\mu_{\max}\tau_{\max}e^{t_4}\left(\alpha_{01}e^{T_1^2}\operatorname{erfc}(T_1) + \sqrt{\frac{2}{\pi}}\left(\frac{(\alpha_{01}-1)t_{11}}{t_{22}^{3/2}} - \frac{\alpha_{01}}{\sqrt{t_{22}}}\right)\right) = \quad (165)$$
  $$0.031242911235461816 \,.$$

- $\frac{\partial \mathcal{J}_{12}}{\partial \nu}$

  For the second term in brackets, we see that $\alpha_{01}\tau_{\min}^2 e^{T_1^2}\operatorname{erfc}(T_1) = 0.465793$ and $\alpha_{01}\tau_{\max}^2 e^{t_1^2}\operatorname{erfc}(t_1) = 1.53644$.

  We now check different values for

  $$\sqrt{\frac{2}{\pi}}\left(\frac{(-1)(\alpha-1)\mu^2\omega^2}{\nu^{5/2}\sqrt{\tau}} + \frac{\sqrt{\tau}(\alpha+\alpha\mu\omega-1)}{\nu^{3/2}} - \frac{\alpha\tau^{3/2}}{\sqrt{\nu}}\right), \quad (166)$$

  where we maximize or minimize all single terms.

  A lower bound on the minimum of this expression is

  $$\sqrt{\frac{2}{\pi}}\left(\frac{(-1)(\alpha_{01}-1)\mu_{\max}^2\omega_{\max}^2}{\nu_{\min}^{5/2}\sqrt{\tau_{\min}}} + \frac{\sqrt{\tau_{\min}}(\alpha_{01}+\alpha_{01}t_{11}-1)}{\nu_{\max}^{3/2}} - \frac{\alpha_{01}\tau_{\max}^{3/2}}{\sqrt{\nu_{\min}}}\right) = \quad (167)$$
  $$- 1.83112 \,.$$

  An upper bound on the maximum of this expression is

  $$\sqrt{\frac{2}{\pi}}\left(\frac{(-1)(\alpha_{01}-1)\mu_{\min}^2\omega_{\min}^2}{\nu_{\max}^{5/2}\sqrt{\tau_{\max}}} + \frac{\sqrt{\tau_{\max}}(\alpha_{01}+\alpha_{01}T_{11}-1)}{\nu_{\min}^{3/2}} - \frac{\alpha_{01}\tau_{\min}^{3/2}}{\sqrt{\nu_{\max}}}\right) = \quad (168)$$
  $$0.0802158 \,.$$

  An upper bound on the maximum is

  $$\frac{1}{8}\lambda_{01}e^{t_4}\left(\sqrt{\frac{2}{\pi}}\left(\frac{(-1)(\alpha_{01}-1)\mu_{\min}^2\omega_{\min}^2}{\nu_{\max}^{5/2}\sqrt{\tau_{\max}}} - \frac{\alpha_{01}\tau_{\min}^{3/2}}{\sqrt{\nu_{\max}}} + \right.\right. \quad (169)$$
  $$\left.\left.\frac{\sqrt{\tau_{\max}}(\alpha_{01}+\alpha_{01}T_{11}-1)}{\nu_{\min}^{3/2}}\right) + \alpha_{01}\tau_{\max}^2 e^{t_1^2}\operatorname{erfc}(t_1)\right) = 0.212328 \,.$$

A lower bound on the minimum is

$$\frac{1}{8}\lambda_{01}e^{t_4}\left(\alpha_{01}\tau_{\min}^2 e^{T_1^2}\operatorname{erfc}(T_1) + \right. \tag{170}$$

$$\left. \sqrt{\frac{2}{\pi}}\left(\frac{(-1)(\alpha_{01}-1)\mu_{\max}^2\omega_{\max}^2}{\nu_{\min}^{5/2}\sqrt{\tau_{\min}}} + \frac{\sqrt{\tau_{\min}}(\alpha_{01}+\alpha_{01}t_{11}-1)}{\nu_{\max}^{3/2}} - \frac{\alpha_{01}\tau_{\max}^{3/2}}{\sqrt{\nu_{\min}}}\right)\right) =$$

$$-0.179318 .$$

Thus, an upper bound on the maximal absolute value is

$$\frac{1}{8}\lambda_{01}e^{t_4}\left(\sqrt{\frac{2}{\pi}}\left(\frac{(-1)(\alpha_{01}-1)\mu_{\min}^2\omega_{\min}^2}{\nu_{\max}^{5/2}\sqrt{\tau_{\max}}} - \frac{\alpha_{01}\tau_{\min}^{3/2}}{\sqrt{\nu_{\max}}} + \right.\right. \tag{171}$$

$$\left.\left. \frac{\sqrt{\tau_{\max}}(\alpha_{01}+\alpha_{01}T_{11}-1)}{\nu_{\min}^{3/2}}\right) + \alpha_{01}\tau_{\max}^2 e^{t_1^2}\operatorname{erfc}(t_1)\right) = 0.21232788238624354 .$$

- $\frac{\partial\mathcal{J}_{12}}{\partial\tau}$

We use Lemma 34 to obtain an upper bound on the maximum of the expression of the lemma:

$$\sqrt{\frac{2}{\pi}}\left(\frac{0.1^2\cdot 0.1^2(-1)(\alpha_{01}-1)}{(0.8\cdot 0.8)^{3/2}} - \sqrt{0.8\cdot 0.8}\alpha_{01} + \frac{(0.1\cdot 0.1)\alpha_{01}-\alpha_{01}+1}{\sqrt{0.8\cdot 0.8}}\right) = -1.72296 . \tag{172}$$

We use Lemma 34 to obtain an lower bound on the minimum of the expression of the lemma:

$$\sqrt{\frac{2}{\pi}}\left(\frac{0.1^2\cdot 0.1^2(-1)(\alpha_{01}-1)}{(1.5\cdot 1.25)^{3/2}} - \sqrt{1.5\cdot 1.25}\alpha_{01} + \frac{(-0.1\cdot 0.1)\alpha_{01}-\alpha_{01}+1}{\sqrt{1.5\cdot 1.25}}\right) = -2.2302 . \tag{173}$$

Next we apply Lemma 37 for the expression $\nu\tau e^{\frac{(\mu\omega+\nu\tau)^2}{2\nu\tau}}\operatorname{erfc}\left(\frac{\mu\omega+\nu\tau}{\sqrt{2}\sqrt{\nu\tau}}\right)$. We use Lemma 37 to obtain an upper bound on the maximum of this expression:

$$1.5\cdot 1.25 e^{\frac{(1.5\cdot 1.25-0.1\cdot 0.1)^2}{2\cdot 1.5\cdot 1.25}}\alpha_{01}\operatorname{erfc}\left(\frac{1.5\cdot 1.25-0.1\cdot 0.1}{\sqrt{2}\sqrt{1.5\cdot 1.25}}\right) = 1.37381 . \tag{174}$$

We use Lemma 37 to obtain an lower bound on the minimum of this expression:

$$0.8\cdot 0.8 e^{\frac{(0.8\cdot 0.8+0.1\cdot 0.1)^2}{2\cdot 0.8\cdot 0.8}}\alpha_{01}\operatorname{erfc}\left(\frac{0.8\cdot 0.8+0.1\cdot 0.1}{\sqrt{2}\sqrt{0.8\cdot 0.8}}\right) = 0.620462 . \tag{175}$$

Next we apply Lemma 23 for $2\alpha e^{\frac{(\mu\omega+\nu\tau)^2}{2\nu\tau}}\operatorname{erfc}\left(\frac{\mu\omega+\nu\tau}{\sqrt{2}\sqrt{\nu\tau}}\right)$. An upper bound on this expression is

$$2e^{\frac{(0.8\cdot 0.8-0.1\cdot 0.1)^2}{2 0.8\cdot 0.8}}\alpha_{01}\operatorname{erfc}\left(\frac{0.8\cdot 0.8-0.1\cdot 0.1}{\sqrt{2}\sqrt{0.8\cdot 0.8}}\right) = 1.96664 . \tag{176}$$

A lower bound on this expression is

$$2e^{\frac{(1.5\cdot1.25+0.1\cdot0.1)^2}{2\cdot1.5\cdot1.25}}\alpha_{01}\operatorname{erfc}\left(\frac{1.5\cdot1.25+0.1\cdot0.1}{\sqrt{2}\sqrt{1.5\cdot1.25}}\right) \;=\; 1.4556\,. \tag{177}$$

The sum of the minimal values of the terms is $-2.23019+0.62046+1.45560 = -0.154133$.

The sum of the maximal values of the terms is $-1.72295 + 1.37380 + 1.96664 = 1.61749$.

Thus, an upper bound on the maximal absolute value is

$$\frac{1}{8}\lambda_{01}e^{t_4}\left(\alpha_{01}T_{22}e^{\frac{(t_{11}+T_{22})^2}{2T_{22}}}\operatorname{erfc}\left(\frac{t_{11}+T_{22}}{\sqrt{2}\sqrt{T_{22}}}\right)\;+\right. \tag{178}$$
$$2\alpha_{01}e^{t_1^2}\operatorname{erfc}(t_1) + \sqrt{\frac{2}{\pi}}\left(-\frac{(\alpha_{01}-1)T_{11}^2}{t_{22}^{3/2}}+\frac{-\alpha_{01}+\alpha_{01}T_{11}+1}{\sqrt{t_{22}}}\;-\right.$$
$$\left.\left. \alpha_{01}\sqrt{t_{22}}\right)\right) \;=\; 0.2124377655377270\,.$$

- $\frac{\partial\mathcal{J}_{21}}{\partial\mu}$

  An upper bound on the maximum is

$$\lambda_{01}^2\omega_{\max}^2\left(\alpha_{01}^2 e^{T_1^2}\left(-e^{-T_4}\right)\operatorname{erfc}(T_1) + 2\alpha_{01}^2 e^{t_2^2}e^{t_4}\operatorname{erfc}(t_2)\;- \tag{179}$$
$$\operatorname{erfc}(T_3) + 2\right) \;=\; 0.0222044\,.$$

  A upper bound on the absolute minimum is

$$\lambda_{01}^2\omega_{\max}^2\left(\alpha_{01}^2 e^{t_1^2}\left(-e^{-t_4}\right)\operatorname{erfc}(t_1) + 2\alpha_{01}^2 e^{T_2^2}e^{T_4}\operatorname{erfc}(T_2)\;- \tag{180}$$
$$\operatorname{erfc}(t_3) + 2\right) \;=\; 0.00894889\,.$$

  Thus, an upper bound on the maximal absolute value is

$$\lambda_{01}^2\omega_{\max}^2\left(\alpha_{01}^2 e^{T_1^2}\left(-e^{-T_4}\right)\operatorname{erfc}(T_1) + 2\alpha_{01}^2 e^{t_2^2}e^{t_4}\operatorname{erfc}(t_2)\;- \tag{181}$$
$$\operatorname{erfc}(T_3) + 2\right) \;=\; 0.02220441024325437\,.$$

- $\frac{\partial\mathcal{J}_{21}}{\partial\omega}$

  An upper bound on the maximum is

$$\lambda_{01}^2\left(\alpha_{01}^2(2T_{11}+1)e^{t_2^2}e^{-t_4}\operatorname{erfc}(t_2) + 2T_{11}(2-\operatorname{erfc}(T_3))\;+ \tag{182}$$
$$\alpha_{01}^2(t_{11}+1)e^{T_1^2}\left(-e^{-T_4}\right)\operatorname{erfc}(T_1) + \sqrt{\frac{2}{\pi}}\sqrt{T_{22}}e^{-t_4}\right) \;=\; 1.14696\,.$$

  A lower bound on the minimum is

$$\lambda_{01}^2\left(\alpha_{01}^2(T_{11}+1)e^{t_1^2}\left(-e^{-t_4}\right)\operatorname{erfc}(t_1)\;+ \tag{183}$$
$$\alpha_{01}^2(2t_{11}+1)e^{T_2^2}e^{-T_4}\operatorname{erfc}(T_2) + 2t_{11}(2-\operatorname{erfc}(T_3))+$$

$$\sqrt{\frac{2}{\pi}}\sqrt{t_{22}}e^{-T_4}\Big) \;=\; -0.359403 \;.$$

Thus, an upper bound on the maximal absolute value is

$$\lambda_{01}^2\left(\alpha_{01}^2(2T_{11}+1)e^{t_2^2}e^{-t_4}\operatorname{erfc}(t_2)+2T_{11}(2-\operatorname{erfc}(T_3))+\right. \tag{184}$$

$$\alpha_{01}^2(t_{11}+1)e^{T_1^2}\left(-e^{-T_4}\right)\operatorname{erfc}(T_1)+\sqrt{\frac{2}{\pi}}\sqrt{T_{22}}e^{-t_4}\Big) \;=\; 1.146955401845684 \;.$$

- $\frac{\partial \mathcal{J}_{21}}{\partial \nu}$

  An upper bound on the maximum is

$$\frac{1}{2}\lambda_{01}^2\tau_{\max}\omega_{\max}e^{-t_4}\left(\alpha_{01}^2\left(-e^{T_1^2}\right)\operatorname{erfc}(T_1)+4\alpha_{01}^2e^{t_2^2}\operatorname{erfc}(t_2)+\right. \tag{185}$$

$$\frac{\sqrt{\frac{2}{\pi}}(-1)\left(\alpha_{01}^2-1\right)}{\sqrt{T_{22}}}\Bigg) \;=\; 0.149834 \;.$$

  A lower bound on the minimum is

$$\frac{1}{2}\lambda_{01}^2\tau_{\max}\omega_{\max}e^{-t_4}\left(\alpha_{01}^2\left(-e^{t_1^2}\right)\operatorname{erfc}(t_1)+4\alpha_{01}^2e^{T_2^2}\operatorname{erfc}(T_2)+\right. \tag{186}$$

$$\frac{\sqrt{\frac{2}{\pi}}(-1)\left(\alpha_{01}^2-1\right)}{\sqrt{t_{22}}}\Bigg) \;=\; -0.0351035 \;.$$

  Thus, an upper bound on the maximal absolute value is

$$\frac{1}{2}\lambda_{01}^2\tau_{\max}\omega_{\max}e^{-t_4}\left(\alpha_{01}^2\left(-e^{T_1^2}\right)\operatorname{erfc}(T_1)+4\alpha_{01}^2e^{t_2^2}\operatorname{erfc}(t_2)+\right. \tag{187}$$

$$\frac{\sqrt{\frac{2}{\pi}}(-1)\left(\alpha_{01}^2-1\right)}{\sqrt{T_{22}}}\Bigg) \;=\; 0.14983446469110305 \;.$$

- $\frac{\partial \mathcal{J}_{21}}{\partial \tau}$

  An upper bound on the maximum is

$$\frac{1}{2}\lambda_{01}^2\nu_{\max}\omega_{\max}e^{-t_4}\left(\alpha_{01}^2\left(-e^{T_1^2}\right)\operatorname{erfc}(T_1)+4\alpha_{01}^2e^{t_2^2}\operatorname{erfc}(t_2)+\right. \tag{188}$$

$$\frac{\sqrt{\frac{2}{\pi}}(-1)\left(\alpha_{01}^2-1\right)}{\sqrt{T_{22}}}\Bigg) \;=\; 0.179801 \;.$$

  A lower bound on the minimum is

$$\frac{1}{2}\lambda_{01}^2\nu_{\max}\omega_{\max}e^{-t_4}\left(\alpha_{01}^2\left(-e^{t_1^2}\right)\operatorname{erfc}(t_1)+4\alpha_{01}^2e^{T_2^2}\operatorname{erfc}(T_2)+\right. \tag{189}$$

$$\left. \frac{\sqrt{\frac{2}{\pi}}(-1)\left(\alpha_{01}^2 - 1\right)}{\sqrt{t_{22}}} \right) = -0.0421242 .$$

Thus, an upper bound on the maximal absolute value is

$$\frac{1}{2}\lambda_{01}^2 \nu_{\max}\omega_{\max}e^{-t_4}\left(\alpha_{01}^2\left(-e^{T_1^2}\right)\operatorname{erfc}(T_1) + 4\alpha_{01}^2 e^{t_2^2}\operatorname{erfc}(t_2) + \tag{190}$$

$$\left. \frac{\sqrt{\frac{2}{\pi}}(-1)\left(\alpha_{01}^2 - 1\right)}{\sqrt{T_{22}}} \right) = 0.17980135762932363 .$$

- $\frac{\partial \mathcal{J}_{22}}{\partial \mu}$

  We use the fact that $\frac{\partial \mathcal{J}_{22}}{\partial \mu} = \frac{\partial \mathcal{J}_{21}}{\partial \nu}$. Thus, an upper bound on the maximal absolute value is

$$\frac{1}{2}\lambda_{01}^2 \tau_{\max}\omega_{\max}e^{-t_4}\left(\alpha_{01}^2\left(-e^{T_1^2}\right)\operatorname{erfc}(T_1) + 4\alpha_{01}^2 e^{t_2^2}\operatorname{erfc}(t_2) + \tag{191}$$

$$\left. \frac{\sqrt{\frac{2}{\pi}}(-1)\left(\alpha_{01}^2 - 1\right)}{\sqrt{T_{22}}} \right) = 0.14983446469110305 .$$

- $\frac{\partial \mathcal{J}_{22}}{\partial \omega}$

  An upper bound on the maximum is

$$\frac{1}{2}\lambda_{01}^2 \mu_{\max}\tau_{\max}e^{-t_4}\left(\alpha_{01}^2\left(-e^{T_1^2}\right)\operatorname{erfc}(T_1) + 4\alpha_{01}^2 e^{t_2^2}\operatorname{erfc}(t_2) + \tag{192}$$

$$\left. \frac{\sqrt{\frac{2}{\pi}}(-1)\left(\alpha_{01}^2 - 1\right)}{\sqrt{T_{22}}} \right) = 0.149834 .$$

A lower bound on the minimum is

$$\frac{1}{2}\lambda_{01}^2 \mu_{\max}\tau_{\max}e^{-t_4}\left(\alpha_{01}^2\left(-e^{t_1^2}\right)\operatorname{erfc}(t_1) + 4\alpha_{01}^2 e^{T_2^2}\operatorname{erfc}(T_2) + \tag{193}$$

$$\left. \frac{\sqrt{\frac{2}{\pi}}(-1)\left(\alpha_{01}^2 - 1\right)}{\sqrt{t_{22}}} \right) = -0.0351035 .$$

Thus, an upper bound on the maximal absolute value is

$$\frac{1}{2}\lambda_{01}^2 \mu_{\max}\tau_{\max}e^{-t_4}\left(\alpha_{01}^2\left(-e^{T_1^2}\right)\operatorname{erfc}(T_1) + 4\alpha_{01}^2 e^{t_2^2}\operatorname{erfc}(t_2) + \tag{194}$$

$$\left. \frac{\sqrt{\frac{2}{\pi}}(-1)\left(\alpha_{01}^2 - 1\right)}{\sqrt{T_{22}}} \right) = 0.14983446469110305 .$$

- $\frac{\partial \mathcal{J}_{22}}{\partial \nu}$

We apply Lemma 35 to the expression $\sqrt{\frac{2}{\pi}}\left(\frac{(\alpha^2-1)\mu\omega}{(\nu\tau)^{3/2}} - \frac{3\alpha^2}{\sqrt{\nu\tau}}\right)$. Using Lemma 35, an upper bound on the maximum is

$$
\frac{1}{4}\lambda_{01}^2\tau_{\max}^2 e^{-t_4}\left(\alpha_{01}^2\left(-e^{T_1^2}\right)\operatorname{erfc}(T_1) + 8\alpha_{01}^2 e^{t_2^2}\operatorname{erfc}(t_2) + \tag{195}
$$
$$
\sqrt{\frac{2}{\pi}}\left(\frac{\left(\alpha_{01}^2 - 1\right)T_{11}}{T_{22}^{3/2}} - \frac{3\alpha_{01}^2}{\sqrt{T_{22}}}\right)\right) = 1.19441\ .
$$

Using Lemma 35, a lower bound on the minimum is

$$
\frac{1}{4}\lambda_{01}^2\tau_{\max}^2 e^{-t_4}\left(\alpha_{01}^2\left(-e^{t_1^2}\right)\operatorname{erfc}(t_1) + 8\alpha_{01}^2 e^{T_2^2}\operatorname{erfc}(T_2) + \tag{196}
$$
$$
\sqrt{\frac{2}{\pi}}\left(\frac{\left(\alpha_{01}^2 - 1\right)t_{11}}{t_{22}^{3/2}} - \frac{3\alpha_{01}^2}{\sqrt{t_{22}}}\right)\right) = -1.80574\ .
$$

Thus, an upper bound on the maximal absolute value is

$$
-\frac{1}{4}\lambda_{01}^2\tau_{\max}^2 e^{-t_4}\left(\alpha_{01}^2\left(-e^{t_1^2}\right)\operatorname{erfc}(t_1) + 8\alpha_{01}^2 e^{T_2^2}\operatorname{erfc}(T_2) + \tag{197}
$$
$$
\sqrt{\frac{2}{\pi}}\left(\frac{\left(\alpha_{01}^2 - 1\right)t_{11}}{t_{22}^{3/2}} - \frac{3\alpha_{01}^2}{\sqrt{t_{22}}}\right)\right) = 1.805740052651535\ .
$$

- $\frac{\partial \mathcal{J}_{22}}{\partial \tau}$

We apply Lemma 36 to the expression $\sqrt{\frac{2}{\pi}}\left(\frac{(\alpha^2-1)\mu\omega}{\sqrt{\nu\tau}} - 3\alpha^2\sqrt{\nu\tau}\right)$.

We apply Lemma 37 to the expression $\nu\tau e^{\frac{(\mu\omega+\nu\tau)^2}{2\nu\tau}}\operatorname{erfc}\left(\frac{\mu\omega+\nu\tau}{\sqrt{2}\sqrt{\nu\tau}}\right)$. We apply Lemma 38 to the expression $\nu\tau e^{\frac{(\mu\omega+2\nu\tau)^2}{2\nu\tau}}\operatorname{erfc}\left(\frac{\mu\omega+2\nu\tau}{\sqrt{2}\sqrt{\nu\tau}}\right)$.

We combine the results of these lemmata to obtain an upper bound on the maximum:

$$
\frac{1}{4}\lambda_{01}^2\left(-\alpha_{01}^2 t_{22} e^{-T_4} e^{\frac{(T_{11}+t_{22})^2}{2t_{22}}}\operatorname{erfc}\left(\frac{T_{11}+t_{22}}{\sqrt{2}\sqrt{t_{22}}}\right) + \tag{198}
$$
$$
8\alpha_{01}^2 T_{22} e^{-t_4} e^{\frac{(t_{11}+2T_{22})^2}{2T_{22}}}\operatorname{erfc}\left(\frac{t_{11}+2T_{22}}{\sqrt{2}\sqrt{T_{22}}}\right) -
$$
$$
2\alpha_{01}^2 e^{T_1^2} e^{-T_4}\operatorname{erfc}(T_1) + 4\alpha_{01}^2 e^{t_2^2} e^{-t_4}\operatorname{erfc}(t_2) + 2(2 - \operatorname{erfc}(T_3)) +
$$
$$
\sqrt{\frac{2}{\pi}} e^{-T_4}\left(\frac{\left(\alpha_{01}^2 - 1\right)T_{11}}{\sqrt{t_{22}}} - 3\alpha_{01}^2\sqrt{t_{22}}\right)\right) = 2.39669\ .
$$

We combine the results of these lemmata to obtain an lower bound on the minimum:

$$
\frac{1}{4}\lambda_{01}^2\left(8\alpha_{01}^2 t_{22} e^{-T_4} e^{\frac{(T_{11}+2t_{22})^2}{2t_{22}}}\operatorname{erfc}\left(\frac{T_{11}+2t_{22}}{\sqrt{2}\sqrt{t_{22}}}\right) + \tag{199}
$$
$$
\alpha_{01}^2 T_{22} e^{-t_4} e^{\frac{(t_{11}+T_{22})^2}{2T_{22}}}\operatorname{erfc}\left(\frac{t_{11}+T_{22}}{\sqrt{2}\sqrt{T_{22}}}\right) -
$$

$$2\alpha_{01}^2 e^{t_1^2} e^{-t_4}\operatorname{erfc}(t_1) + 4\alpha_{01}^2 e^{T_2^2} e^{-T_4}\operatorname{erfc}(T_2) +$$

$$2(2 - \operatorname{erfc}(t_3)) + \sqrt{\frac{2}{\pi}} e^{-t_4}\left(\frac{\left(\alpha_{01}^2 - 1\right)t_{11}}{\sqrt{T_{22}}} - 3\alpha_{01}^2\sqrt{T_{22}}\right)\right) = -1.17154\,.$$

Thus, an upper bound on the maximal absolute value is

$$\frac{1}{4}\lambda_{01}^2\left(-\alpha_{01}^2 t_{22} e^{-T_4} e^{\frac{(T_{11}+t_{22})^2}{2t_{22}}}\operatorname{erfc}\left(\frac{T_{11}+t_{22}}{\sqrt{2}\sqrt{t_{22}}}\right) + \right. \tag{200}$$

$$8\alpha_{01}^2 T_{22} e^{-t_4} e^{\frac{(t_{11}+2T_{22})^2}{2T_{22}}}\operatorname{erfc}\left(\frac{t_{11}+2T_{22}}{\sqrt{2}\sqrt{T_{22}}}\right) -$$

$$2\alpha_{01}^2 e^{T_1^2} e^{-T_4}\operatorname{erfc}(T_1) + 4\alpha_{01}^2 e^{t_2^2} e^{-t_4}\operatorname{erfc}(t_2) + 2(2 - \operatorname{erfc}(T_3)) +$$

$$\left. \sqrt{\frac{2}{\pi}} e^{-T_4}\left(\frac{\left(\alpha_{01}^2 - 1\right)T_{11}}{\sqrt{t_{22}}} - 3\alpha_{01}^2\sqrt{t_{22}}\right)\right) = 2.396685907216327\,.$$

<div align="right">□</div>

**Lemma 40** (Derivatives of the Mapping). *We assume $\alpha = \alpha_{01}$ and $\lambda = \lambda_{01}$. We restrict the range of the variables to the domain $\mu \in [-0.1, 0.1]$, $\omega \in [-0.1, 0.1]$, $\nu \in [0.8, 1.5]$, and $\tau \in [0.8, 1.25]$.*

*The derivative $\frac{\partial}{\partial \mu}\tilde{\mu}(\mu, \omega, \nu, \tau, \lambda, \alpha)$ has the sign of $\omega$.*

*The derivative $\frac{\partial}{\partial \nu}\tilde{\mu}(\mu, \omega, \nu, \tau, \lambda, \alpha)$ is positive.*

*The derivative $\frac{\partial}{\partial \mu}\tilde{\xi}(\mu, \omega, \nu, \tau, \lambda, \alpha)$ has the sign of $\omega$.*

*The derivative $\frac{\partial}{\partial \nu}\tilde{\xi}(\mu, \omega, \nu, \tau, \lambda, \alpha)$ is positive.*

*Proof.*   ∎ $\frac{\partial}{\partial \mu}\tilde{\mu}(\mu, \omega, \nu, \tau, \lambda, \alpha)$

$(2 - \operatorname{erfc}(x) > 0$ according to Lemma 21 and $e^{x^2}\operatorname{erfc}(x)$ is also larger than zero according to Lemma 23. Consequently, has $\frac{\partial}{\partial \mu}\tilde{\mu}(\mu, \omega, \nu, \tau, \lambda, \alpha)$ the sign of $\omega$.

∎ $\frac{\partial}{\partial \nu}\tilde{\mu}(\mu, \omega, \nu, \tau, \lambda, \alpha)$

Lemma 23 says $e^{x^2}\operatorname{erfc}(x)$ is decreasing in $\frac{\mu\omega+\nu\tau}{\sqrt{2}\sqrt{\nu\tau}}$. The first term (negative) is increasing in $\nu\tau$ since it is proportional to minus one over the squared root of $\nu\tau$.

We obtain a lower bound by setting $\frac{\mu\omega+\nu\tau}{\sqrt{2}\sqrt{\nu\tau}} = \frac{1.5\cdot1.25+0.1\cdot0.1}{\sqrt{2}\sqrt{1.5\cdot1.25}}$ for the $e^{x^2}\operatorname{erfc}(x)$ term. The term in brackets is larger than $e^{\left(\frac{1.5\cdot1.25+0.1\cdot0.1}{\sqrt{2}\sqrt{1.5\cdot1.25}}\right)^2}\alpha_{01}\operatorname{erfc}\left(\frac{1.5\cdot1.25+0.1\cdot0.1}{\sqrt{2}\sqrt{1.5\cdot1.25}}\right) - \sqrt{\frac{2}{\pi0.8\cdot0.8}}(\alpha_{01} - 1) = 0.056$ Consequently, the function is larger than zero.

∎ $\frac{\partial}{\partial \mu}\tilde{\xi}(\mu, \omega, \nu, \tau, \lambda, \alpha)$

We consider the sub-function

$$\sqrt{\frac{2}{\pi}}\sqrt{\nu\tau} - \alpha^2\left(e^{\left(\frac{\mu\omega+\nu\tau}{\sqrt{2}\sqrt{\nu\tau}}\right)^2}\operatorname{erfc}\left(\frac{\mu\omega+\nu\tau}{\sqrt{2}\sqrt{\nu\tau}}\right) - e^{\left(\frac{\mu\omega+2\nu\tau}{\sqrt{2}\sqrt{\nu\tau}}\right)^2}\operatorname{erfc}\left(\frac{\mu\omega+2\nu\tau}{\sqrt{2}\sqrt{\nu\tau}}\right)\right)\,. \tag{201}$$

We set $x = \nu\tau$ and $y = \mu\omega$ and obtain

$$\sqrt{\frac{2}{\pi}}\sqrt{x} - \alpha^2 \left( e^{\left(\frac{x+y}{\sqrt{2}\sqrt{x}}\right)^2} \operatorname{erfc}\left(\frac{x+y}{\sqrt{2}\sqrt{x}}\right) - e^{\left(\frac{2x+y}{\sqrt{2}\sqrt{x}}\right)^2} \operatorname{erfc}\left(\frac{2x+y}{\sqrt{2}\sqrt{x}}\right) \right) \ . \qquad (202)$$

The derivative of this sub-function with respect to $y$ is

$$\frac{\alpha^2 \left( e^{\frac{(2x+y)^2}{2x}}(2x+y)\operatorname{erfc}\left(\frac{2x+y}{\sqrt{2}\sqrt{x}}\right) - e^{\frac{(x+y)^2}{2x}}(x+y)\operatorname{erfc}\left(\frac{x+y}{\sqrt{2}\sqrt{x}}\right) \right)}{x} = \qquad (203)$$

$$\frac{\sqrt{2}\alpha^2\sqrt{x}\left( \frac{e^{\frac{(2x+y)^2}{2x}}(x+y)\operatorname{erfc}\left(\frac{x+y}{\sqrt{2}\sqrt{x}}\right)}{\sqrt{2}\sqrt{x}} - \frac{e^{\frac{(x+y)^2}{2x}}(x+y)\operatorname{erfc}\left(\frac{x+y}{\sqrt{2}\sqrt{x}}\right)}{\sqrt{2}\sqrt{x}} \right)}{x} > 0 \ .$$

The inequality follows from Lemma 24, which states that $ze^{z^2}\operatorname{erfc}(z)$ is monotonically increasing in $z$. Therefore the sub-function is increasing in $y$.

The derivative of this sub-function with respect to $x$ is

$$\frac{\sqrt{\pi}\alpha^2 \left( e^{\frac{(2x+y)^2}{2x}}(4x^2-y^2)\operatorname{erfc}\left(\frac{2x+y}{\sqrt{2}\sqrt{x}}\right) - e^{\frac{(x+y)^2}{2x}}(x-y)(x+y)\operatorname{erfc}\left(\frac{x+y}{\sqrt{2}\sqrt{x}}\right) \right) - \sqrt{2}\left(\alpha^2-1\right)x^{3/2}}{2\sqrt{\pi}x^2} \ . \qquad (204)$$

The sub-function is increasing in $x$, since the derivative is larger than zero:

$$\frac{\sqrt{\pi}\alpha^2 \left( e^{\frac{(2x+y)^2}{2x}}(4x^2-y^2)\operatorname{erfc}\left(\frac{2x+y}{\sqrt{2}\sqrt{x}}\right) - e^{\frac{(x+y)^2}{2x}}(x-y)(x+y)\operatorname{erfc}\left(\frac{x+y}{\sqrt{2}\sqrt{x}}\right) \right) - \sqrt{2}x^{3/2}\left(\alpha^2-1\right)}{2\sqrt{\pi}x^2} \geqslant \qquad (205)$$

$$\frac{\sqrt{\pi}\alpha^2 \left( \frac{(2x-y)(2x+y)2}{\sqrt{\pi}\left(\frac{2x+y}{\sqrt{2}\sqrt{x}}+\sqrt{\left(\frac{2x+y}{\sqrt{2}\sqrt{x}}\right)^2+2}\right)} - \frac{(x-y)(x+y)2}{\sqrt{\pi}\left(\frac{x+y}{\sqrt{2}\sqrt{x}}+\sqrt{\left(\frac{x+y}{\sqrt{2}\sqrt{x}}\right)^2+\frac{4}{\pi}}\right)} \right) - \sqrt{2}x^{3/2}\left(\alpha^2-1\right)}{2\sqrt{\pi}x^2} =$$

$$\frac{\sqrt{\pi}\alpha^2 \left( \frac{(2x-y)(2x+y)2\left(\sqrt{2}\sqrt{x}\right)}{\sqrt{\pi}\left(2x+y+\sqrt{(2x+y)^2+4x}\right)} - \frac{(x-y)(x+y)2\left(\sqrt{2}\sqrt{x}\right)}{\sqrt{\pi}\left(x+y+\sqrt{(x+y)^2+\frac{8x}{\pi}}\right)} \right) - \sqrt{2}x^{3/2}\left(\alpha^2-1\right)}{2\sqrt{\pi}x^2} =$$

$$\frac{\sqrt{\pi}\alpha^2 \left( \frac{(2x-y)(2x+y)2}{\sqrt{\pi}\left(2x+y+\sqrt{(2x+y)^2+4x}\right)} - \frac{(x-y)(x+y)2}{\sqrt{\pi}\left(x+y+\sqrt{(x+y)^2+\frac{8x}{\pi}}\right)} \right) - x\left(\alpha^2-1\right)}{\sqrt{2}\sqrt{\pi}x^{3/2}} >$$

$$\frac{\sqrt{\pi}\alpha^2 \left( \frac{(2x-y)(2x+y)2}{\sqrt{\pi}\left(2x+y+\sqrt{(2x+y)^2+2(2x+y)+1}\right)} - \frac{(x-y)(x+y)2}{\sqrt{\pi}\left(x+y+\sqrt{(x+y)^2+0.782\cdot2(x+y)+0.782^2}\right)} \right) - x\left(\alpha^2-1\right)}{\sqrt{2}\sqrt{\pi}x^{3/2}} =$$

$$\frac{\sqrt{\pi}\alpha^2\left(\frac{(2x-y)(2x+y)2}{\sqrt{\pi}\left(2x+y+\sqrt{(2x+y+1)^2}\right)}-\frac{(x-y)(x+y)2}{\sqrt{\pi}\left(x+y+\sqrt{(x+y+0.782)^2}\right)}\right)-x\left(\alpha^2-1\right)}{\sqrt{2}\sqrt{\pi}x^{3/2}}=$$

$$\frac{\sqrt{\pi}\alpha^2\left(\frac{(2x-y)(2x+y)2}{\sqrt{\pi}(2(2x+y)+1)}-\frac{(x-y)(x+y)2}{\sqrt{\pi}(2(x+y)+0.782)}\right)-x\left(\alpha^2-1\right)}{\sqrt{2}\sqrt{\pi}x^{3/2}}=$$

$$\frac{\sqrt{\pi}\alpha^2\left(\frac{(2(x+y)+0.782)(2x-y)(2x+y)2}{\sqrt{\pi}}-\frac{(x-y)(x+y)(2(2x+y)+1)2}{\sqrt{\pi}}\right)}{(2(2x+y)+1)(2(x+y)+0.782)\sqrt{2}\sqrt{\pi}x^{3/2}}+$$

$$\frac{\sqrt{\pi}\alpha^2\left(-x\left(\alpha^2-1\right)(2(2x+y)+1)(2(x+y)+0.782)\right)}{(2(2x+y)+1)(2(x+y)+0.782)\sqrt{2}\sqrt{\pi}x^{3/2}}=$$

$$\frac{8x^3+(12y+2.68657)x^2+(y(4y-6.41452)-1.40745)x+1.22072y^2}{(2(2x+y)+1)(2(x+y)+0.782)\sqrt{2}\sqrt{\pi}x^{3/2}}>$$

$$\frac{8x^3+(2.68657-120.01)x^2+(0.01(-6.41452-40.01)-1.40745)x+1.22072(0.0)^2}{(2(2x+y)+1)(2(x+y)+0.782)\sqrt{2}\sqrt{\pi}x^{3/2}}=$$

$$\frac{8x^2+2.56657x-1.472}{(2(2x+y)+1)(2(x+y)+0.782)\sqrt{2}\sqrt{\pi}\sqrt{x}}=$$

$$\frac{8x^2+2.56657x-1.472}{(2(2x+y)+1)(2(x+y)+0.782)\sqrt{2}\sqrt{\pi}\sqrt{x}}=$$

$$\frac{8(x+0.618374)(x-0.297553)}{(2(2x+y)+1)(2(x+y)+0.782)\sqrt{2}\sqrt{\pi}\sqrt{x}}>0\,.$$

We explain this chain of inequalities:

- First inequality: We applied Lemma 22 two times.
- Equalities factor out $\sqrt{2}\sqrt{x}$ and reformulate.
- Second inequality part 1: we applied

$$0<2y\implies(2x+y)^2+4x+1<(2x+y)^2+2(2x+y)+1=(2x+y+1)^2\,.$$
(206)

- Second inequality part 2: we show that for $a=\frac{1}{20}\left(\sqrt{\frac{2048+169\pi}{\pi}}-13\right)$ following holds: $\frac{8x}{\pi}-\left(a^2+2a(x+y)\right)\geqslant0$. We have $\frac{\partial}{\partial x}\frac{8x}{\pi}-\left(a^2+2a(x+y)\right)=\frac{8}{\pi}-2a>0$ and $\frac{\partial}{\partial y}\frac{8x}{\pi}-\left(a^2+2a(x+y)\right)=-2a>0$. Therefore the minimum is at border for minimal $x$ and maximal $y$:

$$\frac{8\cdot0.64}{\pi}-\left(\frac{2}{20}\left(\sqrt{\frac{2048+169\pi}{\pi}}-13\right)(0.64+0.01)+\left(\frac{1}{20}\left(\sqrt{\frac{2048+169\pi}{\pi}}-13\right)\right)^2\right)=0\,.$$
(207)

Thus

$$\frac{8x}{\pi}\geqslant a^2+2a(x+y)\,.$$
(208)

for $a=\frac{1}{20}\left(\sqrt{\frac{2048+169\pi}{\pi}}-13\right)>0.782$.

- Equalities only solve square root and factor out the resulting terms $(2(2x + y) + 1)$ and $(2(x + y) + 0.782)$.
- We set $\alpha = \alpha_{01}$ and multiplied out. Thereafter we also factored out $x$ in the numerator. Finally a quadratic equations was solved.

The sub-function has its minimal value for minimal $x$ and minimal $y$ $x = \nu\tau = 0.8 \cdot 0.8 = 0.64$ and $y = \mu\omega = -0.1 \cdot 0.1 = -0.01$. We further minimize the function

$$\mu\omega e^{\frac{\mu^2\omega^2}{2\nu\tau}}\left(2 - \operatorname{erfc}\left(\frac{\mu\omega}{\sqrt{2}\sqrt{\nu\tau}}\right)\right) > -0.01 e^{\frac{0.01^2}{20.64}}\left(2 - \operatorname{erfc}\left(\frac{0.01}{\sqrt{2}\sqrt{0.64}}\right)\right) . \quad (209)$$

We compute the minimum of the term in brackets of $\frac{\partial}{\partial\mu}\tilde{\xi}(\mu, \omega, \nu, \tau, \lambda, \alpha)$:

$$\mu\omega e^{\frac{\mu^2\omega^2}{2\nu\tau}}\left(2 - \operatorname{erfc}\left(\frac{\mu\omega}{\sqrt{2}\sqrt{\nu\tau}}\right)\right) + \quad (210)$$

$$\alpha_{01}^2\left(-\left(e^{\left(\frac{\mu\omega+\nu\tau}{\sqrt{2}\sqrt{\nu\tau}}\right)^2}\operatorname{erfc}\left(\frac{\mu\omega + \nu\tau}{\sqrt{2}\sqrt{\nu\tau}}\right) - e^{\left(\frac{\mu\omega+2\nu\tau}{\sqrt{2}\sqrt{\nu\tau}}\right)^2}\operatorname{erfc}\left(\frac{\mu\omega + 2\nu\tau}{\sqrt{2}\sqrt{\nu\tau}}\right)\right)\right) +$$

$$\sqrt{\frac{2}{\pi}}\sqrt{\nu\tau} >$$

$$\alpha_{01}^2\left(-\left(e^{\left(\frac{0.64-0.01}{\sqrt{2}\sqrt{0.64}}\right)^2}\operatorname{erfc}\left(\frac{0.64 - 0.01}{\sqrt{2}\sqrt{0.64}}\right) - e^{\left(\frac{20.64-0.01}{\sqrt{2}\sqrt{0.64}}\right)^2}\operatorname{erfc}\left(\frac{2 \cdot 0.64 - 0.01}{\sqrt{2}\sqrt{0.64}}\right)\right)\right) -$$

$$0.01 e^{\frac{0.01^2}{20.64}}\left(2 - \operatorname{erfc}\left(\frac{0.01}{\sqrt{2}\sqrt{0.64}}\right)\right) + \sqrt{0.64}\sqrt{\frac{2}{\pi}} = 0.0923765 .$$

Therefore the term in brackets is larger than zero.

Thus, $\frac{\partial}{\partial\mu}\tilde{\xi}(\mu, \omega, \nu, \tau, \lambda, \alpha)$ has the sign of $\omega$.

- ■ $\frac{\partial}{\partial\nu}\tilde{\xi}(\mu, \omega, \nu, \tau, \lambda, \alpha)$

  We look at the sub-term

$$2e^{\left(\frac{2x+y}{\sqrt{2}\sqrt{x}}\right)^2}\operatorname{erfc}\left(\frac{2x + y}{\sqrt{2}\sqrt{x}}\right) - e^{\left(\frac{x+y}{\sqrt{2}\sqrt{x}}\right)^2}\operatorname{erfc}\left(\frac{x + y}{\sqrt{2}\sqrt{x}}\right) . \quad (211)$$

  We obtain a chain of inequalities:

$$2e^{\left(\frac{2x+y}{\sqrt{2}\sqrt{x}}\right)^2}\operatorname{erfc}\left(\frac{2x + y}{\sqrt{2}\sqrt{x}}\right) - e^{\left(\frac{x+y}{\sqrt{2}\sqrt{x}}\right)^2}\operatorname{erfc}\left(\frac{x + y}{\sqrt{2}\sqrt{x}}\right) > \quad (212)$$

$$\frac{2 \cdot 2}{\sqrt{\pi}\left(\frac{2x+y}{\sqrt{2}\sqrt{x}} + \sqrt{\left(\frac{2x+y}{\sqrt{2}\sqrt{x}}\right)^2 + 2}\right)} - \frac{2}{\sqrt{\pi}\left(\frac{x+y}{\sqrt{2}\sqrt{x}} + \sqrt{\left(\frac{x+y}{\sqrt{2}\sqrt{x}}\right)^2 + \frac{4}{\pi}}\right)} =$$

$$\frac{2\sqrt{2}\sqrt{x}\left(\frac{2}{\sqrt{(2x+y)^2+4x}+2x+y} - \frac{1}{\sqrt{(x+y)^2+\frac{8x}{\pi}}+x+y}\right)}{\sqrt{\pi}} >$$

$$\frac{2\sqrt{2}\sqrt{x}\left(\frac{2}{\sqrt{(2x+y)^2+2(2x+y)+1}+2x+y} - \frac{1}{\sqrt{(x+y)^2+0.782\cdot2(x+y)+0.782^2}+x+y}\right)}{\sqrt{\pi}} =$$

$$\frac{2\sqrt{2}\sqrt{x}\left(\frac{2}{2(2x+y)+1} - \frac{1}{2(x+y)+0.782}\right)}{\sqrt{\pi}} =$$

$$\frac{\left(2\sqrt{2}\sqrt{x}\right)\left(2(2(x+y)+0.782) - (2(2x+y)+1)\right)}{\sqrt{\pi}((2(x+y)+0.782)(2(2x+y)+1))} =$$

$$\frac{\left(2\sqrt{2}\sqrt{x}\right)(2y + 0.782 \cdot 2 - 1)}{\sqrt{\pi}((2(x+y)+0.782)(2(2x+y)+1))} > 0 \,.$$

We explain this chain of inequalities:

- First inequality: We applied Lemma 22 two times.
- Equalities factor out $\sqrt{2}\sqrt{x}$ and reformulate.
- Second inequality part 1: we applied

$$0 < 2y \implies (2x+y)^2 + 4x + 1 < (2x+y)^2 + 2(2x+y) + 1 = (2x+y+1)^2 \,. \tag{213}$$

- Second inequality part 2: we show that for $a = \frac{1}{20}\left(\sqrt{\frac{2048+169\pi}{\pi}} - 13\right)$ following holds: $\frac{8x}{\pi} - \left(a^2 + 2a(x+y)\right) \geqslant 0$. We have $\frac{\partial}{\partial x} \frac{8x}{\pi} - \left(a^2 + 2a(x+y)\right) = \frac{8}{\pi} - 2a > 0$ and $\frac{\partial}{\partial y} \frac{8x}{\pi} - \left(a^2 + 2a(x+y)\right) = -2a < 0$. Therefore the minimum is at border for minimal $x$ and maximal $y$:

$$\frac{8 \cdot 0.64}{\pi} - \left(\frac{2}{20}\left(\sqrt{\frac{2048+169\pi}{\pi}} - 13\right)(0.64 + 0.01) + \left(\frac{1}{20}\left(\sqrt{\frac{2048+169\pi}{\pi}} - 13\right)\right)^2\right) = 0 \,. \tag{214}$$

Thus

$$\frac{8x}{\pi} \geqslant a^2 + 2a(x+y) \,. \tag{215}$$

for $a = \frac{1}{20}\left(\sqrt{\frac{2048+169\pi}{\pi}} - 13\right) > 0.782$.

- Equalities only solve square root and factor out the resulting terms $(2(2x+y)+1)$ and $(2(x+y)+0.782)$.

We know that $(2 - \text{erfc}(x) > 0$ according to Lemma 21. For the sub-term we derived

$$2e^{\left(\frac{2x+y}{\sqrt{2}\sqrt{x}}\right)^2} \text{erfc}\left(\frac{2x+y}{\sqrt{2}\sqrt{x}}\right) - e^{\left(\frac{x+y}{\sqrt{2}\sqrt{x}}\right)^2} \text{erfc}\left(\frac{x+y}{\sqrt{2}\sqrt{x}}\right) > 0 \,. \tag{216}$$

Consequently, both terms in the brackets of $\frac{\partial}{\partial \nu}\tilde{\xi}(\mu, \omega, \nu, \tau, \lambda, \alpha)$ are larger than zero. Therefore $\frac{\partial}{\partial \nu}\tilde{\xi}(\mu, \omega, \nu, \tau, \lambda, \alpha)$ is larger than zero.

□

Figure S5: The graph of function $\tilde{\mu}$ for low variances $x = \nu\tau$ for $\mu\omega = 0.01$, where $x \in [0,3]$, is displayed in yellow. Lower and upper bounds based on the Abramowitz bounds (Lemma 22) are displayed in green and blue, respectively.

**Lemma 41** (Mean at low variance). *The mapping of the mean $\tilde{\mu}$ (Eq. (5))*

$$\tilde{\mu}(\mu, \omega, \nu, \tau, \lambda, \alpha) \;=\; \frac{1}{2}\lambda\left(-(\alpha + \mu\omega)\,\mathrm{erfc}\left(\frac{\mu\omega}{\sqrt{2}\sqrt{\nu\tau}}\right) + \right. \tag{217}$$

$$\left. \alpha e^{\mu\omega + \frac{\nu\tau}{2}}\,\mathrm{erfc}\left(\frac{\mu\omega + \nu\tau}{\sqrt{2}\sqrt{\nu\tau}}\right) + \sqrt{\frac{2}{\pi}}\sqrt{\nu\tau}\,e^{-\frac{\mu^2\omega^2}{2\nu\tau}} + 2\mu\omega\right)$$

*in the domain $-0.1 \leqslant \mu \leqslant -0.1$, $-0.1 \leqslant \omega \leqslant -0.1$, and $0.02 \leqslant \nu\tau \leqslant 0.5$ is bounded by*

$$|\tilde{\mu}(\mu, \omega, \nu, \tau, \lambda_{01}, \alpha_{01})| < 0.289324 \tag{218}$$

*and*

$$\lim_{\nu \to 0} |\tilde{\mu}(\mu, \omega, \nu, \tau, \lambda_{01}, \alpha_{01})| = \lambda\mu\omega. \tag{219}$$

We can consider $\tilde{\mu}$ with given $\mu\omega$ as a function in $x = \nu\tau$. We show the graph of this function at the maximal $\mu\omega = 0.01$ in the interval $x \in [0, 1]$ in Figure S5.

*Proof.* Since $\tilde{\mu}$ is strictly monotonically increasing with $\mu\omega$

$$\tilde{\mu}(\mu, \omega, \nu, \tau, \lambda, \alpha) \leqslant \tag{220}$$
$$\tilde{\mu}(0.1, 0.1, \nu, \tau, \lambda, \alpha) \leqslant$$

$$\frac{1}{2}\lambda\left(-(\alpha + 0.01)\,\mathrm{erfc}\left(\frac{0.01}{\sqrt{2}\sqrt{\nu\tau}}\right) + \alpha e^{0.01 + \frac{\nu\tau}{2}}\,\mathrm{erfc}\left(\frac{0.01 + \nu\tau}{\sqrt{2}\sqrt{\nu\tau}}\right) + \sqrt{\frac{2}{\pi}}\sqrt{\nu\tau}\,e^{-\frac{0.01^2}{2\nu\tau}} + 2 \cdot 0.01\right) \leqslant$$

$$\frac{1}{2}\lambda_{01}\left(e^{\frac{0.05}{2} + 0.01}\alpha_{01}\,\mathrm{erfc}\left(\frac{0.02 + 0.01}{\sqrt{2}\sqrt{0.02}}\right) - (\alpha_{01} + 0.01)\,\mathrm{erfc}\left(\frac{0.01}{\sqrt{2}\sqrt{0.02}}\right) + e^{-\frac{0.01^2}{2 \cdot 0.5}}\sqrt{0.5}\sqrt{\frac{2}{\pi}} + 0.01 \cdot 2\right)$$

$$< 0.21857,$$

where we have used the monotonicity of the terms in $\nu\tau$.

Similary, we can use the monotonicity of the terms in $\nu\tau$ to show that

$$\tilde{\mu}(\mu, \omega, \nu, \tau, \lambda, \alpha) \geqslant \tag{221}$$

$$\tilde{\mu}(0.1, -0.1, \nu, \tau, \lambda, \alpha) >$$
$$- 0.289324,$$

such that $|\tilde{\mu}| < 0.289324$ at low variances.

Furthermore, when $(\nu\tau) \to 0$, the terms with the arguments of the complementary error functions erfc and the exponential function go to infinity, therefore these three terms converge to zero. Hence, the remaining terms are only $2\mu\omega \cdot \frac{1}{2}\lambda$.                    □

**Lemma 42** (Bounds on derivates of $\tilde{\mu}$ in $\Omega^-$)**.** *The derivatives of the function* $\tilde{\mu}(\mu, \omega, \nu, \tau, \lambda_{01}, \alpha_{01}$ *(Eq. (5)) with respect to* $\mu, \omega, \nu, \tau$ *in the domain* $\Omega^- = \{\mu, \omega, \nu, \tau \mid -0.1 \leqslant \mu \leqslant 0.1, -0.1 \leqslant \omega \leqslant 0.1, 0.05 \leqslant \nu \leqslant 0.24, 0.8 \leqslant \tau \leqslant 1.25\}$ *can be bounded as follows:*

$$\left|\frac{\partial}{\partial\mu}\tilde{\mu}\right| < 0.14 \tag{222}$$

$$\left|\frac{\partial}{\partial\omega}\tilde{\mu}\right| < 0.14$$

$$\left|\frac{\partial}{\partial\nu}\tilde{\mu}\right| < 0.52$$

$$\left|\frac{\partial}{\partial\tau}\tilde{\mu}\right| < 0.11.$$

*Proof.* The expression

$$\frac{\partial}{\partial\mu}\tilde{\mu} = J_{11} = \frac{1}{2}\lambda\omega e^{\frac{-(\mu\omega)^2}{2\nu\tau}} \left( 2e^{\frac{(\mu\omega)^2}{2\nu\tau}} - e^{\frac{(\mu\omega)^2}{2\nu\tau}} \operatorname{erfc}\left(\frac{\mu\omega}{\sqrt{2}\sqrt{\nu\tau}}\right) + \alpha e^{\frac{(\mu\omega+\nu\tau)^2}{2\nu\tau}} \operatorname{erfc}\left(\frac{\mu\omega+\nu\tau}{\sqrt{2}\sqrt{\nu\tau}}\right) \right) \tag{223}$$

contains the terms $e^{\frac{(\mu\omega)^2}{2\nu\tau}} \operatorname{erfc}\left(\frac{\mu\omega}{\sqrt{2}\sqrt{\nu\tau}}\right)$ and $e^{\frac{(\mu\omega+\nu\tau)^2}{2\nu\tau}} \operatorname{erfc}\left(\frac{\mu\omega+\nu\tau}{\sqrt{2}\sqrt{\nu\tau}}\right)$ which are monotonically decreasing in their arguments (Lemma 23). We can therefore obtain their minima and maximal at the minimal and maximal arguments. Since the first term has a negative sign in the expression, both terms reach their maximal value at $\mu\omega = -0.01$, $\nu = 0.05$, and $\tau = 0.8$.

$$\left|\frac{\partial}{\partial\mu}\tilde{\mu}\right| \leqslant \tag{224}$$
$$\frac{1}{2}\left|\lambda\omega\right| \left| \left(2 - e^{0.0353553^2} \operatorname{erfc}(0.0353553) + \alpha e^{0.106066^2} \operatorname{erfc}(0.106066)\right) \right| <$$
$$0.133$$

Since, $\tilde{\mu}$ is symmetric in $\mu$ and $\omega$, these bounds also hold for the derivate to $\omega$.

We use the argumentation that the term with the error function is monotonically decreasing (Lemma 23) again for the expression

$$\frac{\partial}{\partial\nu}\tilde{\mu} = J_{12} = \tag{225}$$

Figure S6: The graph of the function $h(x) = \tilde{\mu}^2(0.1, -0.1, x, 1, \lambda_{01}, \alpha_{01})$ is displayed. It has a local maximum at $x = \nu\tau \approx 0.187342$ and $h(x) \approx 0.00451457$ in the domain $x \in [0, 1]$.

$$= \frac{1}{4}\lambda\tau e^{-\frac{\mu^2\omega^2}{2\nu\tau}}\left(\alpha e^{\frac{(\mu\omega+\nu\tau)^2}{2\nu\tau}}\operatorname{erfc}\left(\frac{\mu\omega+\nu\tau}{\sqrt{2}\sqrt{\nu\tau}}\right) - (\alpha-1)\sqrt{\frac{2}{\pi\nu\tau}}\right) \leqslant$$

$$\left|\frac{1}{4}\lambda\tau\right|\left(|1.1072 - 2.68593|\right) < 0.52.$$

We have used that the term $1.1072 \leqslant \alpha_{01}e^{\frac{(\mu\omega+\nu\tau)^2}{2\nu\tau}}\operatorname{erfc}\left(\frac{\mu\omega+\nu\tau}{\sqrt{2}\sqrt{\nu\tau}}\right) \leqslant 1.49042$ and the term $0.942286 \leqslant (\alpha-1)\sqrt{\frac{2}{\pi\nu\tau}} \leqslant 2.68593$.

Since $\tilde{\mu}$ is symmetric in $\nu$ and $\tau$, we only have to chance outermost term $\left|\frac{1}{4}\lambda\tau\right|$ to $\left|\frac{1}{4}\lambda\nu\right|$ to obtain the estimate $\left|\frac{\partial}{\partial\tau}\tilde{\mu}\right| < 0.11$.

$\square$

**Lemma 43** (Tight bound on $\tilde{\mu}^2$ in $\Omega^-$). *The function $\tilde{\mu}^2(\mu, \omega, \nu, \tau, \lambda_{01}, \alpha_{01})$ (Eq. (5)) is bounded by*

$$\left|\tilde{\mu}^2\right| < 0.005 \tag{226}$$

$$\tag{227}$$

*in the domain $\Omega^- = \{\mu, \omega, \nu, \tau \mid -0.1 \leqslant \mu \leqslant 0.1, -0.1 \leqslant \omega \leqslant 0.1, 0.05 \leqslant \nu \leqslant 0.24, 0.8 \leqslant \tau \leqslant 1.25\}$.*

We visualize the function $\tilde{\mu}^2$ at its maximal $\mu\nu = -0.01$ and for $x = \nu\tau$ in the form $h(x) = \tilde{\mu}^2(0.1, -0.1, x, 1, \lambda_{01}, \alpha_{01})$ in Figure S6.

*Proof.* We use a similar strategy to the one we have used to show the bound on the singular value (Lemmata 10, 11, and 12), where we eveluted the function on a grid and used bounds on the derivatives together with the mean value theorem. Here we have

$$\left|\tilde{\mu}^2(\mu, \omega, \nu, \tau, \lambda_{01}, \alpha_{01}) - \tilde{\mu}^2(\mu+\Delta\mu, \omega+\Delta\omega, \nu+\Delta\nu, \tau+\Delta\tau, \lambda_{01}, \alpha_{01})\right| \leqslant \tag{228}$$

$$\left|\frac{\partial}{\partial\mu}\tilde{\mu}^2\right||\Delta\mu| + \left|\frac{\partial}{\partial\omega}\tilde{\mu}^2\right||\Delta\omega| + \left|\frac{\partial}{\partial\nu}\tilde{\mu}^2\right||\Delta\nu| + \left|\frac{\partial}{\partial\tau}\tilde{\mu}^2\right||\Delta\tau|.$$

We use Lemma 42 and Lemma 41, to obtain

$$\left| \frac{\partial}{\partial \mu} \tilde{\mu}^2 \right| = 2 \cdot |\tilde{\mu}| \cdot \left| \frac{\partial}{\partial \mu} \tilde{\mu} \right| < 2 \cdot 0.289324 \cdot 0.14 = 0.08101072 \tag{229}$$

$$\left| \frac{\partial}{\partial \omega} \tilde{\mu}^2 \right| = 2 \cdot |\tilde{\mu}| \cdot \left| \frac{\partial}{\partial \omega} \tilde{\mu} \right| < 2 \cdot 0.289324 \cdot 0.14 = 0.08101072$$

$$\left| \frac{\partial}{\partial \nu} \tilde{\mu}^2 \right| = 2 \cdot |\tilde{\mu}| \cdot \left| \frac{\partial}{\partial \nu} \tilde{\mu} \right| < 2 \cdot 0.289324 \cdot 0.52 = 0.30089696$$

$$\left| \frac{\partial}{\partial \tau} \tilde{\mu}^2 \right| = 2 \cdot |\tilde{\mu}| \cdot \left| \frac{\partial}{\partial \tau} \tilde{\mu} \right| < 2 \cdot 0.289324 \cdot 0.11 = 0.06365128$$

We evaluated the function $\tilde{\mu}^2$ in a grid $G$ of $\Omega^-$ with $\Delta\mu = 0.001498041$, $\Delta\omega = 0.001498041$, $\Delta\nu = 0.0004033190$, and $\Delta\tau = 0.0019065994$ using a computer and obtained the maximal value $\max_G(\tilde{\mu})^2 = 0.00451457$, therefore the maximal value of $\tilde{\mu}^2$ is bounded by

$$\max_{(\mu,\omega,\nu,\tau)\in\Omega^-} (\tilde{\mu})^2 \leqslant \tag{230}$$

$$0.00451457 + 0.001498041 \cdot 0.08101072 + 0.001498041 \cdot 0.08101072+$$

$$0.0004033190 \cdot 0.30089696 + 0.0019065994 \cdot 0.06365128 < 0.005. \tag{231}$$

Furthermore we used error propagation to estimate the numerical error on the function evaluation. Using the error propagation rules derived in Subsection S3.4.5, we found that the numerical error is smaller than $10^{-13}$ in the worst case. $\qquad\square$

**Lemma 44** (Main subfunction). *For $1.2 \leqslant x \leqslant 20$ and $-0.1 \leqslant y \leqslant 0.1$,*

*the function*

$$e^{\frac{(x+y)^2}{2x}} \operatorname{erfc}\left( \frac{x+y}{\sqrt{2}\sqrt{x}} \right) - 2e^{\frac{(2x+y)^2}{2x}} \operatorname{erfc}\left( \frac{2x+y}{\sqrt{2}\sqrt{x}} \right) \tag{232}$$

*is smaller than zero, is strictly monotonically increasing in $x$, and strictly monotonically decreasing in $y$ for the minimal $x = 12/10 = 1.2$.*

*Proof.* We first consider the derivative of sub-function Eq. (90) with respect to $x$. The derivative of the function

$$e^{\frac{(x+y)^2}{2x}} \operatorname{erfc}\left( \frac{x+y}{\sqrt{2}\sqrt{x}} \right) - 2e^{\frac{(2x+y)^2}{2x}} \operatorname{erfc}\left( \frac{2x+y}{\sqrt{2}\sqrt{x}} \right) \tag{233}$$

with respect to $x$ is

$$\frac{\sqrt{\pi}\left( e^{\frac{(x+y)^2}{2x}}(x-y)(x+y)\operatorname{erfc}\left(\frac{x+y}{\sqrt{2}\sqrt{x}}\right) - 2e^{\frac{(2x+y)^2}{2x}}(4x^2-y^2)\operatorname{erfc}\left(\frac{2x+y}{\sqrt{2}\sqrt{x}}\right) \right) + \sqrt{2}\sqrt{x}(3x-y)}{2\sqrt{\pi}x^2} = \tag{234}$$

$$\frac{\sqrt{\pi}\left( e^{\frac{(x+y)^2}{2x}}(x-y)(x+y)\operatorname{erfc}\left(\frac{x+y}{\sqrt{2}\sqrt{x}}\right) - 2e^{\frac{(2x+y)^2}{2x}}(2x+y)(2x-y)\operatorname{erfc}\left(\frac{2x+y}{\sqrt{2}\sqrt{x}}\right) \right) + \sqrt{2}\sqrt{x}(3x-y)}{2\sqrt{\pi}x^2} =$$

$$\frac{\sqrt{\pi}\left(\frac{e^{\frac{(x+y)^2}{2x}}(x-y)(x+y)\operatorname{erfc}\left(\frac{x+y}{\sqrt{2}\sqrt{x}}\right)}{\sqrt{2}\sqrt{x}} - \frac{2e^{\frac{(2x+y)^2}{2x}}(2x+y)(2x-y)\operatorname{erfc}\left(\frac{2x+y}{\sqrt{2}\sqrt{x}}\right)}{\sqrt{2}\sqrt{x}}\right) + (3x-y)}{2\sqrt{2}\sqrt{\pi}x^2\sqrt{x}} \, .$$

We consider the numerator

$$\sqrt{\pi}\left(\frac{e^{\frac{(x+y)^2}{2x}}(x-y)(x+y)\operatorname{erfc}\left(\frac{x+y}{\sqrt{2}\sqrt{x}}\right)}{\sqrt{2}\sqrt{x}} - \frac{2e^{\frac{(2x+y)^2}{2x}}(2x+y)(2x-y)\operatorname{erfc}\left(\frac{2x+y}{\sqrt{2}\sqrt{x}}\right)}{\sqrt{2}\sqrt{x}}\right) + (3x-y) \, . \tag{235}$$

For bounding this value, we use the approximation

$$e^{z^2}\operatorname{erfc}(z) \approx \frac{2.911}{\sqrt{\pi}(2.911-1)z + \sqrt{\pi z^2 + 2.911^2}} \, . \tag{236}$$

from Ren and MacKenzie (2007). We start with an error analysis of this approximation. According to Ren and MacKenzie (2007) (Figure 1), the approximation error is positive in the range $[0.7, 3.2]$. This range contains all possible arguments of $\operatorname{erfc}$ that we consider. Numerically we maximized and minimized the approximation error of the whole expression

$$E(x,y) = \left(\frac{e^{\frac{(x+y)^2}{2x}}(x-y)(x+y)\operatorname{erfc}\left(\frac{x+y}{\sqrt{2}\sqrt{x}}\right)}{\sqrt{2}\sqrt{x}} - \frac{2e^{\frac{(2x+y)^2}{2x}}(2x-y)(2x+y)\operatorname{erfc}\left(\frac{2x+y}{\sqrt{2}\sqrt{x}}\right)}{\sqrt{2}\sqrt{x}}\right) - \tag{237}$$

$$\left(\frac{2.911(x-y)(x+y)}{\left(\sqrt{2}\sqrt{x}\right)\left(\frac{\sqrt{\pi}(2.911-1)(x+y)}{\sqrt{2}\sqrt{x}} + \sqrt{\pi\left(\frac{x+y}{\sqrt{2}\sqrt{x}}\right)^2 + 2.911^2}\right)} - \right.$$

$$\left.\frac{2\cdot 2.911(2x-y)(2x+y)}{\left(\sqrt{2}\sqrt{x}\right)\left(\frac{\sqrt{\pi}(2.911-1)(2x+y)}{\sqrt{2}\sqrt{x}} + \sqrt{\pi\left(\frac{2x+y}{\sqrt{2}\sqrt{x}}\right)^2 + 2.911^2}\right)}\right) \, .$$

We numerically determined $0.0113556 \leqslant E(x,y) \leqslant 0.0169551$ for $1.2 \leqslant x \leqslant 20$ and $-0.1 \leqslant y \leqslant 0.1$. We used different numerical optimization techniques like gradient based constraint BFGS algorithms and non-gradient-based Nelder-Mead methods with different start points. Therefore our approximation is smaller than the function that we approximate. We subtract an additional safety gap of $0.0131259$ from our approximation to ensure that the inequality via the approximation holds true. With this safety gap the inequality would hold true even for negative $x$, where the approximation error becomes negative and the safety gap would compensate. Of course, the safety gap of $0.0131259$ is not necessary for our analysis but may help or future investigations.

We have the sequences of inequalities using the approximation of Ren and MacKenzie (2007):

$$
(3x - y) + \left( \frac{e^{\frac{(x+y)^2}{2x}} (x-y)(x+y)\, \mathrm{erfc}\left(\frac{x+y}{\sqrt{2}\sqrt{x}}\right)}{\sqrt{2}\sqrt{x}} - \frac{2 e^{\frac{(2x+y)^2}{2x}} (2x-y)(2x+y)\, \mathrm{erfc}\left(\frac{2x+y}{\sqrt{2}\sqrt{x}}\right)}{\sqrt{2}\sqrt{x}} \right) \sqrt{\pi} \geqslant
$$

$$\tag{238}$$

$$
(3x - y) + \left( \frac{2.911(x-y)(x+y)}{\left( \sqrt{\pi\left(\frac{x+y}{\sqrt{2}\sqrt{x}}\right)^2 + 2.911^2} + \frac{(2.911-1)\sqrt{\pi}(x+y)}{\sqrt{2}\sqrt{x}} \right)\left(\sqrt{2}\sqrt{x}\right)} - \right.
$$

$$
\left. \frac{2(2x-y)(2x+y)2.911}{\left(\sqrt{2}\sqrt{x}\right)\left( \sqrt{\pi\left(\frac{2x+y}{\sqrt{2}\sqrt{x}}\right)^2 + 2.911^2} + \frac{(2.911-1)\sqrt{\pi}(2x+y)}{\sqrt{2}\sqrt{x}} \right)} \right) \sqrt{\pi} - 0.0131259 =
$$

$$
(3x - y) + \left( \frac{\left(\sqrt{2}\sqrt{x}\,2.911\right)(x-y)(x+y)}{\left( \sqrt{\pi(x+y)^2 + 2\cdot 2.911^2 x} + (2.911-1)(x+y)\sqrt{\pi} \right)\left(\sqrt{2}\sqrt{x}\right)} - \right.
$$

$$
\left. \frac{2(2x-y)(2x+y)\left(\sqrt{2}\sqrt{x}\,2.911\right)}{\left(\sqrt{2}\sqrt{x}\right)\left( \sqrt{\pi(2x+y)^2 + 2\cdot 2.911^2 x} + (2.911-1)(2x+y)\sqrt{\pi} \right)} \right) \sqrt{\pi} - 0.0131259 =
$$

$$
(3x - y) + 2.911 \left( \frac{(x-y)(x+y)}{(2.911-1)(x+y) + \sqrt{(x+y)^2 + \frac{2\cdot 2.911^2 x}{\pi}}} - \right.
$$

$$
\left. \frac{2(2x-y)(2x+y)}{(2.911-1)(2x+y) + \sqrt{(2x+y)^2 + \frac{2\cdot 2.911^2 x}{\pi}}} \right) - 0.0131259 \geqslant
$$

$$
(3x - y) + 2.911 \left( \frac{(x-y)(x+y)}{(2.911-1)(x+y) + \sqrt{\left(\frac{2.911^2}{\pi}\right)^2 + (x+y)^2 + \frac{2\cdot 2.911^2 x}{\pi} + \frac{2\cdot 2.911^2 y}{\pi}}} - \right.
$$

$$
\left. \frac{2(2x-y)(2x+y)}{(2.911-1)(2x+y) + \sqrt{(2x+y)^2 + \frac{2\cdot 2.911^2 x}{\pi}}} \right) - 0.0131259 =
$$

$$
(3x - y) + 2.911 \left( \frac{(x-y)(x+y)}{(2.911-1)(x+y) + \sqrt{\left(x + y + \frac{2.911^2}{\pi}\right)^2}} - \right.
$$

$$
\left. \frac{2(2x-y)(2x+y)}{(2.911-1)(2x+y) + \sqrt{(2x+y)^2 + \frac{2\cdot 2.911^2 x}{\pi}}} \right) - 0.0131259 =
$$

$$(3x - y) + 2.911 \left( \frac{(x-y)(x+y)}{2.911(x+y) + \frac{2.911^2}{\pi}} - \frac{2(2x-y)(2x+y)}{(2.911-1)(2x+y) + \sqrt{(2x+y)^2 + \frac{2 \cdot 2.911^2 x}{\pi}}} \right) - 0.0131259 =$$

$$(3x - y) + \frac{(x-y)(x+y)}{(x+y) + \frac{2.911}{\pi}} - \frac{2(2x-y)(2x+y)2.911}{(2.911-1)(2x+y) + \sqrt{(2x+y)^2 + \frac{2 \cdot 2.911^2 x}{\pi}}} - 0.0131259 =$$

$$\left( -2(2x-y)2.911 \left( (x+y) + \frac{2.911}{\pi} \right)(2x+y) \; + \right.$$

$$\left( (x+y) + \frac{2.911}{\pi} \right)(3x - y - 0.0131259) \left( (2.911-1)(2x+y) + \sqrt{(2x+y)^2 + \frac{2 \cdot 2.911^2 x}{\pi}} \right) \; +$$

$$(x-y)(x+y) \left. \left( (2.911-1)(2x+y) + \sqrt{(2x+y)^2 + \frac{2 \cdot 2.911^2 x}{\pi}} \right) \right)$$

$$\left( \left( (x+y) + \frac{2.911}{\pi} \right) \left( (2.911-1)(2x+y) + \sqrt{(2x+y)^2 + \frac{2 \cdot 2.911^2 x}{\pi}} \right) \right)^{-1} =$$

$$\left( ((x-y)(x+y) + (3x - y - 0.0131259)(x+y+0.9266)) \left( \sqrt{(2x+y)^2 + 5.39467x} + 3.822x + 1.911y \right) \right. -$$

(239)

$$5.822(2x-y)(x+y+0.9266)(2x+y))$$

$$\left( \left( (x+y) + \frac{2.911}{\pi} \right) \left( (2.911-1)(2x+y) + \sqrt{(2x+y)^2 + \frac{22.911^2 x}{\pi}} \right) \right)^{-1} > 0 \, .$$

We explain this sequence of inequalities:

- First inequality: The approximation of Ren and MacKenzie (2007) and then subtracting a safety gap (which would not be necessary for the current analysis).

- Equalities: The factor $\sqrt{2}\sqrt{x}$ is factored out and canceled.

- Second inequality: adds a positive term in the first root to obtain a binomial form. The term containing the root is positive and the root is in the denominator, therefore the whole term becomes smaller.

- Equalities: solve for the term and factor out.

- Bringing all terms to the denominator $\left( (x+y) + \frac{2.911}{\pi} \right) \left( (2.911-1)(2x+y) + \sqrt{(2x+y)^2 + \frac{2 \cdot 2.911^2 x}{\pi}} \right)$.

- Equalities: Multiplying out and expanding terms.

- Last inequality $> 0$ is proofed in the following sequence of inequalities.

We look at the numerator of the last expression of Eq. (238), which we show to be positive in order to show $> 0$ in Eq. (238). The numerator is

$$((x-y)(x+y) + (3x - y - 0.0131259)(x+y+0.9266)) \left( \sqrt{(2x+y)^2 + 5.39467x} + 3.822x + 1.911y \right) -$$

(240)

$5.822(2x - y)(x + y + 0.9266)(2x + y) =$

$- 5.822(2x - y)(x + y + 0.9266)(2x + y) + (3.822x + 1.911y)((x - y)(x + y)+$

$(3x - y - 0.0131259)(x + y + 0.9266)) + ((x - y)(x + y)+$

$(3x - y - 0.0131259)(x + y + 0.9266))\sqrt{(2x + y)^2 + 5.39467x} =$

$- 8.0x^3 + \left(4x^2 + 2xy + 2.76667x - 2y^2 - 0.939726y - 0.0121625\right)\sqrt{(2x + y)^2 + 5.39467x} -$

$8.0x^2y - 11.0044x^2 + 2.0xy^2 + 1.69548xy - 0.0464849x + 2.0y^3 + 3.59885y^2 - 0.0232425y =$

$- 8.0x^3 + \left(4x^2 + 2xy + 2.76667x - 2y^2 - 0.939726y - 0.0121625\right)\sqrt{(2x + y)^2 + 5.39467x} -$

$8.0x^2y - 11.0044x^2 + 2.0xy^2 + 1.69548xy - 0.0464849x + 2.0y^3 + 3.59885y^2 - 0.0232425y$ .

The factor in front of the root is positive. If the term, that does not contain the root, was positive, then the whole expression would be positive and we would have proofed that the numerator is positive. Therefore we consider the case that the term, that does not contain the root, is negative. The term that contains the root must be larger than the other term in absolute values.

$$- \left(-8.0x^3 - 8.0x^2y - 11.0044x^2 + 2.xy^2 + 1.69548xy - 0.0464849x + 2.y^3 + 3.59885y^2 - 0.0232425y\right) <$$
$$\tag{241}$$

$$\left(4x^2 + 2xy + 2.76667x - 2y^2 - 0.939726y - 0.0121625\right)\sqrt{(2x + y)^2 + 5.39467x}\,.$$

Therefore the squares of the root term have to be larger than the square of the other term to show $> 0$ in Eq. (238). Thus, we have the inequality:

$$\left(-8.0x^3 - 8.0x^2y - 11.0044x^2 + 2.xy^2 + 1.69548xy - 0.0464849x + 2.y^3 + 3.59885y^2 - 0.0232425y\right)^2 <$$
$$\tag{242}$$

$$\left(4x^2 + 2xy + 2.76667x - 2y^2 - 0.939726y - 0.0121625\right)^2 \left((2x + y)^2 + 5.39467x\right)\,.$$

This is equivalent to

$$0 < \left(4x^2 + 2xy + 2.76667x - 2y^2 - 0.939726y - 0.0121625\right)^2 \left((2x + y)^2 + 5.39467x\right) -$$
$$\tag{243}$$

$$\left(-8.0x^3 - 8.0x^2y - 11.0044x^2 + 2.0xy^2 + 1.69548xy - 0.0464849x + 2.0y^3 + 3.59885y^2 - 0.0232425y\right)^2 =$$

$- 1.2227x^5 + 40.1006x^4y + 27.7897x^4 + 41.0176x^3y^2 + 64.5799x^3y + 39.4762x^3 + 10.9422x^2y^3 -$

$13.543x^2y^2 - 28.8455x^2y - 0.364625x^2 + 0.611352xy^4 + 6.83183xy^3 + 5.46393xy^2 +$

$0.121746xy + 0.000798008x - 10.6365y^5 - 11.927y^4 + 0.190151y^3 - 0.000392287y^2$ .

We obtain the inequalities:

$$- 1.2227x^5 + 40.1006x^4y + 27.7897x^4 + 41.0176x^3y^2 + 64.5799x^3y + 39.4762x^3 + 10.9422x^2y^3 -$$
$$\tag{244}$$

$13.543x^2y^2 - 28.8455x^2y - 0.364625x^2 + 0.611352xy^4 + 6.83183xy^3 + 5.46393xy^2 +$

$0.121746xy + 0.000798008x - 10.6365y^5 - 11.927y^4 + 0.190151y^3 - 0.000392287y^2 =$

$- 1.2227x^5 + 27.7897x^4 + 41.0176x^3y^2 + 39.4762x^3 - 13.543x^2y^2 - 0.364625x^2 +$

$y\left(40.1006x^4 + 64.5799x^3 + 10.9422x^2y^2 - 28.8455x^2 + 6.83183xy^2 + 0.121746x - \right.$

$10.6365y^4 + 0.190151y^2) + 0.611352xy^4 + 5.46393xy^2 + 0.000798008x - 11.927y^4 - 0.000392287y^2 \;>$

$-\; 1.2227x^5 + 27.7897x^4 + 41.0176 \cdot (0.0)^2 x^3 + 39.4762x^3 - 13.543 \cdot (0.1)^2 x^2 - 0.364625x^2 -$

$0.1 \cdot \big(40.1006x^4 + 64.5799x^3 + 10.9422 \cdot (0.1)^2 x^2 - 28.8455x^2 + 6.83183 \cdot (0.1)^2 x + 0.121746x \;+$

$10.6365 \cdot (0.1)^4 + 0.190151 \cdot (0.1)^2\big) +$

$0.611352 \cdot (0.0)^4 x + 5.46393 \cdot (0.0)^2 x + 0.000798008x - 11.927 \cdot (0.1)^4 - 0.000392287 \cdot (0.1)^2 \;=$

$-\; 1.2227x^5 + 23.7796x^4 + (20 + 13.0182)x^3 + 2.37355x^2 - 0.0182084x - 0.000194074 \;\geqslant$

$-\; 1.2227x^5 + 24.7796x^4 + 13.0182x^3 + 2.37355x^2 - 0.0182084x - 0.000194074 \;>$

$13.0182x^3 + 2.37355x^2 - 0.0182084x - 0.000194074 \;>\; 0 \,.$

We used $24.7796 \cdot (20)^4 - 1.2227 \cdot (20)^5 = 52090.9 > 0$ and $x \leqslant 20$. We have proofed the last inequality $> 0$ of Eq. (238).

Consequently the derivative is always positive independent of $y$, thus

$$e^{\frac{(x+y)^2}{2x}} \operatorname{erfc}\left(\frac{x+y}{\sqrt{2}\sqrt{x}}\right) - 2e^{\frac{(2x+y)^2}{2x}} \operatorname{erfc}\left(\frac{2x+y}{\sqrt{2}\sqrt{x}}\right) \tag{245}$$

is strictly monotonically increasing in $x$.

**The main subfunction is smaller than zero.**   Next we show that the sub-function Eq. (90) is smaller than zero. We consider the limit:

$$\lim_{x\to\infty} e^{\frac{(x+y)^2}{2x}} \operatorname{erfc}\left(\frac{x+y}{\sqrt{2}\sqrt{x}}\right) - 2e^{\frac{(2x+y)^2}{2x}} \operatorname{erfc}\left(\frac{2x+y}{\sqrt{2}\sqrt{x}}\right) \;=\; 0 \tag{246}$$

The limit follows from Lemma 22. Since the function is monotonic increasing in $x$, it has to approach 0 from below. Thus,

$$e^{\frac{(x+y)^2}{2x}} \operatorname{erfc}\left(\frac{x+y}{\sqrt{2}\sqrt{x}}\right) - 2e^{\frac{(2x+y)^2}{2x}} \operatorname{erfc}\left(\frac{2x+y}{\sqrt{2}\sqrt{x}}\right) \tag{247}$$

is smaller than zero.

**Behavior of the main subfunction with respect to $y$ at minimal $x$.**   We now consider the derivative of sub-function Eq. (90) with respect to $y$. We proofed that sub-function Eq. (90) is strictly monotonically increasing independent of $y$. In the proof of Theorem 16, we need the minimum of sub-function Eq. (90). Therefore we are only interested in the derivative of sub-function Eq. (90) with respect to $y$ for the minimum $x = 12/10 = 1.2$

Consequently, we insert the minimum $x = 12/10 = 1.2$ into the sub-function Eq. (90). The main terms become

$$\frac{x+y}{\sqrt{2}\sqrt{x}} \;=\; \frac{y+1.2}{\sqrt{2}\sqrt{1.2}} \;=\; \frac{y}{\sqrt{2}\sqrt{1.2}} + \frac{\sqrt{1.2}}{\sqrt{2}} \;=\; \frac{5y+6}{2\sqrt{15}} \tag{248}$$

and

$$\frac{2x+y}{\sqrt{2}\sqrt{x}} \;=\; \frac{y+1.2 \cdot 2}{\sqrt{2}\sqrt{1.2}} \;=\; \frac{y}{\sqrt{2}\sqrt{1.2}} + \sqrt{1.2}\sqrt{2} \;=\; \frac{5y+12}{2\sqrt{15}} \,. \tag{249}$$

Sub-function Eq. ([90](#)) becomes:

$$e^{\left(\frac{y}{\sqrt{2}\sqrt{\frac{12}{10}}}+\frac{\sqrt{\frac{12}{10}}}{\sqrt{2}}\right)^2}\operatorname{erfc}\left(\frac{y}{\sqrt{2}\sqrt{\frac{12}{10}}}+\frac{\sqrt{\frac{12}{10}}}{\sqrt{2}}\right)-2e^{\left(\frac{y}{\sqrt{2}\sqrt{\frac{12}{10}}}+\sqrt{2}\sqrt{\frac{12}{10}}\right)^2}\operatorname{erfc}\left(\frac{y}{\sqrt{2}\sqrt{\frac{12}{10}}}+\sqrt{2}\sqrt{\frac{12}{10}}\right)\ .$$

$$(250)$$

The derivative of this function with respect to $y$ is

$$\frac{\sqrt{15\pi}\left(e^{\frac{1}{60}(5y+6)^2}(5y+6)\operatorname{erfc}\left(\frac{5y+6}{2\sqrt{15}}\right)-2e^{\frac{1}{60}(5y+12)^2}(5y+12)\operatorname{erfc}\left(\frac{5y+12}{2\sqrt{15}}\right)\right)+30}{6\sqrt{15\pi}}\ .\quad(251)$$

We again will use the approximation of Ren and MacKenzie (2007)

$$e^{z^2}\operatorname{erfc}(z)\ =\ \frac{2.911}{\sqrt{\pi}(2.911-1)z+\sqrt{\pi z^2+2.911^2}}\ .\qquad(252)$$

Therefore we first perform an error analysis. We estimated the maximum and minimum of

$$\sqrt{15\pi}\left(\frac{2\cdot 2.911(5y+12)}{\frac{\sqrt{\pi}(2.911-1)(5y+12)}{2\sqrt{15}}+\sqrt{\pi\left(\frac{5y+12}{2\sqrt{15}}\right)^2+2.911^2}}-\frac{2.911(5y+6)}{\frac{\sqrt{\pi}(2.911-1)(5y+6)}{2\sqrt{15}}+\sqrt{\pi\left(\frac{5y+6}{2\sqrt{15}}\right)^2+2.911^2}}\right)+30+$$

$$(253)$$

$$\sqrt{15\pi}\left(e^{\frac{1}{60}(5y+6)^2}(5y+6)\operatorname{erfc}\left(\frac{5y+6}{2\sqrt{15}}\right)-2e^{\frac{1}{60}(5y+12)^2}(5y+12)\operatorname{erfc}\left(\frac{5y+12}{2\sqrt{15}}\right)\right)+30\ .$$

We obtained for the maximal absolute error the value $0.163052$. We added an approximation error of $0.2$ to the approximation of the derivative. Since we want to show that the approximation upper bounds the true expression, the addition of the approximation error is required here. We get a sequence of inequalities:

$$\sqrt{15\pi}\left(e^{\frac{1}{60}(5y+6)^2}(5y+6)\operatorname{erfc}\left(\frac{5y+6}{2\sqrt{15}}\right)-2e^{\frac{1}{60}(5y+12)^2}(5y+12)\operatorname{erfc}\left(\frac{5y+12}{2\sqrt{15}}\right)\right)+30\ \leqslant$$

$$(254)$$

$$\sqrt{15\pi}\left(\frac{2.911(5y+6)}{\frac{\sqrt{\pi}(2.911-1)(5y+6)}{2\sqrt{15}}+\sqrt{\pi\left(\frac{5y+6}{2\sqrt{15}}\right)^2+2.911^2}}-\frac{2\cdot 2.911(5y+12)}{\frac{\sqrt{\pi}(2.911-1)(5y+12)}{2\sqrt{15}}+\sqrt{\pi\left(\frac{5y+12}{2\sqrt{15}}\right)^2+2.911^2}}\right)+$$

$$30+0.2\ =$$

$$\frac{(30\cdot 2.911)(5y+6)}{(2.911-1)(5y+6)+\sqrt{(5y+6)^2+\left(\frac{2\sqrt{15}\cdot 2.911}{\sqrt{\pi}}\right)^2}}-\frac{2(30\cdot 2.911)(5y+12)}{(2.911-1)(5y+12)+\sqrt{(5y+12)^2+\left(\frac{2\sqrt{15}\cdot 2.911}{\sqrt{\pi}}\right)^2}}+$$

$$30+0.2\ =$$

$$\left( (0.2 + 30) \left( (2.911 - 1)(5y + 12) + \sqrt{(5y + 12)^2 + \left( \frac{2\sqrt{15} \cdot 2.911}{\sqrt{\pi}} \right)^2} \right) \right.$$

$$\left( (2.911 - 1)(5y + 6) + \sqrt{(5y + 6)^2 + \left( \frac{2\sqrt{15} \cdot 2.911}{\sqrt{\pi}} \right)^2} \right) -$$

$$2 \cdot 30 \cdot 2.911(5y + 12) \left( (2.911 - 1)(5y + 6) + \sqrt{(5y + 6)^2 + \left( \frac{2\sqrt{15} \cdot 2.911}{\sqrt{\pi}} \right)^2} \right) +$$

$$\left. 2.911 \cdot 30(5y + 6) \left( (2.911 - 1)(5y + 12) + \sqrt{(5y + 12)^2 + \left( \frac{2\sqrt{15} \cdot 2.911}{\sqrt{\pi}} \right)^2} \right) \right)$$

$$\left( \left( (2.911 - 1)(5y + 6) + \sqrt{(5y + 6)^2 + \left( \frac{2\sqrt{15} \cdot 2.911}{\sqrt{\pi}} \right)^2} \right) \right.$$

$$\left. \left( (2.911 - 1)(5y + 12) + \sqrt{(5y + 12)^2 + \left( \frac{2\sqrt{15} \cdot 2.911}{\sqrt{\pi}} \right)^2} \right) \right)^{-1} \; < \; 0 \, .$$

We explain this sequence of inequalities.

- First inequality: The approximation of Ren and MacKenzie (2007) and then adding the error bound to ensure that the approximation is larger than the true value.

- First equality: The factor $2\sqrt{15}$ and $2\sqrt{\pi}$ are factored out and canceled.

- Second equality: Bringing all terms to the denominator

$$\left( (2.911 - 1)(5y + 6) + \sqrt{(5y + 6)^2 + \left( \frac{2\sqrt{15}2.911}{\sqrt{\pi}} \right)^2} \right) \tag{255}$$

$$\left( (2.911 - 1)(5y + 12) + \sqrt{(5y + 12)^2 + \left( \frac{2\sqrt{15} \cdot 2.911}{\sqrt{\pi}} \right)^2} \right) \, .$$

- Last inequality $< 0$ is proofed in the following sequence of inequalities.

We look at the numerator of the last term in Eq. (254). We have to proof that this numerator is smaller than zero in order to proof the last inequality of Eq. (254). The numerator is

$$(0.2 + 30) \left( (2.911 - 1)(5y + 12) + \sqrt{(5y + 12)^2 + \left( \frac{2\sqrt{15} \cdot 2.911}{\sqrt{\pi}} \right)^2} \right) \tag{256}$$

$$\left( (2.911 - 1)(5y + 6) + \sqrt{(5y + 6)^2 + \left( \frac{2\sqrt{15} \cdot 2.911}{\sqrt{\pi}} \right)^2} \right) -$$

$$2 \cdot 30 \cdot 2.911(5y + 12) \left( (2.911 - 1)(5y + 6) + \sqrt{(5y + 6)^2 + \left( \frac{2\sqrt{15} \cdot 2.911}{\sqrt{\pi}} \right)^2} \right) +$$

$$2.911 \cdot 30(5y + 6) \left( (2.911 - 1)(5y + 12) + \sqrt{(5y + 12)^2 + \left( \frac{2\sqrt{15} \cdot 2.911}{\sqrt{\pi}} \right)^2} \right) .$$

We now compute upper bounds for this numerator:

$$(0.2 + 30) \left( (2.911 - 1)(5y + 12) + \sqrt{(5y + 12)^2 + \left( \frac{2\sqrt{15} \cdot 2.911}{\sqrt{\pi}} \right)^2} \right) \tag{257}$$

$$\left( (2.911 - 1)(5y + 6) + \sqrt{(5y + 6)^2 + \left( \frac{2\sqrt{15} \cdot 2.911}{\sqrt{\pi}} \right)^2} \right) -$$

$$2 \cdot 30 \cdot 2.911(5y + 12) \left( (2.911 - 1)(5y + 6) + \sqrt{(5y + 6)^2 + \left( \frac{2\sqrt{15} \cdot 2.911}{\sqrt{\pi}} \right)^2} \right) +$$

$$2.911 \cdot 30(5y + 6) \left( (2.911 - 1)(5y + 12) + \sqrt{(5y + 12)^2 + \left( \frac{2\sqrt{15} \cdot 2.911}{\sqrt{\pi}} \right)^2} \right) =$$

$$-1414.99y^2 - 584.739\sqrt{(5y + 6)^2 + 161.84}y + 725.211\sqrt{(5y + 12)^2 + 161.84}y -$$

$$5093.97y - 1403.37\sqrt{(5y + 6)^2 + 161.84} + 30.2\sqrt{(5y + 6)^2 + 161.84}\sqrt{(5y + 12)^2 + 161.84} +$$

$$870.253\sqrt{(5y + 12)^2 + 161.84} - 4075.17 \ <$$

$$-1414.99y^2 - 584.739\sqrt{(5y + 6)^2 + 161.84}y + 725.211\sqrt{(5y + 12)^2 + 161.84}y -$$

$$5093.97y - 1403.37\sqrt{(6 + 5 \cdot (-0.1))^2 + 161.84} + 30.2\sqrt{(6 + 5 \cdot 0.1)^2 + 161.84}\sqrt{(12 + 5 \cdot 0.1)^2 + 161.84} +$$

$$870.253\sqrt{(12 + 5 \cdot 0.1)^2 + 161.84} - 4075.17 \ =$$

$$-1414.99y^2 - 584.739\sqrt{(5y + 6)^2 + 161.84}y + 725.211\sqrt{(5y + 12)^2 + 161.84}y - 5093.97y - 309.691 \ <$$

$$y \left( -584.739\sqrt{(5y + 6)^2 + 161.84} + 725.211\sqrt{(5y + 12)^2 + 161.84} - 5093.97 \right) - 309.691 \ <$$

$$-0.1 \left( 725.211\sqrt{(12 + 5 \cdot (-0.1))^2 + 161.84} - 584.739\sqrt{(6 + 5 \cdot 0.1)^2 + 161.84} - 5093.97 \right) - 309.691 \ =$$

$$-208.604 \ .$$

For the first inequality we choose $y$ in the roots, so that positive terms maximally increase and negative terms maximally decrease. The second inequality just removed the $y^2$ term which is always negative, therefore increased the expression. For the last inequality, the term in brackets is

negative for all settings of $y$. Therefore we make the brackets as negative as possible and make the whole term positive by multiplying with $y = -0.1$.

Consequently

$$e^{\frac{(x+y)^2}{2x}} \operatorname{erfc}\left(\frac{x+y}{\sqrt{2}\sqrt{x}}\right) - 2e^{\frac{(2x+y)^2}{2x}} \operatorname{erfc}\left(\frac{2x+y}{\sqrt{2}\sqrt{x}}\right) \tag{258}$$

is strictly monotonically decreasing in $y$ for the minimal $x = 1.2$. $\qquad\square$

**Lemma 45** (Main subfunction below). *For $0.007 \leqslant x \leqslant 0.875$ and $-0.01 \leqslant y \leqslant 0.01$, the function*

$$e^{\frac{(x+y)^2}{2x}} \operatorname{erfc}\left(\frac{x+y}{\sqrt{2}\sqrt{x}}\right) - 2e^{\frac{(2x+y)^2}{2x}} \operatorname{erfc}\left(\frac{2x+y}{\sqrt{2}\sqrt{x}}\right) \tag{259}$$

*smaller than zero, is strictly monotonically increasing in $x$ and strictly monotonically increasing in $y$ for the minimal $x = 0.007 = 0.00875 \cdot 0.8$, $x = 0.56 = 0.7 \cdot 0.8$, $x = 0.128 = 0.16 \cdot 0.8$, and $x = 0.216 = 0.24 \cdot 0.9$ (lower bound of $0.9$ on $\tau$).*

*Proof.* We first consider the derivative of sub-function Eq. (100) with respect to $x$. The derivative of the function

$$e^{\frac{(x+y)^2}{2x}} \operatorname{erfc}\left(\frac{x+y}{\sqrt{2}\sqrt{x}}\right) - 2e^{\frac{(2x+y)^2}{2x}} \operatorname{erfc}\left(\frac{2x+y}{\sqrt{2}\sqrt{x}}\right) \tag{260}$$

with respect to $x$ is

$$\frac{\sqrt{\pi}\left(e^{\frac{(x+y)^2}{2x}}(x-y)(x+y)\operatorname{erfc}\left(\frac{x+y}{\sqrt{2}\sqrt{x}}\right) - 2e^{\frac{(2x+y)^2}{2x}}\left(4x^2 - y^2\right)\operatorname{erfc}\left(\frac{2x+y}{\sqrt{2}\sqrt{x}}\right)\right) + \sqrt{2}\sqrt{x}(3x - y)}{2\sqrt{\pi}x^2} = \tag{261}$$

$$\frac{\sqrt{\pi}\left(e^{\frac{(x+y)^2}{2x}}(x-y)(x+y)\operatorname{erfc}\left(\frac{x+y}{\sqrt{2}\sqrt{x}}\right) - 2e^{\frac{(2x+y)^2}{2x}}(2x+y)(2x-y)\operatorname{erfc}\left(\frac{2x+y}{\sqrt{2}\sqrt{x}}\right)\right) + \sqrt{2}\sqrt{x}(3x - y)}{2\sqrt{\pi}x^2} =$$

$$\frac{\sqrt{\pi}\left(\frac{e^{\frac{(x+y)^2}{2x}}(x-y)(x+y)\operatorname{erfc}\left(\frac{x+y}{\sqrt{2}\sqrt{x}}\right)}{\sqrt{2}\sqrt{x}} - \frac{2e^{\frac{(2x+y)^2}{2x}}(2x+y)(2x-y)\operatorname{erfc}\left(\frac{2x+y}{\sqrt{2}\sqrt{x}}\right)}{\sqrt{2}\sqrt{x}}\right) + (3x - y)}{\sqrt{2}2\sqrt{\pi}\sqrt{x}x^2}.$$

We consider the numerator

$$\sqrt{\pi}\left(\frac{e^{\frac{(x+y)^2}{2x}}(x-y)(x+y)\operatorname{erfc}\left(\frac{x+y}{\sqrt{2}\sqrt{x}}\right)}{\sqrt{2}\sqrt{x}} - \frac{2e^{\frac{(2x+y)^2}{2x}}(2x+y)(2x-y)\operatorname{erfc}\left(\frac{2x+y}{\sqrt{2}\sqrt{x}}\right)}{\sqrt{2}\sqrt{x}}\right) + (3x - y). \tag{262}$$

For bounding this value, we use the approximation

$$e^{z^2}\operatorname{erfc}(z) \approx \frac{2.911}{\sqrt{\pi}(2.911 - 1)z + \sqrt{\pi z^2 + 2.911^2}}. \tag{263}$$

from Ren and MacKenzie (2007). We start with an error analysis of this approximation. According to Ren and MacKenzie (2007) (Figure 1), the approximation error is both positive and negative in the range $[0.175, 1.33]$. This range contains all possible arguments of erfc that we consider in this subsection. Numerically we maximized and minimized the approximation error of the whole expression

$$E(x, y) = \left( \frac{e^{\frac{(x+y)^2}{2x}}(x-y)(x+y)\,\mathrm{erfc}\left(\frac{x+y}{\sqrt{2}\sqrt{x}}\right)}{\sqrt{2}\sqrt{x}} - \frac{2e^{\frac{(2x+y)^2}{2x}}(2x-y)(2x+y)\,\mathrm{erfc}\left(\frac{2x+y}{\sqrt{2}\sqrt{x}}\right)}{\sqrt{2}\sqrt{x}} \right) -$$

(264)

$$\left( \frac{2.911(x-y)(x+y)}{\left(\sqrt{2}\sqrt{x}\right)\left( \frac{\sqrt{\pi}(2.911-1)(x+y)}{\sqrt{2}\sqrt{x}} + \sqrt{\pi\left(\frac{x+y}{\sqrt{2}\sqrt{x}}\right)^2 + 2.911^2} \right)} - \right.$$

$$\left. \frac{2 \cdot 2.911(2x-y)(2x+y)}{\left(\sqrt{2}\sqrt{x}\right)\left( \frac{\sqrt{\pi}(2.911-1)(2x+y)}{\sqrt{2}\sqrt{x}} + \sqrt{\pi\left(\frac{2x+y}{\sqrt{2}\sqrt{x}}\right)^2 + 2.911^2} \right)} \right).$$

We numerically determined $-0.000228141 \leqslant E(x, y) \leqslant 0.00495688$ for $0.08 \leqslant x \leqslant 0.875$ and $-0.01 \leqslant y \leqslant 0.01$. We used different numerical optimization techniques like gradient based constraint BFGS algorithms and non-gradient-based Nelder-Mead methods with different start points. Therefore our approximation is smaller than the function that we approximate.

We use an error gap of $-0.0003$ to countermand the error due to the approximation. We have the sequences of inequalities using the approximation of Ren and MacKenzie (2007):

$$(3x - y) + \left( \frac{e^{\frac{(x+y)^2}{2x}}(x-y)(x+y)\,\mathrm{erfc}\left(\frac{x+y}{\sqrt{2}\sqrt{x}}\right)}{\sqrt{2}\sqrt{x}} - \frac{2e^{\frac{(2x+y)^2}{2x}}(2x-y)(2x+y)\,\mathrm{erfc}\left(\frac{2x+y}{\sqrt{2}\sqrt{x}}\right)}{\sqrt{2}\sqrt{x}} \right)\sqrt{\pi} \geqslant$$

(265)

$$(3x - y) + \left( \frac{2.911(x-y)(x+y)}{\left( \sqrt{\pi\left(\frac{x+y}{\sqrt{2}\sqrt{x}}\right)^2 + 2.911^2} + \frac{(2.911-1)\sqrt{\pi}(x+y)}{\sqrt{2}\sqrt{x}} \right)\left(\sqrt{2}\sqrt{x}\right)} - \right.$$

$$\left. \frac{2(2x-y)(2x+y)2.911}{\left(\sqrt{2}\sqrt{x}\right)\left( \sqrt{\pi\left(\frac{2x+y}{\sqrt{2}\sqrt{x}}\right)^2 + 2.911^2} + \frac{(2.911-1)\sqrt{\pi}(2x+y)}{\sqrt{2}\sqrt{x}} \right)} \right)\sqrt{\pi} - 0.0003 =$$

$$(3x - y) + \left( \frac{\left(\sqrt{2}\sqrt{x}2.911\right)(x-y)(x+y)}{\left(\sqrt{\pi(x+y)^2 + 2\cdot 2.911^2 x} + (2.911-1)(x+y)\sqrt{\pi}\right)\left(\sqrt{2}\sqrt{x}\right)} - \right.$$

$$\left. \frac{2(2x-y)(2x+y)\left(\sqrt{2}\sqrt{x}2.911\right)}{\left(\sqrt{2}\sqrt{x}\right)\left(\sqrt{\pi(2x+y)^2 + 2\cdot 2.911^2 x} + (2.911-1)(2x+y)\sqrt{\pi}\right)} \right) \sqrt{\pi} - 0.0003 =$$

$$(3x - y) + 2.911\left( \frac{(x-y)(x+y)}{(2.911-1)(x+y) + \sqrt{(x+y)^2 + \frac{2\cdot 2.911^2 x}{\pi}}} - \right.$$

$$\left. \frac{2(2x-y)(2x+y)}{(2.911-1)(2x+y) + \sqrt{(2x+y)^2 + \frac{2\cdot 2.911^2 x}{\pi}}} \right) - 0.0003 \geqslant$$

$$(3x - y) + 2.911\left( \frac{(x-y)(x+y)}{(2.911-1)(x+y) + \sqrt{\left(\frac{2.911^2}{\pi}\right)^2 + (x+y)^2 + \frac{2\cdot 2.911^2 x}{\pi} + \frac{2\cdot 2.911^2 y}{\pi}}} - \right.$$

$$\left. \frac{2(2x-y)(2x+y)}{(2.911-1)(2x+y) + \sqrt{(2x+y)^2 + \frac{2\cdot 2.911^2 x}{\pi}}} \right) - 0.0003 =$$

$$(3x - y) + 2.911\left( \frac{(x-y)(x+y)}{(2.911-1)(x+y) + \sqrt{\left(x+y + \frac{2.911^2}{\pi}\right)^2}} - \right.$$

$$\left. \frac{2(2x-y)(2x+y)}{(2.911-1)(2x+y) + \sqrt{(2x+y)^2 + \frac{2\cdot 2.911^2 x}{\pi}}} \right) - 0.0003 =$$

$$(3x - y) + 2.911\left( \frac{(x-y)(x+y)}{2.911(x+y) + \frac{2.911^2}{\pi}} - \frac{2(2x-y)(2x+y)}{(2.911-1)(2x+y) + \sqrt{(2x+y)^2 + \frac{2\cdot 2.911^2 x}{\pi}}} \right) - 0.0003 =$$

$$(3x - y) + \frac{(x-y)(x+y)}{(x+y) + \frac{2.911}{\pi}} - \frac{2(2x-y)(2x+y)2.911}{(2.911-1)(2x+y) + \sqrt{(2x+y)^2 + \frac{2\cdot 2.911^2 x}{\pi}}} - 0.0003 =$$

$$(3x - y) + \frac{(x-y)(x+y)}{(x+y) + \frac{2.911}{\pi}} - \frac{2(2x-y)(2x+y)2.911}{(2.911-1)(2x+y) + \sqrt{(2x+y)^2 + \frac{2\cdot 2.911^2 x}{\pi}}} - 0.0003 =$$

$$\left( -2(2x-y)2.911\left((x+y) + \frac{2.911}{\pi}\right)(2x+y) + \right.$$

$$\left((x+y) + \frac{2.911}{\pi}\right)(3x - y - 0.0003)\left((2.911-1)(2x+y) + \sqrt{(2x+y)^2 + \frac{2\cdot 2.911^2 x}{\pi}}\right) +$$

$$\left. (x-y)(x+y)\left((2.911-1)(2x+y) + \sqrt{(2x+y)^2 + \frac{2\cdot 2.911^2 x}{\pi}}\right)\right)$$

$$\left(\left((x+y)+\frac{2.911}{\pi}\right)\left((2.911-1)(2x+y)+\sqrt{(2x+y)^2+\frac{2\cdot2.911^2x}{\pi}}\right)\right)^{-1} =$$

$$\left(-8x^3-8x^2y+4x^2\sqrt{(2x+y)^2+5.39467x}-10.9554x^2+2xy^2-2y^2\sqrt{(2x+y)^2+5.39467x}+\right.$$

$$1.76901xy+2xy\sqrt{(2x+y)^2+5.39467x}+2.7795x\sqrt{(2x+y)^2+5.39467x}-$$

$$0.9269y\sqrt{(2x+y)^2+5.39467x}-0.00027798\sqrt{(2x+y)^2+5.39467x}-0.00106244x+$$

$$\left.2y^3+3.62336y^2-0.00053122y\right)$$

$$\left(\left((x+y)+\frac{2.911}{\pi}\right)\left((2.911-1)(2x+y)+\sqrt{(2x+y)^2+\frac{2\cdot2.911^2x}{\pi}}\right)\right)^{-1} =$$

$$\left(-8x^3+\left(4x^2+2xy+2.7795x-2y^2-0.9269y-0.00027798\right)\sqrt{(2x+y)^2+5.39467x}-\right.$$

$$\left.8x^2y-10.9554x^2+2xy^2+1.76901xy-0.00106244x+2y^3+3.62336y^2-0.00053122y\right)$$

$$\left(\left((x+y)+\frac{2.911}{\pi}\right)\left((2.911-1)(2x+y)+\sqrt{(2x+y)^2+\frac{2\cdot2.911^2x}{\pi}}\right)\right)^{-1} > 0\,.$$

We explain this sequence of inequalities:

- First inequality: The approximation of Ren and MacKenzie (2007) and then subtracting an error gap of 0.0003.

- Equalities: The factor $\sqrt{2}\sqrt{x}$ is factored out and canceled.

- Second inequality: adds a positive term in the first root to obtain a binomial form. The term containing the root is positive and the root is in the denominator, therefore the whole term becomes smaller.

- Equalities: solve for the term and factor out.

- Bringing all terms to the denominator $\left((x+y)+\frac{2.911}{\pi}\right)\left((2.911-1)(2x+y)+\sqrt{(2x+y)^2+\frac{2\cdot2.911^2x}{\pi}}\right)$.

- Equalities: Multiplying out and expanding terms.

- Last inequality $>0$ is proofed in the following sequence of inequalities.

We look at the numerator of the last expression of Eq. (265), which we show to be positive in order to show $>0$ in Eq. (265). The numerator is

$$-8x^3+\left(4x^2+2xy+2.7795x-2y^2-0.9269y-0.00027798\right)\sqrt{(2x+y)^2+5.39467x}-$$
$$\tag{266}$$
$$8x^2y-10.9554x^2+2xy^2+1.76901xy-0.00106244x+2y^3+3.62336y^2-0.00053122y\,.$$

The factor $4x^2+2xy+2.7795x-2y^2-0.9269y-0.00027798$ in front of the root is positive:

$$4x^2+2xy+2.7795x-2y^2-0.9269y-0.00027798 > \tag{267}$$
$$-2y^2+0.007\cdot2y-0.9269y+4\cdot0.007^2+2.7795\cdot0.007-0.00027798 =$$

$$-2y^2 - 0.9129y + 2.77942 \;=\; -2(y + 1.42897)(y - 0.972523) \;>\; 0\,.$$

If the term that does not contain the root would be positive, then everything is positive and we have proofed the the numerator is positive. Therefore we consider the case that the term that does not contain the root is negative. The term that contains the root must be larger than the other term in absolute values.

$$-\left(-8x^3 - 8x^2y - 10.9554x^2 + 2xy^2 + 1.76901xy - 0.00106244x + 2y^3 + 3.62336y^2 - 0.00053122y\right) \;<$$
(268)
$$\left(4x^2 + 2xy + 2.7795x - 2y^2 - 0.9269y - 0.00027798\right)\sqrt{(2x+y)^2 + 5.39467x}\,.$$

Therefore the squares of the root term have to be larger than the square of the other term to show $> 0$ in Eq. (265). Thus, we have the inequality:

$$\left(-8x^3 - 8x^2y - 10.9554x^2 + 2xy^2 + 1.76901xy - 0.00106244x + 2y^3 + 3.62336y^2 - 0.00053122y\right)^2 \;<$$
(269)
$$\left(4x^2 + 2xy + 2.7795x - 2y^2 - 0.9269y - 0.00027798\right)^2\left((2x+y)^2 + 5.39467x\right)\,.$$

This is equivalent to

$$0 \;<\; \left(4x^2 + 2xy + 2.7795x - 2y^2 - 0.9269y - 0.00027798\right)^2\left((2x+y)^2 + 5.39467x\right) -$$
(270)
$$\left(-8x^3 - 8x^2y - 10.9554x^2 + 2xy^2 + 1.76901xy - 0.00106244x + 2y^3 + 3.62336y^2 - 0.00053122y\right)^2 \;=$$
$$x \cdot 4.168614250 \cdot 10^{-7} - y^2 2.049216091 \cdot 10^{-7} - 0.0279456x^5 +$$
$$43.0875x^4y + 30.8113x^4 + 43.1084x^3y^2 + 68.989x^3y + 41.6357x^3 + 10.7928x^2y^3 - 13.1726x^2y^2 -$$
$$27.8148x^2y - 0.00833715x^2 + 0.0139728xy^4 + 5.47537xy^3 +$$
$$4.65089xy^2 + 0.00277916xy - 10.7858y^5 - 12.2664y^4 + 0.00436492y^3\,.$$

We obtain the inequalities:

$$x \cdot 4.168614250 \cdot 10^{-7} - y^2 2.049216091 \cdot 10^{-7} - 0.0279456x^5 + \qquad\qquad (271)$$
$$43.0875x^4y + 30.8113x^4 + 43.1084x^3y^2 + 68.989x^3y + 41.6357x^3 + 10.7928x^2y^3 -$$
$$13.1726x^2y^2 - 27.8148x^2y - 0.00833715x^2 +$$
$$0.0139728xy^4 + 5.47537xy^3 + 4.65089xy^2 + 0.00277916xy - 10.7858y^5 - 12.2664y^4 + 0.00436492y^3 \;>$$
$$x \cdot 4.168614250 \cdot 10^{-7} - (0.01)^2 2.049216091 \cdot 10^{-7} - 0.0279456x^5 +$$
$$0.0 \cdot 43.0875x^4 + 30.8113x^4 + 43.1084(0.0)^2x^3 + 0.0 \cdot 68.989x^3 + 41.6357x^3 +$$
$$10.7928(0.0)^3x^2 - 13.1726(0.01)^2x^2 - 27.8148(0.01)x^2 - 0.00833715x^2 +$$
$$0.0139728(0.0)^4x + 5.47537(0.0)^3x + 4.65089(0.0)^2x +$$
$$0.0 \cdot 0.00277916x - 10.7858(0.01)^5 - 12.2664(0.01)^4 + 0.00436492(0.0)^3 \;=$$
$$x \cdot 4.168614250 \cdot 10^{-7} - 1.237626189 \cdot 10^{-7} - 0.0279456x^5 + 30.8113x^4 + 41.6357x^3 - 0.287802x^2 \;>$$
$$-\left(\frac{x}{0.007}\right)^3 1.237626189 \cdot 10^{-7} + 30.8113x^4 - (0.875)\cdot 0.0279456x^4 + 41.6357x^3 - \frac{(0.287802x)x^2}{0.007} \;=$$

$$30.7869x^4 + 0.160295x^3 \; > \; 0 \; .$$

We used $x \geqslant 0.007$ and $x \leqslant 0.875$ (reducing the negative $x^4$-term to a $x^3$-term). We have proofed the last inequality $> 0$ of Eq. (265).

Consequently the derivative is always positive independent of $y$, thus

$$e^{\frac{(x+y)^2}{2x}} \operatorname{erfc}\left(\frac{x+y}{\sqrt{2}\sqrt{x}}\right) - 2e^{\frac{(2x+y)^2}{2x}} \operatorname{erfc}\left(\frac{2x+y}{\sqrt{2}\sqrt{x}}\right) \tag{272}$$

is strictly monotonically increasing in $x$.

Next we show that the sub-function Eq. (100) is smaller than zero. We consider the limit:

$$\lim_{x\to\infty} e^{\frac{(x+y)^2}{2x}} \operatorname{erfc}\left(\frac{x+y}{\sqrt{2}\sqrt{x}}\right) \; - \; 2e^{\frac{(2x+y)^2}{2x}} \operatorname{erfc}\left(\frac{2x+y}{\sqrt{2}\sqrt{x}}\right) \; = \; 0 \tag{273}$$

The limit follows from Lemma 22. Since the function is monotonic increasing in $x$, it has to approach 0 from below. Thus,

$$e^{\frac{(x+y)^2}{2x}} \operatorname{erfc}\left(\frac{x+y}{\sqrt{2}\sqrt{x}}\right) - 2e^{\frac{(2x+y)^2}{2x}} \operatorname{erfc}\left(\frac{2x+y}{\sqrt{2}\sqrt{x}}\right) \tag{274}$$

is smaller than zero.

We now consider the derivative of sub-function Eq. (100) with respect to $y$. We proofed that sub-function Eq. (100) is strictly monotonically increasing independent of $y$. In the proof of Theorem 3, we need the minimum of sub-function Eq. (100). First, we are interested in the derivative of sub-function Eq. (100) with respect to $y$ for the minimum $x = 0.007 = 7/1000$.

Consequently, we insert the minimum $x = 0.007 = 7/1000$ into the sub-function Eq. (100):

$$e^{\left(\frac{y}{\sqrt{2}\sqrt{\frac{7}{1000}}}+\frac{\sqrt{\frac{7}{1000}}}{\sqrt{2}}\right)^2} \operatorname{erfc}\left(\frac{y}{\sqrt{2}\sqrt{\frac{7}{1000}}}+\frac{\sqrt{\frac{7}{1000}}}{\sqrt{2}}\right) - \tag{275}$$

$$2e^{\left(\frac{y}{\sqrt{2}\sqrt{\frac{7}{1000}}}+\sqrt{2}\sqrt{\frac{7}{1000}}\right)^2} \operatorname{erfc}\left(\frac{y}{\sqrt{2}\sqrt{\frac{7}{1000}}}+\sqrt{2}\sqrt{\frac{7}{1000}}\right) =$$

$$e^{\frac{500y^2}{7}+y+\frac{7}{2000}} \operatorname{erfc}\left(\frac{1000y+7}{20\sqrt{35}}\right) - 2e^{\frac{(500y+7)^2}{3500}} \operatorname{erfc}\left(\frac{500y+7}{10\sqrt{35}}\right) \; .$$

The derivative of this function with respect to $y$ is

$$\left(\frac{1000y}{7}+1\right) e^{\frac{500y^2}{7}+y+\frac{7}{2000}} \operatorname{erfc}\left(\frac{1000y+7}{20\sqrt{35}}\right) - \tag{276}$$

$$\frac{1}{7}4e^{\frac{(500y+7)^2}{3500}}(500y+7)\operatorname{erfc}\left(\frac{500y+7}{10\sqrt{35}}\right) + 20\sqrt{\frac{5}{7\pi}} \; >$$

$$\left(1+\frac{1000\cdot(-0.01)}{7}\right) e^{-0.01+\frac{7}{2000}+\frac{500\cdot(-0.01)^2}{7}} \operatorname{erfc}\left(\frac{7+1000+(-0.01)}{20\sqrt{35}}\right) -$$

$$\frac{1}{7}4e^{\frac{(7+500\cdot0.01)^2}{3500}}(7+500\cdot0.01)\operatorname{erfc}\left(\frac{7+500\cdot0.01}{10\sqrt{35}}\right) + 20\sqrt{\frac{5}{7\pi}} \; > \; 3.56 \; .$$

For the first inequality, we use Lemma 24. Lemma 24 says that the function $xe^{x^2}\operatorname{erfc}(x)$ has the sign of $x$ and is monotonically increasing to $\frac{1}{\sqrt{\pi}}$. Consequently, we inserted the maximal $y = 0.01$ to make the negative term more negative and the minimal $y = -0.01$ to make the positive term less positive.

Consequently

$$e^{\frac{(x+y)^2}{2x}}\operatorname{erfc}\left(\frac{x+y}{\sqrt{2}\sqrt{x}}\right) - 2e^{\frac{(2x+y)^2}{2x}}\operatorname{erfc}\left(\frac{2x+y}{\sqrt{2}\sqrt{x}}\right) \tag{277}$$

is strictly monotonically increasing in $y$ for the minimal $x = 0.007$.

Next, we consider $x = 0.7 \cdot 0.8 = 0.56$, which is the maximal $\nu = 0.7$ and minimal $\tau = 0.8$. We insert the minimum $x = 0.56 = 56/100$ into the sub-function Eq. (100):

$$e^{\left(\frac{y}{\sqrt{2}\sqrt{\frac{56}{100}}}+\frac{\sqrt{\frac{56}{100}}}{\sqrt{2}}\right)^2}\operatorname{erfc}\left(\frac{y}{\sqrt{2}\sqrt{\frac{56}{100}}}+\frac{\sqrt{\frac{56}{100}}}{\sqrt{2}}\right) - \tag{278}$$

$$2e^{\left(\frac{y}{\sqrt{2}\sqrt{\frac{56}{100}}}+\sqrt{2}\sqrt{\frac{56}{100}}\right)^2}\operatorname{erfc}\left(\frac{y}{\sqrt{2}\sqrt{\frac{56}{100}}}+\sqrt{2}\sqrt{\frac{56}{100}}\right).$$

The derivative with respect to $y$ is:

$$\frac{5e^{\left(\frac{5y}{2\sqrt{7}}+\frac{\sqrt{7}}{5}\right)^2}\left(\frac{5y}{2\sqrt{7}}+\frac{\sqrt{7}}{5}\right)\operatorname{erfc}\left(\frac{5y}{2\sqrt{7}}+\frac{\sqrt{7}}{5}\right)}{\sqrt{7}} - \tag{279}$$

$$\frac{10e^{\left(\frac{5y}{2\sqrt{7}}+\frac{2\sqrt{7}}{5}\right)^2}\left(\frac{5y}{2\sqrt{7}}+\frac{2\sqrt{7}}{5}\right)\operatorname{erfc}\left(\frac{5y}{2\sqrt{7}}+\frac{2\sqrt{7}}{5}\right)}{\sqrt{7}} + \frac{5}{\sqrt{7\pi}} >$$

$$\frac{5e^{\left(\frac{\sqrt{7}}{5}-\frac{0.01\cdot5}{2\sqrt{7}}\right)^2}\left(\frac{\sqrt{7}}{5}-\frac{0.01\cdot5}{2\sqrt{7}}\right)\operatorname{erfc}\left(\frac{\sqrt{7}}{5}-\frac{0.01\cdot5}{2\sqrt{7}}\right)}{\sqrt{7}} -$$

$$\frac{10e^{\left(\frac{2\sqrt{7}}{5}+\frac{0.01\cdot5}{2\sqrt{7}}\right)^2}\left(\frac{2\sqrt{7}}{5}+\frac{0.01\cdot5}{2\sqrt{7}}\right)\operatorname{erfc}\left(\frac{2\sqrt{7}}{5}+\frac{0.01\cdot5}{2\sqrt{7}}\right)}{\sqrt{7}} + \frac{5}{\sqrt{7\pi}} > 0.00746\,.$$

For the first inequality we applied Lemma 24 which states that the function $xe^{x^2}\operatorname{erfc}(x)$ is monotonically increasing. Consequently, we inserted the maximal $y = 0.01$ to make the negative term more negative and the minimal $y = -0.01$ to make the positive term less positive.

Consequently

$$e^{\frac{(x+y)^2}{2x}}\operatorname{erfc}\left(\frac{x+y}{\sqrt{2}\sqrt{x}}\right) - 2e^{\frac{(2x+y)^2}{2x}}\operatorname{erfc}\left(\frac{2x+y}{\sqrt{2}\sqrt{x}}\right) \tag{280}$$

is strictly monotonically increasing in $y$ for $x = 0.56$.

Next, we consider $x = 0.16 \cdot 0.8 = 0.128$, which is the minimal $\tau = 0.8$. We insert the minimum $x = 0.128 = 128/1000$ into the sub-function Eq. (100):

$$e^{\left(\frac{y}{\sqrt{2}\sqrt{\frac{128}{1000}}} + \frac{\sqrt{\frac{128}{1000}}}{\sqrt{2}}\right)^2} \operatorname{erfc}\left(\frac{y}{\sqrt{2}\sqrt{\frac{128}{1000}}} + \frac{\sqrt{\frac{128}{1000}}}{\sqrt{2}}\right) - \tag{281}$$

$$2e^{\left(\frac{y}{\sqrt{2}\sqrt{\frac{128}{1000}}} + \sqrt{2}\sqrt{\frac{128}{1000}}\right)^2} \operatorname{erfc}\left(\frac{y}{\sqrt{2}\sqrt{\frac{128}{1000}}} + \sqrt{2}\sqrt{\frac{128}{1000}}\right) =$$

$$e^{\frac{125y^2}{32} + y + \frac{8}{125}} \operatorname{erfc}\left(\frac{125y + 16}{20\sqrt{10}}\right) - 2e^{\frac{(125y+32)^2}{4000}} \operatorname{erfc}\left(\frac{125y + 32}{20\sqrt{10}}\right) .$$

The derivative with respect to $y$ is:

$$\frac{1}{16}\left(e^{\frac{125y^2}{32} + y + \frac{8}{125}}(125y + 16)\operatorname{erfc}\left(\frac{125y + 16}{20\sqrt{10}}\right) - \tag{282}$$

$$2e^{\frac{(125y+32)^2}{4000}}(125y + 32)\operatorname{erfc}\left(\frac{125y + 32}{20\sqrt{10}}\right) + 20\sqrt{\frac{10}{\pi}}\right) >$$

$$\frac{1}{16}\left((16 + 125(-0.01))e^{-0.01 + \frac{8}{125} + \frac{125(-0.01)^2}{32}}\operatorname{erfc}\left(\frac{16 + 125(-0.01)}{20\sqrt{10}}\right) - $$

$$2e^{\frac{(32+1250.01)^2}{4000}}(32 + 1250.01)\operatorname{erfc}\left(\frac{32 + 1250.01}{20\sqrt{10}}\right) + 20\sqrt{\frac{10}{\pi}}\right) > 0.4468 .$$

For the first inequality we applied Lemma 24 which states that the function $xe^{x^2}\operatorname{erfc}(x)$ is monotonically increasing. Consequently, we inserted the maximal $y = 0.01$ to make the negative term more negative and the minimal $y = -0.01$ to make the positive term less positive.

Consequently

$$e^{\frac{(x+y)^2}{2x}}\operatorname{erfc}\left(\frac{x + y}{\sqrt{2}\sqrt{x}}\right) - 2e^{\frac{(2x+y)^2}{2x}}\operatorname{erfc}\left(\frac{2x + y}{\sqrt{2}\sqrt{x}}\right) \tag{283}$$

is strictly monotonically increasing in $y$ for $x = 0.128$.

Next, we consider $x = 0.24 \cdot 0.9 = 0.216$, which is the minimal $\tau = 0.9$ (here we consider 0.9 as lower bound for $\tau$). We insert the minimum $x = 0.216 = 216/1000$ into the sub-function Eq. (100):

$$e^{\left(\frac{y}{\sqrt{2}\sqrt{\frac{216}{1000}}} + \frac{\sqrt{\frac{216}{1000}}}{\sqrt{2}}\right)^2} \operatorname{erfc}\left(\frac{y}{\sqrt{2}\sqrt{\frac{216}{1000}}} + \frac{\sqrt{\frac{216}{1000}}}{\sqrt{2}}\right) - \tag{284}$$

$$2e^{\left(\frac{y}{\sqrt{2}\sqrt{\frac{216}{1000}}} + \sqrt{2}\sqrt{\frac{216}{1000}}\right)^2} \operatorname{erfc}\left(\frac{y}{\sqrt{2}\sqrt{\frac{216}{1000}}} + \sqrt{2}\sqrt{\frac{216}{1000}}\right) =$$

$$e^{\frac{(125y+27)^2}{6750}} \operatorname{erfc}\left(\frac{125y + 27}{15\sqrt{30}}\right) - 2e^{\frac{(125y+54)^2}{6750}} \operatorname{erfc}\left(\frac{125y + 54}{15\sqrt{30}}\right)$$

The derivative with respect to $y$ is:

$$\frac{1}{27}\left(e^{\frac{(125y+27)^2}{6750}}(125y+27)\operatorname{erfc}\left(\frac{125y+27}{15\sqrt{30}}\right)- \tag{285}$$

$$2e^{\frac{(125y+54)^2}{6750}}(125y+54)\operatorname{erfc}\left(\frac{125y+54}{15\sqrt{30}}\right)+15\sqrt{\frac{30}{\pi}}\right)>$$

$$\frac{1}{27}\left((27+125(-0.01))e^{\frac{(27+125(-0.01))^2}{6750}}\operatorname{erfc}\left(\frac{27+125(-0.01)}{15\sqrt{30}}\right)-$$

$$2e^{\frac{(54+1250.01)^2}{6750}}(54+1250.01)\operatorname{erfc}\left(\frac{54+1250.01}{15\sqrt{30}}\right)+15\sqrt{\frac{30}{\pi}}\right))>\ 0.211288\ .$$

For the first inequality we applied Lemma 24 which states that the function $xe^{x^2}\operatorname{erfc}(x)$ is monotonically increasing. Consequently, we inserted the maximal $y=0.01$ to make the negative term more negative and the minimal $y=-0.01$ to make the positive term less positive.

Consequently

$$e^{\frac{(x+y)^2}{2x}}\operatorname{erfc}\left(\frac{x+y}{\sqrt{2}\sqrt{x}}\right)-2e^{\frac{(2x+y)^2}{2x}}\operatorname{erfc}\left(\frac{2x+y}{\sqrt{2}\sqrt{x}}\right) \tag{286}$$

is strictly monotonically increasing in $y$ for $x=0.216$.     $\square$

**Lemma 46** (Monotone Derivative). *For $\lambda=\lambda_{01}$, $\alpha=\alpha_{01}$ and the domain $-0.1\leqslant\mu\leqslant0.1$, $-0.1\leqslant\omega\leqslant0.1$, $0.00875\leqslant\nu\leqslant0.7$, and $0.8\leqslant\tau\leqslant1.25$. We are interested of the derivative of*

$$\tau\left(e^{\left(\frac{\mu\omega+\nu\tau}{\sqrt{2}\sqrt{\nu\tau}}\right)^2}\operatorname{erfc}\left(\frac{\mu\omega+\nu\tau}{\sqrt{2}\sqrt{\nu\tau}}\right)-2e^{\left(\frac{\mu\omega+2\cdot\nu\tau}{\sqrt{2}\sqrt{\nu\tau}}\right)^2}\operatorname{erfc}\left(\frac{\mu\omega+2\cdot\nu\tau}{\sqrt{2}\sqrt{\nu\tau}}\right)\right)\ . \tag{287}$$

*The derivative of the equation above with respect to*

- *$\nu$ is larger than zero;*

- *$\tau$ is smaller than zero for maximal $\nu=0.7$, $\nu=0.16$, and $\nu=0.24$ (with $0.9\leqslant\tau$);*

- *$y=\mu\omega$ is larger than zero for $\nu\tau=0.00875\cdot0.8=0.007$, $\nu\tau=0.7\cdot0.8=0.56$, $\nu\tau=0.16\cdot0.8=0.128$, and $\nu\tau=0.24\cdot0.9=0.216$.*

*Proof.* We consider the domain: $-0.1\leqslant\mu\leqslant0.1$, $-0.1\leqslant\omega\leqslant0.1$, $0.00875\leqslant\nu\leqslant0.7$, and $0.8\leqslant\tau\leqslant1.25$.

We use Lemma 17 to determine the derivatives. **Consequently,** the derivative of

$$\tau\left(e^{\left(\frac{\mu\omega+\nu\tau}{\sqrt{2}\sqrt{\nu\tau}}\right)^2}\operatorname{erfc}\left(\frac{\mu\omega+\nu\tau}{\sqrt{2}\sqrt{\nu\tau}}\right)-2e^{\left(\frac{\mu\omega+2\cdot\nu\tau}{\sqrt{2}\sqrt{\nu\tau}}\right)^2}\operatorname{erfc}\left(\frac{\mu\omega+2\cdot\nu\tau}{\sqrt{2}\sqrt{\nu\tau}}\right)\right) \tag{288}$$

with respect to $\nu$ is larger than zero, which follows directly from Lemma 17 using the chain rule.

**Consequently,** the derivative of

$$\tau\left(e^{\left(\frac{\mu\omega+\nu\tau}{\sqrt{2}\sqrt{\nu\tau}}\right)^2}\operatorname{erfc}\left(\frac{\mu\omega+\nu\tau}{\sqrt{2}\sqrt{\nu\tau}}\right)-2e^{\left(\frac{\mu\omega+2\cdot\nu\tau}{\sqrt{2}\sqrt{\nu\tau}}\right)^2}\operatorname{erfc}\left(\frac{\mu\omega+2\cdot\nu\tau}{\sqrt{2}\sqrt{\nu\tau}}\right)\right) \tag{289}$$

with respect to $y = \mu\omega$ is larger than zero for $\nu\tau = 0.00875 \cdot 0.8 = 0.007$, $\nu\tau = 0.7 \cdot 0.8 = 0.56$, $\nu\tau = 0.16 \cdot 0.8 = 0.128$, and $\nu\tau = 0.24 \cdot 0.9 = 0.216$, which also follows directly from Lemma 17.

We now consider the derivative with respect to $\tau$, which is not trivial since $\tau$ is a factor of the whole expression. The sub-expression should be maximized as it appears with negative sign in the mapping for $\nu$.

First, we consider the function for the largest $\nu = 0.7$ and the largest $y = \mu\omega = 0.01$ for determining the derivative with respect to $\tau$.

The expression becomes

$$\tau \left( e^{\left( \frac{\frac{7\cdot\tau}{10} + \frac{1}{100}}{\sqrt{2}\sqrt{\frac{7\cdot\tau}{10}}} \right)^2} \operatorname{erfc}\left( \frac{\frac{7\cdot\tau}{10} + \frac{1}{100}}{\sqrt{2}\sqrt{\frac{7\cdot\tau}{10}}} \right) - \right. \tag{290}$$
$$\left. 2e^{\left( \frac{\frac{2\cdot 7\cdot\tau}{10} + \frac{1}{100}}{\sqrt{2}\sqrt{\frac{7\cdot\tau}{10}}} \right)^2} \operatorname{erfc}\left( \frac{\frac{2\cdot 7\cdot\tau}{10} + \frac{1}{100}}{\sqrt{2}\sqrt{\frac{7\cdot\tau}{10}}} \right) \right) .$$

The derivative with respect to $\tau$ is

$$\left( \sqrt{\pi} \left( e^{\frac{(70\tau+1)^2}{14000\tau}} (700\tau(7\cdot\tau+20)-1) \operatorname{erfc}\left( \frac{70\tau+1}{20\sqrt{35}\sqrt{\tau}} \right) - \tag{291} \right. \right.$$
$$\left. 2e^{\frac{(140\tau+1)^2}{14000\tau}} (2800\tau(7\cdot\tau+5)-1) \operatorname{erfc}\left( \frac{140\tau+1}{20\sqrt{35}\sqrt{\tau}} \right) \right) + 20\sqrt{35}(210\tau-1)\sqrt{\tau} \right)$$
$$\left( 14000\sqrt{\pi}\tau \right)^{-1} .$$

We are considering only the numerator and use again the approximation of Ren and MacKenzie (2007). The error analysis on the whole numerator gives an approximation error $97 < E < 186$. Therefore we add 200 to the numerator when we use the approximation Ren and MacKenzie (2007). We obtain the inequalities:

$$\sqrt{\pi} \left( e^{\frac{(70\tau+1)^2}{14000\tau}} (700\tau(7\cdot\tau+20)-1) \operatorname{erfc}\left( \frac{70\tau+1}{20\sqrt{35}\sqrt{\tau}} \right) - \tag{292} \right.$$
$$\left. 2e^{\frac{(140\tau+1)^2}{14000\tau}} (2800\tau(7\cdot\tau+5)-1) \operatorname{erfc}\left( \frac{140\tau+1}{20\sqrt{35}\sqrt{\tau}} \right) \right) + 20\sqrt{35}(210\tau-1)\sqrt{\tau} \leqslant$$

$$\sqrt{\pi} \left( \frac{2.911(700\tau(7\cdot\tau+20)-1)}{\frac{\sqrt{\pi}(2.911-1)(70\tau+1)}{20\sqrt{35}\sqrt{\tau}} + \sqrt{\pi\left(\frac{70\tau+1}{20\sqrt{35}\sqrt{\tau}}\right)^2 + 2.911^2}} - \right.$$

$$\left. \frac{2\cdot 2.911(2800\tau(7\cdot\tau+5)-1)}{\frac{\sqrt{\pi}(2.911-1)(140\tau+1)}{20\sqrt{35}\sqrt{\tau}} + \sqrt{\pi\left(\frac{140\tau+1}{20\sqrt{35}\sqrt{\tau}}\right)^2 + 2.911^2}} \right)$$
$$+ 20\sqrt{35}(210\tau-1)\sqrt{\tau} + 200 =$$

$$\sqrt{\pi}\left(\frac{(700\tau(7\cdot\tau+20)-1)\left(20\cdot\sqrt{35}\cdot2.911\sqrt{\tau}\right)}{\sqrt{\pi}(2.911-1)(70\tau+1)+\sqrt{\left(20\cdot2.911\sqrt{35}\sqrt{\tau}\right)^2+\pi(70\tau+1)^2}}-\right.$$

$$\left.\frac{2(2800\tau(7\cdot\tau+5)-1)\left(20\cdot\sqrt{35}\cdot2.911\sqrt{\tau}\right)}{\sqrt{\pi}(2.911-1)(140\tau+1)+\sqrt{\left(20\cdot\sqrt{35}\cdot2.911\sqrt{\tau}\right)^2+\pi(140\tau+1)^2}}\right)+$$

$$\left(20\sqrt{35}(210\tau-1)\sqrt{\tau}+200\right)=$$

$$\left(\left(20\sqrt{35}(210\tau-1)\sqrt{\tau}+200\right)\left(\sqrt{\pi}(2.911-1)(70\tau+1)+\sqrt{\left(20\cdot\sqrt{35}\cdot2.911\sqrt{\tau}\right)^2+\pi(70\tau+1)^2}\right)\right.$$

$$\left(\sqrt{\pi}(2.911-1)(140\tau+1)+\sqrt{\left(20\cdot\sqrt{35}\cdot2.911\sqrt{\tau}\right)^2+\pi(140\tau+1)^2}\right)+$$

$$2.911\cdot20\sqrt{35}\sqrt{\pi}(700\tau(7\cdot\tau+20)-1)\sqrt{\tau}$$

$$\left(\sqrt{\pi}(2.911-1)(140\tau+1)+\sqrt{\left(20\cdot\sqrt{35}\cdot2.911\sqrt{\tau}\right)^2+\pi(140\tau+1)^2}\right)-$$

$$\sqrt{\pi}2\cdot20\cdot\sqrt{35}\cdot2.911(2800\tau(7\cdot\tau+5)-1)$$

$$\sqrt{\tau}\left(\sqrt{\pi}(2.911-1)(70\tau+1)+\sqrt{\left(20\cdot\sqrt{35}\cdot2.911\sqrt{\tau}\right)^2+\pi(70\tau+1)^2}\right)\right)$$

$$\left(\left(\sqrt{\pi}(2.911-1)(70\tau+1)+\sqrt{\left(20\sqrt{35}\cdot2.911\cdot\sqrt{\tau}\right)^2+\pi(70\tau+1)^2}\right)\right.$$

$$\left.\left(\sqrt{\pi}(2.911-1)(140\tau+1)+\sqrt{\left(20\sqrt{35}\cdot2.911\cdot\sqrt{\tau}\right)^2+\pi(140\tau+1)^2}\right)\right)^{-1}.$$

After applying the approximation of Ren and MacKenzie (2007) and adding 200, we first factored out $20\sqrt{35}\sqrt{\tau}$. Then we brought all terms to the same denominator.

We now consider the numerator:

$$\left(20\sqrt{35}(210\tau-1)\sqrt{\tau}+200\right)\left(\sqrt{\pi}(2.911-1)(70\tau+1)+\sqrt{\left(20\cdot\sqrt{35}\cdot2.911\sqrt{\tau}\right)^2+\pi(70\tau+1)^2}\right)$$

$$\tag{293}$$

$$\left(\sqrt{\pi}(2.911-1)(140\tau+1)+\sqrt{\left(20\cdot\sqrt{35}\cdot2.911\sqrt{\tau}\right)^2+\pi(140\tau+1)^2}\right)+$$

$$2.911\cdot20\sqrt{35}\sqrt{\pi}(700\tau(7\cdot\tau+20)-1)\sqrt{\tau}$$

$$\left(\sqrt{\pi}(2.911-1)(140\tau+1)+\sqrt{\left(20\cdot\sqrt{35}\cdot2.911\sqrt{\tau}\right)^2+\pi(140\tau+1)^2}\right)-$$

$$\sqrt{\pi}2\cdot20\cdot\sqrt{35}\cdot2.911(2800\tau(7\cdot\tau+5)-1)\sqrt{\tau}$$

$$\left(\sqrt{\pi}(2.911-1)(70\tau+1)+\sqrt{\left(20\cdot\sqrt{35}\cdot2.911\sqrt{\tau}\right)^2+\pi(70\tau+1)^2}\right)=$$

$$-1.70658 \times 10^7 \sqrt{\pi(70\tau+1)^2 + 118635\tau}\,\tau^{3/2}+$$

$$4200\sqrt{35}\sqrt{\pi(70\tau+1)^2+118635\tau}\sqrt{\pi(140\tau+1)^2+118635\tau}\,\tau^{3/2}+$$

$$8.60302 \times 10^6 \sqrt{\pi(140\tau+1)^2+118635\tau}\,\tau^{3/2} - 2.89498 \times 10^7 \tau^{3/2}-$$

$$1.21486 \times 10^7 \sqrt{\pi(70\tau+1)^2+118635\tau}\,\tau^{5/2} + 8.8828 \times 10^6 \sqrt{\pi(140\tau+1)^2+118635\tau}\,\tau^{5/2}-$$

$$2.43651 \times 10^7 \tau^{5/2} - 1.46191 \times 10^9 \tau^{7/2} + 2.24868 \times 10^7 \tau^2 + 94840.5\sqrt{\pi(70\tau+1)^2+118635\tau}\,\tau +$$

$$47420.2\sqrt{\pi(140\tau+1)^2+118635\tau}\,\tau + 481860\tau + 710.354\sqrt{\tau}+$$

$$820.213\sqrt{\tau}\sqrt{\pi(70\tau+1)^2+118635\tau} + 677.432\sqrt{\pi(70\tau+1)^2+118635\tau}-$$

$$1011.27\sqrt{\tau}\sqrt{\pi(140\tau+1)^2+118635\tau}-$$

$$20\sqrt{35}\sqrt{\tau}\sqrt{\pi(70\tau+1)^2+118635\tau}\sqrt{\pi(140\tau+1)^2+118635\tau}+$$

$$200\sqrt{\pi(70\tau+1)^2+118635\tau}\sqrt{\pi(140\tau+1)^2+118635\tau}+$$

$$677.432\sqrt{\pi(140\tau+1)^2+118635\tau} + 2294.57 \ =$$

$$-2.89498 \times 10^7 \tau^{3/2} - 2.43651 \times 10^7 \tau^{5/2} - 1.46191 \times 10^9 \tau^{7/2}+$$

$$\left(-1.70658 \times 10^7 \tau^{3/2} - 1.21486 \times 10^7 \tau^{5/2} + 94840.5\tau + 820.213\sqrt{\tau} + 677.432\right)$$

$$\sqrt{\pi(70\tau+1)^2+118635\tau}+$$

$$\left(8.60302 \times 10^6 \tau^{3/2} + 8.8828 \times 10^6 \tau^{5/2} + 47420.2\tau - 1011.27\sqrt{\tau} + 677.432\right)$$

$$\sqrt{\pi(140\tau+1)^2+118635\tau}+$$

$$\left(4200\sqrt{35}\tau^{3/2} - 20\sqrt{35}\sqrt{\tau} + 200\right)\sqrt{\pi(70\tau+1)^2+118635\tau}\sqrt{\pi(140\tau+1)^2+118635\tau}+$$

$$2.24868 \times 10^7 \tau^2 + 481860.\tau + 710.354\sqrt{\tau} + 2294.57 \ \leqslant$$

$$-2.89498 \times 10^7 \tau^{3/2} - 2.43651 \times 10^7 \tau^{5/2} - 1.46191 \times 10^9 \tau^{7/2}+$$

$$\left(-1.70658 \times 10^7 \tau^{3/2} - 1.21486 \times 10^7 \tau^{5/2} + 820.213\sqrt{1.25} + 1.25 \cdot 94840.5 + 677.432\right)$$

$$\sqrt{\pi(70\tau+1)^2+118635\tau}+$$

$$\left(8.60302 \times 10^6 \tau^{3/2} + 8.8828 \times 10^6 \tau^{5/2} - 1011.27\sqrt{0.8} + 1.25 \cdot 47420.2 + 677.432\right)$$

$$\sqrt{\pi(140\tau+1)^2+118635\tau}+$$

$$\left(4200\sqrt{35}\tau^{3/2} - 20\sqrt{35}\sqrt{\tau} + 200\right)$$

$$\sqrt{\pi(70\tau+1)^2+118635\tau}\sqrt{\pi(140\tau+1)^2+118635\tau}+$$

$$2.24868 \times 10^7 \tau^2 + 710.354\sqrt{1.25} + 1.25 \cdot 481860. + 2294.57 \ =$$

$$-2.89498 \times 10^7 \tau^{3/2} - 2.43651 \times 10^7 \tau^{5/2} - 1.46191 \times 10^9 \tau^{7/2}+$$

$$\left(-1.70658 \times 10^7 \tau^{3/2} - 1.21486 \times 10^7 \tau^{5/2} + 120145.\right)\sqrt{\pi(70\tau+1)^2+118635\tau}+$$

$$\left(8.60302 \times 10^6 \tau^{3/2} + 8.8828 \times 10^6 \tau^{5/2} + 59048.2\right)\sqrt{\pi(140\tau+1)^2+118635\tau}+$$

$$\left(4200\sqrt{35}\tau^{3/2} - 20\sqrt{35}\sqrt{\tau} + 200\right)\sqrt{\pi(70\tau+1)^2+118635\tau}\sqrt{\pi(140\tau+1)^2+118635\tau}+$$

$$2.24868 \times 10^7 \tau^2 + 605413 \ =$$

$$-2.89498 \times 10^7 \tau^{3/2} - 2.43651 \times 10^7 \tau^{5/2} - 1.46191 \times 10^9 \tau^{7/2} +$$

$$\left(8.60302 \times 10^6 \tau^{3/2} + 8.8828 \times 10^6 \tau^{5/2} + 59048.2\right) \sqrt{19600\pi(\tau + 1.94093)(\tau + 0.0000262866)} +$$

$$\left(-1.70658 \times 10^7 \tau^{3/2} - 1.21486 \times 10^7 \tau^{5/2} + 120145.\right) \sqrt{4900\pi(\tau + 7.73521)(\tau + 0.0000263835)} +$$

$$\left(4200\sqrt{35}\tau^{3/2} - 20\sqrt{35}\sqrt{\tau} + 200\right)$$

$$\sqrt{19600\pi(\tau + 1.94093)(\tau + 0.0000262866)}\sqrt{4900\pi(\tau + 7.73521)(\tau + 0.0000263835)} +$$

$$2.24868 \times 10^7 \tau^2 + 605413 \leqslant$$

$$-2.89498 \times 10^7 \tau^{3/2} - 2.43651 \times 10^7 \tau^{5/2} - 1.46191 \times 10^9 \tau^{7/2} +$$

$$\left(8.60302 \times 10^6 \tau^{3/2} + 8.8828 \times 10^6 \tau^{5/2} + 59048.2\right) \sqrt{19600\pi(\tau + 1.94093)\tau} +$$

$$\left(-1.70658 \times 10^7 \tau^{3/2} - 1.21486 \times 10^7 \tau^{5/2} + 120145.\right) \sqrt{4900\pi 1.00003(\tau + 7.73521)\tau} +$$

$$\left(4200\sqrt{35}\tau^{3/2} - 20\sqrt{35}\sqrt{\tau} + 200\right) \sqrt{19600\pi 1.00003(\tau + 1.94093)\tau}$$

$$\sqrt{4900\pi 1.00003(\tau + 7.73521)\tau} +$$

$$2.24868 \times 10^7 \tau^2 + 605413 \ =$$

$$-2.89498 \times 10^7 \tau^{3/2} - 2.43651 \times 10^7 \tau^{5/2} - 1.46191 \times 10^9 \tau^{7/2} +$$

$$\left(-3.64296 \times 10^6 \tau^{3/2} + 7.65021 \times 10^8 \tau^{5/2} + 6.15772 \times 10^6 \tau\right)$$

$$\sqrt{\tau + 1.94093}\sqrt{\tau + 7.73521} + 2.24868 \times 10^7 \tau^2 +$$

$$\left(2.20425 \times 10^9 \tau^3 + 2.13482 \times 10^9 \tau^2 + 1.46527 \times 10^7 \sqrt{\tau}\right) \sqrt{\tau + 1.94093} +$$

$$\left(-1.5073 \times 10^9 \tau^3 - 2.11738 \times 10^9 \tau^2 + 1.49066 \times 10^7 \sqrt{\tau}\right) \sqrt{\tau + 7.73521} + 605413 \leqslant$$

$$\sqrt{1.25 + 1.94093}\sqrt{1.25 + 7.73521} \left(-3.64296 \times 10^6 \tau^{3/2} + 7.65021 \times 10^8 \tau^{5/2} + 6.15772 \times 10^6 \tau\right) +$$

$$\sqrt{1.25 + 1.94093} \left(2.20425 \times 10^9 \tau^3 + 2.13482 \times 10^9 \tau^2 + 1.46527 \times 10^7 \sqrt{\tau}\right) +$$

$$\sqrt{0.8 + 7.73521} \left(-1.5073 \times 10^9 \tau^3 - 2.11738 \times 10^9 \tau^2 + 1.49066 \times 10^7 \sqrt{\tau}\right) -$$

$$2.89498 \times 10^7 \tau^{3/2} - 2.43651 \times 10^7 \tau^{5/2} - 1.46191 \times 10^9 \tau^{7/2} + 2.24868 \times 10^7 \tau^2 + 605413 \ =$$

$$-4.84561 \times 10^7 \tau^{3/2} + 4.07198 \times 10^9 \tau^{5/2} - 1.46191 \times 10^9 \tau^{7/2} -$$

$$4.66103 \times 10^8 \tau^3 - 2.34999 \times 10^9 \tau^2 +$$

$$3.29718 \times 10^7 \tau + 6.97241 \times 10^7 \sqrt{\tau} + 605413 \leqslant$$

$$\frac{605413\tau^{3/2}}{0.8^{3/2}} - 4.84561 \times 10^7 \tau^{3/2} +$$

$$4.07198 \times 10^9 \tau^{5/2} - 1.46191 \times 10^9 \tau^{7/2} -$$

$$4.66103 \times 10^8 \tau^3 - 2.34999 \times 10^9 \tau^2 + \frac{3.29718 \times 10^7 \sqrt{\tau}\tau}{\sqrt{0.8}} + \frac{6.97241 \times 10^7 \tau\sqrt{\tau}}{0.8} \ =$$

$$\tau^{3/2} \left(-4.66103 \times 10^8 \tau^{3/2} - 1.46191 \times 10^9 \tau^2 - 2.34999 \times 10^9 \sqrt{\tau} +\right.$$

$$\left.4.07198 \times 10^9 \tau + 7.64087 \times 10^7\right) \leqslant$$

$$\tau^{3/2} \left(-4.66103 \times 10^8 \tau^{3/2} - 1.46191 \times 10^9 \tau^2 + \frac{7.64087 \times 10^7 \sqrt{\tau}}{\sqrt{0.8}} -\right.$$

$$2.34999 \times 10^9 \sqrt{\tau} + 4.07198 \times 10^9 \tau) \;=\;$$
$$\tau^2 \left( -1.46191 \times 10^9 \tau^{3/2} + 4.07198 \times 10^9 \sqrt{\tau} - 4.66103 \times 10^8 \tau - 2.26457 \times 10^9 \right) \;\leqslant\;$$
$$\left( -2.26457 \times 10^9 + 4.07198 \times 10^9 \sqrt{0.8} - 4.66103 \times 10^8 0.8 - 1.46191 \times 10^9 0.8^{3/2} \right) \tau^2 \;=\;$$
$$-\,4.14199 \times 10^7 \tau^2 \;<\; 0 \,.$$

First we expanded the term (multiplied it out). The we put the terms multiplied by the same square root into brackets. The next inequality sign stems from inserting the maximal value of $1.25$ for $\tau$ for some positive terms and value of $0.8$ for negative terms. These terms are then expanded at the $=$-sign. The next equality factors the terms under the squared root. We decreased the negative term by setting $\tau = \tau + 0.0000263835$ under the root. We increased positive terms by setting $\tau + 0.000026286 = 1.00003\tau$ and $\tau + 0.000026383 = 1.00003\tau$ under the root for positive terms. The positive terms are increase, since $\frac{0.8 + 0.000026383}{0.8} = 1.00003$, thus $\tau + 0.000026286 < \tau + 0.000026383 \leqslant 1.00003\tau$. For the next inequality we decreased negative terms by inserting $\tau = 0.8$ and increased positive terms by inserting $\tau = 1.25$. The next equality expands the terms. We use upper bound of $1.25$ and lower bound of $0.8$ to obtain terms with corresponding exponents of $\tau$.

For the last $\leqslant$-sign we used the function

$$-1.46191 \times 10^9 \tau^{3/2} + 4.07198 \times 10^9 \sqrt{\tau} - 4.66103 \times 10^8 \tau - 2.26457 \times 10^9 \tag{294}$$

The derivative of this function is

$$-2.19286 \times 10^9 \sqrt{\tau} + \frac{2.03599 \times 10^9}{\sqrt{\tau}} - 4.66103 \times 10^8 \tag{295}$$

and the second order derivative is

$$-\frac{1.01799 \times 10^9}{\tau^{3/2}} - \frac{1.09643 \times 10^9}{\sqrt{\tau}} \;<\; 0 \,. \tag{296}$$

The derivative at $0.8$ is smaller than zero:

$$-\,2.19286 \times 10^9 \sqrt{0.8} - 4.66103 \times 10^8 + \frac{2.03599 \times 10^9}{\sqrt{0.8}} \;= \tag{297}$$
$$-\,1.51154 \times 10^8 \;<\; 0 \,.$$

Since the second order derivative is negative, the derivative decreases with increasing $\tau$. Therefore the derivative is negative for all values of $\tau$ that we consider, that is, the function Eq. (294) is strictly monotonically decreasing. The maximum of the function Eq. (294) is therefore at $0.8$. We inserted $0.8$ to obtain the maximum.

**Consequently**, the derivative of

$$\tau \left( e^{\left( \frac{\mu\omega + \nu\tau}{\sqrt{2}\sqrt{\nu\tau}} \right)^2} \operatorname{erfc}\left( \frac{\mu\omega + \nu\tau}{\sqrt{2}\sqrt{\nu\tau}} \right) - 2 e^{\left( \frac{\mu\omega + 2\cdot\nu\tau}{\sqrt{2}\sqrt{\nu\tau}} \right)^2} \operatorname{erfc}\left( \frac{\mu\omega + 2\cdot\nu\tau}{\sqrt{2}\sqrt{\nu\tau}} \right) \right) \tag{298}$$

with respect to $\tau$ is smaller than zero for maximal $\nu = 0.7$.

Next, we consider the function for the largest $\nu = 0.16$ and the largest $y = \mu\omega = 0.01$ for determining the derivative with respect to $\tau$.

The expression becomes

$$\tau \left( e^{\left( \frac{\frac{16\tau}{100} + \frac{1}{100}}{\sqrt{2}\sqrt{\frac{16\tau}{100}}} \right)^2} \operatorname{erfc}\left( \frac{\frac{16\tau}{100} + \frac{1}{100}}{\sqrt{2}\sqrt{\frac{16\tau}{100}}} \right) - \right. \tag{299}$$
$$\left. 2e^{\left( \frac{2\frac{16\tau}{100} + \frac{1}{100}}{\sqrt{2}\sqrt{\frac{16\tau}{100}}} \right)^2} \operatorname{erfc}\left( \frac{2\frac{16\tau}{100} + \frac{1}{100}}{\sqrt{2}\sqrt{\frac{16\tau}{100}}} \right) \right) .$$

The derivative with respect to $\tau$ is

$$\left( \sqrt{\pi} \left( e^{\frac{(16\tau+1)^2}{3200\tau}} (128\tau(2 \cdot \tau + 25) - 1) \operatorname{erfc}\left( \frac{16\tau + 1}{40\sqrt{2}\sqrt{\tau}} \right) - \right. \right. \tag{300}$$
$$\left. \left. 2e^{\frac{(32\cdot\tau+1)^2}{3200\tau}} (128\tau(8\tau + 25) - 1) \operatorname{erfc}\left( \frac{32 \cdot \tau + 1}{40\sqrt{2}\sqrt{\tau}} \right) \right) + 40\sqrt{2}(48\tau - 1)\sqrt{\tau} \right)$$
$$\left( 3200\sqrt{\pi}\tau \right)^{-1} .$$

We are considering only the numerator and use again the approximation of Ren and MacKenzie (2007). The error analysis on the whole numerator gives an approximation error $1.1 < E < 12$. Therefore we add 20 to the numerator when we use the approximation of Ren and MacKenzie (2007). We obtain the inequalities:

$$\sqrt{\pi} \left( e^{\frac{(16\tau+1)^2}{3200\tau}} (128\tau(2 \cdot \tau + 25) - 1) \operatorname{erfc}\left( \frac{16\tau + 1}{40\sqrt{2}\sqrt{\tau}} \right) - \right. \tag{301}$$
$$\left. 2e^{\frac{(32\cdot\tau+1)^2}{3200\tau}} (128\tau(8\tau + 25) - 1) \operatorname{erfc}\left( \frac{32 \cdot \tau + 1}{40\sqrt{2}\sqrt{\tau}} \right) \right) + 40\sqrt{2}(48\tau - 1)\sqrt{\tau} \leqslant$$

$$\sqrt{\pi} \left( \frac{2.911(128\tau(2 \cdot \tau + 25) - 1)}{\frac{\sqrt{\pi}(2.911-1)(16\tau+1)}{40\sqrt{2}\sqrt{\tau}} + \sqrt{\pi \left( \frac{16\tau+1}{40\sqrt{2}\sqrt{\tau}} \right)^2 + 2.911^2}} - \right.$$

$$\left. \frac{2 \cdot 2.911(128\tau(8\tau + 25) - 1)}{\frac{\sqrt{\pi}(2.911-1)(32\cdot\tau+1)}{40\sqrt{2}\sqrt{\tau}} + \sqrt{\pi \left( \frac{32\cdot\tau+1}{40\sqrt{2}\sqrt{\tau}} \right)^2 + 2.911^2}} \right)$$
$$+ 40\sqrt{2}(48\tau - 1)\sqrt{\tau} + 20 =$$
$$\sqrt{\pi} \left( \frac{(128\tau(2 \cdot \tau + 25) - 1)\left( 40\sqrt{2}2.911\sqrt{\tau} \right)}{\sqrt{\pi}(2.911 - 1)(16\tau + 1) + \sqrt{\left( 40\sqrt{2}2.911\sqrt{\tau} \right)^2 + \pi(16\tau + 1)^2}} - \right.$$

$$\left. \frac{2(128\tau(8\tau + 25) - 1)\left( 40\sqrt{2}2.911\sqrt{\tau} \right)}{\sqrt{\pi}(2.911 - 1)(32 \cdot \tau + 1) + \sqrt{\left( 40\sqrt{2}2.911\sqrt{\tau} \right)^2 + \pi(32 \cdot \tau + 1)^2}} \right) +$$

$$40\sqrt{2}(48\tau - 1)\sqrt{\tau} + 20 \;=$$

$$\left(\left(40\sqrt{2}(48\tau - 1)\sqrt{\tau} + 20\right)\left(\sqrt{\pi}(2.911 - 1)(16\tau + 1) + \sqrt{\left(40\sqrt{2}2.911\sqrt{\tau}\right)^2 + \pi(16\tau + 1)^2}\right)\right.$$

$$\left(\sqrt{\pi}(2.911 - 1)(32 \cdot \tau + 1) + \sqrt{\left(40\sqrt{2}2.911\sqrt{\tau}\right)^2 + \pi(32 \cdot \tau + 1)^2}\right) + +$$

$$2.911 \cdot 40\sqrt{2}\sqrt{\pi}(128\tau(2 \cdot \tau + 25) - 1)\sqrt{\tau}$$

$$\left(\sqrt{\pi}(2.911 - 1)(32 \cdot \tau + 1) + \sqrt{\left(40\sqrt{2}2.911\sqrt{\tau}\right)^2 + \pi(32 \cdot \tau + 1)^2}\right) -$$

$$2\sqrt{\pi}40\sqrt{2}2.911(128\tau(8\tau + 25) - 1)$$

$$\sqrt{\tau}\left(\sqrt{\pi}(2.911 - 1)(16\tau + 1) + \sqrt{\left(40\sqrt{2}2.911\sqrt{\tau}\right)^2 + \pi(16\tau + 1)^2}\right)\right)$$

$$\left(\left(\sqrt{\pi}(2.911 - 1)(32 \cdot \tau + 1) + \sqrt{\left(40\sqrt{2}2.911\sqrt{\tau}\right)^2 + \pi(32 \cdot \tau + 1)^2}\right)\right.$$

$$\left.\left(\sqrt{\pi}(2.911 - 1)(32 \cdot \tau + 1) + \sqrt{\left(40\sqrt{2}2.911\sqrt{\tau}\right)^2 + \pi(32 \cdot \tau + 1)^2}\right)\right)^{-1}.$$

After applying the approximation of Ren and MacKenzie (2007) and adding 20, we first factored out $40\sqrt{2}\sqrt{\tau}$. Then we brought all terms to the same denominator.

We now consider the numerator:

$$\left(40\sqrt{2}(48\tau - 1)\sqrt{\tau} + 20\right)\left(\sqrt{\pi}(2.911 - 1)(16\tau + 1) + \sqrt{\left(40\sqrt{2}2.911\sqrt{\tau}\right)^2 + \pi(16\tau + 1)^2}\right)$$

$$\tag{302}$$

$$\left(\sqrt{\pi}(2.911 - 1)(32 \cdot \tau + 1) + \sqrt{\left(40\sqrt{2}2.911\sqrt{\tau}\right)^2 + \pi(32 \cdot \tau + 1)^2}\right) +$$

$$2.911 \cdot 40\sqrt{2}\sqrt{\pi}(128\tau(2 \cdot \tau + 25) - 1)\sqrt{\tau}$$

$$\left(\sqrt{\pi}(2.911 - 1)(32 \cdot \tau + 1) + \sqrt{\left(40\sqrt{2}2.911\sqrt{\tau}\right)^2 + \pi(32 \cdot \tau + 1)^2}\right) -$$

$$2\sqrt{\pi}40\sqrt{2}2.911(128\tau(8\tau + 25) - 1)\sqrt{\tau}$$

$$\left(\sqrt{\pi}(2.911 - 1)(16\tau + 1) + \sqrt{\left(40\sqrt{2}2.911\sqrt{\tau}\right)^2 + \pi(16\tau + 1)^2}\right) =$$

$$-1.86491 \times 10^6\sqrt{\pi(16\tau + 1)^2 + 27116.5\tau}\tau^{3/2} +$$

$$1920\sqrt{2}\sqrt{\pi(16\tau + 1)^2 + 27116.5\tau}\sqrt{\pi(32 \cdot \tau + 1)^2 + 27116.5\tau}\tau^{3/2} +$$

$$940121\sqrt{\pi(32 \cdot \tau + 1)^2 + 27116.5\tau}\tau^{3/2} - 3.16357 \times 10^6\tau^{3/2} -$$

$$303446\sqrt{\pi(16\tau + 1)^2 + 27116.5\tau}\tau^{5/2} + 221873\sqrt{\pi(32 \cdot \tau + 1)^2 + 27116.5\tau}\tau^{5/2} - 608588\tau^{5/2} -$$

$$8.34635 \times 10^6\tau^{7/2} + 117482.\tau^2 + 2167.78\sqrt{\pi(16\tau + 1)^2 + 27116.5\tau}\tau +$$

$$1083.89\sqrt{\pi(32\cdot\tau+1)^2+27116.5\tau}\tau+$$

$$11013.9\tau+339.614\sqrt{\tau}+392.137\sqrt{\tau}\sqrt{\pi(16\tau+1)^2+27116.5\tau}+$$

$$67.7432\sqrt{\pi(16\tau+1)^2+27116.5\tau}-483.478\sqrt{\tau}\sqrt{\pi(32\cdot\tau+1)^2+27116.5\tau}-$$

$$40\sqrt{2}\sqrt{\tau}\sqrt{\pi(16\tau+1)^2+27116.5\tau}\sqrt{\pi(32\cdot\tau+1)^2+27116.5\tau}+$$

$$20\sqrt{\pi(16\tau+1)^2+27116.5\tau}\sqrt{\pi(32\cdot\tau+1)^2+27116.5\tau}+$$

$$67.7432\sqrt{\pi(32\cdot\tau+1)^2+27116.5\tau}+229.457\ =$$

$$-3.16357\times10^6\tau^{3/2}-608588\tau^{5/2}-8.34635\times10^6\tau^{7/2}+$$

$$\left(-1.86491\times10^6\tau^{3/2}-303446\tau^{5/2}+2167.78\tau+392.137\sqrt{\tau}+67.7432\right)$$

$$\sqrt{\pi(16\tau+1)^2+27116.5\tau}+$$

$$\left(940121\tau^{3/2}+221873\tau^{5/2}+1083.89\tau-483.478\sqrt{\tau}+67.7432\right)$$

$$\sqrt{\pi(32\cdot\tau+1)^2+27116.5\tau}+$$

$$\left(1920\sqrt{2}\tau^{3/2}-40\sqrt{2}\sqrt{\tau}+20\right)\sqrt{\pi(16\tau+1)^2+27116.5\tau}\sqrt{\pi(32\cdot\tau+1)^2+27116.5\tau}+$$

$$117482.\tau^2+11013.9\tau+339.614\sqrt{\tau}+229.457\ \leqslant$$

$$-3.16357\times10^6\tau^{3/2}-608588\tau^{5/2}-8.34635\times10^6\tau^{7/2}+$$

$$\left(-1.86491\times10^6\tau^{3/2}-303446\tau^{5/2}+392.137\sqrt{1.25}+1.252167.78+67.7432\right)$$

$$\sqrt{\pi(16\tau+1)^2+27116.5\tau}+$$

$$\left(940121\tau^{3/2}+221873\tau^{5/2}-483.478\sqrt{0.8}+1.251083.89+67.7432\right)$$

$$\sqrt{\pi(32\cdot\tau+1)^2+27116.5\tau}+$$

$$\left(1920\sqrt{2}\tau^{3/2}-40\sqrt{2}\sqrt{\tau}+20\right)\sqrt{\pi(16\tau+1)^2+27116.5\tau}\sqrt{\pi(32\cdot\tau+1)^2+27116.5\tau}+$$

$$117482.\tau^2+339.614\sqrt{1.25}+1.2511013.9+229.457\ =$$

$$-3.16357\times10^6\tau^{3/2}-608588\tau^{5/2}-8.34635\times10^6\tau^{7/2}+$$

$$\left(-1.86491\times10^6\tau^{3/2}-303446\tau^{5/2}+3215.89\right)\sqrt{\pi(16\tau+1)^2+27116.5\tau}+$$

$$\left(940121\tau^{3/2}+221873\tau^{5/2}+990.171\right)\sqrt{\pi(32\cdot\tau+1)^2+27116.5\tau}+$$

$$\left(1920\sqrt{2}\tau^{3/2}-40\sqrt{2}\sqrt{\tau}+20\right)\sqrt{\pi(16\tau+1)^2+27116.5\tau}\sqrt{\pi(32\cdot\tau+1)^2+27116.5\tau}+$$

$$117482\tau^2+14376.6\ =$$

$$-3.16357\times10^6\tau^{3/2}-608588\tau^{5/2}-8.34635\times10^6\tau^{7/2}+$$

$$\left(940121\tau^{3/2}+221873\tau^{5/2}+990.171\right)\sqrt{1024\pi(\tau+8.49155)(\tau+0.000115004)}+$$

$$\left(-1.86491\times10^6\tau^{3/2}-303446\tau^{5/2}+3215.89\right)\sqrt{256\pi(\tau+33.8415)(\tau+0.000115428)}+$$

$$\left(1920\sqrt{2}\tau^{3/2}-40\sqrt{2}\sqrt{\tau}+20\right)\sqrt{1024\pi(\tau+8.49155)(\tau+0.000115004)}$$

$$\sqrt{256\pi(\tau+33.8415)(\tau+0.000115428)}+$$

$$117482.\tau^2+14376.6\ \leqslant$$

$$- 3.16357 \times 10^6 \tau^{3/2} - 608588 \tau^{5/2} - 8.34635 \times 10^6 \tau^{7/2} +$$

$$\left( 940121 \tau^{3/2} + 221873 \tau^{5/2} + 990.171 \right) \sqrt{1024 \pi 1.00014 (\tau + 8.49155) \tau} +$$

$$\left( 1920 \sqrt{2} \tau^{3/2} - 40 \sqrt{2} \sqrt{\tau} + 20 \right) \sqrt{256 \pi 1.00014 (\tau + 33.8415) \tau} \sqrt{1024 \pi 1.00014 (\tau + 8.49155) \tau} +$$

$$\left( -1.86491 \times 10^6 \tau^{3/2} - 303446 \tau^{5/2} + 3215.89 \right) \sqrt{256 \pi (\tau + 33.8415) \tau} +$$

$$117482. \tau^2 + 14376.6 =$$

$$- 3.16357 \times 10^6 \tau^{3/2} - 608588 \tau^{5/2} - 8.34635 \times 10^6 \tau^{7/2} +$$

$$\left( -91003 \tau^{3/2} + 4.36814 \times 10^6 \tau^{5/2} + 32174.4 \tau \right) \sqrt{\tau + 8.49155} \sqrt{\tau + 33.8415} + 117482. \tau^2 +$$

$$\left( 1.25852 \times 10^7 \tau^3 + 5.33261 \times 10^7 \tau^2 + 56165.1 \sqrt{\tau} \right) \sqrt{\tau + 8.49155} +$$

$$\left( -8.60549 \times 10^6 \tau^3 - 5.28876 \times 10^7 \tau^2 + 91200.4 \sqrt{\tau} \right) \sqrt{\tau + 33.8415} + 14376.6 \leqslant$$

$$\sqrt{1.25 + 8.49155} \sqrt{1.25 + 33.8415} \left( -91003 \tau^{3/2} + 4.36814 \times 10^6 \tau^{5/2} + 32174.4 \tau \right) +$$

$$\sqrt{1.25 + 8.49155} \left( 1.25852 \times 10^7 \tau^3 + 5.33261 \times 10^7 \tau^2 + 56165.1 \sqrt{\tau} \right) +$$

$$\sqrt{0.8 + 33.8415} \left( -8.60549 \times 10^6 \tau^3 - 5.28876 \times 10^7 \tau^2 + 91200.4 \sqrt{\tau} \right) -$$

$$3.16357 \times 10^6 \tau^{3/2} - 608588 \tau^{5/2} - 8.34635 \times 10^6 \tau^{7/2} + 117482. \tau^2 + 14376.6 =$$

$$- 4.84613 \times 10^6 \tau^{3/2} + 8.01543 \times 10^7 \tau^{5/2} - 8.34635 \times 10^6 \tau^{7/2} -$$

$$1.13691 \times 10^7 \tau^3 - 1.44725 \times 10^8 \tau^2 +$$

$$594875. \tau + 712078. \sqrt{\tau} + 14376.6 \leqslant$$

$$\frac{14376.6 \tau^{3/2}}{0.8^{3/2}} - 4.84613 \times 10^6 \tau^{3/2} +$$

$$8.01543 \times 10^7 \tau^{5/2} - 8.34635 \times 10^6 \tau^{7/2} -$$

$$1.13691 \times 10^7 \tau^3 - 1.44725 \times 10^8 \tau^2 + \frac{594875. \sqrt{\tau} \tau}{\sqrt{0.8}} + \frac{712078. \tau \sqrt{\tau}}{0.8} =$$

$$- 3.1311 \cdot 10^6 \tau^{3/2} - 1.44725 \cdot 10^8 \tau^2 + 8.01543 \cdot 10^7 \tau^{5/2} - 1.13691 \cdot 10^7 \tau^3 -$$

$$8.34635 \cdot 10^6 \tau^{7/2} \leqslant$$

$$- 3.1311 \times 10^6 \tau^{3/2} + \frac{8.01543 \times 10^7 \sqrt{1.25} \tau^{5/2}}{\sqrt{\tau}} -$$

$$8.34635 \times 10^6 \tau^{7/2} - 1.13691 \times 10^7 \tau^3 - 1.44725 \times 10^8 \tau^2 =$$

$$- 3.1311 \times 10^6 \tau^{3/2} - 8.34635 \times 10^6 \tau^{7/2} - 1.13691 \times 10^7 \tau^3 - 5.51094 \times 10^7 \tau^2 2 < 0 .$$

First we expanded the term (multiplied it out). The we put the terms multiplied by the same square root into brackets. The next inequality sign stems from inserting the maximal value of 1.25 for $\tau$ for some positive terms and value of 0.8 for negative terms. These terms are then expanded at the =-sign. The next equality factors the terms under the squared root. We decreased the negative term by setting $\tau = \tau + 0.00011542$ under the root. We increased positive terms by setting $\tau + 0.00011542 = 1.00014 \tau$ and $\tau + 0.000115004 = 1.00014 \tau$ under the root for positive terms. The positive terms are increase, since $\frac{0.8 + 0.00011542}{0.8} < 1.000142$, thus $\tau + 0.000115004 < \tau + 0.00011542 \leqslant 1.00014 \tau$. For the next inequality we decreased negative terms by inserting $\tau = 0.8$ and increased positive terms by inserting $\tau = 1.25$. The next equality expands the terms.

We use upper bound of $1.25$ and lower bound of $0.8$ to obtain terms with corresponding exponents of $\tau$.

**Consequently**, the derivative of

$$\tau\left(e^{\left(\frac{\mu\omega+\nu\tau}{\sqrt{2}\sqrt{\nu\tau}}\right)^2}\operatorname{erfc}\left(\frac{\mu\omega+\nu\tau}{\sqrt{2}\sqrt{\nu\tau}}\right) - 2e^{\left(\frac{\mu\omega+2\cdot\nu\tau}{\sqrt{2}\sqrt{\nu\tau}}\right)^2}\operatorname{erfc}\left(\frac{\mu\omega+2\cdot\nu\tau}{\sqrt{2}\sqrt{\nu\tau}}\right)\right) \tag{303}$$

with respect to $\tau$ is smaller than zero for maximal $\nu = 0.16$.

Next, we consider the function for the largest $\nu = 0.24$ and the largest $y = \mu\omega = 0.01$ for determining the derivative with respect to $\tau$. However we assume $0.9 \leqslant \tau$, in order to restrict the domain of $\tau$.

The expression becomes

$$\tau\left(e^{\left(\frac{\frac{24\tau}{100}+\frac{1}{100}}{\sqrt{2}\sqrt{\frac{24\tau}{100}}}\right)^2}\operatorname{erfc}\left(\frac{\frac{24\tau}{100}+\frac{1}{100}}{\sqrt{2}\sqrt{\frac{24\tau}{100}}}\right) - \right. \tag{304}$$
$$\left. 2e^{\left(\frac{2\frac{24\tau}{100}+\frac{1}{100}}{\sqrt{2}\sqrt{\frac{24\tau}{100}}}\right)^2}\operatorname{erfc}\left(\frac{2\frac{24\tau}{100}+\frac{1}{100}}{\sqrt{2}\sqrt{\frac{24\tau}{100}}}\right)\right)\ .$$

The derivative with respect to $\tau$ is

$$\left(\sqrt{\pi}\left(e^{\frac{(24\tau+1)^2}{4800\tau}}\left(192\cdot\tau(3\tau+25)-1\right)\operatorname{erfc}\left(\frac{24\tau+1}{40\sqrt{3}\sqrt{\tau}}\right)-\right.\right. \tag{305}$$
$$\left.\left. 2e^{\frac{(48\tau+1)^2}{4800\tau}}\left(192\cdot\tau(12\cdot\tau+25)-1\right)\operatorname{erfc}\left(\frac{48\tau+1}{40\sqrt{3}\sqrt{\tau}}\right)\right) + 40\sqrt{3}(72\cdot\tau-1)\sqrt{\tau}\right)$$
$$\left(4800\sqrt{\pi}\tau\right)^{-1}\ .$$

We are considering only the numerator and use again the approximation of Ren and MacKenzie (2007). The error analysis on the whole numerator gives an approximation error $14 < E < 32$. Therefore we add 32 to the numerator when we use the approximation of Ren and MacKenzie (2007). We obtain the inequalities:

$$\sqrt{\pi}\left(e^{\frac{(24\tau+1)^2}{4800\tau}}\left(192\cdot\tau(3\tau+25)-1\right)\operatorname{erfc}\left(\frac{24\tau+1}{40\sqrt{3}\sqrt{\tau}}\right)-\right. \tag{306}$$
$$\left. 2e^{\frac{(48\tau+1)^2}{4800\tau}}\left(192\cdot\tau(12\cdot\tau+25)-1\right)\operatorname{erfc}\left(\frac{48\tau+1}{40\sqrt{3}\sqrt{\tau}}\right)\right) + 40\sqrt{3}(72\cdot\tau-1)\sqrt{\tau} \leqslant$$

$$\sqrt{\pi}\left(\frac{2.911\left(192\cdot\tau(3\tau+25)-1\right)}{\frac{\sqrt{\pi}(2.911-1)(24\tau+1)}{40\sqrt{3}\sqrt{\tau}}+\sqrt{\pi\left(\frac{24\tau+1}{40\sqrt{3}\sqrt{\tau}}\right)^2+2.911^2}}-\right.$$
$$\left.\frac{2\cdot 2.911\left(192\cdot\tau(12\cdot\tau+25)-1\right)}{\frac{\sqrt{\pi}(2.911-1)(48\tau+1)}{40\sqrt{3}\sqrt{\tau}}+\sqrt{\pi\left(\frac{48\tau+1}{40\sqrt{3}\sqrt{\tau}}\right)^2+2.911^2}}\right) +$$

$$40\sqrt{3}(72 \cdot \tau - 1)\sqrt{\tau} + 32 =$$

$$\sqrt{\pi}\left(\frac{(192 \cdot \tau(3\tau + 25) - 1)\left(40\sqrt{3}2.911\sqrt{\tau}\right)}{\sqrt{\pi}(2.911 - 1)(24\tau + 1) + \sqrt{\left(40\sqrt{3}2.911\sqrt{\tau}\right)^2 + \pi(24\tau + 1)^2}} - \right.$$

$$\left.\frac{2(192 \cdot \tau(12 \cdot \tau + 25) - 1)\left(40\sqrt{3}2.911\sqrt{\tau}\right)}{\sqrt{\pi}(2.911 - 1)(48\tau + 1) + \sqrt{\left(40\sqrt{3}2.911\sqrt{\tau}\right)^2 + \pi(48\tau + 1)^2}}\right) +$$

$$40\sqrt{3}(72 \cdot \tau - 1)\sqrt{\tau} + 32 =$$

$$\left(\left(40\sqrt{3}(72 \cdot \tau - 1)\sqrt{\tau} + 32\right)\left(\sqrt{\pi}(2.911 - 1)(24\tau + 1) + \sqrt{\left(40\sqrt{3}2.911\sqrt{\tau}\right)^2 + \pi(24\tau + 1)^2}\right)\right.$$

$$\left(\sqrt{\pi}(2.911 - 1)(48\tau + 1) + \sqrt{\left(40\sqrt{3}2.911\sqrt{\tau}\right)^2 + \pi(48\tau + 1)^2}\right) +$$

$$2.911 \cdot 40\sqrt{3}\sqrt{\pi}(192 \cdot \tau(3\tau + 25) - 1)\sqrt{\tau}$$

$$\left(\sqrt{\pi}(2.911 - 1)(48\tau + 1) + \sqrt{\left(40\sqrt{3}2.911\sqrt{\tau}\right)^2 + \pi(48\tau + 1)^2}\right) -$$

$$2\sqrt{\pi}40\sqrt{3}2.911(192 \cdot \tau(12 \cdot \tau + 25) - 1)$$

$$\left.\sqrt{\tau}\left(\sqrt{\pi}(2.911 - 1)(24\tau + 1) + \sqrt{\left(40\sqrt{3}2.911\sqrt{\tau}\right)^2 + \pi(24\tau + 1)^2}\right)\right)$$

$$\left(\left(\sqrt{\pi}(2.911 - 1)(24\tau + 1) + \sqrt{\left(40\sqrt{3}2.911\sqrt{\tau}\right)^2 + \pi(24\tau + 1)^2}\right)\right.$$

$$\left.\left(\sqrt{\pi}(2.911 - 1)(48\tau + 1) + \sqrt{\left(40\sqrt{3}2.911\sqrt{\tau}\right)^2 + \pi(48\tau + 1)^2}\right)\right)^{-1}.$$

After applying the approximation of Ren and MacKenzie (2007) and adding 200, we first factored out $40\sqrt{3}\sqrt{\tau}$. Then we brought all terms to the same denominator.

We now consider the numerator:

$$\left(40\sqrt{3}(72 \cdot \tau - 1)\sqrt{\tau} + 32\right)\left(\sqrt{\pi}(2.911 - 1)(24\tau + 1) + \sqrt{\left(40\sqrt{3}2.911\sqrt{\tau}\right)^2 + \pi(24\tau + 1)^2}\right)$$

$$\tag{307}$$

$$\left(\sqrt{\pi}(2.911 - 1)(48\tau + 1) + \sqrt{\left(40\sqrt{3}2.911\sqrt{\tau}\right)^2 + \pi(48\tau + 1)^2}\right) +$$

$$2.911 \cdot 40\sqrt{3}\sqrt{\pi}(192 \cdot \tau(3\tau + 25) - 1)\sqrt{\tau}$$

$$\left(\sqrt{\pi}(2.911 - 1)(48\tau + 1) + \sqrt{\left(40\sqrt{3}2.911\sqrt{\tau}\right)^2 + \pi(48\tau + 1)^2}\right) -$$

$$2\sqrt{\pi}40\sqrt{3}2.911(192 \cdot \tau(12 \cdot \tau + 25) - 1)\sqrt{\tau}$$

$$\left(\sqrt{\pi}(2.911 - 1)(24\tau + 1) + \sqrt{\left(40\sqrt{3}2.911\sqrt{\tau}\right)^2 + \pi(24\tau + 1)^2}\right) =$$

$$-\,3.42607 \times 10^6 \sqrt{\pi(24\tau+1)^2 + 40674.8\tau}\,\tau^{3/2}+$$

$$2880\sqrt{3}\sqrt{\pi(24\tau+1)^2 + 40674.8\tau}\,\sqrt{\pi(48\tau+1)^2 + 40674.8\tau}\,\tau^{3/2}+$$

$$1.72711 \times 10^6 \sqrt{\pi(48\tau+1)^2 + 40674.8\tau}\,\tau^{3/2} - 5.81185 \times 10^6 \tau^{3/2}-$$

$$836198\sqrt{\pi(24\tau+1)^2 + 40674.8\tau}\,\tau^{5/2} + 611410\sqrt{\pi(48\tau+1)^2 + 40674.8\tau}\,\tau^{5/2}-$$

$$1.67707 \times 10^6 \tau^{5/2}-$$

$$3.44998 \times 10^7 \tau^{7/2} + 422935.\tau^2 + 5202.68\sqrt{\pi(24\tau+1)^2 + 40674.8\tau}\,\tau+$$

$$2601.34\sqrt{\pi(48\tau+1)^2 + 40674.8\tau}\,\tau +$$

$$26433.4\tau + 415.94\sqrt{\tau} + 480.268\sqrt{\tau}\sqrt{\pi(24\tau+1)^2 + 40674.8\tau} +$$

$$108.389\sqrt{\pi(24\tau+1)^2 + 40674.8\tau} - 592.138\sqrt{\tau}\sqrt{\pi(48\tau+1)^2 + 40674.8\tau}-$$

$$40\sqrt{3}\sqrt{\tau}\sqrt{\pi(24\tau+1)^2 + 40674.8\tau}\,\sqrt{\pi(48\tau+1)^2 + 40674.8\tau} +$$

$$32\sqrt{\pi(24\tau+1)^2 + 40674.8\tau}\,\sqrt{\pi(48\tau+1)^2 + 40674.8\tau} +$$

$$108.389\sqrt{\pi(48\tau+1)^2 + 40674.8\tau} + 367.131 \;=$$

$$-\,5.81185 \times 10^6 \tau^{3/2} - 1.67707 \times 10^6 \tau^{5/2} - 3.44998 \times 10^7 \tau^{7/2}+$$

$$\left(-3.42607 \times 10^6 \tau^{3/2} - 836198\tau^{5/2} + 5202.68\tau + 480.268\sqrt{\tau} + 108.389\right)$$

$$\sqrt{\pi(24\tau+1)^2 + 40674.8\tau}+$$

$$\left(1.72711 \times 10^6 \tau^{3/2} + 611410\tau^{5/2} + 2601.34\tau - 592.138\sqrt{\tau} + 108.389\right)$$

$$\sqrt{\pi(48\tau+1)^2 + 40674.8\tau}+$$

$$\left(2880\sqrt{3}\tau^{3/2} - 40\sqrt{3}\sqrt{\tau} + 32\right)\sqrt{\pi(24\tau+1)^2 + 40674.8\tau}\,\sqrt{\pi(48\tau+1)^2 + 40674.8\tau}+$$

$$422935.\tau^2 + 26433.4\tau + 415.94\sqrt{\tau} + 367.131 \;\leqslant$$

$$-\,5.81185 \times 10^6 \tau^{3/2} - 1.67707 \times 10^6 \tau^{5/2} - 3.44998 \times 10^7 \tau^{7/2}+$$

$$\left(-3.42607 \times 10^6 \tau^{3/2} - 836198\tau^{5/2} + 480.268\sqrt{1.25} + 1.255202.68 + 108.389\right)$$

$$\sqrt{\pi(24\tau+1)^2 + 40674.8\tau}+$$

$$\left(1.72711 \times 10^6 \tau^{3/2} + 611410\tau^{5/2} - 592.138\sqrt{0.9} + 1.252601.34 + 108.389\right)$$

$$\sqrt{\pi(48\tau+1)^2 + 40674.8\tau}+$$

$$\left(2880\sqrt{3}\tau^{3/2} - 40\sqrt{3}\sqrt{\tau} + 32\right)\sqrt{\pi(24\tau+1)^2 + 40674.8\tau}\,\sqrt{\pi(48\tau+1)^2 + 40674.8\tau}+$$

$$422935\tau^2 + 415.94\sqrt{1.25} + 1.2526433.4 + 367.131 \;=$$

$$-\,5.81185 \times 10^6 \tau^{3/2} - 1.67707 \times 10^6 \tau^{5/2} - 3.44998 \times 10^7 \tau^{7/2}+$$

$$\left(-3.42607 \times 10^6 \tau^{3/2} - 836198\tau^{5/2} + 7148.69\right)\sqrt{\pi(24\tau+1)^2 + 40674.8\tau}+$$

$$\left(1.72711 \times 10^6 \tau^{3/2} + 611410\tau^{5/2} + 2798.31\right)\sqrt{\pi(48\tau+1)^2 + 40674.8\tau}+$$

$$\left(2880\sqrt{3}\tau^{3/2} - 40\sqrt{3}\sqrt{\tau} + 32\right)\sqrt{\pi(24\tau+1)^2 + 40674.8\tau}\,\sqrt{\pi(48\tau+1)^2 + 40674.8\tau}+$$

$$422935\tau^2 + 33874 \;=$$

$$- 5.81185 \times 10^6 \tau^{3/2} - 1.67707 \times 10^6 \tau^{5/2} - 3.44998 \times 10^7 \tau^{7/2} +$$

$$\left(1.72711 \times 10^6 \tau^{3/2} + 611410 \tau^{5/2} + 2798.31\right) \sqrt{2304\pi(\tau + 5.66103)(\tau + 0.0000766694)} +$$

$$\left(-3.42607 \times 10^6 \tau^{3/2} - 836198 \tau^{5/2} + 7148.69\right) \sqrt{576\pi(\tau + 22.561)(\tau + 0.0000769518)} +$$

$$\left(2880\sqrt{3}\tau^{3/2} - 40\sqrt{3}\sqrt{\tau} + 32\right) \sqrt{2304\pi(\tau + 5.66103)(\tau + 0.0000766694)}$$

$$\sqrt{576\pi(\tau + 22.561)(\tau + 0.0000769518)} +$$

$$422935\tau^2 + 33874 \ \leqslant$$

$$- 5.81185 10^6 \tau^{3/2} - 1.67707 \times 10^6 \tau^{5/2} - 3.44998 \times 10^7 \tau^{7/2} +$$

$$\left(1.72711 \times 10^6 \tau^{3/2} + 611410 \tau^{5/2} + 2798.31\right) \sqrt{2304\pi 1.0001(\tau + 5.66103)\tau} +$$

$$\left(2880\sqrt{3}\tau^{3/2} - 40\sqrt{3}\sqrt{\tau} + 32\right) \sqrt{2304\pi 1.0001(\tau + 5.66103)\tau} \sqrt{576\pi 1.0001(\tau + 22.561)\tau} +$$

$$\left(-3.42607 \times 10^6 \tau^{3/2} - 836198 \tau^{5/2} + 7148.69\right)$$

$$\sqrt{576\pi(\tau + 22.561)\tau} +$$

$$422935\tau^2 + 33874. \ =$$

$$- 5.81185 10^6 \tau^{3/2} - 1.67707 \times 10^6 \tau^{5/2} - 3.44998 \times 10^7 \tau^{7/2} +$$

$$\left(-250764.\tau^{3/2} + 1.8055 \times 10^7 \tau^{5/2} + 115823.\tau\right)$$

$$\sqrt{\tau + 5.66103}\sqrt{\tau + 22.561} + 422935.\tau^2 +$$

$$\left(5.20199 \times 10^7 \tau^3 + 1.46946 \times 10^8 \tau^2 + 238086.\sqrt{\tau}\right) \sqrt{\tau + 5.66103} +$$

$$\left(-3.55709 \times 10^7 \tau^3 - 1.45741 \times 10^8 \tau^2 + 304097.\sqrt{\tau}\right) \sqrt{\tau + 22.561} + 33874. \ \leqslant$$

$$\sqrt{1.25 + 5.66103}\sqrt{1.25 + 22.561} \left(-250764.\tau^{3/2} + 1.8055 \times 10^7 \tau^{5/2} + 115823.\tau\right) +$$

$$\sqrt{1.25 + 5.66103} \left(5.20199 \times 10^7 \tau^3 + 1.46946 \times 10^8 \tau^2 + 238086.\sqrt{\tau}\right) +$$

$$\sqrt{0.9 + 22.561} \left(-3.55709 \times 10^7 \tau^3 - 1.45741 \times 10^8 \tau^2 + 304097.\sqrt{\tau}\right) -$$

$$5.81185 10^6 \tau^{3/2} - 1.67707 \times 10^6 \tau^{5/2} - 3.44998 \times 10^7 \tau^{7/2} + 422935.\tau^2 + 33874. \ \leqslant$$

$$\frac{33874.\tau^{3/2}}{0.9^{3/2}} - 9.02866 \times 10^6 \tau^{3/2} + 2.29933 \times 10^8 \tau^{5/2} - 3.44998 \times 10^7 \tau^{7/2} -$$

$$3.5539 \times 10^7 \tau^3 - 3.19193 \times 10^8 \tau^2 + \frac{1.48578 \times 10^6 \sqrt{\tau}\tau}{\sqrt{0.9}} + \frac{2.09884 \times 10^6 \tau\sqrt{\tau}}{0.9} \ =$$

$$- 5.09079 \times 10^6 \tau^{3/2} + 2.29933 \times 10^8 \tau^{5/2} -$$

$$3.44998 \times 10^7 \tau^{7/2} - 3.5539 \times 10^7 \tau^3 - 3.19193 \times 10^8 \tau^2 \ \leqslant$$

$$- 5.09079 \times 10^6 \tau^{3/2} + \frac{2.29933 \times 10^8 \sqrt{1.25}\tau^{5/2}}{\sqrt{\tau}} - 3.44998 \times 10^7 \tau^{7/2} -$$

$$3.5539 \times 10^7 \tau^3 - 3.19193 \times 10^8 \tau^2 \ =$$

$$- 5.09079 \times 10^6 \tau^{3/2} - 3.44998 \times 10^7 \tau^{7/2} - 3.5539 \times 10^7 \tau^3 - 6.21197 \times 10^7 \tau^2 \ < \ 0 \, .$$

First we expanded the term (multiplied it out). The we put the terms multiplied by the same square root into brackets. The next inequality sign stems from inserting the maximal value of

1.25 for $\tau$ for some positive terms and value of 0.9 for negative terms. These terms are then expanded at the =-sign. The next equality factors the terms under the squared root. We decreased the negative term by setting $\tau = \tau + 0.0000769518$ under the root. We increased positive terms by setting $\tau + 0.0000769518 = 1.0000962\tau$ and $\tau + 0.0000766694 = 1.0000962\tau$ under the root for positive terms. The positive terms are increase, since $\frac{0.8+0.0000769518}{0.8} < 1.0000962$, thus $\tau + 0.0000766694 < \tau + 0.0000769518 \leqslant 1.0000962\tau$. For the next inequality we decreased negative terms by inserting $\tau = 0.9$ and increased positive terms by inserting $\tau = 1.25$. The next equality expands the terms. We use upper bound of 1.25 and lower bound of 0.9 to obtain terms with corresponding exponents of $\tau$.

**Consequently**, the derivative of

$$\tau \left( e^{\left(\frac{\mu\omega+\nu\tau}{\sqrt{2}\sqrt{\nu\tau}}\right)^2} \operatorname{erfc}\left( \frac{\mu\omega+\nu\tau}{\sqrt{2}\sqrt{\nu\tau}} \right) - 2e^{\left(\frac{\mu\omega+2\cdot\nu\tau}{\sqrt{2}\sqrt{\nu\tau}}\right)^2} \operatorname{erfc}\left( \frac{\mu\omega+2\cdot\nu\tau}{\sqrt{2}\sqrt{\nu\tau}} \right) \right) \tag{308}$$

with respect to $\tau$ is smaller than zero for maximal $\nu = 0.24$ and the domain $0.9 \leqslant \tau \leqslant 1.25$.   □

**Lemma 47.** *In the domain* $-0.01 \leqslant y \leqslant 0.01$ *and* $0.64 \leqslant x \leqslant 1.875$, *the function* $f(x,y) = e^{\frac{1}{2}(2y+x)} \operatorname{erfc}\left( \frac{x+y}{\sqrt{2x}} \right)$ *has a global maximum at* $y = 0.64$ *and* $x = -0.01$ *and a global minimum at* $y = 1.875$ *and* $x = 0.01$.

*Proof.* $f(x,y) = e^{\frac{1}{2}(2y+x)} \operatorname{erfc}\left( \frac{x+y}{\sqrt{2x}} \right)$ is strictly monotonically decreasing in $x$, since its derivative with respect to $x$ is negative:

$$\frac{e^{-\frac{y^2}{2x}} \left( \sqrt{\pi}x^{3/2} e^{\frac{(x+y)^2}{2x}} \operatorname{erfc}\left( \frac{x+y}{\sqrt{2}\sqrt{x}} \right) + \sqrt{2}(y-x) \right)}{2\sqrt{\pi}x^{3/2}} < 0$$

$$\Longleftrightarrow \quad \sqrt{\pi}x^{3/2} e^{\frac{(x+y)^2}{2x}} \operatorname{erfc}\left( \frac{x+y}{\sqrt{2}\sqrt{x}} \right) + \sqrt{2}(y-x) < 0$$

$$\sqrt{\pi}x^{3/2} e^{\frac{(x+y)^2}{2x}} \operatorname{erfc}\left( \frac{x+y}{\sqrt{2}\sqrt{x}} \right) + \sqrt{2}(y-x) \leqslant$$

$$\frac{2x^{3/2}}{\frac{x+y}{\sqrt{2}\sqrt{x}} + \sqrt{\frac{(x+y)^2}{2x} + \frac{4}{\pi}}} + y\sqrt{2} - x\sqrt{2} \leqslant$$

$$\frac{2 \cdot 0.64^{3/2}}{\frac{0.01+0.64}{\sqrt{2}\sqrt{0.64}} + \sqrt{\frac{(0.01+0.64)^2}{2\cdot0.64} + \frac{4}{\pi}}} + 0.01\sqrt{2} - 0.64\sqrt{2} = -0.334658 < 0. \tag{309}$$

The two last inqualities come from applying Abramowitz bounds [22] and from the fact that the expression $\frac{2x^{3/2}}{\frac{x+y}{\sqrt{2}\sqrt{x}}+\sqrt{\frac{(x+y)^2}{2x}+\frac{4}{\pi}}} + y\sqrt{2} - x\sqrt{2}$ does not change monotonicity in the domain and hence the maximum must be found at the border. For $x = 0.64$ that maximizes the function $f(x,y)$ is monotonically in $y$, because its derivative w.r.t. $y$ at $x = 0.64$ is

$$e^y \left( 1.37713 \operatorname{erfc}(0.883883y + 0.565685) - 1.37349e^{-0.78125(y+0.64)^2} \right) < 0$$

$$\Longleftrightarrow \quad \left( 1.37713 \operatorname{erfc}(0.883883y + 0.565685) - 1.37349e^{-0.78125(y+0.64)^2} \right) < 0$$

$$\left(1.37713\,\mathrm{erfc}(0.883883y + 0.565685) - 1.37349e^{-0.78125(y+0.64)^2}\right) \leqslant$$

$$\left(1.37713\,\mathrm{erfc}(0.883883 \cdot -0.01 + 0.565685) - 1.37349e^{-0.78125(0.01+0.64)^2}\right) =$$

$$0.5935272325870631 - 0.987354705867739 < 0. \tag{310}$$

Therefore, the values $y = 0.64$ and $x = -0.01$ give a global maximum of the function $f(x, y)$ in the domain $-0.01 \leqslant y \leqslant 0.01$ and $0.64 \leqslant x \leqslant 1.875$ and the values $y = 1.875$ and $x = 0.01$ give the global minimum. $\qquad\square$

# S4   Additional information on experiments

In this section, we report the hyperparameters that were considered for each method and data set and give details on the processing of the data sets.

## S4.1   121 UCI Machine Learning Repository data sets: Hyperparameters

For the UCI data sets, the best hyperparameter setting was determined by a grid-search over all hyperparameter combinations using 15% of the training data as validation set. The early stopping parameter was determined on the smoothed learning curves of 100 epochs of the validation set. Smoothing was done using moving averages of 10 consecutive values. We tested "rectangular" and "conic" layers – rectangular layers have constant number of hidden units in each layer, conic layers start with the given number of hidden units in the first layer and then decrease the number of hidden units to the size of the output layer according to the geometric progression. If multiple hyperparameters provided identical performance on the validation set, we preferred settings with a higher number of layers, lower learning rates and higher dropout rates. All methods had the chance to adjust their hyperparameters to the data set at hand.

Table S1: Hyperparameters considered for self-normalizing networks in the UCI data sets.

| Hyperparameter | Considered values |
|---|---|
| Number of hidden units | {1024, 512, 256} |
| Number of hidden layers | {2, 3, 4, 8, 16, 32} |
| Learning rate | {0.01, 0.1, 1} |
| Dropout rate | {0.05, 0} |
| Layer form | {rectangular, conic} |

Table S2: Hyperparameters considered for ReLU networks with MS initialization in the UCI data sets.

| Hyperparameter | Considered values |
|---|---|
| Number of hidden units | {1024, 512, 256} |
| Number of hidden layers | {2,3,4,8,16,32} |
| Learning rate | {0.01, 0.1, 1} |
| Dropout rate | {0.5, 0} |
| Layer form | {rectangular, conic} |

Table S3: Hyperparameters considered for batch normalized networks in the UCI data sets.

| Hyperparameter | Considered values |
|---|---|
| Number of hidden units | {1024, 512, 256} |
| Number of hidden layers | {2, 3, 4, 8, 16, 32} |
| Learning rate | {0.01, 0.1, 1} |
| Normalization | {Batchnorm} |
| Layer form | {rectangular, conic} |

Table S4: Hyperparameters considered for weight normalized networks in the UCI data sets.

| Hyperparameter | Considered values |
|---|---|
| Number of hidden units | {1024, 512, 256} |
| Number of hidden layers | {2, 3, 4, 8, 16, 32} |
| Learning rate | {0.01, 0.1, 1} |
| Normalization | {Weightnorm} |
| Layer form | {rectangular, conic} |

Table S5: Hyperparameters considered for layer normalized networks in the UCI data sets.

| Hyperparameter | Considered values |
|---|---|
| Number of hidden units | {1024, 512, 256} |
| Number of hidden layers | {2, 3, 4, 8, 16, 32} |
| Learning rate | {0.01, 0.1, 1} |
| Normalization | {Layernorm} |
| Layer form | {rectangular, conic} |

Table S6: Hyperparameters considered for Highway networks in the UCI data sets.

| Hyperparameter | Considered values |
|---|---|
| Number of hidden layers | {2, 3, 4, 8, 16, 32} |
| Learning rate | {0.01, 0.1, 1} |
| Dropout rate | {0, 0.5} |

Table S7: Hyperparameters considered for Residual networks in the UCI data sets.

| Hyperparameter | Considered values |
|---|---|
| Number of blocks | {2, 3, 4, 8, 16} |
| Number of neurons per blocks | {1024, 512, 256} |
| Block form | {rectangular, diavolo} |
| Bottleneck | {25%, 50%} |
| Learning rate | {0.01, 0.1, 1} |

## S4.2   121 UCI Machine Learning Repository data sets: detailed results

**Methods compared.**   We used data sets and preprocessing scripts by Fernández-Delgado et al. (2014) for data preparation and defining training and test sets. With several flaws in the method comparison(Wainberg et al., 2016) that we avoided, the authors compared 179 machine learning methods of 17 groups in their experiments. The method groups were defined by Fernández-Delgado et al. (2014) as follows: Support Vector Machines, RandomForest, Multivariate adaptive regression splines (MARS), Boosting, Rule-based, logistic and multinomial regression, Discriminant Analysis (DA), Bagging, Nearest Neighbour, DecisionTree, other Ensembles, Neural Networks, Bayesian, Other Methods, generalized linear models (GLM), Partial least squares and principal component regression (PLSR), and Stacking. However, many of methods assigned to those groups were merely different implementations of the same method. Therefore, we selected one representative of each of the 17 groups for method comparison. The representative method was chosen as the group's method with the median performance across all tasks. Finally, we included 17 other machine learning methods of Fernández-Delgado et al. (2014), and 6 FNNs, BatchNorm, WeightNorm, LayerNorm, Highway, Residual and MSRAinit networks, and self-normalizing neural networks (SNNs) giving a total of 24 compared methods.

**Results of FNN methods for all 121 data sets.**   The results of the compared FNN methods can be found in Table S8.

**Small and large data sets.**   We assigned each of the 121 UCI data sets into the group "large datasets" or "small datasets" if the had more than 1,000 data points or less, respectively. We expected that Deep Learning methods require large data sets to competitive to other machine learning methods. This resulted in 75 small and 46 large data sets.

Table S8: Comparison of FNN methods on all 121 UCI data sets.. The table reports the accuracy of FNN methods at each individual task of the 121 UCI data sets. The first column gives the name of the data set, the second the number of training data points $N$, the third the number of features $M$ and the consecutive columns the accuracy values of self-normalizing networks (SNNs), ReLU networks without normalization and with MSRA initialization (MS), Highway networks (HW), Residual Networks (ResNet), networks with batch normalization (BN), weight normalization (WN), and layer normalization (LN).

| dataset | $N$ | $M$ | SNN | MS | HW | ResNet | BN | WN | LN |
|---|---|---|---|---|---|---|---|---|---|
| abalone | 4177 | 9 | 0.6657 | 0.6284 | 0.6427 | 0.6466 | 0.6303 | 0.6351 | 0.6178 |
| acute-inflammation | 120 | 7 | 1.0000 | 1.0000 | 1.0000 | 1.0000 | 1.0000 | 1.0000 | 0.9000 |
| acute-nephritis | 120 | 7 | 1.0000 | 1.0000 | 1.0000 | 1.0000 | 1.0000 | 1.0000 | 1.0000 |
| adult | 48842 | 15 | 0.8476 | 0.8487 | 0.8453 | 0.8484 | 0.8499 | 0.8453 | 0.8517 |
| annealing | 898 | 32 | 0.7600 | 0.7300 | 0.3600 | 0.2600 | 0.1200 | 0.6500 | 0.5000 |
| arrhythmia | 452 | 263 | 0.6549 | 0.6372 | 0.6283 | 0.6460 | 0.5929 | 0.6018 | 0.5752 |
| audiology-std | 196 | 60 | 0.8000 | 0.6800 | 0.7200 | 0.8000 | 0.6400 | 0.7200 | 0.8000 |
| balance-scale | 625 | 5 | 0.9231 | 0.9231 | 0.9103 | 0.9167 | 0.9231 | 0.9551 | 0.9872 |
| balloons | 16 | 5 | 1.0000 | 0.5000 | 0.2500 | 1.0000 | 1.0000 | 0.0000 | 0.7500 |
| bank | 4521 | 17 | 0.8903 | 0.8876 | 0.8885 | 0.8796 | 0.8823 | 0.8850 | 0.8920 |
| blood | 748 | 5 | 0.7701 | 0.7754 | 0.7968 | 0.8021 | 0.7647 | 0.7594 | 0.7112 |
| breast-cancer | 286 | 10 | 0.7183 | 0.6901 | 0.7465 | 0.7465 | 0.7324 | 0.6197 | 0.6620 |
| breast-cancer-wisc | 699 | 10 | 0.9714 | 0.9714 | 0.9771 | 0.9714 | 0.9829 | 0.9657 | 0.9714 |
| breast-cancer-wisc-diag | 569 | 31 | 0.9789 | 0.9718 | 0.9789 | 0.9507 | 0.9789 | 0.9718 | 0.9648 |
| breast-cancer-wisc-prog | 198 | 34 | 0.6735 | 0.7347 | 0.8367 | 0.8163 | 0.7755 | 0.8367 | 0.7959 |
| breast-tissue | 106 | 10 | 0.7308 | 0.4615 | 0.6154 | 0.4231 | 0.4615 | 0.5385 | 0.5769 |
| car | 1728 | 7 | 0.9838 | 0.9861 | 0.9560 | 0.9282 | 0.9606 | 0.9769 | 0.9907 |
| cardiotocography-10clases | 2126 | 22 | 0.8399 | 0.8418 | 0.8456 | 0.8173 | 0.7910 | 0.8606 | 0.8362 |
| cardiotocography-3clases | 2126 | 22 | 0.9153 | 0.8964 | 0.9171 | 0.9021 | 0.9096 | 0.8945 | 0.9021 |
| chess-krvk | 28056 | 7 | 0.8805 | 0.8606 | 0.5255 | 0.8543 | 0.8781 | 0.7673 | 0.8938 |
| chess-krvkp | 3196 | 37 | 0.9912 | 0.9900 | 0.9900 | 0.9912 | 0.9862 | 0.9912 | 0.9875 |
| congressional-voting | 435 | 17 | 0.6147 | 0.6055 | 0.5872 | 0.5963 | 0.5872 | 0.5872 | 0.5780 |
| conn-bench-sonar-mines-rocks | 208 | 61 | 0.7885 | 0.8269 | 0.8462 | 0.8077 | 0.7115 | 0.8269 | 0.6731 |
| conn-bench-vowel-deterding | 990 | 12 | 0.9957 | 0.9935 | 0.9784 | 0.9935 | 0.9610 | 0.9524 | 0.9935 |
| connect-4 | 67557 | 43 | 0.8807 | 0.8831 | 0.8599 | 0.8716 | 0.8729 | 0.8833 | 0.8856 |
| contrac | 1473 | 10 | 0.5190 | 0.5136 | 0.5054 | 0.5136 | 0.4538 | 0.4755 | 0.4592 |
| credit-approval | 690 | 16 | 0.8430 | 0.8430 | 0.8547 | 0.8430 | 0.8721 | 0.9070 | 0.8547 |
| cylinder-bands | 512 | 36 | 0.7266 | 0.7656 | 0.7969 | 0.7734 | 0.7500 | 0.7578 | 0.7578 |
| dermatology | 366 | 35 | 0.9231 | 0.9121 | 0.9780 | 0.9231 | 0.9341 | 0.9451 | 0.9451 |
| echocardiogram | 131 | 11 | 0.8182 | 0.8485 | 0.6061 | 0.8485 | 0.8485 | 0.7879 | 0.8182 |
| ecoli | 336 | 8 | 0.8929 | 0.8333 | 0.8690 | 0.8214 | 0.8214 | 0.8452 | 0.8571 |
| energy-y1 | 768 | 9 | 0.9583 | 0.9583 | 0.8802 | 0.8177 | 0.8646 | 0.9010 | 0.9479 |
| energy-y2 | 768 | 9 | 0.9063 | 0.8958 | 0.9010 | 0.8750 | 0.8750 | 0.8906 | 0.8802 |
| fertility | 100 | 10 | 0.9200 | 0.8800 | 0.8800 | 0.8400 | 0.6800 | 0.6800 | 0.8800 |
| flags | 194 | 29 | 0.4583 | 0.4583 | 0.4375 | 0.3750 | 0.4167 | 0.4167 | 0.3542 |
| glass | 214 | 10 | 0.7358 | 0.6038 | 0.6415 | 0.6415 | 0.5849 | 0.6792 | 0.6981 |
| haberman-survival | 306 | 4 | 0.7368 | 0.7237 | 0.6447 | 0.6842 | 0.7368 | 0.7500 | 0.6842 |
| hayes-roth | 160 | 4 | 0.6786 | 0.4643 | 0.7857 | 0.7143 | 0.7500 | 0.5714 | 0.8929 |
| heart-cleveland | 303 | 14 | 0.6184 | 0.6053 | 0.6316 | 0.5658 | 0.5789 | 0.5658 | 0.5789 |
| heart-hungarian | 294 | 13 | 0.7945 | 0.8356 | 0.7945 | 0.8082 | 0.8493 | 0.7534 | 0.8493 |
| heart-switzerland | 123 | 13 | 0.3548 | 0.3871 | 0.5806 | 0.3226 | 0.3871 | 0.2581 | 0.5161 |
| heart-va | 200 | 13 | 0.3600 | 0.2600 | 0.4000 | 0.2600 | 0.2800 | 0.2200 | 0.2400 |
| hepatitis | 155 | 20 | 0.7692 | 0.7692 | 0.6667 | 0.7692 | 0.8718 | 0.8462 | 0.7436 |
| hill-valley | 1212 | 101 | 0.5248 | 0.5116 | 0.5000 | 0.5396 | 0.5050 | 0.4934 | 0.5050 |
| horse-colic | 368 | 26 | 0.8088 | 0.8529 | 0.7794 | 0.8088 | 0.8529 | 0.7059 | 0.7941 |
| ilpd-indian-liver | 583 | 10 | 0.6986 | 0.6644 | 0.6781 | 0.6712 | 0.5959 | 0.6918 | 0.6986 |

| | | | | | | | | |
|---|---|---|---|---|---|---|---|---|
| image-segmentation | 2310 | 19 | 0.9114 | 0.9090 | 0.9024 | 0.8919 | 0.8481 | 0.8938 | 0.8838 |
| ionosphere | 351 | 34 | 0.8864 | 0.9091 | 0.9432 | 0.9545 | 0.9432 | 0.9318 | 0.9432 |
| iris | 150 | 5 | 0.9730 | 0.9189 | 0.8378 | 0.9730 | 0.9189 | 1.0000 | 0.9730 |
| led-display | 1000 | 8 | 0.7640 | 0.7200 | 0.7040 | 0.7160 | 0.6280 | 0.6920 | 0.6480 |
| lenses | 24 | 5 | 0.6667 | 1.0000 | 1.0000 | 0.6667 | 0.8333 | 0.8333 | 0.6667 |
| letter | 20000 | 17 | 0.9726 | 0.9712 | 0.8984 | 0.9762 | 0.9796 | 0.9580 | 0.9742 |
| libras | 360 | 91 | 0.7889 | 0.8667 | 0.8222 | 0.7111 | 0.7444 | 0.8000 | 0.8333 |
| low-res-spect | 531 | 101 | 0.8571 | 0.8496 | 0.9023 | 0.8647 | 0.8571 | 0.8872 | 0.8947 |
| lung-cancer | 32 | 57 | 0.6250 | 0.3750 | 0.1250 | 0.2500 | 0.5000 | 0.5000 | 0.2500 |
| lymphography | 148 | 19 | 0.9189 | 0.7297 | 0.7297 | 0.6757 | 0.7568 | 0.7568 | 0.7838 |
| magic | 19020 | 11 | 0.8692 | 0.8629 | 0.8673 | 0.8723 | 0.8713 | 0.8690 | 0.8620 |
| mammographic | 961 | 6 | 0.8250 | 0.8083 | 0.7917 | 0.7833 | 0.8167 | 0.8292 | 0.8208 |
| miniboone | 130064 | 51 | 0.9307 | 0.9250 | 0.9270 | 0.9254 | 0.9262 | 0.9272 | 0.9313 |
| molec-biol-promoter | 106 | 58 | 0.8462 | 0.7692 | 0.6923 | 0.7692 | 0.7692 | 0.6923 | 0.4615 |
| molec-biol-splice | 3190 | 61 | 0.9009 | 0.8482 | 0.8833 | 0.8557 | 0.8519 | 0.8494 | 0.8607 |
| monks-1 | 556 | 7 | 0.7523 | 0.6551 | 0.5833 | 0.7546 | 0.9074 | 0.5000 | 0.7014 |
| monks-2 | 601 | 7 | 0.5926 | 0.6343 | 0.6389 | 0.6273 | 0.3287 | 0.6644 | 0.5162 |
| monks-3 | 554 | 7 | 0.6042 | 0.7454 | 0.5880 | 0.5833 | 0.5278 | 0.5231 | 0.6991 |
| mushroom | 8124 | 22 | 1.0000 | 1.0000 | 1.0000 | 1.0000 | 0.9990 | 0.9995 | 0.9995 |
| musk-1 | 476 | 167 | 0.8739 | 0.8655 | 0.8992 | 0.8739 | 0.8235 | 0.8992 | 0.8992 |
| musk-2 | 6598 | 167 | 0.9891 | 0.9945 | 0.9915 | 0.9964 | 0.9982 | 0.9927 | 0.9951 |
| nursery | 12960 | 9 | 0.9978 | 0.9988 | 1.0000 | 0.9994 | 0.9994 | 0.9966 | 0.9966 |
| oocytes_merluccius_nucleus_4d | 1022 | 42 | 0.8235 | 0.8196 | 0.7176 | 0.8000 | 0.8078 | 0.8078 | 0.7686 |
| oocytes_merluccius_states_2f | 1022 | 26 | 0.9529 | 0.9490 | 0.9490 | 0.9373 | 0.9333 | 0.9020 | 0.9412 |
| oocytes_trisopterus_nucleus_2f | 912 | 26 | 0.7982 | 0.8728 | 0.8289 | 0.7719 | 0.7456 | 0.7939 | 0.8202 |
| oocytes_trisopterus_states_5b | 912 | 33 | 0.9342 | 0.9430 | 0.9342 | 0.8947 | 0.8947 | 0.9254 | 0.8991 |
| optical | 5620 | 63 | 0.9711 | 0.9666 | 0.9644 | 0.9627 | 0.9716 | 0.9638 | 0.9755 |
| ozone | 2536 | 73 | 0.9700 | 0.9732 | 0.9716 | 0.9669 | 0.9669 | 0.9748 | 0.9716 |
| page-blocks | 5473 | 11 | 0.9583 | 0.9708 | 0.9656 | 0.9605 | 0.9613 | 0.9730 | 0.9708 |
| parkinsons | 195 | 23 | 0.8980 | 0.9184 | 0.8367 | 0.9184 | 0.8571 | 0.8163 | 0.8571 |
| pendigits | 10992 | 17 | 0.9706 | 0.9714 | 0.9671 | 0.9708 | 0.9734 | 0.9620 | 0.9657 |
| pima | 768 | 9 | 0.7552 | 0.7656 | 0.7188 | 0.7135 | 0.7188 | 0.6979 | 0.6927 |
| pittsburg-bridges-MATERIAL | 106 | 8 | 0.8846 | 0.8462 | 0.9231 | 0.9231 | 0.8846 | 0.8077 | 0.9231 |
| pittsburg-bridges-REL-L | 103 | 8 | 0.6923 | 0.7692 | 0.6923 | 0.8462 | 0.7692 | 0.6538 | 0.7308 |
| pittsburg-bridges-SPAN | 92 | 8 | 0.6957 | 0.5217 | 0.5652 | 0.5652 | 0.5652 | 0.6522 | 0.6087 |
| pittsburg-bridges-T-OR-D | 102 | 8 | 0.8400 | 0.8800 | 0.8800 | 0.8800 | 0.8800 | 0.8800 | 0.8800 |
| pittsburg-bridges-TYPE | 105 | 8 | 0.6538 | 0.6538 | 0.5385 | 0.6538 | 0.1154 | 0.4615 | 0.6538 |
| planning | 182 | 13 | 0.6889 | 0.6667 | 0.6000 | 0.7111 | 0.6222 | 0.6444 | 0.6889 |
| plant-margin | 1600 | 65 | 0.8125 | 0.8125 | 0.8375 | 0.7975 | 0.7600 | 0.8175 | 0.8425 |
| plant-shape | 1600 | 65 | 0.7275 | 0.6350 | 0.6325 | 0.5150 | 0.2850 | 0.6575 | 0.6775 |
| plant-texture | 1599 | 65 | 0.8125 | 0.7900 | 0.7900 | 0.8000 | 0.8200 | 0.8175 | 0.8350 |
| post-operative | 90 | 9 | 0.7273 | 0.7273 | 0.5909 | 0.7273 | 0.5909 | 0.5455 | 0.7727 |
| primary-tumor | 330 | 18 | 0.5244 | 0.5000 | 0.4512 | 0.3902 | 0.5122 | 0.5000 | 0.4512 |
| ringnorm | 7400 | 21 | 0.9751 | 0.9843 | 0.9692 | 0.9811 | 0.9843 | 0.9719 | 0.9827 |
| seeds | 210 | 8 | 0.8846 | 0.8654 | 0.9423 | 0.8654 | 0.8654 | 0.8846 | 0.8846 |
| semeion | 1593 | 257 | 0.9196 | 0.9296 | 0.9447 | 0.9146 | 0.9372 | 0.9322 | 0.9447 |
| soybean | 683 | 36 | 0.8511 | 0.8723 | 0.8617 | 0.8670 | 0.8883 | 0.8537 | 0.8484 |
| spambase | 4601 | 58 | 0.9409 | 0.9461 | 0.9435 | 0.9461 | 0.9426 | 0.9504 | 0.9513 |
| spect | 265 | 23 | 0.6398 | 0.6183 | 0.6022 | 0.6667 | 0.6344 | 0.6398 | 0.6720 |
| spectf | 267 | 45 | 0.4973 | 0.6043 | 0.8930 | 0.7005 | 0.2299 | 0.4545 | 0.5561 |
| statlog-australian-credit | 690 | 15 | 0.5988 | 0.6802 | 0.6802 | 0.6395 | 0.6802 | 0.6860 | 0.6279 |
| statlog-german-credit | 1000 | 25 | 0.7560 | 0.7280 | 0.7760 | 0.7720 | 0.7520 | 0.7400 | 0.7400 |

| | | | | | | | | |
|---|---|---|---|---|---|---|---|---|
| statlog-heart | 270 | 14 | 0.9254 | 0.8358 | 0.7761 | 0.8657 | 0.7910 | 0.8657 | 0.7910 |
| statlog-image | 2310 | 19 | 0.9549 | 0.9757 | 0.9584 | 0.9584 | 0.9671 | 0.9515 | 0.9757 |
| statlog-landsat | 6435 | 37 | 0.9100 | 0.9075 | 0.9110 | 0.9055 | 0.9040 | 0.8925 | 0.9040 |
| statlog-shuttle | 58000 | 10 | 0.9990 | 0.9983 | 0.9977 | 0.9992 | 0.9988 | 0.9988 | 0.9987 |
| statlog-vehicle | 846 | 19 | 0.8009 | 0.8294 | 0.7962 | 0.7583 | 0.7583 | 0.8009 | 0.7915 |
| steel-plates | 1941 | 28 | 0.7835 | 0.7567 | 0.7608 | 0.7629 | 0.7031 | 0.7856 | 0.7588 |
| synthetic-control | 600 | 61 | 0.9867 | 0.9800 | 0.9867 | 0.9600 | 0.9733 | 0.9867 | 0.9733 |
| teaching | 151 | 6 | 0.5000 | 0.6053 | 0.5263 | 0.5526 | 0.5000 | 0.3158 | 0.6316 |
| thyroid | 7200 | 22 | 0.9816 | 0.9770 | 0.9708 | 0.9799 | 0.9778 | 0.9807 | 0.9752 |
| tic-tac-toe | 958 | 10 | 0.9665 | 0.9833 | 0.9749 | 0.9623 | 0.9833 | 0.9707 | 0.9791 |
| titanic | 2201 | 4 | 0.7836 | 0.7909 | 0.7927 | 0.7727 | 0.7800 | 0.7818 | 0.7891 |
| trains | 10 | 30 | NA | NA | NA | NA | 0.5000 | 0.5000 | 1.0000 |
| twonorm | 7400 | 21 | 0.9805 | 0.9778 | 0.9708 | 0.9735 | 0.9757 | 0.9730 | 0.9724 |
| vertebral-column-2clases | 310 | 7 | 0.8312 | 0.8701 | 0.8571 | 0.8312 | 0.8312 | 0.6623 | 0.8442 |
| vertebral-column-3clases | 310 | 7 | 0.8312 | 0.8052 | 0.7922 | 0.7532 | 0.7792 | 0.7403 | 0.8312 |
| wall-following | 5456 | 25 | 0.9098 | 0.9076 | 0.9230 | 0.9223 | 0.9333 | 0.9274 | 0.9128 |
| waveform | 5000 | 22 | 0.8480 | 0.8312 | 0.8320 | 0.8360 | 0.8360 | 0.8376 | 0.8448 |
| waveform-noise | 5000 | 41 | 0.8608 | 0.8328 | 0.8696 | 0.8584 | 0.8480 | 0.8640 | 0.8504 |
| wine | 178 | 14 | 0.9773 | 0.9318 | 0.9091 | 0.9773 | 0.9773 | 0.9773 | 0.9773 |
| wine-quality-red | 1599 | 12 | 0.6300 | 0.6250 | 0.5625 | 0.6150 | 0.5450 | 0.5575 | 0.6100 |
| wine-quality-white | 4898 | 12 | 0.6373 | 0.6479 | 0.5564 | 0.6307 | 0.5335 | 0.5482 | 0.6544 |
| yeast | 1484 | 9 | 0.6307 | 0.6173 | 0.6065 | 0.5499 | 0.4906 | 0.5876 | 0.6092 |
| zoo | 101 | 17 | 0.9200 | 1.0000 | 0.8800 | 1.0000 | 0.7200 | 0.9600 | 0.9600 |

**Results.** The results of the method comparison are given in Tables S9 and S10 for small and large data sets, respectively. On small data sets, SVMs performed best followed by RandomForest and SNNs. On large data sets, SNNs are the best method followed by SVMs and Random Forest.

Table S9: UCI comparison reporting the average rank of a method on 75 classification task of the UCI machine learning repository with less than 1000 data points. For each dataset, the 24 compared methods, were ranked by their accuracy and the ranks were averaged across the tasks. The first column gives the method group, the second the method, the third the average rank , and the last the $p$-value of a paired Wilcoxon test whether the difference to the best performing method is significant. SNNs are ranked third having been outperformed by Random Forests and SVMs.

| methodGroup | method | avg. rank | $p$-value |
|---|---|---|---|
| SVM | LibSVM_weka | 9.3 | |
| RandomForest | RRFglobal_caret | 9.6 | 2.5e-01 |
| SNN | SNN | 9.6 | 3.8e-01 |
| LMR | SimpleLogistic_weka | 9.9 | 1.5e-01 |
| NeuralNetworks | lvq_caret | 10.1 | 1.0e-01 |
| MARS | gcvEarth_caret | 10.7 | 3.6e-02 |
| MSRAinit | MSRAinit | 11.0 | 4.0e-02 |
| LayerNorm | LayerNorm | 11.3 | 7.2e-02 |
| Highway | Highway | 11.5 | 8.9e-03 |
| DiscriminantAnalysis | mda_R | 11.8 | 2.6e-03 |
| Boosting | LogitBoost_weka | 11.9 | 2.4e-02 |
| Bagging | ctreeBag_R | 12.1 | 1.8e-03 |
| ResNet | ResNet | 12.3 | 3.5e-03 |
| BatchNorm | BatchNorm | 12.6 | 4.9e-04 |
| Rule-based | JRip_caret | 12.9 | 1.7e-04 |
| WeightNorm | WeightNorm | 13.0 | 8.3e-05 |
| DecisionTree | rpart2_caret | 13.6 | 7.0e-04 |
| OtherEnsembles | Dagging_weka | 13.9 | 3.0e-05 |
| Nearest Neighbour | NNge_weka | 14.0 | 7.7e-04 |
| OtherMethods | pam_caret | 14.2 | 1.5e-04 |
| PLSR | simpls_R | 14.3 | 4.6e-05 |
| Bayesian | NaiveBayes_weka | 14.6 | 1.2e-04 |
| GLM | bayesglm_caret | 15.0 | 1.6e-06 |
| Stacking | Stacking_weka | 20.9 | 2.2e-12 |

Table S10: UCI comparison reporting the average rank of a method on 46 classification task of the UCI machine learning repository with more than 1000 data points. For each dataset, the 24 compared methods, were ranked by their accuracy and the ranks were averaged across the tasks. The first column gives the method group, the second the method, the third the average rank , and the last the $p$-value of a paired Wilcoxon test whether the difference to the best performing method is significant. SNNs are ranked first having outperformed diverse machine learning methods and other FNNs.

| methodGroup | method | avg. rank | $p$-value |
|---|---|---|---|
| SNN | SNN | 5.8 | |
| SVM | LibSVM_weka | 6.1 | 5.8e-01 |
| RandomForest | RRFglobal_caret | 6.6 | 2.1e-01 |
| MSRAinit | MSRAinit | 7.1 | 4.5e-03 |
| LayerNorm | LayerNorm | 7.2 | 7.1e-02 |
| Highway | Highway | 7.9 | 1.7e-03 |
| ResNet | ResNet | 8.4 | 1.7e-04 |
| WeightNorm | WeightNorm | 8.7 | 5.5e-04 |
| BatchNorm | BatchNorm | 9.7 | 1.8e-04 |
| MARS | gcvEarth_caret | 9.9 | 8.2e-05 |
| Boosting | LogitBoost_weka | 12.1 | 2.2e-07 |
| LMR | SimpleLogistic_weka | 12.4 | 3.8e-09 |
| Rule-based | JRip_caret | 12.4 | 9.0e-08 |
| Bagging | ctreeBag_R | 13.5 | 1.6e-05 |
| DiscriminantAnalysis | mda_R | 13.9 | 1.4e-10 |
| Nearest Neighbour | NNge_weka | 14.1 | 1.6e-10 |
| DecisionTree | rpart2_caret | 15.5 | 2.3e-08 |
| OtherEnsembles | Dagging_weka | 16.1 | 4.4e-12 |
| NeuralNetworks | lvq_caret | 16.3 | 1.6e-12 |
| Bayesian | NaiveBayes_weka | 17.9 | 1.6e-12 |
| OtherMethods | pam_caret | 18.3 | 2.8e-14 |
| GLM | bayesglm_caret | 18.7 | 1.5e-11 |
| PLSR | simpls_R | 19.0 | 3.4e-11 |
| Stacking | Stacking_weka | 22.5 | 2.8e-14 |

### S4.3   Tox21 challenge data set: Hyperparameters

For the Tox21 data set, the best hyperparameter setting was determined by a grid-search over all hyperparameter combinations using the validation set defined by the challenge winners (Mayr et al., 2016). The hyperparameter space was chosen to be similar to the hyperparameters that were tested by Mayr et al. (2016). The early stopping parameter was determined on the smoothed learning curves of 100 epochs of the validation set. Smoothing was done using moving averages of 10 consecutive values. We tested "rectangular" and "conic" layers – rectangular layers have constant number of hidden units in each layer, conic layers start with the given number of hidden units in the first layer and then decrease the number of hidden units to the size of the output layer according to the geometric progession. All methods had the chance to adjust their hyperparameters to the data set at hand.

Table S11: Hyperparameters considered for self-normalizing networks in the Tox21 data set.

| Hyperparameter | Considered values |
| --- | --- |
| Number of hidden units | {1024, 2048} |
| Number of hidden layers | {2,3,4,6,8,16,32} |
| Learning rate | {0.01, 0.05, 0.1} |
| Dropout rate | {0.05, 0.10} |
| Layer form | {rectangular, conic} |
| L2 regularization parameter | {0.001,0.0001,0.00001} |

Table S12: Hyperparameters considered for ReLU networks with MS initialization in the Tox21 data set.

| Hyperparameter | Considered values |
| --- | --- |
| Number of hidden units | {1024, 2048} |
| Number of hidden layers | {2,3,4,6,8,16,32} |
| Learning rate | {0.01, 0.05, 0.1} |
| Dropout rate | {0.5, 0} |
| Layer form | {rectangular, conic} |
| L2 regularization parameter | {0.001,0.0001,0.00001} |

Table S13: Hyperparameters considered for batch normalized networks in the Tox21 data set.

| Hyperparameter | Considered values |
|---|---|
| Number of hidden units | {1024, 2048} |
| Number of hidden layers | {2, 3, 4, 6, 8, 16, 32} |
| Learning rate | {0.01, 0.05, 0.1} |
| Normalization | {Batchnorm} |
| Layer form | {rectangular, conic} |
| L2 regularization parameter | {0.001, 0.0001, 0.00001} |

Table S14: Hyperparameters considered for weight normalized networks in the Tox21 data set.

| Hyperparameter | Considered values |
|---|---|
| Number of hidden units | {1024, 2048} |
| Number of hidden layers | {2, 3, 4, 6, 8, 16, 32} |
| Learning rate | {0.01, 0.05, 0.1} |
| Normalization | {Weightnorm} |
| Dropout rate | {0, 0.5} |
| Layer form | {rectangular, conic} |
| L2 regularization parameter | {0.001, 0.0001, 0.00001} |

Table S15: Hyperparameters considered for layer normalized networks in the Tox21 data set.

| Hyperparameter | Considered values |
|---|---|
| Number of hidden units | {1024, 2048} |
| Number of hidden layers | {2, 3, 4, 6, 8, 16, 32} |
| Learning rate | {0.01, 0.05, 0.1} |
| Normalization | {Layernorm} |
| Dropout rate | {0, 0.5} |
| Layer form | {rectangular, conic} |
| L2 regularization parameter | {0.001, 0.0001, 0.00001} |

Table S16: Hyperparameters considered for Highway networks in the Tox21 data set.

| Hyperparameter | Considered values |
|---|---|
| Number of hidden layers | {2, 3, 4, 6, 8, 16, 32} |
| Learning rate | {0.01, 0.05, 0.1} |
| Dropout rate | {0, 0.5} |
| L2 regularization parameter | {0.001, 0.0001, 0.00001} |

Figure S7: Distribution of network inputs of an SNN for the Tox21 data set. The plots show the distribution of network inputs $z$ of the second layer of a typical Tox21 network. The red curves display a kernel density estimator of the network inputs and the black curve is the density of a standard normal distribution. **Left panel:** At initialization time before learning. The distribution of network inputs is close to a standard normal distribution. **Right panel:** After 40 epochs of learning. The distributions of network inputs is close to a normal distribution.

Table S17: Hyperparameters considered for Residual networks in the Tox21 data set.

| Hyperparameter | Considered values |
|---|---|
| Number of blocks | {2, 3, 4, 6, 8, 16} |
| Number of neurons per blocks | {1024, 2048} |
| Block form | {rectangular, diavolo} |
| Bottleneck | {25%, 50%} |
| Learning rate | {0.01, 0.05, 0.1} |
| L2 regularization parameter | {0.001,0.0001,0.00001} |

**Distribution of network inputs.**    We empirically checked the assumption that the distribution of network inputs can well be approximated by a normal distribution. To this end, we investigated the density of the network inputs before and during learning and found that these density are close to normal distributions (see Figure S7).

## S4.4  HTRU2 data set: Hyperparameters

For the HTRU2 data set, the best hyperparameter setting was determined by a grid-search over all hyperparameter combinations using one of the 9 non-testing folds as validation fold in a nested cross-validation procedure. Concretely, if $M$ was the testing fold, we used $M - 1$ as validation fold, and for $M = 1$ we used fold 10 for validation. The early stopping parameter was determined on the smoothed learning curves of 100 epochs of the validation set. Smoothing was done using moving averages of 10 consecutive values. We tested "rectangular" and "conic" layers – rectangular layers have constant number of hidden units in each layer, conic layers start with the given number of hidden units in the first layer and then decrease the number of hidden units to the size of the output layer according to the geometric progression. All methods had the chance to adjust their hyperparameters to the data set at hand.

Table S18: Hyperparameters considered for self-normalizing networks on the HTRU2 data set.

| Hyperparameter | Considered values |
|---|---|
| Number of hidden units | {256, 512, 1024} |
| Number of hidden layers | {2, 4, 8, 16, 32} |
| Learning rate | {0.1, 0.01, 1} |
| Dropout rate | { 0, 0.05} |
| Layer form | {rectangular, conic} |

Table S19: Hyperparameters considered for ReLU networks with Microsoft initialization on the HTRU2 data set.

| Hyperparameter | Considered values |
|---|---|
| Number of hidden units | {256, 512, 1024} |
| Number of hidden layers | {2, 4, 8, 16, 32} |
| Learning rate | {0.1, 0.01, 1} |
| Dropout rate | {0, 0.5} |
| Layer form | {rectangular, conic} |

Table S20: Hyperparameters considered for BatchNorm networks on the HTRU2 data set.

| Hyperparameter | Considered values |
|---|---|
| Number of hidden units | {256, 512, 1024} |
| Number of hidden layers | {2, 4, 8, 16, 32} |
| Learning rate | {0.1, 0.01, 1} |
| Normalization | {Batchnorm} |
| Layer form | {rectangular, conic} |

Table S21: Hyperparameters considered for WeightNorm networks on the HTRU2 data set.

| Hyperparameter | Considered values |
|---|---|
| Number of hidden units | {256, 512, 1024} |
| Number of hidden layers | {2, 4, 8, 16, 32} |
| Learning rate | {0.1, 0.01, 1} |
| Normalization | {Weightnorm} |
| Layer form | {rectangular, conic} |

Table S22: Hyperparameters considered for LayerNorm networks on the HTRU2 data set.

| Hyperparameter | Considered values |
|---|---|
| Number of hidden units | {256, 512, 1024} |
| Number of hidden layers | {2, 4, 8, 16, 32} |
| Learning rate | {0.1, 0.01, 1} |
| Normalization | {Layernorm} |
| Layer form | {rectangular, conic} |

Table S23: Hyperparameters considered for Highway networks on the HTRU2 data set.

| Hyperparameter | Considered values |
|---|---|
| Number of Hidden Layers | {2, 4, 8, 16, 32} |
| Learning rate | {0.1, 0.01, 1} |
| Dropout rate | {0, 0.5} |

Table S24: Hyperparameters considered for Residual networks on the HTRU2 data set.

| Hyperparameter | Considered values |
|---|---|
| Number of hidden units | {256, 512, 1024} |
| Number of residual blocks | {2, 3, 4, 8, 16} |
| Learning rate | {0.1, 0.01, 1} |
| Block form | {rectangular, diavolo} |
| Bottleneck | {0.25, 0.5} |

# S5   Appendix

In this section we report bounds on previously discussed expressions as determined by numerical methods (min and max have been computed).

$$0_{(\mu=0.06,\omega=0,\nu=1.35,\tau=1.12)} < \frac{\partial \mathcal{J}_{11}}{\partial \mu} < .00182415_{(\mu=-0.1,\omega=0.1,\nu=1.47845,\tau=0.883374)}$$
$$(311)$$

$$0.905413_{(\mu=0.1,\omega=-0.1,\nu=1.5,\tau=1.25)} < \frac{\partial \mathcal{J}_{11}}{\partial \omega} < 1.04143_{(\mu=0.1,\omega=0.1,\nu=0.8,\tau=0.8)}$$

$$-0.0151177_{(\mu=-0.1,\omega=0.1,\nu=0.8,\tau=1.25)} < \frac{\partial \mathcal{J}_{11}}{\partial \nu} < 0.0151177_{(\mu=0.1,\omega=-0.1,\nu=0.8,\tau=1.25)}$$

$$-0.015194_{(\mu=-0.1,\omega=0.1,\nu=0.8,\tau=1.25)} < \frac{\partial \mathcal{J}_{11}}{\partial \tau} < 0.015194_{(\mu=0.1,\omega=-0.1,\nu=0.8,\tau=1.25)}$$

$$-0.0151177_{(\mu=-0.1,\omega=0.1,\nu=0.8,\tau=1.25)} < \frac{\partial \mathcal{J}_{12}}{\partial \mu} < 0.0151177_{(\mu=0.1,\omega=-0.1,\nu=0.8,\tau=1.25)}$$

$$-0.0151177_{(\mu=0.1,\omega=-0.1,\nu=0.8,\tau=1.25)} < \frac{\partial \mathcal{J}_{12}}{\partial \omega} < 0.0151177_{(\mu=0.1,\omega=-0.1,\nu=0.8,\tau=1.25)}$$

$$-0.00785613_{(\mu=0.1,\omega=-0.1,\nu=1.5,\tau=1.25)} < \frac{\partial \mathcal{J}_{12}}{\partial \nu} < 0.0315805_{(\mu=0.1,\omega=0.1,\nu=0.8,\tau=0.8)}$$

$$0.0799824_{(\mu=0.1,\omega=-0.1,\nu=1.5,\tau=1.25)} < \frac{\partial \mathcal{J}_{12}}{\partial \tau} < 0.110267_{(\mu=-0.1,\omega=0.1,\nu=0.8,\tau=0.8)}$$

$$0_{(\mu=0.06,\omega=0,\nu=1.35,\tau=1.12)} < \frac{\partial \mathcal{J}_{21}}{\partial \mu} < 0.0174802_{(\mu=0.1,\omega=0.1,\nu=0.8,\tau=0.8)}$$

$$0.0849308_{(\mu=0.1,\omega=-0.1,\nu=0.8,\tau=0.8)} < \frac{\partial \mathcal{J}_{21}}{\partial \omega} < 0.695766_{(\mu=0.1,\omega=0.1,\nu=1.5,\tau=1.25)}$$

$$-0.0600823_{(\mu=0.1,\omega=-0.1,\nu=0.8,\tau=1.25)} < \frac{\partial \mathcal{J}_{21}}{\partial \nu} < 0.0600823_{(\mu=-0.1,\omega=0.1,\nu=0.8,\tau=1.25)}$$

$$-0.0673083_{(\mu=0.1,\omega=-0.1,\nu=1.5,\tau=0.8)} < \frac{\partial \mathcal{J}_{21}}{\partial \tau} < 0.0673083_{(\mu=-0.1,\omega=0.1,\nu=1.5,\tau=0.8)}$$

$$-0.0600823_{(\mu=0.1,\omega=-0.1,\nu=0.8,\tau=1.25)} < \frac{\partial \mathcal{J}_{22}}{\partial \mu} < 0.0600823_{(\mu=-0.1,\omega=0.1,\nu=0.8,\tau=1.25)}$$

$$-0.0600823_{(\mu=0.1,\omega=-0.1,\nu=0.8,\tau=1.25)} < \frac{\partial \mathcal{J}_{22}}{\partial \omega} < 0.0600823_{(\mu=-0.1,\omega=0.1,\nu=0.8,\tau=1.25)}$$

$$-0.276862_{(\mu=-0.01,\omega=-0.01,\nu=0.8,\tau=1.25)} < \frac{\partial \mathcal{J}_{22}}{\partial \nu} < -0.084813_{(\mu=-0.1,\omega=0.1,\nu=1.5,\tau=0.8)}$$

$$0.562302_{(\mu=0.1,\omega=-0.1,\nu=1.5,\tau=1.25)} < \frac{\partial \mathcal{J}_{22}}{\partial \tau} < 0.664051_{(\mu=0.1,\omega=0.1,\nu=0.8,\tau=0.8)}$$

$$\left| \frac{\partial \mathcal{J}_{11}}{\partial \mu} \right| < 0.00182415(0.0031049101995398316) \qquad (312)$$

$$\left| \frac{\partial \mathcal{J}_{11}}{\partial \omega} \right| < 1.04143(1.055872374194189)$$

$$\left| \frac{\partial \mathcal{J}_{11}}{\partial \nu} \right| < 0.0151177(0.031242911235461816)$$

$$\left|\frac{\partial \mathcal{J}_{11}}{\partial \tau}\right| < 0.015194(0.03749149348255419)$$

$$\left|\frac{\partial \mathcal{J}_{12}}{\partial \mu}\right| < 0.0151177(0.031242911235461816)$$

$$\left|\frac{\partial \mathcal{J}_{12}}{\partial \omega}\right| < 0.0151177(0.031242911235461816)$$

$$\left|\frac{\partial \mathcal{J}_{12}}{\partial \nu}\right| < 0.0315805(0.21232788238624354)$$

$$\left|\frac{\partial \mathcal{J}_{12}}{\partial \tau}\right| < 0.110267(0.2124377655377270)$$

$$\left|\frac{\partial \mathcal{J}_{21}}{\partial \mu}\right| < 0.0174802(0.02220441024325437)$$

$$\left|\frac{\partial \mathcal{J}_{21}}{\partial \omega}\right| < 0.695766(1.146955401845684)$$

$$\left|\frac{\partial \mathcal{J}_{21}}{\partial \nu}\right| < 0.0600823(0.14983446469110305)$$

$$\left|\frac{\partial \mathcal{J}_{21}}{\partial \tau}\right| < 0.0673083(0.17980135762932363)$$

$$\left|\frac{\partial \mathcal{J}_{22}}{\partial \mu}\right| < 0.0600823(0.14983446469110305)$$

$$\left|\frac{\partial \mathcal{J}_{22}}{\partial \omega}\right| < 0.0600823(0.14983446469110305)$$

$$\left|\frac{\partial \mathcal{J}_{22}}{\partial \nu}\right| < 0.562302(1.805740052651535)$$

$$\left|\frac{\partial \mathcal{J}_{22}}{\partial \tau}\right| < 0.664051(2.396685907216327)$$