[Reviews · NeurIPS 2017]

Reviewer 1



I am putting "accept" because this paper already seems to be attracting a lot of attention in the DL community and I see no reason to squash it. However I do have some reservations. There is all this theory and a huge derivation, showing that under certain conditions this particular type of unit will lead to activation norms that don't blow up or get small. Basically it seems to come from the property of SELUs, that on at least one side, as the input to the nonlinearity gets farther from zero, the output gets compressed, so making the input larger doesn't make the output larger. Intuitively it seems to me that many nonlinearities should have this property, including sigmoid-type nonlinearities-- so what relation does this theory have to the experiments? Also, there are ways to deal with this problem (batch-norm, layer-norm etc.), so all this theory seems kind of orthogonal to the practical question of whether SELUs are actually better for deep learning. And although you go to all this trouble to set a particular value of "alpha" in the SELU, I don't see that it's fundamentally any different from a regular ELU, only you'd need to select different variances of the parameters to get your theory to work. As for the experimental results, they all seem to be on extremely small, unstructured datasets ("unstructured" == there is no sequence structure, for instance). It could very well be that ELUs (OK, SELUs) work well for this type of data. Anyway, I'm recommending accept; my only peeve is that the huge derivation, and the amount of attention this paper seems to be attracting, are IMO a little out of proportion to the actual contribution of paper. [Response to something mentioned in the rebuttal: your points about the derivative needing to be > 1 seem to assume that the parameters a Glorot-style covariance. For derivatives less than one, you'd just have to have larger parameter values.]

Reviewer 2



I appreciate the amount of work that enter in this paper. I am however not convinced about the effects of SELU activations in practice. I think the theory build behind them is nice, but, as it happens a lot of times, I’m not sure how much bearing it has in practice. I think the construction here is unstable because of the initial assumptions. Some more detailed comments: 1. Batch norm and other normalization for CNN and RNN vs MLPs. I am not convinced by the authors arguments. One can easily increase the minibatch size to to reduce variance, not? And then they can apply batch norm. Why is this not a solution? 2. Based on the theory we know that, under some constraints on the weights, using SELU you have a fixed point attractor where the activation are normalized. How fast do you converge to this attractor? How many layers? I could easily imagine that you need to have 100 layers to see the activation converge to 0 mean and variance. I might be wrong, but I didn’t felt like we have a bounds or any guarantee of seeing the activations be normalized after just 2 layers or something small like this. 3. Also while the layers on top of such a deep MLP are normalized, the ones on the bottom are not (because you haven’t applied the map sufficiently many times). This means that while the gradients are well behaved through the top layers, they most likely are not on the bottom few layers and can still vanish/explode. 4. This might not be the authors fault, but I’m not familiar with any of the datasets (UCI are a bit toyish to be fair). The numbers don’t seem that impressive to me (e.g. table 2) though I’m not sure I’m reading the numbers correctly. Unfortunately this doesn’t help with my confidence in the approach. Minor detail. Another approach for normalizing activation is the one given by https://arxiv.org/abs/1507.00210, which I feel will be more stable regardless to minibatch size.

Reviewer 3



The paper proposes a new (class of) activation function f to more efficiently train very deep feed-forward neural networks (networks with f are called SNNs). f is a combination of scaled ReLU (positive x) and an exponential (negative x). The authors argue that 1) SNNs converge towards normalized activation distributions 2) SGD is more stable as the SNN approx preserves variance from layer to layer. In fact, f is part of a family of activation functions, for which theoretical guarantees for fixed-point convergence exist. These functions are contraction mappings and characterized by mean/variance preservation across layers at the fixed point -- solving these constraints allows finding other "self-normalizing" f, in principle. Whether f converges to a fixed point, is sensitive to the choice of hyper-parameters: the authors demonstrate certain weight initializations and parameters settings that give the fixed-point behavior. Also, authors analyze the performance of SNNs with a modified version of dropout, which preserves mean/variance while using f. In support, authors provide theoretical proofs and experimental results on many datasets, although these vary in size (and quality?). As such, I think the paper warrants an accept -- although I have questions / concerns that I hope the authors can address. Questions / concerns: 1. Most exps are run on small datasets. This is understandable since only FNNs are considered, but do we get similar results on networks that also include e.g. convolutional / recurrent layers? How can the theoretical analysis work / apply in that case (e.g. convolutions / pooling / memory cells are not always contraction mappings)? 2. Ad point 1. What are the results for SNNs on MNIST / Cifar-10? 3. The authors only considered layers without bias units. What happens when these are included? (This likely make it harder to preserve domains, for instance) 4. During training, the mean/var of the weights do change, and are not guaranteed to be in a basin of attraction (see Thm 1) anymore. Why is there still convergence to a fixed-point, even though no e.g. explicit weight clipping / normalization is applied (to make Thm 1 apply)? 5. How does this analysis interact with different optimizers? What is the difference between using momentum-based / Adagrad etc for SNNs? 6. Hyper-parameters of f need to be set precisely, otherwise f is not domain-preserving / contraction mapping anymore. How sensitive is the model to this choice? 7. It is unclear how the fixed-point analysis theoretically relates to the generalization performance of the found solution. Is there a theoretical connection?